

SciPost Phys. Lect. Notes 77 (2023)

# Covariant canonical formulations of classical field theories

**François Gieres⋆**

Institut de Physique des 2 Infinis de Lyon, Université Claude Bernard Lyon 1 and
CNRS/IN2P3, Bat. P. Dirac, 4 rue Enrico Fermi, F-69622-Villeurbanne (France)

⋆ gieres@ipnl.in2p3.fr

## Abstract

We review in simple terms the covariant approaches to the canonical formulation of classical relativistic field theories (in particular gauge field theories and general relativity) and we discuss the relationships between these approaches as well as the relation with the standard (non-covariant) Hamiltonian formulation. Particular attention is paid to conservation laws (notably related to geometric symmetries) within the different approaches. Moreover, for each of these approaches, the impact of space-time boundaries is also addressed. To make the text accessible to a wider audience, we have included an outline of Poisson and symplectic geometry for both classical mechanics and field theory.


---

## Contents

Dedicated to the memory of Isadore Manuel Singer (1924-2021)

*who established many original and profound results in pure mathematics which found important applications in physics to which he also contributed substantially by his work and his action. By his wonderful lectures on field theory he conveyed his broad knowledge while highlighting the essential points and casting the physical concepts in their appropriate mathematical setting. An outstanding, undogmatic and open-minded mathematician of great simplicity and generosity left us while leaving a lasting impression on all those who had the chance to meet him.*

# 1   Introduction

## 1.1   Motivation and scope

In view of the quantization of a given classical Lagrangian field theory, one is interested in a *canonical formulation* of the theory and in particular in Poisson brackets, notably of observables (gauge invariant functionals) in a gauge field theory. Since time derivatives of fields are treated differently from spatial derivatives in the standard Hamiltonian approach to classical field theory on Minkowski space-time,[1] Lorentz covariance is not manifest. By way of consequence, the proof of relativistic invariance of physical results may require a fair amount of cumbersome technical work. Similarly, in curved space-time, the general covariance of field theory described by an approach which distinguishes the time coordinate is at the origin of messy details to be handled [1]. All of these difficulties thus appear to be artificial and due to the chosen approach. For this reason, covariant canonical formulations (that do not distinguish any of the space-time coordinates) have been sought for which retain as much as possible the advantages of the standard Hamiltonian approach. Two approaches (based on the notions of *multiphase space* and of *covariant phase space,* respectively) have attracted a lot of attention during the last decades following ideas put forward, in particular, towards 1970 by the Warsaw school [2] (notably J. Kijowski [3], K. Gawędzki [4] and W. M. Tulczyjew [5]) and independently by the Spanish school [6,7] as well as H. Goldschmidt and S. Sternberg [8]. Other formulations (introduction of *Peierls bracket* and *variational bicomplex* approach) have been considered in part with the same motivation in mind. In the present text, we briefly outline the corresponding approaches while having in mind the treatment of symmetries and in particular the conserved currents/charges associated to Poincaré invariance in gauge field theories. We note that the presentations given in the literature often rely on a great amount of mathematical machinery, but we will try to give an as simple as possible local formulation while referring to the literature for the mathematical refinements. *Apart from giving an overview, our main goal is to emphasize the numerous and important relations between the presented approaches* some of which have already been considered in the pioneering works, though they are often ignored in the literature. It turns out that the variational bicomplex approach not only represents an all-embracing mathematical framework for the covariant approaches, but also corresponds to the local formulation that is often considered in the physics literature. Our considerations are restricted to the classical theory and the quantization will only be commented upon.

---

[1]In this respect, we recall that for a collection $x \mapsto \varphi(x) \equiv (\varphi^a)_{a=1,\dots,N}$ of classical relativistic fields on $n$-dimensional Minkowski space-time parametrized by $x \equiv (x^\mu)_{\mu=0,1,\dots,n-1}$ and for a given Lagrangian density $\mathcal{L}(\varphi, \partial_\mu \varphi, x)$ (with $\partial_\mu \varphi \equiv \partial \varphi / \partial x^\mu$), the *Hamiltonian density* is defined by $\mathcal{H}(\varphi^a, \pi_a, \partial_k \varphi^a, x) \equiv \pi_a \dot{\varphi}^a - \mathcal{L}$ where $\dot{\varphi}^a \equiv \partial_0 \varphi^a$ and $\pi_a \equiv \partial \mathcal{L} / \partial \dot{\varphi}^a$, the index $k \in \{1, \dots, n-1\}$ labeling the spatial coordinates.

## 1.2   Outline of text

In section 2, we provide a short pre-/overview of the (relations between the) different approaches discussed in the sequel of the text in the form of a synthetic flow diagram whose content is outlined. In sections 3-6, we successively present the different covariant approaches to the canonical formulation of classical relativistic field theories (i.e. *multisymplectic geometry,* the *Peierls bracket,* the *covariant phase space* and the *variational bicomplex*) together with the relationships between them. In section 6, we focus on symmetries and conservation laws, in particular in relationship with gauge symmetries (notably diffeomorphism invariance in gravity): this presentation relies on the language and framework of the variational bicomplex as well as covariant phase space. The presentations given in the different sections are largely independent of each other and the level of details provided is somewhat uneven: this reflects our desire to limit the corresponding mathematical subtleties as well as our endeavor to focus on the features of the different formulations which can be directly related to each other.

Four appendices provide some mathematical and physical background. In fact, various textbooks on mathematical physics provide a detailed and extensive introduction to the mathematical/geometric formulation of classical mechanics and some also include a discussion of field theory (infinite-dimensional Hamiltonian systems). In the appendices C and D, we briefly recall the formulation of Hamiltonian mechanics and field theory which is familiar to physicists and relate it to the geometric description by introducing only those notions which are necessary for having a background and mathematical underpinning for the points addressed in the main body of the text. The appendices A and B gather some general notions of differential geometry as well as a synthetic introduction to the different differentials which are considered in the space of fields over space-time.

## 1.3   About space-times and boundaries close by or far away

Though we limit ourselves (for the sake of simplicity) to Minkowski space-time in many of our considerations, we also address curved space-times in relationship with gravitational theories. Therefore, some general comments are in order concerning the mathematical assumptions made for space-time manifolds as well as the eventual presence of boundaries.

**Structure of space-time:** Quite generally, space-time is assumed to represent a *n-dimensional Lorentzian manifold* $(M, g)$ i.e. a real smooth $n$-manifold endowed with a Lorentzian metric tensor field $(g_{\mu\nu})$. More precisely, this manifold is supposed to be oriented and time-oriented [9–12]. Moreover, for some considerations $(M, g)$ will be assumed to be *globally hyperbolic.* The latter property implies that there is a foliation of $M$ by Cauchy hypersurfaces. More precisely, $M$ is diffeomorphic to $\mathbb{R} \times \Sigma$ ($\cong \mathbb{R} \times \Sigma_t$ for any $t \in \mathbb{R}$): here $\Sigma_t$ represents a Cauchy hypersurface for $M$ at $t \in \mathbb{R}$ (i.e. $\Sigma_t$ is a smooth, boundaryless submanifold of codimension 1 which is such that any inextendible causal curve intersects $\Sigma_t$ exactly once) [9, 11, 12]. Different illustrations of this mathematical structure of space-time are given by Figures 3 and 5 of Section 5 as well as Figure 8 of Section 7 (where boundaries are assumed to be present). We note that the class of globally hyperbolic space-times contains most, if not all, examples of physical interest, in particular the Minkowski, Schwarzschild, de Sitter, Friedmann-Robertson-Walker,... space-times [13].

**About boundaries and boundary conditions/terms:** Minkowski space-time $M = \mathbb{R}^n$ is unbounded and specific physical or mathematical considerations may lead us to consider some fall-off conditions for fields or symmetry parameters as one approaches infinity in some direction, e.g. for $|\vec{x}| \equiv (x_1^2 + \cdots + x_{n-1}^2)^{1/2} \to \infty$. These conditions can be viewed as conditions imposed at a "space-time boundary at infinity". More generally, for any space-time manifold

$M$, the presence of distinguished hypersurfaces (like isolated horizons or entanglement surfaces) generally requires the specification of *boundary conditions* and possibly the introduction of *boundary terms* (e.g. into the action functional). Just as the introduction of boundary conditions has a crucial impact on the properties of observables in quantum mechanics (notably on their spectrum [14]), such conditions also have important mathematical and physical implications in field theory (e.g. for the uniqueness of the solutions of field equations or of the Green functions associated to differential operators, for the existence and properties of global charges,...).

More specifically, in the context of general relativity, the interest into boundaries and boundary terms at infinity has several motivations. First of all, the fact that the Einstein-Hilbert Lagrangian depends on second order derivatives of the metric tensor leads to the introduction of a boundary term [15,16] which ensures that the variational principle yields Einstein's field equations, e.g. see reference [17]. (For gauge theories and gravity, the systematics of boundary actions was quite recently addressed by the authors of reference [18] in the framework of the covariant phase space formalism.) The definition of a conserved charge as an integral over a hypersurface at infinity calls for an investigation of these hypersurfaces and related properties of fields. Furthermore, it has been realized that there exist various interesting relationships between concepts or theories defined in the *bulk* and concepts or theories defined on its *boundary*.

**Boundary/bulk correspondence:** While this idea of a holographic description already finds a basic expression in Noether's theorems (which imply a relation between gauge symmetries and conserved charges given by surface integrals), different dualities have ben pointed out like the so-called *AdS/CFT correspondence* or the so-called *fluid/gravity duality* as well as various generalizations like *celestial holography*. For an overview of these ideas (including several hundred references to the literature), we refer to the guiding introduction of the recent notes of L. Ciambelli [19] dedicated to asymptotic symmetries and the so-called *corner proposal*.

**Point of view adopted in these notes:** A space-time region may have boundary components which are space-like, time-like or null. Moreover, there may be corners [16,20] where different components of the boundary join each other, see Figure 8 of Subsection 7.8. Since a full-fledged treatment of general boundaries or boundary conditions requires the introduction of extra mathematical technicalities (e.g. see reference [16] for the gravitational a action) we have refrained from addressing this issue in a systematic manner in these introductory notes. In this respect we also note that some of the related issues still continue to represent a quite active field of investigation without a complete consensus on the physical interpretations or fundamental character of the different proposals. Accordingly, for each of the different approaches to covariant canonical formulations of classical field theory that we consider, we have limited ourselves to providing an outline of procedures to tackle boundaries along with some some indications to the literature (subsections 3.5, 4.5, 6.7 and 7.8).

## 1.4   On the notation and conventions

We generally use the *notation* that is standard in the physics literature. An exception is our presentation of the variational bicomplex where we prefer to follow the mathematical literature while spelling out the relationship with the notation considered in theoretical physics.

We focus on relativistic theories and our signature of the Minkowski metric is "mostly minus" $(+-\cdots-)$. We generally use the natural system of units (where $c \equiv 1 \equiv \hbar$) and denote the space-time coordinates by $x = (x^\mu)_{\mu=0,1,\dots,n-1} \in \mathbb{R}^n$. The explicit derivative with respect

to $x^\mu$ is denoted by $\partial/\partial x^\mu$ and the total derivative by $\partial_\mu$, e.g.

$$(\partial_\mu \mathcal{L})\big(\varphi(x), \partial_\nu \varphi(x), x\big) \equiv \frac{\partial \mathcal{L}}{\partial \varphi} \partial_\mu \varphi + \frac{\partial \mathcal{L}}{\partial(\partial_\nu \varphi)} \partial_\mu(\partial_\nu \varphi) + \frac{\partial \mathcal{L}}{\partial x^\mu}, \qquad \text{where} \;\; \partial_\mu \varphi = \frac{\partial \varphi}{\partial x^\mu}.$$

The Lagrangian densities $\mathcal{L}$ are generally assumed to be of *first order* though some comments are made on higher orders. For our illustrations of general results, we consider bosonic fields (scalar and vector fields): at the classical level, *spinor fields* are described by anticommuting variables so that one has to take into account signs as well as the distinction between right and left functional derivatives, etc. The DeWitt notation [21] allows for a unified formulation of all fields with the appropriate signs, but we refrain from using it in our introductory overview.

For the *functional derivative* of a first order action functional $S[\varphi] \equiv \int d^n x\, \mathcal{L}(\varphi(x), \partial_\mu \varphi(x), x)$ (with $\varphi \equiv (\varphi^a)_{a=1,\dots,N}$) with respect to a field $\varphi^a$, we use the notation $\delta S/\delta \varphi^a$, i.e.

$$\frac{\delta S}{\delta \varphi^a} \equiv \frac{\partial \mathcal{L}}{\partial \varphi^a} - \partial_\mu \left( \frac{\partial \mathcal{L}}{\partial(\partial_\mu \varphi^a)} \right).$$

We remark that this expression is also denoted by $\frac{\delta \mathcal{L}}{\delta \varphi^a}$ in the literature and then referred to as the *Euler-Lagrange derivative*, e.g. see [22] (and [23] for a general discussion in the framework of classical mechanics).

## 1.5 Note on the references

We generally refer to the pioneering works (from the seventies on) which are devoted to field theory as well as to recent assessments. Quite generally , the subject is rooted in the *geometric formulation of variational calculus* which has been addressed by many authors over the years, in particular J.-L. Lagrange [24], V. Volterra [25], C. Carathéodory [26], T. De Donder [27], H. Weyl [28], T. Lepage [29], P. Dedecker [30]. The historical evolution of ideas has been retraced in a certain number of recent works, e.g. [31–38] and the more recent evolution related to field theory will be outlined in section 2.

## 2 Pre-/overview of results and historical evolution of the subject

At the end of this section, we present the different approaches (as well as some of the main relationships between them) in the form of a synthetic flow diagram. This diagram provides a schematic overview and partial summary of the facts and results to be discussed in the following sections. We will presently outline the content of the flow diagram while trying to convey already some of the basic concepts and ideas (within their historical context), the details being postponed to the later parts of the notes.

Our starting point is a classical, relativistic Lagrangian field theory (e.g. a gauge field theory) on Minkowski space-time $\mathbb{R}^n$ or more generally on a $n$-dimensional Lorentzian manifold $M$ which is supposed to be globally hyperbolic. Thus, we have a collection of fields $x \mapsto \varphi(x) \equiv (\varphi^a(x))$ with $a \in \{1,\dots,N\}$ on space-time $M$ and a Lagrangian density $\mathcal{L}(\varphi, \partial_\mu \varphi, x)$ (generally assumed to be of first order) determining the dynamics of fields. In the simplest instance on which we focus in general, the field $x \mapsto \varphi(x)$ can be viewed as a section in the trivial fibre (more precisely vector) bundle $E \equiv \mathbb{R}^n \times \mathbb{R}^N \to M \equiv \mathbb{R}^n$ over Minkowski space-time $M$, i.e. it amounts to a smooth map

$$\begin{aligned} s : M &\longrightarrow E \\ x &\longmapsto s(x) \equiv \big(x, \varphi(x)\big), \qquad \text{with} \quad \varphi(x) \in \mathbb{R}^N. \end{aligned} \tag{1}$$

The space $E = \mathbb{R}^n \times \mathbb{R}^N$ is finite-dimensional and parametrized by local coordinates $(x^\mu, q^a)$ where the variables $q^a$ correspond to the value of the field $\varphi^a$ at $x \in M$. Since a first order Lagrangian density $\mathcal{L}(\varphi, \partial_\mu \varphi, x)$ not only depends on $x$ and $\varphi$, but also on the first order derivatives of $\varphi$, it represents a function on the so-called *1-jet bundle* $J^1 E \equiv JE = \mathbb{R}^n \times \mathbb{R}^N \times \mathbb{R}^{nN}$ over $M$ which is parametrized by $(x^\mu, q^a, q_\mu^a)$ where $q_\mu^a$ corresponds to the value of $\partial_\mu \varphi^a$ at $x \in M$, see Figure 1 in Section 3. The fibre bundle $E$ itself may be viewed as 0-jet bundle, i.e. $J^0 E = E$. Higher jet-bundles $J^p E$ (with $p = 2, 3, \dots$) amount to the introduction of higher order derivatives, e.g. for the field equations it is appropriate to consider the 2-jet bundle $J^2 E$ which involves extra coordinates $q_{\mu\nu}^a = q_{\nu\mu}^a$. The notion of jet-bundle has been introduced by the mathematician C. Ehresmann in the fifties [39] and obviously provides the adequate mathematical framework for formulating Lagrangian field theories and more generally partial differential equations on a base manifold $M$, e.g. see Subsection 6.1 or references [40–42] for more details on jet-bundles. Indeed, the finite-dimensional space $JE$ and the so-called *extended multiphase space* $P \equiv J^{\circledcirc} E$ (which represents a kind of dual of $JE$, see the synthetic Figure 1 in Section 3 and the related comments after Eqn. (11)) are at the heart of the *multisymplectic* or *multiphase space approach* to the canonical formulation of field theory. The so-called *variational bicomplex approach* (discussed in Section 6) rather relies on the infinite-jet bundle $J^\infty E$ over $M$ which is parametrized by $(x^\mu, q^a, q_\mu^a, q_{\mu\nu}^a, \dots)$. In the following, we outline the content of the flow diagram. Each of the lines of the latter is elaborated upon in the section or appendix which is indicated in parenthesis.

**Line (1) of flow diagram (Appendix D):[2]** The **standard Hamiltonian formulation** of classical relativistic field theories (put forward in 1929 by W. Heisenberg and W. Pauli [44]) relies on the introduction of the *conjugate variables* $\pi_a \equiv \partial \mathcal{L} / \partial \dot{\varphi}^a$ and of the *canonical Hamiltonian* $H \equiv \int d^{n-1}x \, \mathcal{H}(\varphi^a, \pi_a, \partial_k \varphi^a, x)$, the Lagrangian equations of motion becoming the Hamiltonian equations $\dot{\varphi}^a = \delta H / \delta \pi_a$, $\dot{\pi}_a = -\delta H / \delta \varphi^a$. As was already realized by Heisenberg and Pauli, the presence of gauge symmetries yields *constraints,* e.g. $0 = \pi_0 \equiv \partial \mathcal{L} / \partial \dot{A}^0$ for the Lagrangian density $\mathcal{L} \equiv -\frac{1}{4} F^{\mu\nu} F_{\mu\nu}$ of the free electromagnetic field. A procedure to tackle this problem was put forward by L. Rosenfeld [45] in 1930 as well as by P. Bergmann (and his collaborators) and in particular by P. A. M. Dirac in the fifties (who rediscovered and refined various results of Rosenfeld [46]). Some standard references for *Dirac's formulation of constrained Hamiltonian systems* are [23, 47] and we will refer more specifically to [43] for the treatment of Poincaré symmetries in this setting. Here, we only note that the main ingredients of this canonical formulation of classical field theory are the inclusion of *constraints* into the canonical Hamiltonian by virtue of *Lagrange multiplier fields* (leading to the so-called *extended Hamiltonian $H_E$*) and the consideration of *gauge fixing functions* (leading to the so-called *Dirac brackets* of functions on phase space): for finite-dimensional dynamical systems, the geometric treatment of constraints and the definition of the Dirac bracket are described in detail in Appendix C.6.

**Line (2).*a* of flow diagram (Section 3):** Two classes of covariant canonical approaches to classical field theory have already been considered in the pioneering works of the Polish school [3–5]. The first one is the so-called **multiphase** or **multisymplectic approach** which generalizes the geometric formulation of *classical mechanics* based on the "dynamical" Poincaré-Cartan 1-form $p_a dq^a - \mathcal{H} dt = (p_a \dot{q}^a - \mathcal{H}) dt = \mathcal{L} dt$ (see Eqn. (C.30) of Appendix C.5). In *field theory,* the relativistic covariance is now ensured from the beginning on by associating a *momentum vector field* $\pi_a^\mu \equiv \partial \mathcal{L} / \partial (\partial_\mu \varphi^a)$ to each field $\varphi^a$ and by considering the so-called

---

[2]Appendix D provides a review of the Hamiltonian formulation of classical field theories. A concise introduction to the modification of this formulation which is brought about by constraint equations (that appear in particular in gauge field theories) can be found in sections 5 and 7 of reference [43].

*covariant (or De Donder-Weyl) equations* $\partial_\mu \varphi^a = \partial\mathcal{H}/\partial\pi^\mu_a$, $\partial_\mu\pi^\mu_a = -\partial\mathcal{H}/\partial\varphi^a$ involving the *covariant Hamiltonian density* $\mathcal{H} \equiv \pi^\mu_a \partial_\mu \varphi^a - \mathcal{L}$ (see Subsection 3.1 for more details). The values $q^a$ and $p^\mu_a$ of the fields $\varphi^a$ and $\pi^\mu_a$ at a point $x$ then represent a finite-dimensional space: together with the space-time coordinates $x^\mu$ and an energy-type variable $p$ (corresponding to the value of the Hamiltonian scalar $\mathcal{H}$), these coordinates parametrize the so-called *extended multiphase space* $P = J^{\circlearrowright} E$ mentioned above. The "kinematical" Poincaré-Cartan 1-form $\theta = p_a dq^a - E dt$ of mechanics (see Eqn. (C.26)) generalizes in field theory to a naturally given $n$-form on $P$ whose exterior derivative is the so-called *multisymplectic $(n + 1)$-form* $\omega \equiv -d\theta$ on $P$: the latter generalizes the symplectic 2-form $\omega = -d\theta = dq^a \wedge dp_a + dE \wedge dt$ of classical mechanics (see Eqn. (C.26)) and it also gives rise to the so-called *multisymplectic brackets* [48] of differential forms on $P$ (to be discussed in Subsection 3.3 below). Thus, one has in particular a Poisson bracket of $(n-1)$-forms, the latter forms corresponding to current densities ($j^\mu$) whose conservation laws are generally associated to global symmetries of the theory by virtue of Noether's first theorem.

**Line ②.*b* of flow diagram (Section 5):**   An alternative covariant approach, referred to as the **covariant phase space approach**, consist of viewing a phase space like the one of classical mechanics parametrized by $(q^a, p_a) \in \mathbb{R}^{2N}$ as the space of trajectories, i.e. the *space of solutions of the classical equations of motion.* Denoting this infinite-dimensional space in field theory by $Z$, one can introduce a non-degenerate Poisson bracket for functionals on this space or, equivalently, define a symplectic form on it: such a Poisson structure was already introduced in 1952 by R. Peierls [49] in his attempt to construct a covariant canonical approach to field theory and it was elaborated in detail by B. S. DeWitt [21] (see Subsection 4.2.2 for the details). More recently, it has been applied for devising a rigorous approach to the perturbative renormalization of field theory, see [50] and references therein. The symplectic form $\Omega$ which is associated to the Peierls bracket was introduced directly (i.e. without reference to the Peierls bracket) by the Polish school [3, 4] and it involves the multisymplectic $(n + 1)$-form mentioned above (see Eqn. (158) of Subsection 5.2). The fact that the symplectic form $\Omega$ is actually associated to the Peierls bracket was first established by a Hamiltonian analysis in reference [51] and corroborated by more general arguments in reference [52] (see also references [53, 54] for the case of gauge field theories). The fact that the Peierls bracket and the Dirac bracket coincide for physical observables (i.e. gauge invariant functionals) in gauge field theory was also shown in reference [51] (cf. Subsection 5.3 below). More recently, the relationship between the multisymplectic bracket of $(n-1)$-forms mentioned above and the Peierls bracket has been elucidated as well [55].

Concerning the historic development of the subject, we note that the covariant phase space approach has a long and complex history which can be traced back to J.-L. Lagrange (see reference [33]). Precursory ideas in classical field theory include the work of I. E. Segal [56]. Covariant phase space was introduced more precisely in 1970 in the context of the symplectic formulation of classical mechanics by J.-M. Souriau [57] and, shortly thereafter, in the context of classical field theory by the Polish school [3,4]. It was rediscovered by E.Witten [58] and by G. Zuckerman [59] and other authors in the eighties. In the sequel, it has been further elaborated and applied, in particular in the context of gravity following the work of R. Wald [60–65], A. Ashthekar [38] and their collaborators as well as various other authors, e.g. see [66] and references therein.

**Line ② of flow diagram (Section 6):**   While multisymplectic geometry relies on the 1-jet bundle $JE$ over $M$, the **variational bicomplex** (introduced independently by I. M. Gel'fand and his collaborators [67], A. M. Vinogradov [68], W. M. Tulczyjew [69] and F. Takens [70] at the end of the Seventies) rather relies on the infinite-jet bundle $J^\infty E$ over $M$ which is parametrized

by $(x^\mu, q^a, q^a_\mu, q^a_{\mu\nu}, \dots)$. In this approach it is natural (both from the mathematical and physical points of view) to decompose the exterior derivative (acting on differential forms) on $J^\infty E$, i.e. the differential

$$d = dx^\mu \frac{\partial}{\partial x^\mu} + dq^a \frac{\partial}{\partial q^a} + dq^a_\mu \frac{\partial}{\partial q^a_\mu} + \dots,$$

into horizontal and vertical parts: one writes

$$d = d_{\mathrm{h}} + d_{\mathrm{v}}, \qquad \text{where} \quad \begin{cases} d_{\mathrm{h}} \equiv dx^\mu \partial_\mu, & \text{with} \quad \partial_\mu \equiv \frac{\partial}{\partial x^\mu} + q^a_\mu \frac{\partial}{\partial q^a} + \cdots, \\ d_{\mathrm{v}} \equiv \theta^a \frac{\partial}{\partial q^a} + \dots, & \text{with} \quad \theta^a \equiv dq^a - q^a_\mu dx^\mu + \dots, \end{cases}$$

see Eqn. (177) below for more details. Here, $\partial_\mu$ represents the total derivative with respect to $x^\mu$ and the differential $d_{\mathrm{v}}$ describes an infinitesimal field variation. Remarkably enough, this enlarged mathematical framework not only allows us [59,71,72] to give a precise mathematical formulation of the covariant phase space approach, it also encompasses (and even simplifies [73]) the approach of multisymplectic geometry (see Subsections 6.4 and 6.5). Henceforth, it provides a *unified treatment of the different covariant approaches* and it also allows us [74] to rephrase the traditional formulation of field theory [23] conveniently in the language of jet-bundles, thereby clarifying its mathematical underpinnings (cf. Sections 6 and 7). Indeed, the consideration of the infinite jet-bundle $J^\infty E$ (i.e. of an arbitrary high order of derivatives of fields $\varphi^a$) provides a general mathematical setting for dealing with symmetries and conservation laws as well as to tackle the inverse problem of variational calculus. Moreover, the above-mentioned horizontal and vertical parts of the differential $d$ on $J^\infty E$ represent the exterior derivative and field variation which are usually introduced in the physics literature in the context of classical field theory.

# 3 Multiphase (or multisymplectic) approach

Our starting point is a relativistic first order Lagrangian field theory on an $n$-dimensional space-time manifold $M$ which we assume to be given, for simplicity, by Minkowski space-time $M = \mathbb{R}^n$ and parametrized by coordinates $x \equiv (x^\mu) \equiv (t, \vec{x})$. Thus, we have a Lagrangian density $\mathcal{L}(\varphi^a, \partial_\mu \varphi^a, x)$ depending on a collection of classical relativistic fields $\varphi^a : M \to \mathbb{R}$ with $a \in \{1, \dots, N\}$ and we assume that all fields and their derivatives fall off sufficiently fast at spatial infinity. For some considerations (e.g. the conservation of energy and momentum addressed in Subsection 3.2), we suppose that $\mathcal{L}$ does not explicitly depend on $x$. Some general references for the present section are given by [33,75–77].

## 3.1 Generalities

An explicitly Lorentz covariant Hamiltonian formulation can be achieved by associating to each field $\varphi^a$ (with $a \in \{1, \dots, N\}$) of the Lagrangian formulation a *canonical momentum vector field* defined by

$$\pi^\mu_a \equiv \frac{\partial \mathcal{L}}{\partial(\partial_\mu \varphi^a)}, \tag{2}$$

and then considering the so-called *covariant* (or *De Donder-Weyl*) *Hamiltonian*

$$\mathcal{H}(x^\mu, \varphi^a, \pi^\mu_a) \equiv \left(\pi^\mu_a \partial_\mu \varphi^a - \mathcal{L}\right)\big|_{\partial_\mu \varphi^a = \text{function of } (\pi^\mu_a, \varphi^a, x^\mu)}, \tag{3}$$

coincide for gauge invariant fctls.

Hamiltonian $H_E$ + Dirac bracket of fctls. on $P$

Multisymplectic Poisson bracket of $(n-1)$-forms

← related →

Peierls bracket of fctls. on $Z$

← coincide →

Symplectic Poisson bracket of fctls. on $Z$

For constraints: Lagrangian multipliers + gauge fixing

Multisymplectic bracket of forms $\{\cdot,\cdot\}_{ms}$

Symplectic 2-form $\Omega$ on $Z$
$$\Omega(X,Y) = \int_\Sigma \phi^*(i_Y\, i_X\, \omega_\mathcal{H})$$

phase space $P = \{\varphi^a, \pi_a \equiv \frac{\partial \mathcal{L}}{\partial \dot\varphi^a}\}$
+ Hamiltonian e.o.m.
+ constraints for gauge theories
($\leadsto$ reduced phase space)

multisymplectic approach
$\{\phi = (\varphi^a, \pi_a^\mu \equiv \frac{\partial \mathcal{L}}{\partial(\partial_\mu\varphi^a)})\}$
+ covar. Hamiltonian e.o.m.
$\leadsto$ finite-dim. multiphase space
$\{(q^a, p_a^\mu)\}$ with multisymplectic $(n+1)$-form $\omega$
$\leadsto \omega_\mathcal{H} \equiv \mathcal{H}^*\omega$

covariant phase space $Z$
$\equiv$ {solutions of e.o.m.}
($=$ infinite-dimensional)

non covar. canonical approach

variational bicomplex

2.a

2.b

① 

② 

Classical Lagr. field (gauge) theory

covariant canonical approaches

e.g. see references [3, 33, 37, 75–83]. (Here, we mention in particular the comprehensive *GIMmsy* papers [78] which addressed the relation of multisymplectic geometry with the standard canonical approach to field theory that is generally considered for quantization.)

Thereby, the Lagrangian equations of motion $0 = \frac{\partial \mathcal{L}}{\partial \varphi^a} - \partial_\mu \left( \frac{\partial \mathcal{L}}{\partial (\partial_\mu \varphi^a)} \right)$ are equivalent to the so-called *covariant Hamiltonian* or *De Donder-Weyl equations*

$$\boxed{\partial_\mu \varphi^a = \frac{\partial \mathcal{H}}{\partial \pi_a^\mu}, \qquad \partial_\mu \pi_a^\mu = -\frac{\partial \mathcal{H}}{\partial \varphi^a}.} \tag{4}$$

Very much like the standard Hamiltonian equations, these equations follow from the variational principle $0 = \delta \int_M \mathcal{L} \, d^n x$ (by varying the fields $\varphi^a$ and $\pi_a^\mu$ independently), i.e. by virtue of (3),

$$0 = \delta \int_M \mathcal{L} \, d^n x, \qquad \text{with } \mathcal{L} \, d^n x = \left( \pi_a^\mu \partial_\mu \varphi^a - \mathcal{H} \right) d^n x. \tag{5}$$

Indeed, we have

$$\delta \int_M \mathcal{L} \, d^n x = \int_M d^n x \left[ \delta \pi_a^\mu \left( \partial_\mu \varphi^a - \frac{\partial \mathcal{H}}{\partial \pi_a^\mu} \right) - \delta \varphi^a \left( \partial_\mu \pi_a^\mu + \frac{\partial \mathcal{H}}{\partial \varphi^a} \right) \right] + \int_M d^n x \, \partial_\mu (\pi_a^\mu \delta \varphi^a), \tag{6}$$

where the last term vanishes for field variations that vanish at infinity.

We note that the $n$-form $\mathcal{H} \, d^n x$ in expression (5) determines the

Lagrangian $n$-form: $\qquad \boxed{\mathcal{L} \, d^n x = \pi_a^\mu \, d\varphi^a \wedge d^{n-1} x_\mu - \mathcal{H}(\varphi, \pi) \, d^n x,} \tag{7}$

with

$$d^n x \equiv dx^0 \wedge dx^1 \wedge \cdots \wedge dx^{n-1}, \qquad d^{n-1} x_\mu \equiv i_{\partial_\mu} d^n x.$$

In the latter equation, $i_{\partial_\mu} d^n x$ denotes the interior product (contraction) of the differential form $d^n x$ by the vector field $\partial_\mu$ (see Appendix A for these mathematical notions). In this respect, we note the useful relation $dx^\nu \wedge d^{n-1} x_\mu = \delta_\mu^\nu d^n x$.

In the present context, we recall that the *standard Hamiltonian approach* to classical field theory relies on the idea that fields

$$x \equiv (t, \vec{x}) \longmapsto \varphi^a(x) = \varphi^a(t, \vec{x}) \equiv \varphi_{\vec{x}}^a(t), \tag{8}$$

generalize the configuration space coordinates $t \mapsto \vec{q}(t)$ of classical non-relativistic mechanics; thereby, they distinguish the time coordinate and, for any given time $t$, they yield an infinite-dimensional space of fields labeled by $\vec{x} \in \mathbb{R}^{n-1}$ (as well as the discrete index $a \in \{1, \ldots, N\}$). The *covariant approach* to the canonical formalism views the fields $\varphi^a$ and the associated momenta $\pi_a^\mu$ as functions of $x \in \mathbb{R}^n$:

$$x \longmapsto \left( \varphi^a(x), \pi_a^\mu(x) \right). \tag{9}$$

For any given space-time point $x$, one thus has a finite-dimensional space $\{(q^a, p_a^\mu)\}$. On the latter (extended by an energy-type variable $p$), we have the following canonically given differential forms: the so-called (cf. Eqn. (7))

"kinematical" multicanonical $n$-form: $\qquad \theta \equiv p_a^\mu \, dq^a \wedge d^{n-1} x_\mu - p \, d^n x, \tag{10}$

and the associated $(n+1)$-form, i.e. the

"kinematical" multisymplectic form: $\qquad \boxed{\omega \equiv -d\theta = dq^a \wedge dp_a^\mu \wedge d^{n-1} x_\mu + dp \wedge d^n x.} \tag{11}$

These differential forms do not depend on the dynamics (i.e. on the Hamiltonian $\mathcal{H}$) and they depend on the variables $q^a, p_a^\mu$ and $p$ which parametrize the so-called *"extended multiphase space"* $P$, this space representing a fibre bundle over the space-time manifold[3] which is parametrized by $(x^\mu)$: the variables $q^a, p_a^\mu$ and $p$ correspond to the values of $\varphi^a, \pi_a^\mu$ and $\mathcal{H}$, respectively (fields, momenta and energy-type variable).

In this geometric set-up (see Figure 1 below), a Hamiltonian $\mathcal{H}$ can be viewed as a map $\mathcal{H} : \tilde{P} \to P$ from the so-called *"ordinary multiphase space"* $\tilde{P}$ (parametrized by $(x^\mu, q^a, p_a^\mu)$) to the extended multiphase space $P$ (parametrized by $(x^\mu, q^a, p_a^\mu, p)$). By pulling back the $n$-form (10) (which is defined on $P$) to $\tilde{P}$ by virtue of the Hamiltonian $\mathcal{H} : \tilde{P} \to P$, we obtain the so-called *Poincaré-Cartan $n$-form* or

*"dynamical" multicanonical $n$-form:*
$$\boxed{\theta_\mathcal{H} \equiv \mathcal{H}^* \theta = p_a^\mu \, dq^a \wedge d^{n-1}x_\mu - \mathcal{H}(x^\mu, q^a, p_a^\mu) \, d^n x \,.}$$
(12)

This form and its differential $\omega_\mathcal{H} \equiv -d\,\theta_\mathcal{H} = -d(\mathcal{H}^*\theta) = -\mathcal{H}^*(d\theta) = \mathcal{H}^*\omega$, i.e. the

*"dynamical" multisymplectic form:*
$$\omega_\mathcal{H} = dq^a \wedge dp_a^\mu \wedge d^{n-1}x_\mu + d\mathcal{H} \wedge d^n x \,,$$
(13)

determine both the *canonical structure* and the *dynamics of the classical field theory* under consideration.

Concerning the latter point, we note that from the geometric point of view (see Figure 1), a field $\varphi \equiv (\varphi^a)$ represents a smooth section of some vector (or more generally fibre) bundle $E$ over $M$, i.e. $\varphi : M \to E$. (Here and elsewhere, we make a slight abuse of notation: for a section $s : M \to E$, we have $x \mapsto s(x) = (x, \varphi(x))$ which implies that $q^a = \varphi^a(x)$ as assumed above.) The collection of fields $\phi \equiv (\varphi^a, \pi_a^\mu)$ represents a smooth section of ordinary multiphase space $\tilde{P}$, i.e.

$$\phi : M \longrightarrow \tilde{P}$$
$$x \longmapsto \big(x, \varphi^a(x), \pi_a^\mu(x)\big),$$
(14)

where $\big(\varphi^a(x), \pi_a^\mu(x)\big) \equiv (q^a, p_a^\mu)$ denote the coordinates in the fibre $\tilde{P}_x$ over $x \in M$. The field $\phi : M \to \tilde{P}$ allows us to pull back the $n$-form (12), which is defined on $\tilde{P}$, to $M$ (see Appendix A for the definition and properties of the pullback): this yields the

*Lagrangian $n$-form:*
$$\phi^* \theta_\mathcal{H} = \pi_a^\mu \, d\varphi^a \wedge d^{n-1}x_\mu - \mathcal{H}(\varphi, \pi) \, d^n x = \mathcal{L} \, d^n x \,.$$
(15)

This result was to be expected since it is the Lagrangian $n$-form on $M$ which motivated the expression of the multicanonical $n$-form $\theta$ on $P$, cf. Eqn. (10).

Accordingly, the present approach allows us to formulate classical field theory in a finite-dimensional setting without considering functional derivatives on an infinite-dimensional space as in the standard approach. For this so-called *multisymplectic approach,* one can introduce numerous variants (like the polysymplectic approach [84]), e.g. see references [76, 85] for a partial overview. However, many of these variants rely on an *a priori* given connection which is not natural for gauge field theories where connections represent dynamical variables. We will not further describe the considered mathematical framework here (though we will expand on some aspects of it in Section 6) and only provide a synthetic table [48] (see Table 1) in which a *comparison* is made *with the geometric formulation of explicitly time-dependent systems in classical mechanics.*[4] Indeed, the latter case amounts to choosing $n = 1$

---

[3]From the mathematical point of view, the extended multiphase space $P = J^{\circleddash}E$ represents the so-called *twisted affine dual* of the space $JE$, see references [40, 48] for more details.

[4]In view of this comparison we use the notation $(q^a, p_a)$ for phase space coordinates in mechanics rather than the traditional notation $(q^i, p_i)$ considered in Appendix C.

in the field theoretic setting: for instance, the multisymplectic $(n+1)$-form (11) then becomes $\omega = dq^a \wedge dp_a + dp \wedge dt$, i.e. the symplectic 2-form in mechanics for a time-dependent system (see the end of Appendix C, equations (C.26)-(C.30)).

By way of illustration, we consider the free, neutral Klein-Gordon field $\varphi$ described by the Lagrangian density $\mathcal{L}(\varphi, \partial_\mu \varphi) = \frac{1}{2}(\partial^\mu \varphi)(\partial_\mu \varphi) - \frac{m^2}{2}\varphi^2$. The canonical momentum vector then reads $\pi^\mu \equiv \partial \mathcal{L}/\partial(\partial_\mu \varphi) = \partial^\mu \varphi$ and the covariant Hamiltonian writes $\mathcal{H}(\varphi, \pi^\mu) = \frac{1}{2}\pi^\mu \pi_\mu + \frac{m^2}{2}\varphi^2$. By contrast to the standard Hamiltonian density $\mathcal{H}_{can}$ which represents the energy density (its integral $\int d^{n-1}x\, \mathcal{H}_{can} \equiv P^0$ being the time-component of the energy-momentum four-vector ($P^\mu$)), the covariant Hamiltonian $\mathcal{H}(\varphi, \pi^\mu)$ is a scalar field.

A partial list of works/applications (in which further references are indicated) is:

- Non-relativistic mechanics [5, 7, 78]

- Relativistic mechanics [3]

- Hydrodynamics [86, 87]

- Gauge field theories [3–5, 34, 54, 78, 87–91]

- Chern-Simons theory [90]

- Spinor fields [92]

- Gravity [5, 89, 93–97]

- String theory [78, 87, 90]

- Branes [98]

- Massive spin-2 field [97]

- Topological field theories [78, 99]

- Korteweg-De Vries equation [100]

- WZW conformal field theory [101]

- Rarita-Schwinger field [92]

- Supersymmetric sigma-models in two dimensions [102]

- Fronsdal theory for massless fields of arbitrary integer spin [97]

- Carollian scalar field theory [90]

To conclude we mention the geometric approach to (gauge) field theories developed by M. Grigoriev and his collaborators which is related to the multisymplectic and covariant phase space formulations of field theory [97, 103]: this approach originates from (and in some sense reduces to) the so-called AKSZ sigma model formulation [104] providing a geometric description of topological field theories.

Figure 1: Geometric set-up of multisymplectic geometry.

Table 1: *Classical (time-dependent) mechanics versus classical field theory.*

| Classical mechanics ($n = 1$) | Classical field theory |
| --- | --- |
| Extended configuration space $\mathbb{R} \times \mathcal{Q}$ (Trivial bundle $\mathbb{R} \times \mathcal{Q} \longrightarrow \mathbb{R}$ over the time axis $\mathbb{R}$, with fiber $\mathcal{Q}$ = configuration space manifold) | "Field configuration bundle" $E \longrightarrow M$ (Vector bundle over an $n$-dimensional space-time manifold $M$, with typical fiber $Q$) |
| Coordinates of $\mathbb{R} \times \mathcal{Q}$ : $t, q^a$ | Coordinates of $E$ : $x^\mu, q^a$ |
| Extended velocity space $\mathbb{R} \times T\mathcal{Q}$ with coordinates $t, q^a, \dot{q}^a$ | Velocity bundle: 1-jet bundle $JE$ with coordinates $x^\mu, q^a, q^a_\mu$ |
| Doubly extended phase space $\mathcal{P} \equiv T^*(\mathbb{R} \times \mathcal{Q}) = \mathbb{R}^2 \times T^*\mathcal{Q}$ | "Extended multiphase space": Twisted affine dual $P \equiv J^{\circledcirc}E$ of 1-jet bundle $JE$ |
| Coordinates of $\mathcal{P}$ : $t, q^a, p_a, E$ ($E$ = energy) | Coordinates of $P$ : $x^\mu, q^a, p^\mu_a, p$ ($p$ = energy-type variable) |
| "Kinematical" Poincaré-Cartan 1-form on $\mathcal{P}$: $\theta = p_a \, dq^a - E \, dt$ | "Kinematical" Poincaré-Cartan $n$-form on $P$: $\theta = p^\mu_a \, dq^a \wedge d^{n-1}x_\mu - p \, d^n x$ |
| Symplectic 2-form on $\mathcal{P}$: $\omega \equiv -d\theta = dq^a \wedge dp_a + dE \wedge dt$ | Multisymplectic $(n+1)$-form on $P$: $\omega \equiv -d\theta = dq^a \wedge dp^\mu_a \wedge d^{n-1}x_\mu + dp \wedge d^n x$ |
| Extended phase space $\tilde{\mathcal{P}} \subset \mathcal{P}$ defined by $E = H(t, \vec{q}, \vec{p})$ | Ordinary multiphase space $\tilde{P} \equiv \vec{J}^{\circledcirc}E$ defined by $p = \mathcal{H}(x^\mu, q^a, p^\mu_a)$ |
| Coordinates of $\tilde{\mathcal{P}}$ : $t, q^a, p_a$ | Coordinates of $\tilde{P}$ : $x^\mu, q^a, p^\mu_a$ |
| "Dynamical" Poincaré-Cartan 1-form on $\tilde{\mathcal{P}}$: $\theta_\mathcal{H} = p_a \, dq^a - H \, dt$ | "Dynamical" Poincaré-Cartan $n$-form on $\tilde{P}$: $\theta_\mathcal{H} = p^\mu_a \, dq^a \wedge d^{n-1}x_\mu - \mathcal{H} \, d^n x$ |
| "Dynamical" symplectic 2-form on $\tilde{\mathcal{P}}$: $\omega_\mathcal{H} \equiv -d\theta_\mathcal{H} = dq_a \wedge dp^a + dH \wedge dt$ | "Dynamical" multisymplectic $(n+1)$-form on $\tilde{P}$: $\omega_\mathcal{H} \equiv -d\theta_\mathcal{H} = dq^a \wedge dp^\mu_a \wedge d^{n-1}x_\mu + d\mathcal{H} \wedge d^n x$ |

## 3.2 Energy-momentum tensor

Let us briefly sketch the description of geometric symmetries and conservation laws for the dynamical system given by a free neutral Klein-Gordon field $\varphi$. The fact that the corresponding Lagrangian density $\mathcal{L}(\varphi, \partial_\mu \varphi)$ does not explicitly depend on the space-time coordinates $x^\mu$ is equivalent to the invariance condition $L_{\partial_\mu} \theta_{\mathcal{H}} = 0$ (where $L_{\partial_\mu}$ denotes the Lie derivative with respect to the vector field $\partial_\mu$) for the Poincaré-Cartan $n$-form $\theta_{\mathcal{H}}$ corresponding to the Lagrangian $n$-form $\phi^* \theta_{\mathcal{H}} = \mathcal{L} d^n x$. It now follows [3] from Noether's theorem that one has conserved quantities given by the $(n-1)$-forms $\alpha_\mu \equiv (i_{\partial_\mu} \theta)\big|$ where $\theta$ is the "kinematical" multicanonical form (10) and where the bar denotes the restriction to the space of solutions $\{(\varphi, \pi^\mu)\}$ of the Hamiltonian equations of motion (4). A short calculation in two space-time dimensions (giving results that generalize to $n$ dimensions) yields [3]

$$\alpha_\mu \equiv (i_{\partial_\mu} \theta)\bigg| = -\sum_{\nu=0}^{n-1} (-1)^\nu T^{\;\nu}_\mu \, dx^0 \wedge \cdots \wedge \widehat{dx^\nu} \wedge \cdots \wedge dx^{n-1}, \quad \text{with} \quad T^{\;\nu}_\mu = (\partial^\nu \varphi)(\partial_\mu \varphi) - \delta^\nu_\mu \mathcal{L}.$$
(16)

Here, $\widehat{dx^\nu}$ denotes the omission of the monomial $dx^\nu$ and the quantities $T^{\;\nu}_\mu$ are the components of the EMT (energy-momentum tensor) of the scalar field $\varphi$. The treatment of Lorentz transformations is more complex, but it can be dealt with along the same lines.

For gauge field theories, constraints again appear. For instance, for the free Maxwell Lagrangian $\mathcal{L} = -\frac{1}{4} F^{\mu\nu} F_{\mu\nu}$ we have the canonical momentum tensor $\pi^{\mu\nu} \equiv \partial \mathcal{L}/\partial(\partial_\mu A_\nu) = -F^{\mu\nu}$ satisfying the constraint equation $\pi^{\mu\nu} + \pi^{\nu\mu} = 0$ which leads to complications related to the gauge symmetry [4, 5, 54, 88, 91].

## 3.3 Multisymplectic brackets and quantization

**Multisymplectic brackets:** By extending the multisymplectic Poisson brackets introduced in the pioneering works [3, 4, 8] (and being motivated by the work of I. V. Kanatchikov [79, 105]) M. Forger, C. Paufler and H. Römer [48] introduced a graded bracket of (certain) forms on extended multiphase space $P = J^\bigcirc E$ (endowed with the multisymplectic $(n+1)$-form $\omega = -d\theta$). The bracket of functions (0-forms) on $P$ vanishes and thereby this bracket has some similarities with the so-called Koszul bracket of differential forms on a symplectic manifold [106]. (For any two functions $f, g$ on a symplectic manifold, their Poisson bracket is related to the Koszul bracket $\{\cdot, \cdot\}_K$ of the one-forms $df, dg$ by $\{df, dg\}_K = -d\{f, g\}$ while $\{f, g\}_K = 0$.)

Since a current density $(j^\mu)$ on an $n$-dimensional space-time manifold $M$ is the Hodge dual of a $(n-1)$-form on $M$, the latter forms and their brackets are of particular interest in field theory. A $(n-1)$-form $\alpha$ on the multisymplectic manifold $(P, \omega)$ is said to be Hamiltonian or a *Hamiltonian form* [48] if there exists a vector field $X$ on $P$ such that $i_X \omega = d\alpha$. The (uniquely defined) vector field $X$ which is associated in this way to the $(n-1)$-form $\alpha$ is denoted by $X_\alpha$. For any two Hamiltonian $(n-1)$-forms $\alpha, \beta$ on $P$, the aforementioned *multisymplectic bracket* is defined by

$$\{\alpha, \beta\}_{\mathrm{ms2}} \equiv i_{X_\beta} i_{X_\alpha} \omega + d\left[ i_{X_\beta} \alpha - i_{X_\alpha} \beta - i_{X_\beta} i_{X_\alpha} \theta \right].$$
(17)

This bracket has all of the properties which are required to hold for a Poisson bracket (see (C.10)) apart from the fact that it is does not satisfy the derivation (Leibniz) rule (C.11). Actually, there is no reasonable candidate for an associative product on the space of Hamiltonian $(n-1)$-forms on $P$.

The fact that the multisymplectic bracket of forms on extended multiphase space $P$ does not have all of the defining properties of a Poisson bracket can be better understood [55] by relating these multisymplectic brackets to the Peierls bracket discussed in Section 4.

**On the quantization:** Concerning the quantization, we note that the lack of canonical pairs of dynamical variables in the multisymplectic approach has led to the consideration of alternative quantization procedures like geometric quantization or Schrödinger's functional approach, e.g. see references [107–110] as well as [84, 111, 112] for the polysymplectic formulation. Quantization procedures starting from the De Donder-Weyl equations or from the De Donder-Weyl-Christodoulou formulation of covariant Hamilton-Jacobi equations of classical field theory [80] have also been addressed in other works (see [82, 113–116] and references therein) while using or relating in part to the work of I. V. Kanatchikov [111].

### 3.4 Relationship with Dirac's procedure

Here, we wish to describe how the expressions and results of Dirac's procedure for the standard Hamiltonian approach (in particular the FCC's and the gauge transformations that they generate, as discussed for instance in section 5.3 of reference [43]) are encoded in the multisymplectic approach to free Maxwell theory in four space-time dimensions. In this respect, we note that the general mathematical formulation of YM-theories has been addressed in the unpublished third part of the GIMmsy papers [78] and that the case of non-Abelian topological BF-theory in four space-time dimensions has been investigated quite recently in reference [99].

To start with, we specify the multicanonical form (10) for the free Maxwell theory: for $n = 4$, we have the 4-form

$$\theta = p^{\mu\nu} \, da_\nu \wedge d^3 x_\mu - p \, d^4 x \, , \tag{18}$$

where $a_\nu$ corresponds to the values of the gauge field $A_\nu$. The local gauge transformations parametrized by an arbitrary function $x \mapsto \epsilon(x)$ are generated on *extended multiphase space P* (i.e. a finite-dimensional space parametrized by the local coordinates $(x^\mu, a_\nu, p_{\mu\nu}, p)$) by the vector field

$$\xi = (\partial_\mu \epsilon) \frac{\partial}{\partial a_\mu} + (\partial_\mu \partial_\nu \epsilon) p^{\mu\nu} \frac{\partial}{\partial p} \, ,$$

where $\partial_\mu \epsilon$ reflects the usual gauge transformation of the gauge potential $A_\mu$. The local gauge invariance of the theory finds its expression in the relation

$$L_\xi \theta = 0 \, , \tag{19}$$

where $L_\xi \equiv i_\xi d + d \, i_\xi$ denotes the Lie derivative of differential forms on $P$ with respect to the vector field $\xi$. One says that one has an *exact* or a *natural symmetry*. (As a matter of fact, equation (19) is a generalization of the description of symmetries in classical non-relativistic mechanics [117]: in that case, the phase space with local coordinates $(q^i, p_i)$ (i.e. the cotangent bundle $T^*Q$ associated to the configuration space manifold $Q$ which is parametrized by local coordinates $(q^i)$) is endowed with a canonically given 1-form $\theta = p_i \, dq^i$ and natural symmetries of the theory leave $\theta$ invariant, i.e. $L_\xi \theta = 0$ where $\xi$ denotes the symmetry generating vector field.) The symmetry of Maxwell's theory described by (19) leads to a conserved current density $(j^\mu)$ which is presently described by the following 3-form on $P$:

$$J(\xi) \equiv i_\xi \theta = p^{\mu\nu} i_\xi(da_\nu) \wedge d^3 x_\mu = p^{\mu\nu} \partial_\nu \epsilon \, d^3 x_\mu \, . \tag{20}$$

Actually, the components $J^\mu(\xi) = p^{\mu\nu} \partial_\nu \epsilon$ of the 3-form $J(\xi)$ are the values of the vector field $j^\mu = F^{\mu\nu} \partial_\nu \epsilon$ which represents the (on-shell) conserved current density associated to local gauge invariance (see Eqn. (236) below).

Vector fields $\xi$ on $P$ which leave the multicanonical form $\theta$ invariant, i.e. satisfy (19), are referred to as *exact Hamiltonian vector fields* [48]. The map $J : \xi \mapsto i_\xi \theta$ associating the 3-form $i_\xi \theta$ to an exact Hamiltonian vector field $\xi$ is referred to as *covariant momentum map* of

the theory. Indeed, quite generally, conserved Noether currents of the Lagrangian formulation appear in the Hamiltonian formulation under the disguise of momentum maps.[5]

To relate the multisymplectic approach to the standard ("instantaneous") Hamiltonian approach, one has to consider a *space plus time* (3+1) *decomposition,* i.e. a foliation of the space-time manifold into space-like hypersurfaces [78, 119]. To do so, we consider 3-dimensional hypersurfaces $\Sigma_t$ in space-time corresponding to fixed values of time $t$. Restriction of the covariant momentum map $J$ defined by (20) to such a hypersurface amounts to the restriction $\sigma$ of fields to $\Sigma_t$: this yields [78, 99] a map $J_t$ whose action on $\xi$ writes

$$\langle J_t(A,\pi),\xi\rangle = \int_{\Sigma_t} \sigma^* J(\xi) = \int_{\Sigma_t} \sigma^*(p^{0\nu}\,\partial_\nu\epsilon\,d^3x_0)\,.$$

Since $\sigma^* p^{0\nu} \equiv \pi^\nu$ represents the canonical momentum associated to the gauge field $A_\nu$, we obtain

$$\langle J_t(A,\pi),\xi\rangle = \int_{\Sigma_t} d^3x\,\pi^\nu\,\partial_\nu\epsilon\,. \tag{21}$$

This expression is nothing but the gauge generator for the (Lagrangian) gauge transformations in Dirac's formulation of the constrained Hamiltonian system, e.g. see equation (5.15) of reference [43]. *In summary,* the restriction of the covariant momentum map (of the multisymplectic formulation), that is determined by the invariance of the theory under local gauge transformations, yields upon a $(3+1)$-decomposition the generator of (Lagrangian) gauge transformations of the standard Hamiltonian approach – see references [78, 99] for other examples and for the mathematical underpinnings.

As a matter of fact, the *extended Hamiltonian* of Dirac's formulation (e.g. see equation (5.9) of reference [43]) can also be recovered from the multisymplectic approach by considering the vector field $\zeta \equiv \partial_0$ on $P$ which generates the $(3+1)$-slicing of space-time:

$$H_{t,\zeta} \equiv -\int_{\Sigma_t} \sigma^* i_\zeta \theta = -\int_{\Sigma_t} \sigma^*\big[p^{\mu\nu}\,da_\nu \wedge i_{\partial_0}(d^3x_\mu) - \mathcal{H}\,d^3x_0\big]\,. \tag{22}$$

The details of this calculation (which uses the field equations) are more involved (see [99] for BF-theories), but one can readily identify in (22) the gauge fixing terms appearing in the extended Hamiltonian. It should be interesting to explore further if and how all results of Dirac's formulation (e.g. see [43] and references therein) are encoded in the multisymplectic formulation.

## 3.5   About space-times with boundaries

In this subsection, we will indicate the modifications brought about the consideration of space-time manifolds $M$ with a boundary $\partial M$ while following references [120, 121]. In view of the relations between the multisymplectic and covariant phase space approaches (cf. equations (158)-(159) below), we mention that some of the underlying ideas and concepts have been put forward earlier (and applied to quantization) by K. Gawędzki and his collaborators in their study of sigma models in the framework of covariant phase space [122].

**Generalities:**   For a Lorentzian manifold $M$ endowed with a metric tensor $(g_{\mu\nu})$, the volume form on $M$ as written in terms of local coordinates $(x^\mu)$ reads $\mathrm{vol}_M = \sqrt{|g|}\,d^nx$, but the local coordinates can be chosen such that $\mathrm{vol}_M = d^nx$. With this choice [120], we still have the

---

[5]The origin of the name "momentum map" can be traced back to the fact that the invariance of a mechanical system under translations and rotations leads to momentum maps whose values are given by the linear and angular momentum, respectively [117, 118].

local expressions encountered in Subsection 3.1, e.g. in equations (4) and (6). If the local coordinates on $\partial M$ are given by $x^1, \ldots, x^{n-1}$, then the boundary term in Eqn. (6), i.e.

$$S_{\partial M} \equiv \int_M \partial_\mu(\pi_a^\mu \delta\varphi^a) \, d^n x = \int_{\partial M} \pi_a^\mu \delta\varphi^a \, d^{n-1}x_\mu \,, \tag{23}$$

reads as follows:

$$S_{\partial M} = \int_{\partial M} \pi_a \delta\varphi^a \, d^{n-1}x \equiv \Theta \,, \qquad \text{with } \pi_a \equiv \pi_a^0\big|_{\partial M} \,, \quad \varphi^a \equiv \varphi^a\big|_{\partial M} \,. \tag{24}$$

(A more precise mathematical formulation is obtained by considering the inclusion map $i : \partial M \hookrightarrow M$, i.e. by writing the fields on $\partial M$ as $\pi_a^0 \circ i$ and $\varphi^a \circ i$, but we will not use this notation here.)

Expression (24) represents the *canonical 1-form* on the phase space $\mathcal{P}_{\partial M}$ over the manifold $\partial M$ which is parametrized by $\varphi^a$ and $\pi_a$ and which is encountered in the standard Hamiltonian formulation (e.g. see Eqn. (D.17) for the symplectic 2-form

$$\Omega \equiv -\delta\Theta = \int_{\partial M} \delta\varphi^a \wedge \delta\pi_a \, d^{n-1}x \,, \tag{25}$$

associated to the canonical 1-form $\Theta$). Thus, *there is a natural relation between the (variation of the) action functional for the fields in the bulk of a manifold with boundary and the canonical symplectic structure on the phase space of boundary fields.* We also note that the integrand $j^\mu \equiv \pi_a^\mu \delta\varphi^a$ in expression (23) represents the *"symplectic potential" current density*: this quantity plays a basic role in the covariant phase space approach, cf. Eqn. (83) below.

**Example:** By way of illustration [120] of the general setting based on equations (6) and (24), we consider the case of *real scalar fields* $\varphi^a$ on a $n$-dimensional space-time manifold $(M, g)$. More precisely, this manifold is supposed to be globally hyperbolic and to have a boundary $\partial M$ given by a Cauchy hypersurface $\Sigma$, i.e. $M \cong \, ]-\infty, a] \times \partial M$ for some $a \in \mathbb{R}$. The boundary of $M$ is then parametrized by local coordinates $(x^1, \ldots, x^{n-1})$ and the *phase space of boundary fields* is parametrized by $\varphi^a|_{\partial M}$ and $\pi_a|_{\partial M}$ (with $\pi_a \equiv \partial\mathcal{L}/\partial\dot\varphi^a$), the associated canonical 1-form being given by (24). Near the boundary $\partial M$ (i.e. in a collar $U_\varepsilon \equiv \, ]-\varepsilon, 0] \times \partial M$ with $\varepsilon > 0$ small) one can consider local coordinates $t = x^0$ and $(x^1, \ldots, x^{n-1})$ where $t$ represents the time evolution parameter. The space-time line element may be assumed to be given by

$$ds^2 = dt^2 + 2g_{0i}(x)dt\,dx^i - h_{ij}(x)dx^i dx^j \,, \quad \text{in} \quad U_\varepsilon = \, ]-\varepsilon, 0] \times \partial M \,, \tag{26}$$

where $(h_{ij})$ denotes a Riemannian metric on $\partial M$. Near the boundary, we now have an *extended phase space* $\mathcal{P}_M$ which is parametrized by $\varphi^a$ and $\pi_a^\mu$, i.e. it involves the degrees of freedom corresponding to the boundary fields (i.e. to the elements of $\mathcal{P}_{\partial M}$) as well as extra degrees of freedom which correspond to the *transversal components* $\pi_a^1, \cdots \pi_a^{n-1}$ *of the momentum vector field* $(\pi_a^\mu)$. We can use the canonical projection $\rho : \mathcal{P}_M \to \mathcal{P}_{\partial M}$ onto the boundary fields to pull back the 2-form (25) from $\mathcal{P}_{\partial M}$ to $\mathcal{P}_M$, i.e. consider the closed 2-form $\tilde\Omega \equiv \rho^*\Omega$ on the extended phase space $\mathcal{P}_M$. The transversal components $\pi_a^k$ span the kernel of this *presymplectic 2-form* $\tilde\Omega$ since the contraction of $\tilde\Omega$ with the phase space vector fields $\delta/\delta\pi_a^k$ vanishes, i.e. $i_{\delta/\delta\pi_a^k}\tilde\Omega = 0$ (cf. Eqn. (C.40) for the finite-dimensional case and Eqn. (110) below for a detailed discussion of gauge field theories).

The subset of the De Donder-Weyl equations (4) on $M$ given by the relations $\partial_k\varphi^a = \partial\mathcal{H}/\partial\pi_a^k$ presently corresponds to *constraint equations* on extended phase space while the other equations describe the time evolution of $\varphi^a$ and $\pi_a^0$. Accordingly, the geometric

treatment of constrained Hamiltonian systems (as outlined in Appendix C.6 for the finite-dimensional case) applies. For instance, for a single free neutral scalar field $(\varphi^a) \equiv \varphi$ on the space-time $M$ with line element (26) near the boundary, we have the Lagrangian density $\mathcal{L} \equiv \frac{1}{2} g^{\mu\nu} \partial_\mu \varphi \, \partial_\nu \varphi - \frac{m^2}{2} \varphi^2$, hence $\pi^\mu \equiv \partial \mathcal{L}/\partial(\partial_\mu \varphi) = g^{\mu\nu} \partial_\nu \varphi$ and

$$
\begin{aligned}
\mathcal{H} \equiv \pi^\mu \partial_\mu \varphi - \mathcal{L} &= \frac{1}{2} g_{\mu\nu} \pi^\mu \pi^\nu + \frac{m^2}{2} \varphi^2 \\
&= \frac{1}{2} \pi^0 \pi^0 + g_{0i} \pi^0 \pi^i - \frac{1}{2} h_{ij} \pi^i \pi^j + \frac{m^2}{2} \varphi^2 .
\end{aligned}
\tag{27}
$$

In this example, the constraint equations

$$
0 = \partial_i \varphi - \frac{\partial \mathcal{H}}{\partial \pi^i} = \partial_i \varphi - g_{0i} \pi^0 + h_{ij} \pi^j ,
$$

can be solved for $\pi^i$ in terms of the phase space boundary fields $\varphi$ and $\pi^0$:

$$
\pi^i = h^{ij}(g_{0j} \pi^0 - \partial_j \varphi) .
\tag{28}
$$

The restriction of the extended phase space $\mathcal{P}_M$ (parametrized by $\varphi$ and $\pi^\mu$) to the subspace of fields satisfying relation (28) yields a space $\mathcal{C} \subset \mathcal{P}_M$ on which the presymplectic 2-form $\tilde{\Omega} = \rho^* \Omega$ is no longer degenerate. Thus, one has obtained a *symplectic diffeomorphism between the boundary phase space $\mathcal{P}_{\partial M}$* (endowed with the symplectic form $\Omega$) *and the reduced phase space $\mathcal{C}$* (endowed with the symplectic form $\tilde{\Omega}_\mathcal{C}$ which is obtained by restricting $\tilde{\Omega}$ from $\mathcal{P}_M$ to $\mathcal{C}$), i.e.

$$
(\mathcal{P}_{\partial M}, \Omega) \cong (\mathcal{C}, \tilde{\Omega}_\mathcal{C}), \qquad \text{with} \begin{cases} \mathcal{C} \subset \mathcal{P}_M \text{ defined by relation (28)}, \\ \tilde{\Omega}_\mathcal{C} \equiv (\rho^* \Omega)|_\mathcal{C} \text{ with the projection map } \rho : \mathcal{P}_M \to \mathcal{P}_{\partial M} . \end{cases}
$$

The systematic study of Lagrangian field theories on manifolds with boundary and their (BV or BFV) quantization has been initiated in reference [121] and developed in the multisymplectic [120] and other frameworks while applying it to various classes of field theories (Yang-Mills theories, gravity, Poisson sigma models, string theory, topological models,...). These approaches lead in particular to a geometric understanding of boundary conditions [120,123] (and of their admissibility in gauge field type theories), thereby providing an adequate basis for the canonical quantization.

## 3.6 Brief summary

The multisymplectic approach has some quite attractive features from the mathematical and physical points of view: *it relies on a finite-dimensional phase space, enjoys manifest Lorentz covariance, ensures locality and allows for a global formulation on generic space-time manifolds.* Yet, a definite consensual formulation is still lacking, the construction of observables in generic theories is still under study [86, 124–126] the Poisson bracket does not have all the familiar properties and a covariant quantization procedure still needs to be elaborated in full detail.

## 4 Peierls bracket in field theory

The so-called Peierls bracket has been introduced in 1952 by R. E. Peierls [49] in his attempt to provide a (relativistically) covariant canonical formulation of field theory, e.g. see references [21, 127, 128] for detailed discussions and [129] for an overview of the literature. Since the notion of simultaneity is not relativistically invariant, Peierls replaced the canonical Poisson

bracket which involves fields at a fixed time (thus leading to commutators of field operators at equal times in quantum theory) by a new bracket. Peierls did not consider the case of gauge field theories which was investigated later on, in particular by B. DeWitt. Our presentation is mainly based on the comprehensive investigation made by the latter author [21]. In order to relate the subject to basic and familiar notions and expressions, we start with an elementary introduction by considering the case of a free neutral Klein-Gordon field $\varphi$ on Minkowski space-time $M = \mathbb{R}^4$.

## 4.1 Free scalar field on Minkowski space-time

The action functional $S[\varphi] = \int_{\mathbb{R}^4} d^4x \, \mathcal{L}(\varphi, \partial_\mu \varphi)$ with $\mathcal{L}(\varphi, \partial_\mu \varphi) \equiv \frac{1}{2}(\partial^\mu \varphi)(\partial_\mu \varphi) - \frac{1}{2} m^2 \varphi^2$ yields the equation of motion $(\Box + m^2)\varphi = 0$. The basic solutions of this equation are given by the plane waves $\varphi(x) \propto e^{\pm ipx}$ with $p \equiv (p^0, \vec{p})$ and $p^0 = \omega_{\vec{p}} \equiv \sqrt{\vec{p}^2 + m^2}$. Thereby, the general solution of the field equation is a "continuous superposition" of these basic solutions (e.g. see reference [130]):

$$\varphi(x) = \int_{\mathbb{R}^3} d^3\tilde{p} \left[ a_{\vec{p}} \, e^{-ipx} + a_{\vec{p}}^* \, e^{ipx} \right]\bigg|_{p^0 = \omega_{\vec{p}}}, \quad \text{with } d^3\tilde{p} \equiv \frac{d^3p}{(2\pi)^{3/2}\sqrt{2\omega_{\vec{p}}}}. \tag{29}$$

Here, $\vec{p} \mapsto a_{\vec{p}}$ is an arbitrary complex-valued function of $\vec{p} \in \mathbb{R}^3$ which can be viewed as a functional of the initial values $\varphi(0, \vec{x})$ and $\dot{\varphi}(0, \vec{x})$:

$$a_{\vec{p}} = \int_{\mathbb{R}^3} \frac{d^3x}{(2\pi)^{3/2}} \left[ \sqrt{\frac{\omega_{\vec{p}}}{2}} \, e^{i\vec{p}\cdot\vec{x}} \, \varphi(0, \vec{x}) + \frac{i}{\sqrt{2\omega_{\vec{p}}}} \, e^{-i\vec{p}\cdot\vec{x}} \, \dot{\varphi}(0, \vec{x}) \right]. \tag{30}$$

For the field $\varphi$ and the associated canonical momentum $\pi \equiv \frac{\partial \mathcal{L}}{\partial \dot{\varphi}} = \dot{\varphi}$, the canonical Poisson bracket at fixed time $t$ (in particular at $t = 0$) is given by expression (D.29), i.e.

$$\{\varphi(t, \vec{x}), \pi(t, \vec{y})\} = \delta(\vec{x} - \vec{y}).$$

For the "annihilation function" $a_{\vec{p}}$ and the "creation function" $a_{\vec{p}}^*$, it follows from (30) that the previous relation is equivalent to

$$\{a_{\vec{p}}, a_{\vec{p}'}^*\} = -i\,\delta(\vec{p} - \vec{p}'),$$

all other brackets between the functions $a_{\vec{p}}, a_{\vec{p}}^*$ vanishing. From this result and the expansion (29), one readily deduces the classical (so-called *Pauli-Jordan*) *commutator function* [131] for the scalar field $\varphi$:

$$\{\varphi(x), \varphi(y)\} = \tilde{G}(x - y), \qquad \text{with} \quad \boxed{\tilde{G}(x) \equiv -\int_{\mathbb{R}^3} \frac{d^3p}{(2\pi)^3} \frac{\sin px}{\omega_{\vec{p}}}\bigg|_{p^0 = \omega_{\vec{p}}}.} \tag{31}$$

This bracket of free fields at different space-time points is their *Peierls bracket* $\{\cdot, \cdot\}_{\mathrm{P}}$, i.e.

$$\boxed{\{\varphi(x), \varphi(y)\}_{\mathrm{P}} = \tilde{G}(x - y).} \tag{32}$$

Thereby, the *Peierls bracket of two functionals $F, G$ of the field $\varphi$* is given by

$$\boxed{\{F, G\}_{\mathrm{P}} \equiv \int_{\mathbb{R}^4} d^4x \int_{\mathbb{R}^4} d^4y \, \frac{\delta F}{\delta \varphi(x)} \, \tilde{G}(x - y) \, \frac{\delta G}{\delta \varphi(y)}.} \tag{33}$$

*The Peierls bracket has all the properties of a Poisson bracket.* At equal times, it reduces to the canonical Poisson bracket. Yet, by contrast to the canonical Poisson bracket, the bracket (33) is not universal since *it depends on the dynamics* by virtue of the commutator function $\tilde{G}$.

We note that the quantization of the theory proceeds along the usual lines, i.e. by replacing the Peierls bracket $\{\cdot,\cdot\}_{\mathrm{p}}$ by $1/(\mathrm{i}\hbar)$ times the commutator of the corresponding operators, i.e. for the field operators $\hat{\varphi}(x)$ in the Heisenberg picture we have

$$[\hat{\varphi}(x),\hat{\varphi}(y)] = \mathrm{i}\hbar\,\tilde{G}(x-y)\mathbb{1}\,. \tag{34}$$

For $x^0 = y^0$, we now recover the canonical (equal-time) commutation relation $[\hat{\varphi}(x),\hat{\varphi}(y)]|_{x^0=y^0} = 0$. The fact that the function $x \mapsto \tilde{G}(x)$ is invariant under Lorentz transformations $x \rightsquigarrow x' = \Lambda x$ then implies that the bosonic field operators $\hat{\varphi}(x)$ and $\hat{\varphi}(y)$ commute if the space-time points $x$ and $y$ are space-like separated, i.e. if $(x-y)^2 < 0$ (fundamental property of *local causality* in quantum field theory). Instead of the canonical quantization, one can also consider the so-called *deformation quantization* [132], e.g. see references [133–135] for reviews of the latter.

As indicated by the foregoing discussion of classical field theory, the Peierls bracket coincides with the Poisson bracket at different (space-)time points which follows from the standard Hamiltonian formulation (compare equations (31) and (32)). Yet, it is also defined in the case where the Hamiltonian formulation is problematic, e.g. for derivative couplings or for non-local interactions – see next section as well as references [50,53,136] for a general treatment based on the action functional.

Let us come back once more to the commutator function $\tilde{G}$ for a free scalar field as given by (31). This function satisfies the field equation as well as specific initial conditions:

$$\text{Field equation :} \qquad 0 = A\tilde{G} \equiv (\Box + m^2)\tilde{G}\,. \tag{35}$$

$$\text{Initial conditions :} \qquad \tilde{G}(0,\vec{x}) = 0\,, \qquad \dot{\tilde{G}}(0,\vec{x}) = -\delta(\vec{x})\,.$$

Thus, it is a solution of a particular Cauchy problem for the dynamical system under consideration which is described by the hyperbolic operator $A \equiv \Box + m^2$. Indeed, *the commutator function $\tilde{G}$ represents the difference of the retarded and advanced Green functions* of the free Klein-Gordon field:

$$\boxed{\tilde{G} = G_{\mathrm{ret}} - G_{\mathrm{adv}}\,,} \quad \text{with} \quad G_{\mathrm{ret}}(x) = -\theta(x^0)\int_{\mathbb{R}^3}\frac{d^3p}{(2\pi)^3}\,\frac{\sin px}{\omega_{\vec{p}}}\bigg|_{p^0=\omega_{\vec{p}}}\,, \quad G_{\mathrm{adv}}(x) = G_{\mathrm{ret}}(-x)\,. \tag{36}$$

Here, $\theta$ denotes the Heaviside function and we have $AG_{\mathrm{ret}} = -\delta$ and $AG_{\mathrm{adv}} = -\delta$ where $\delta$ denotes the Dirac distribution. The generalized function $G_{\mathrm{ret}}$ satisfies the initial conditions $G_{\mathrm{ret}}(0,\vec{x}) = 0$ and $\dot{G}_{\mathrm{ret}}(0^+,\vec{x}) = -\delta(\vec{x})$ (where the coefficient $(-1)$ reflects [137] the coefficient $(-1)$ of $\partial_t^2$ in the operator $-A = -\partial_t^2 + \Delta - m^2$). By abuse of terminology, the function $\tilde{G}$ which solves the homogenous equation $A\tilde{G} = 0$ is sometimes referred to as *causal "Green function"*, see reference [138] for a general discussion of Green functions on a Lorentzian manifold.[6]

## 4.2 Generic field theories

Our starting point is a classical field theory whose dynamics is described by an action functional $S[\varphi]$. For concreteness and simplicity, we consider a collection $\varphi \equiv (\varphi^a)$ of real-valued

---

[6]We note that the relative sign in the definition (36) of $\tilde{G}$ depends on the global sign which is chosen in the definition of $G_{\mathrm{ret}}$ or, equivalently, of the operator $A$.

classical fields $\varphi^a$ on Minkowski space-time $M \equiv \mathbb{R}^n \equiv \mathbb{R}^{d+1}$, but we emphasize that the presentation given in this section generalizes to fields viewed as sections in a non-trivial fibre bundle $E$ over a Lorentzian manifold $M$ which is globally hyperbolic [21]. An important point will be that time does not play a particular role in this approach, hence it provides manifestly covariant expressions which are useful in different approaches to quantization, in particular the functional integral formulation [21] and the rigorous perturbative quantum field theory approach [50].

### 4.2.1 Jacobi fields, Jacobi equation, Jacobi operator

For a given solution $\varphi$ of the field equation (i.e. $\frac{\delta S}{\delta \varphi^a}[\varphi] = 0$ for all $a$), we consider a neighboring solution $\varphi + \delta\varphi$: we have $\frac{\delta S}{\delta \varphi^a}[\varphi + \delta\varphi] = 0$ if $\delta\varphi$ is a solution of the

$$\text{linearized equation of motion:} \qquad \int_M d^n y \, \mathcal{J}_{ab}^{x,y}[\varphi] \, \delta\varphi^b(y) = 0, \qquad (37)$$

where

$$\mathcal{J}_{ab}^{x,y}[\varphi] \equiv \frac{\delta^2 S}{\delta\varphi^a(x)\delta\varphi^b(y)}[\varphi]. \qquad (38)$$

In the literature, the variation $(\delta\varphi^a)$ is referred to as *Jacobi field* (relative to the on-shell field $\varphi$), the linearized equation of motion (37) as *Jacobi equation* (or *homogeneous equation of small disturbances*) and the expression (38) as *Jacobi operator*.

By way of example, we consider a collection of non-interacting scalar fields $\varphi^a$ of equal mass $m$, i.e. the action functional

$$S[\varphi] = \frac{1}{2} \int_M d^n x \left( \partial^\mu \varphi^a \, \partial_\mu \varphi^a - m^2 \varphi^a \varphi^a \right). \qquad (39)$$

In this case, the Jacobi operator has the simple expression

$$\mathcal{J}_{ab}^{x,y}[\varphi] = -\delta_{ab} (\Box + m^2)\delta(x-y),$$

and thereby the Jacobi equation (37) has the same form as the equation of motion of $\varphi^a$, namely $(\Box + m^2)\delta\varphi^a = 0$.

### 4.2.2 Peierls bracket

**Small perturbation of the action by another functional:** Let $\varphi \mapsto A[\varphi]$ be a real-valued functional and $\varepsilon \in \mathbb{R}$ a small parameter. We view $\varepsilon A[\varphi]$ as effect of a weak external agent producing a small disturbance $S[\varphi] \rightsquigarrow S_A[\varphi] \equiv S[\varphi] + \varepsilon A[\varphi]$ of the action functional $S[\varphi]$. Let $\varphi$ be a solution of the equation of motion determined by $S[\varphi]$ and $\varphi + \delta\varphi$ a solution of the equation of motion determined by $S_A[\varphi]$, i.e. $\frac{\delta S_A}{\delta \varphi^a}[\varphi + \delta\varphi] = 0$: expansion of this equation to order $\varepsilon$ yields the so-called *inhomogeneous equation of small disturbances* or

$$\text{inhomogeneous Jacobi equation:} \qquad \boxed{\int_M d^n y \, \mathcal{J}_{ab}^{x,y}[\varphi] \, \delta\varphi^b(y) = -\varepsilon \frac{\delta A}{\delta\varphi^a(x)}[\varphi].} \qquad (40)$$

E.g. for the action of free scalar fields given by (39), the inhomogeneous Jacobi equation writes

$$(\Box + m^2)\delta\varphi^a(x) = -\varepsilon \frac{\delta A}{\delta\varphi^a(x)}[\varphi]. \qquad (41)$$

The latter equation represents an inhomogeneous linear partial differential for the Jacobi fields $\delta\varphi^a$ and in the following we are interested in the solution of this equation and, more generally, in the solution of Eqn. (40) for $\delta\varphi^a$. A particular solution of the inhomogeneous Jacobi equation can be obtained by the method of Green functions: if the Jacobi operator is invertible, i.e. admits a Green function, then the *convolution* of the latter with the inhomogeneous term yields a solution of the equation. As we will discuss below, the existence of a local invariance of the action $S[\varphi]$ implies that the Jacobi operator is not invertible so that some extra work is required for applying the method of Green functions. Henceforth, we first investigate the case of field theories which do not have local symmetries like the model described by (39) which leads to Eqn. (41).

**Field theories without local symmetries:**   Consider an inhomogeneous Jacobi equation like (41) for which the Jacobi operator admits a Green function. For a hyperbolic operator like $\Box + m^2$, we will consider the *retarded Green function* $G_{\text{ret}} \equiv G^-$ and the *advanced Green function* $G_{\text{adv}} \equiv G^+$, i.e.

$$\int_M d^n y \, \mathcal{J}^{x,y}_{ac}[\varphi] \, G^{\pm cb}_{y,z}[\varphi] = -\delta^b_a \, \delta(x-z), \tag{42}$$

and the corresponding boundary conditions for the regular distributions $G^{\pm}$. Then, we have the following *particular solutions of the inhomogeneous Jacobi equation:*

$$\delta\varphi^{\pm a}(x) = \varepsilon \int_M d^n y \, G^{\pm ab}_{x,y}[\varphi] \, \frac{\delta A}{\delta\varphi^b(y)}[\varphi]. \tag{43}$$

We note that the symmetry of the Jacobi operator (38), i.e. the relation $\mathcal{J}^{x,y}_{ab} = \mathcal{J}^{y,x}_{ba}$, implies the

reciprocity relations for the Green functions:     $G^{\pm ab}_{x,y} = G^{\mp ba}_{y,x}$. $\qquad$ (44)

Now, let $\varphi \mapsto B[\varphi]$ be another real-valued functional. The small disturbance $\varphi \rightsquigarrow \varphi + \delta\varphi^{\pm}$ (caused by the change of action functional $S[\varphi] \rightsquigarrow S_A[\varphi] = S[\varphi] + \varepsilon A[\varphi]$) induces a small change of $B$:

$$B[\varphi] \rightsquigarrow B[\varphi] + \delta^{\pm}B[\varphi], \qquad \text{with} \quad \delta^{\pm}B[\varphi] = \int_M d^n x \, \frac{\delta B}{\delta\varphi^a(x)}[\varphi] \, \delta\varphi^{\pm a}(x).$$

After substituting the explicit expression (43) of $\delta\varphi^{\pm a}(x)$ into the latter expression and using the reciprocity relations (44), we find that

$$\delta^{\pm}B[\varphi] = \varepsilon \, D^{\pm}_A B, \qquad \text{with} \quad D^{\pm}_A B = \int_M d^n x \int_M d^n y \, \frac{\delta A}{\delta\varphi^a(x)} \, G^{\mp ab}_{x,y} \, \frac{\delta B}{\delta\varphi^b(y)}. \tag{45}$$

For these expressions one uses the following terminology:

$$\begin{aligned} D^-_A B &: \quad \text{retarded effect of } A \text{ on } B, \\ D^+_A B &: \quad \text{advanced effect of } A \text{ on } B. \end{aligned} \tag{46}$$

Concerning the Green functions we recall that

$$\tilde{G}^{ab}_{x,y} \equiv G^{-ab}_{x,y} - G^{+ab}_{x,y} \; = \; \text{commutator function}.$$

The **Peierls bracket of the real-valued functionals (observables) $A$ and $B$** is now defined by

$$\{A,B\}_P \equiv D_A^+ B - D_A^- B = \int_M d^n x \int_M d^n y \, \frac{\delta A}{\delta \varphi^a(x)} \, \tilde{G}_{x,y}^{ab} \, \frac{\delta A}{\delta \varphi^b(y)}, \tag{47}$$

where we substituted the foregoing equations to obtain the last expression. Obviously, we have

$$\{\varphi^a(x), \varphi^b(y)\}_P = \tilde{G}_{x,y}^{ab}. \tag{48}$$

For a free scalar field $\varphi$, the results (47)-(48) reduce to those given in equations (32)-(33), the explicit expression of $\tilde{G}$ being then given by (31).

**Field theories with local symmetries:** We now consider an action functional $S[\varphi]$ which is invariant under *local symmetry transformations* that are described at the infinitesimal level by

$$\varphi^a(x) \rightsquigarrow \varphi^a(x) + \delta\varphi^a(x), \qquad \text{with} \quad \boxed{\delta\varphi^a(x) = \int_M d^n y \, Q_r^a(x,y) \, \xi^r(y).} \tag{49}$$

Here, $x \mapsto \xi^r(x)$ represents a smooth real-valued function (of compact support or appropriate fall-off properties) and, for the physically interesting cases, the expression $Q_r^a(x,y)$ has the form

$$Q_r^a(x,y) = \bar{Q}_{(0)r}^a(x) \, \delta(x-y) + \bar{Q}_{(1)r}^{a\mu}(x) \, \partial_\mu \delta(x-y), \tag{50}$$

where the coefficients $\bar{Q}$ are typically functions of the fields and their first order derivatives.

The invariance of the action functional $S[\varphi]$ writes

$$0 = \delta S[\varphi] = \int_M d^n x \, \frac{\delta S}{\delta\varphi^a(x)} \, \delta\varphi^a(x) = \int_M d^n x \int_M d^n y \, \frac{\delta S}{\delta\varphi^a(x)} \, Q_r^a(x,y) \, \xi^r(y).$$

Since this identity holds for any function $\xi^r$, we conclude that the local invariance of $S[\varphi]$ is tantamount to the so-called

$$\text{Noether identities :} \qquad \boxed{0 = \int_M d^n x \, \frac{\delta S}{\delta\varphi^a(x)} \, Q_r^a(x,y).} \tag{51}$$

In fact, Noether's second theorem states that there is a one-to-one correspondence between gauge symmetries and Noether identities – see Section 7.3 for more details and some explicit examples, in particular YM-theories and general relativity.

Any invariant, real-valued functional $\varphi \mapsto A[\varphi]$ is referred to as a *physical observable* (though this does not necessarily mean that it represents a measurable physical quantity). Thus, in a theory without local symmetries, any real-valued functional is an observable and thereby the Peierls bracket (47) then represents a well defined bracket for these observables. We now wish to discuss the definition of the Peierls bracket for the case that we have a local symmetry of the action functional, i.e. the identity (51). First, we note that by functionally differentiating the identity (51), we get the relation

$$\int_M d^n x \, \mathcal{J}_{ab}^{x,y} \, Q_r^a(x,y) = 0, \qquad \text{on-shell (i.e. for } \tfrac{\delta S}{\delta\varphi^a(x)} = 0). \tag{52}$$

This equation simply expresses the invariance of the linearized equations of motion. The relation also means that $Q_r^a(x, y) \xi^r(y)$ *is a null eigenvector* (i.e. eigenvector associated to the eigenvalue zero) *of the Jacobi operator* $\mathcal{J}_{ab}^{x,y}$. By way of consequence, this operator is not invertible in a theory with local symmetries and thereby does not admit Green functions. To overcome this problem, we note that for any solution of the inhomogeneous Jacobi equation (40), the field $\varphi + \delta_\xi \varphi$ (with $\delta_\xi \varphi$ given by (49)) is another solution. This redundancy can be removed by imposing a gauge fixing condition for the on-shell invariance of the linearized equations of motion: For the small disturbances, we impose the so-called

$$\text{supplementary condition}: \qquad 0 = \int_M d^n x \, P_a^r(x) \, \delta \varphi^a(x), \qquad (53)$$

with

$$P_a^r(y) = \left[ M_a^{r\mu}(x) \partial_\mu + N_a^r(x) \right] \delta(x - y),$$

where $M$ and $N$ are functions of $\varphi^b$ and $\partial_\mu \varphi^b$. The operator

$$F_{ab}^{x,y} \equiv \mathcal{J}_{ab}^{x,y} + P_a^r(x) \kappa_{r,s}^{x,y} \, P_b^s(y),$$

then has the same symmetry properties as $\mathcal{J}_{ab}^{x,y}$, i.e. $F_{ab}^{x,y} = F_{ba}^{y,x}$. Moreover, with an appropriate choice of $M, N$ and $\kappa$, and with the assumption that all null eigenvectors of $\mathcal{J}_{ab}^{x,y}$ are of the form $Q_r^a(x, y) \xi^r(y)$, the operator $F_{ab}^{x,y}$ is invertible. When the supplementary condition (53) is imposed on the small disturbances $\delta \varphi$, we can obviously replace $\mathcal{J}_{ab}^{x,y}$ by $F_{ab}^{x,y}$ in the inhomogeneous Jacobi equation (40).

In conclusion, we now have the same set-up as for a theory without local symmetries. Accordingly, *the Peierls bracket* $\{A, B\}_P$ *of any two observables $A, B$* is again defined by (47) where the *Green functions* are now the ones *of the operator* $F_{ab}^{x,y}$. One can then show [21] that $D_A^\pm B$ (and thereby $\{A, B\}_P$) does not depend on the choice of $M, N, \kappa$ and that $\{A, B\}_P$ *is again a physical observable*. Moreover, the Peierls bracket $\{A, B\}_P$ is not changed by adding a linear combination of equation of motion functions to $A, B$:

$$\{A, B\}_P = \{\bar{A}, \bar{B}\}_P, \qquad \text{for} \quad \begin{cases} \bar{A} \equiv A + \int_M d^n x \, \mathsf{a}^a(x) \frac{\delta S}{\delta \varphi^a(x)}, \\ \bar{B} \equiv B + \int_M d^n x \, \mathsf{b}^a(x) \frac{\delta S}{\delta \varphi^a(x)}, \end{cases} \qquad (54)$$

where the functions $\mathsf{a}^a, \mathsf{b}^a$ have appropriate fall-off properties. This implies that *the use of Peierls brackets commutes with the application of on-shell conditions.* If no local symmetries are present, then the real-valued fields $\varphi^a(x), \varphi^b(y)$ are themselves physical observables, hence their Peierls bracket is given by (47) which yields (48).

**Concluding remarks:** The elimination of the zero modes originating from local symmetries manifests itself in slightly different disguises in alternative formulations of the theory: we discuss it in the framework of symplectic geometry in Appendix C.6 and in the covariant phase space approach in Section 5.1.

The action functional $S[\varphi] = \int_M d^n x \, \mathcal{L}(\varphi, \partial_\mu \varphi)$ is equivalent to the action functional

$$S[\varphi, \pi] = \int_M d^n x \left[ \pi_a^\mu \partial_\mu \varphi^a - \mathcal{H}(\varphi^a, \pi_a^\mu) \right],$$

see equations (3)-(5): in Subsection 5.2.2, we will use this form of the action to investigate the relation of the Peierls bracket with the symplectic 2-form appearing in the covariant phase space approach to field theory. Finally, we note that a generalization of the Peierls bracket for non-Lagrangian field theories can be introduced [139] by using the concept of *Lagrange anchor.*

## 4.3   Geometric symmetries and conservation laws

As before, we only consider bosonic fields so as to avoid the issue of signs related to anticommuting variables as well as the introduction of tetrad fields upon coupling to gravity. We are interested in the local conservation laws associated to geometric symmetries and in the corresponding algebra of conserved charges for bosonic matter fields ($\varphi^a$). These can be obtained by coupling the matter fields to gravity described by a metric tensor field $\mathfrak{g}(x) \equiv \big(g_{\mu\nu}(x)\big)$ and by considering the (conformal) Killing vector fields for this metric. (The coupling of matter fields to gauge fields as well as more general instances can be discussed along the same lines [21].) We first present an elementary account illustrated by concrete examples before applying the description of local symmetries outlined above to the case of diffeomorphisms of the space-time manifold $(M, \mathfrak{g})$.

**Basic example:**   For concreteness, we focus on a *free, neutral, scalar field $\phi$ of mass m coupled to gravity:* the dynamics of this field-theoretic system is determined by the action functional

$$S[\phi, \mathfrak{g}] \equiv S_{\text{grav}}[\mathfrak{g}] + S_M[\phi, \mathfrak{g}], \qquad \text{with } S_{\text{grav}}[\mathfrak{g}] \equiv \frac{1}{2\kappa} \int_M d^n x \sqrt{|g|}\, R \tag{55}$$

(where $g \equiv \det \mathfrak{g}$ and $\kappa \equiv 8\pi G$, $G$ being Newton's constant), and

$$\begin{aligned} S_M[\phi, \mathfrak{g}] &\equiv \frac{1}{2} \int_M d^n x \sqrt{|g|}\, \big[ (\nabla^\mu \phi)(\nabla_\mu \phi) - m^2 \phi^2 \big] \\ &= \frac{1}{2} \int_M d^n x \sqrt{|g|}\, \big[ g^{\mu\nu}(\partial_\mu \phi)(\partial_\nu \phi) - m^2 \phi^2 \big]. \end{aligned} \tag{56}$$

Here, $\nabla_\mu$ denotes the covariant derivative (of tensor fields) with respect to the Levi-Civita-connection. The equations of motion of the metric field components are *Einstein's field equations:*

$$0 = \frac{\delta S}{\delta g^{\mu\nu}} = \frac{\delta S_{\text{grav}}}{\delta g^{\mu\nu}} + \frac{\delta S_M}{\delta g^{\mu\nu}} = \frac{\sqrt{|g|}}{2\kappa} \big( G_{\mu\nu} + \kappa T_{\mu\nu} \big), \qquad \text{i.e. } G_{\mu\nu} = -\kappa T_{\mu\nu}, \tag{57}$$

with

$$G_{\mu\nu} \equiv R_{\mu\nu} - \frac{1}{2} g_{\mu\nu} R, \qquad \boxed{T^{\mu\nu}[\phi, \mathfrak{g}] \equiv \frac{-2}{\sqrt{|g|}} \frac{\delta S_M[\phi, \mathfrak{g}]}{\delta g_{\mu\nu}}.} \tag{58}$$

Here, $G_{\mu\nu}$ represents the *Einstein tensor* and $T^{\mu\nu}$ is the so-called *metric* or *Einstein-Hilbert EMT* (in curved space). The identity $\nabla^\mu G_{\mu\nu} = 0$ now implies the *"covariant conservation law"* $\nabla_\mu T^{\mu\nu} = 0$ for the solutions of Einstein's field equations, this relation representing a consistency condition for the field equations (57).

   We note that the action functional (56) yields the following explicit expression for the EMT of the scalar field $\phi$:

$$T^{\mu\nu} = (\nabla^\mu \phi)(\nabla^\nu \phi) - g^{\mu\nu} \mathcal{L}_M, \qquad \text{with } \mathcal{L}_M \equiv \frac{1}{2} \big[ (\nabla^\mu \phi)(\nabla_\mu \phi) - m^2 \phi^2 \big]. \tag{59}$$

This expression is *symmetric* in its indices (as a direct consequence of its definition (58)) and it is *covariantly conserved* (by virtue of the equation of motion of $\phi$ which reads $(\nabla^\mu \nabla_\mu + m^2)\phi = 0$). Moreover, this EMT is *traceless* for a massless field in two space-dimensions since we have, for $m = 0$,

$$T^\mu_{\ \mu} \equiv g_{\mu\nu} T^{\mu\nu} = \frac{2-n}{2} (\nabla^\mu \phi)(\nabla_\mu \phi). \tag{60}$$

In this respect, we mention [140] that tracelessless of the EMT may be ensured for the free massless field $\phi$ in any space-time dimension $n \geq 2$ if one considers the non-minimal *conformally invariant* coupling of the matter field $\phi$ to gravity: the latter consists of adding to the matter field Lagrangian density $\mathcal{L}_M$ the term $-\frac{1}{2}\xi_n R\phi^2$ where $R$ denotes the curvature scalar and $\xi_n \equiv \frac{1}{4}\frac{n-2}{n-1}$, i.e. one considers the Lagrangian density

$$\mathcal{L}_{\text{conf}} \equiv \frac{1}{2}\left[(\nabla^\mu \phi)(\nabla_\mu \phi) - \xi_n R\phi^2\right]. \tag{61}$$

The EMT of $\phi$ is then modified by some extra curvature dependent terms [140], the resulting expression being denoted by $T^{\mu\nu}_{\text{conf}}$.

**Geometric symmetries:** The properties of the EMT for the matter field $\phi$ can be better apprehended by treating the metric field $\mathfrak{g}$ as a fixed background field $\mathring{\mathfrak{g}}$, i.e. an important instance for real physical situations [21]. Then, we do not have a dynamical term for gravity in the action functional, henceforth the latter reduces to the one of the matter field $\phi$ coupled to the external gravitational field: $S[\phi, \mathfrak{g}] = S_M[\phi, \mathring{\mathfrak{g}}]$. The EMT now represents the variation of the total action with respect to the external gravitational field:

$$T^{\mu\nu}[\phi] \equiv \left(\frac{-2}{\sqrt{|g|}}\frac{\delta S}{\delta g_{\mu\nu}}\right)\Bigg|_{\mathfrak{g}=\mathring{\mathfrak{g}}}. \tag{62}$$

The covariant conservation law $\nabla_\mu T^{\mu\nu} = 0$ may presently be viewed as a consequence of the *invariance of the action under diffeomorphisms:* a diffeomorphism $x^\mu \rightsquigarrow x'^\mu(x) \simeq x^\mu - \xi^\mu(x)$ is generated by a smooth vector field $\xi \equiv \xi^\mu \partial_\mu$ and acts on the metric tensor field as

$$\delta_\xi g_{\mu\nu} = \nabla_\mu \xi_\nu + \nabla_\nu \xi_\mu. \tag{63}$$

From the invariance of the action under infinitesimal diffeomorphisms and the use of the equations of motion of matter fields it follows that

$$\begin{aligned}
0 = \delta_\xi S &= \int_M d^n x \left(\underbrace{\frac{\delta S}{\delta \phi}}_{\approx 0}\delta_\xi \phi + \frac{\delta S}{\delta g_{\mu\nu}}\delta_\xi g_{\mu\nu}\right)\\
&= \int_M d^n x \frac{\delta S}{\delta g_{\mu\nu}}2\nabla_\mu \xi_\nu\\
&= \int_M d^n x \sqrt{|g|}\,\xi_\nu \nabla_\mu T^{\mu\nu}.
\end{aligned} \tag{64}$$

From the arbitrariness of $\xi_\nu$ one thus concludes that $\nabla_\mu T^{\mu\nu} = 0$ for the solutions of the matter field equations. Similarly, the tracelessness of the EMT for a free massless field $\phi$ in two space-time dimensions follows from the *Weyl invariance of the action,* the Weyl transformation (or Weyl rescaling) of metric and scalar fields being defined for $n \geq 2$ by

$$\tilde{g}_{\mu\nu} = e^{2\sigma} g_{\mu\nu}, \qquad \tilde{\phi} = e^{-\sigma\frac{n-2}{2}}\phi, \tag{65}$$

where $\sigma$ denotes a smooth real-valued function: with $\delta_\sigma \phi = 0$ for $n = 2$, we have

$$0 = \delta_\sigma S = \int_M d^2 x \frac{\delta S}{\delta g_{\mu\nu}}\delta_\sigma g_{\mu\nu} = \int_M d^2 x \frac{\delta S}{\delta g_{\mu\nu}}2\sigma g_{\mu\nu}, \quad \text{hence } 0 = g_{\mu\nu}T^{\mu\nu} = T^\mu{}_\mu.$$

For a generic value of $n$, the action functional associated to the Lagrangian density (61) for a massless scalar field is also invariant under diffeomorphisms as well as under the Weyl rescalings (65) and thereby the corresponding EMT $T^{\mu\nu}_{\text{conf}}$ is symmetric, covariantly conserved and traceless according to the previous line of arguments.

**Conservation laws:** Let us again come back to the general framework (55)-(59) of a matter field $\phi$ which is minimally coupled to a dynamical gravitational field. As noted after Eqn. (58), the covariant conservation law $\nabla_\mu T^{\mu\nu} = 0$ merely represents a consistency condition for Einstein's field equations on curved space. Yet, in some cases, an ordinary conservation law $\partial_\mu(\ldots)^\mu = 0$ can be derived from $\nabla_\mu T^{\mu\nu} = 0$. In particular, this is the case if the space-time manifold $(M, \mathfrak{g})$ admits *conformal isometries* described at the infinitesimal level by *conformal Killing vector fields (CKVF's)*, i.e. solutions $\xi \equiv \xi^\mu \partial_\mu$ of the

$$\text{conformal Killing equation}: \qquad \nabla_\mu \xi_\nu + \nabla_\nu \xi_\mu = \frac{2}{n}(\nabla_\rho \xi^\rho) g_{\mu\nu}. \qquad (66)$$

This case includes the one of *isometries* described at the infinitesimal level by *Killing vector fields (KVF's)*, i.e. solutions $\xi \equiv \xi^\mu \partial_\mu$ of the

$$\text{Killing equation}: \qquad \nabla_\mu \xi_\nu + \nabla_\nu \xi_\mu = 0 \qquad (\text{i.e. } \delta_\xi g_{\mu\nu} = 0). \qquad (67)$$

More precisely, we consider the

$$\text{current density associated with a CKVF } \xi = \xi^\mu \partial_\mu: \qquad \boxed{j_\xi^\mu \equiv T^{\mu\nu} \xi_\nu}, \qquad (68)$$

where $T^{\mu\nu}$ are the components of the EMT of the matter field. Since the EMT is symmetric, covariantly conserved and traceless (for a free massless scalar field $\phi$ in two space-time dimensions), we have

$$\nabla_\mu j_\xi^\mu = \underbrace{(\nabla_\mu T^{\mu\nu})}_{\approx 0} \xi_\nu + T^{\mu\nu}(\nabla_\mu \xi_\nu) = \frac{1}{2} T^{\mu\nu} \underbrace{(\nabla_\mu \xi_\nu + \nabla_\nu \xi_\mu)}_{= \frac{2}{n}(\nabla_\rho \xi^\rho) g_{\mu\nu}} = \frac{1}{n}(\nabla_\rho \xi^\rho) \underbrace{T^\mu_{\ \mu}}_{= 0} = 0. \qquad (69)$$

By virtue of the general relation $\nabla_\mu j_\xi^\mu = \frac{1}{\sqrt{|g|}} \partial_\mu(\sqrt{|g|}\, j_\xi^\mu)$, we thus have the

$$\text{local conservation law}: \qquad \partial_\mu(\sqrt{|g|}\, j_\xi^\mu) = 0, \qquad (70)$$

for the solutions of the field equations.

For the conformally invariant coupling of a massless scalar field $\phi$, the calculation (69) holds true for any value $n \geq 2$ if one replaces $T^{\mu\nu}$ in the current density (68) by $T_{\text{conf}}^{\mu\nu}$. By virtue of Eqn. (69), we also have the result $\nabla_\mu j_\xi^\mu = 0$ if the EMT $T^{\mu\nu}$ is not traceless provided we limit ourselves to a KVF $\xi$ in the expression for $j_\xi^\mu$.

**Example of Minkowski space-time:** In general, space-time manifolds $(M, \mathfrak{g})$ do not admit any conformal isometries, i.e. there are no non-zero CKVF's. However, a maximal number $\frac{1}{2}(n+1)(n+2)$ of such symmetries exists in flat $n$-dimensional space-time $(\mathbb{R}^n, \eta)$. In this case, the covariant derivatives $\nabla_\mu$ reduce to the ordinary derivatives $\partial_\mu$ and the total Lagrangian density describing the dynamics of the matter field $\phi$ becomes $\mathcal{L} = \frac{1}{2}\left[(\partial^\mu \phi)(\partial_\mu \phi) - m^2 \phi^2\right]$. The EMT (59) then reduces to the so-called *canonical EMT* $T_{\text{can}}^{\mu\nu}$ (of a scalar field in Minkowski space-time) which reads

$$T_{\text{can}}^{\mu\nu} = (\partial^\mu \phi)(\partial^\nu \phi) - \eta^{\mu\nu} \mathcal{L}. \qquad (71)$$

If one considers the conformally invariant coupling of a massless scalar field $\phi$ to gravity (as given by the Lagrangian density (61)), then the extra curvature dependent terms appearing

in the corresponding EMT $T^{\mu\nu}_{\text{conf}}$ contribute an additional term to $T^{\mu\nu}_{\text{can}}$ in the flat space limit. More precisely, $T^{\mu\nu}_{\text{conf}}$ reduces to the so-called *new improved* or *CCJ EMT* [141],

$$T^{\mu\nu}_{\text{conf}} = T^{\mu\nu}_{\text{can}} - \xi_n (\partial^\mu \partial^\nu - \eta^{\mu\nu} \Box)\phi^2, \qquad \text{with } \xi_n = \frac{1}{4}\frac{n-2}{n-1}. \tag{72}$$

This expression differs from the canonical EMT (of a massless field) by a superpotential term, see Eqn. (240) below for a general discussion of such terms. (Here, we put forward the result (72) which is not explicitly addressed in reference [21] and we refer to [142] for further details concerning the new improved EMT.)

For Minkowski space-time $(\mathbb{R}^n, \eta)$, the general solution of the conformal Killing equation (66) reads

$$\xi_\mu = a_\mu + \varepsilon_{\mu\nu} x^\nu + \rho\, x_\mu + 2\,(c\cdot x)\, x_\mu - c_\mu x^2, \tag{73}$$

where $a_\mu, \rho, c_\mu$ and $\varepsilon_{\mu\nu} = -\varepsilon_{\nu\mu}$ are constant real parameters. More precisely, the variables $a_\mu, \varepsilon_{\mu\nu}$ parametrize infinitesimal Poincaré transformations (i.e. KVF's) and $\rho$ labels *scale transformations (dilatations)* while $(c_\mu)$ labels *special conformal transformations (SCT's)* which are also referred to as *conformal boosts*. The set of these infinitesimal transformations represents the Lie algebra of the *conformal group associated to* $(\mathbb{R}^n, \eta)$, this Lie group having the dimension $\frac{1}{2}(n+1)(n+2)$.

By virtue of equations (68) and (69), we have *conserved current densities*

$$j^\mu_\xi = T^{\mu\nu}_{\text{can}} \xi_\nu, \qquad \text{if } \xi = \text{KVF}, \tag{74}$$

$$j^\mu_\xi = T^{\mu\nu}_{\text{conf}} \xi_\nu, \qquad \text{if } \xi = \text{CKVF and } \phi = \text{massless field}.$$

These results represent the so-called *Besselhagen form* of the conserved current densities associated to the Poincaré group and to the conformal group of $(\mathbb{R}^n, \eta)$, respectively [142, 143]. As it is always the case, these current densities are determined up to improvement terms, see Eqn. (240) below.

**Background field approach and Peierls bracket of charges:** While relying on the elaborations and on the notation of reference [21], we now present general results (for the scalar matter field coupled to gravity) and relate them to the explicit expressions that we discussed above. More general physical situations can be addressed along the same lines by decomposing fields into a background field configuration and a finite disturbance according to the background field approach to field theory [21].

The local symmetry transformation (63) of the metric field under an infinitesimal diffeomorphism $x^\mu \rightsquigarrow x'^\mu(x) \simeq x^\mu - \xi^\mu(x)$ has the general form (49)-(50) with $a \doteq (\mu\nu)$ and $r \doteq \sigma$:

$$\delta g_{\mu\nu}(x) = \int_M d^n y\; {}_{\mu\nu}Q_\sigma(x,y)\, \xi^\sigma(y), \qquad \text{with } {}_{\mu\nu}Q_\sigma(x,y) = g_{\mu\tau}(y)\nabla_\nu \delta^\tau_\sigma\, \delta(x-y) + (\mu \leftrightarrow \nu). \tag{75}$$

Now suppose that the scalar matter field is coupled to a *gravitational background* $\overset{\circ}{\mathfrak{g}}$. We again denote the response (62) of the dynamical system to the variation of the metric by $T^{\mu\nu}$ (this setting including the particular instance where $T^{\mu\nu} = T^{\mu\nu}_{\text{conf}}$ is the EMT for the conformally invariant coupling of a massless scalar field to gravity). Moreover, let us suppose that the background metric $\overset{\circ}{\mathfrak{g}}$ admits some linearly independent CKVF's $K_A \equiv K^\mu_A \partial_\mu$ labeled by the index $A$. With expression (68) in mind, we presently consider the following *current densities*

$j_A \equiv (j_A^\mu)$ *associated to the matter field coupled to the background metric:*

$$\text{current densities}: \quad \boxed{j_A^\nu \equiv \frac{1}{2} T^{\rho\sigma}{}_{\rho\sigma} \bar{Q}^\nu_{(1)\tau} K_A^\tau.} \tag{76}$$

In the latter expression, $\bar{Q}_{(1)}$ represents the derivative term of the symmetry transformation (75) (see expansion (50) of $Q$), e.g. for the flat space metric $\overset{\circ}{\mathfrak{g}} = \eta$ we have $_{\rho\sigma}\bar{Q}^\nu_{(1)\tau} = \eta_{\rho\tau}\delta^\nu_\sigma + (\rho \leftrightarrow \sigma)$. By construction, the current densities $j_A \equiv (j_A^\mu)$ satisfy the

$$\text{local conservation laws}: \quad \boxed{\partial_\mu j_A^\mu = 0,} \quad \text{on-shell}. \tag{77}$$

Their integral over a complete space-like Cauchy hypersurface $\Sigma$ (or a smooth local deformation thereof) yields

$$\text{conserved charges}: \quad \boxed{Q_A \equiv \int_\Sigma d\Sigma_\mu j_A^\mu.} \tag{78}$$

The latter are independent of $\Sigma$ (by virtue of the local conservation laws (77)) and they represent physical observables.

The set of all CKVF's defines a Lie algebra, the commutator being the usual Lie bracket of vector fields. If $f_{AB}{}^C$ denotes the structure constants of this Lie algebra (with respect to a basis $\{K_A\}$ of complete CKVF's $K_A$), then one can show (by a tricky calculation which involves a delicate use of the kinematics of Green's functions [21]) that the *linear space of conserved charges* (78) *endowed with the Peierls bracket* defines a Lie algebra which is homeomorphic to the Lie algebra of CKVF's:

$$\text{algebra of charges}: \quad \boxed{\{Q_A, Q_B\}_P = f_{AB}{}^C Q_C.} \tag{79}$$

Exponentiation of this (finite-dimensional) Lie algebra of charges yields a Lie group which is referred to as *charge group*. The existence and structure of this group is determined by the special symmetries characterizing the background metric $\overset{\circ}{\mathfrak{g}}$: the group depends on the choice of background by virtue of the CKVF's. For further discussion we refer to the work [21] whose author also provides a general expression for the conserved charges $Q_A$ as an integral over an $(n-2)$-dimensional surface at spatial infinity (cf. pages 90-92 of reference [21]).

Here, we only come back to the particular instance of a *flat background metric,* i.e. to the explicit expressions for the conserved current densities that one recovers from the general formula (76) for the case of a massless scalar field $\phi$ on Minkowski space-time ($\mathbb{R}^n, \eta$). The CKVF's $K_A = K_A^\sigma \partial_\sigma$ presently have the following explicit expression (cf. (73)):

$$\begin{aligned}
\text{translations}: & \quad K_\mu \equiv K_\mu^\sigma \partial_\sigma = \delta_\mu^\sigma \partial_\sigma = \partial_\mu, \\
\text{Lorentz transformations}: & \quad K_{\mu\nu} \equiv K_{\mu\nu}^\sigma \partial_\sigma = (\eta_{\mu\tau}\delta_\nu^\sigma - \eta_{\nu\tau}\delta_\mu^\sigma)x^\tau \partial_\sigma = x_\mu \partial_\nu - x_\nu \partial_\mu, \\
\text{Dilatations}: & \quad K \equiv K^\mu \partial_\mu = x^\mu \partial_\mu, \\
\text{conformal boosts}: & \quad \tilde{K}_\mu \equiv \tilde{K}_\mu^\nu \partial_\nu = \left(x_\mu x^\nu - \frac{1}{2}\delta_\mu^\nu x^2\right)\partial_\nu.
\end{aligned} \tag{80}$$

The corresponding conserved current densities $(j_A^\mu)$ have the Besselhagen form (74),

$$j_\mu{}^\nu = t_\mu{}^\nu, \qquad j_{\mu\lambda}{}^\nu = x_\mu t_\lambda{}^\nu - x_\lambda t_\mu{}^\nu, \qquad j^\mu = x^\nu t_\nu{}^\mu, \qquad \tilde{j}_\mu^\nu = x_\mu x^\sigma t_\sigma{}^\nu - \frac{1}{2}x^2 t_\mu{}^\nu, \tag{81}$$

where $t^{\mu\nu} \equiv T^{\mu\nu}_{\text{conf}}$ denotes the new improved EMT given by Eqn. (72).

## 4.4 Mathematical underpinnings

Some twenty years ago, a new (and mathematically rigorous) algebraic approach to classical relativistic field theories [11] and to their perturbative quantization and renormalization [144] was put forward by R. Brunetti, M. Dütsch, K. Fredenhagen, S. Hollands, K. Rejzner and R. Wald with some advice by R. Stora (cf. foreword of K. Fredenhagen to the monograph [50]). In the sequel, this approach was further elaborated as well as reviewed, e.g. see [11, 50, 145] and references therein. One of the main goals of this formulation is the precise definition and construction of *local* observables on globally hyperbolic space-times. A basic ingredient is given by the Peierls bracket and the deformation quantization based on this bracket. In this context it was rigorously proven [11] that the Peierls bracket represents a Poisson bracket (see also reference [53] for related considerations). The case of space-time manifolds with a boundary will be commented upon in the next subsection.

## 4.5 Commutator function and Peierls bracket on manifolds with a boundary

On a Lorentzian manifold $(M, g)$ with a time-like boundary $\partial M$, the commutator function $\tilde{G} = G_{\text{ret}} - G_{\text{adv}}$ of a scalar field does not only satisfy the *field equation* and the characteristic *boundary conditions* (i.e. the generalization of equation (35) to $(M, g)$), but it also has to satisfy eventual *boundary conditions* that are imposed on the boundary $\partial M$ (e.g. Dirichlet, Neumann, Robin,... conditions) [137]. Even for the simple case of a free real scalar field one cannot generally expect the existence and uniqueness of retarded/advanced Green functions (of the Klein-Gordon operator $\Box + m^2$ with $\Box \equiv g^{\mu\nu}\partial_\mu\partial_\nu$) on a *generic* space-time manifold $M$ with a time-like boundary. And even with the additional assumption that this manifold is globally hyperbolic [146], general results do not appear to exist to date [147]. However, with this assumption uniquely defined Green functions have been constructed [148] for a certain number of instances of physical interest (like AdS space-times or Casimir effect configurations). In general, this construction uses appropriately chosen Fourier mode expansions.

The definition of the *Peierls bracket* in terms of the commutator function (cf. Eqn. (47)) on space-times with a boundary requires a careful choice of functional spaces, e.g. see references [13, 147, 149]. The degeneracy of the Peierls bracket for Abelian gauge fields on a manifold with boundary has been addressed in particular for the Aharonov-Bohm configuration, i.e. for the outside of an infinitely long solenoid containing a constant magnetic flux [150]. This degeneracy of the Peierls bracket leads to a quantum theory with a non-local behaviour which can be related to the Gauss law of electromagnetism. Yet, this physical system requires a subtle analysis of the notion of gauge equivalence, see reference [150].

# 5 Covariant phase space approach

For a Lagrangian system in classical non-relativistic mechanics defined on a configuration space parametrized by $(q^i)$, the phase space is parametrized by local coordinates $(q^i, p_i)$ which can be viewed as the values of position and momentum at a given time, say $t = 0$. This description referring to a fixed value of time is not manifestly covariant, in particular in the case of classical field theory where one considers the Cauchy data of fields on some space-like hypersurface in space-time (i.e. one relies on the introduction of the splitting of space and time). Yet, the initial data $(q^i, p_i)$ determine a unique trajectory of the dynamical system by virtue of the classical equations of motion and thereby the *set Z of all solutions of the equations of motion* can be viewed as a *covariant definition of phase space* [3, 4, 56–59]. The geometric set-up can be illustrated for instance by the harmonic oscillator $\ddot{x} + \omega^2 x = 0$ (with unit mass so that $p = \dot{x}$). For a given energy, the trajectory in phase space parametrized by $(x, p)$ is given by an

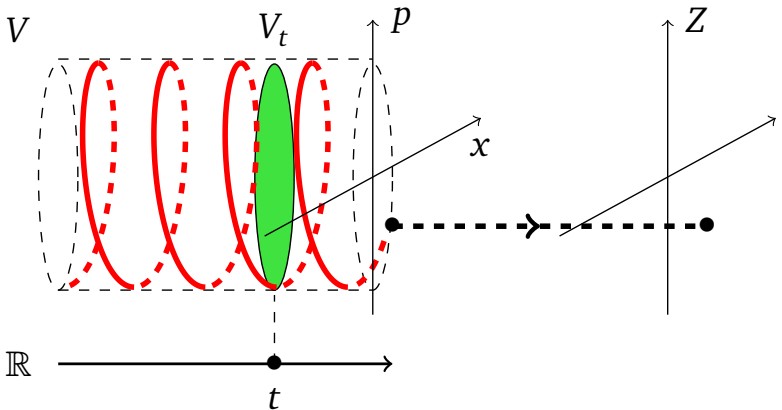

Figure 2: Covariant phase space of harmonic oscillator.

ellipse. In the course of time, the particle winds around this ellipse: in the extended velocity space $V = \mathbb{R} \times \mathbb{R}^2$ (with coordinates $(t, x, p)$), we thus have a helicoidal motion, see Figure 2.

There is a one-to-one correspondence between the space $Z$ of all solutions of the equation of motion (referred to as the *space of motions* by J.-M. Souriau [57]) and the space $V_t \equiv \{t\} \times \mathbb{R}^2$ (the "nontemporal" phase space [57]) for any fixed value $t$ of time.

Starting from this view-point for classical mechanics or field theory, we can choose a particular system of coordinates/fields and a solution of the equations of motion written in terms of these variables: the initial values of this solution are then uniquely defined and they correspond to the usual non-covariant description of phase space.[7] We note that in the case of field theory where the phase space coordinates $\left( \varphi^a(t, \vec{x}), \pi_a(t, \vec{x}) \right)$ are given by fields defined on space-time, one has to take into account (e.g. for the discussion of geometric structures like Poisson brackets or symplectic forms) the regularity properties and the *boundary conditions* characterizing the $\vec{x}$-dependence of these fields. The latter conditions are mathematical choices [151] which generally depend on the physical system under consideration, e.g. we may have a decreasing-type condition, a quasi-periodic-type condition or a finite density-type condition [151]. We will generally suppose that we have fields decreasing sufficiently fast for $|\vec{x}| \rightarrow \infty$, but we will also make some comments on the periodic case for the KdV equation in Appendix D.4.

In the following we will discuss this point of view in more detail for classical relativistic field theories while assuming that the equations of motion of the dynamical system are given by the Lagrangian equations.

---

[7]For mathematical subtleties concerning the equivalence of the different interpretations of phase space, we refer to [52]. E.g. the space-time manifold should be assumed to be a globally hyperbolic Lorentzian manifold so as to ensure the existence of Cauchy surfaces (and thereby of Cauchy data) in it.

## 5.1 Symplectic structure in covariant phase space

In the context of field theories, the covariant phase space approach goes back to the work [3,4] and was reinvented later on [58,59] with further elaborations for instance in references [38, 60–62, 152, 153]. The cohomological method for the derivation of conservation laws in diffeomorphism invariant theories which was put forward in this framework by R. Wald and his collaborators [63–65] as well as the related cohomological method of G. Barnich and F. Brandt [154] (see also [155–157]) which relies on the variational bicomplex have found many applications for field theories with local symmetries in flat or curved space-time, e.g. see [74,158] and references therein. These cohomological approaches to conservation laws will be discussed in Section 7. In the following, we will present the covariant phase space approach in elementary terms and then relate it to the other approaches (Subsection 5.2 and Subsection 5.3). As we will see, *the construction of the symplectic structure on covariant phase space* (see equations (84)-(90)) *as well as of the associated Poisson structure* (see Eqn. (172)) *essentially relies on the Lagrangian* defining the field content and dynamics of the theory. The construction does not refer to the non-covariant canonical approach and thereby leads to covariant results even in the presence of local symmetries. Besides the symplectic structure, we will discuss the conservation law of energy and momentum for Yang-Mills theory and general relativity within the present approach. As we will subsequently see in Section 6 (and in particular in Section 6.4), a mathematically rigorous formulation of the covariant phase space approach is provided by the variational bicomplex.

Our starting point is a relativistic first order Lagrangian field theory on a space-time manifold $M$. Thus, we have a Lagrangian density $\mathcal{L}(\varphi^a, \partial_\mu \varphi^a)$ depending on a collection of classical relativistic fields $\varphi^a : M \to \mathbb{R}$ with $a \in \{1, \ldots, N\}$. To start with, we consider $n$-dimensional Minkowski space-time $M = \mathbb{R}^n$ parametrized by coordinates $x \equiv (x^\mu) \equiv (t, \vec{x})$ while supposing that $\mathcal{L}$ does not explicitly depend on $x$ and that all fields and their derivatives fall off sufficiently fast at spatial infinity. As usual, the Poincaré invariant action functional is denoted by $S[\varphi] \equiv \int_M d^n x \, \mathcal{L}(\varphi^a, \partial_\mu \varphi^a)$. The case of curved space-time will be addressed in subsection 5.1.2 and the formulation in terms of differential forms on space-time will be discussed in the framework of the variational bicomplex in Section 6 as well as in Subsection 7.7.

### 5.1.1 Lagrangian field theory in Minkowski space-time

**Generalities:**  An infinitesimal variation of fields at fixed $x$ ('vertical' or 'active' transformation),

$$\varphi^a(x) \rightsquigarrow \varphi^a(x) + \delta \varphi^a(x),$$

induces the following variation of the Lagrangian:

$$\delta \mathcal{L}(\varphi^a, \partial_\mu \varphi^a) = \frac{\partial \mathcal{L}}{\partial \varphi^a} \delta \varphi^a + \frac{\partial \mathcal{L}}{\partial (\partial_\mu \varphi^a)} \delta \partial_\mu \varphi^a.$$

From $\delta \partial_\mu \varphi^a = \partial_\mu \delta \varphi^a$ and application of the Leibniz rule to the last term of the previous equation, it follows that

$$\boxed{\delta \mathcal{L} = \frac{\delta S}{\delta \varphi^a} \delta \varphi^a + \partial_\mu j^\mu} \qquad \text{(``first variational formula'')}, \qquad (82)$$

where $\frac{\delta S}{\delta \varphi^a} \equiv \frac{\partial \mathcal{L}}{\partial \varphi^a} - \partial_\mu \left( \frac{\partial \mathcal{L}}{\partial (\partial_\mu \varphi^a)} \right)$ is the *equation of motion function* and where we have the so-called

"symplectic potential" current density $\qquad j^\mu \equiv \frac{\partial \mathcal{L}}{\partial (\partial_\mu \varphi^a)} \delta \varphi^a, \qquad$ i.e. $\qquad \boxed{j^\mu = \pi_a^\mu \, \delta \varphi^a.}$

$$(83)$$

In the last equality, we substituted the definition of the canonical momentum vector field $\pi_a^\mu \equiv \partial \mathcal{L}/\partial(\partial_\mu \varphi^a)$ associated to the field $\varphi^a$, see Section 3.1. In this respect, we recall that the Lagrangian equations of motion are equivalent to the covariant Hamiltonian equations (4) which involve the momentum vector field $\pi_a^\mu$.

By definition, **covariant phase space** $Z$ is the infinite-dimensional space of solutions

$$\phi \equiv (\varphi^a, \pi_a^\mu),$$

of the covariant Hamiltonian equations (4). Accordingly, we rely on the view-point (9) which we already considered in the multisymplectic approach to classical field theory. We interpret the variation $\delta$ as a *differential* on this infinite-dimensional space (commuting with $\partial_\mu$). Thus, $\delta \varphi^a$ and $\delta \pi_a^\mu$ represent odd elements of the differential algebra $\Omega^\bullet(Z) \equiv \oplus_{p \in \mathbb{Z}} \Omega^p(Z)$ of all forms on $Z$ (where $\Omega^p(Z) = 0$ for $p < 0$), this space being endowed with the exterior product denoted by $\wedge$. The differential $\delta$ acts on 0-forms on $Z$, i.e. on functionals $F : Z \to \mathbb{R}$ according to

$$\boxed{(\delta F)(\phi) \equiv \int_M d^n x \, \delta\phi(x) \frac{\delta F}{\delta \phi(x)}.}$$

Quite generally, application of $\delta$ to a $p$-form on $Z$ yields a $(p+1)$-form. The linear operator $\delta$ satisfies the

$$\text{graded Leibniz rule:} \qquad \delta(P \wedge Q) = \delta P \wedge Q + (-1)^{\deg P} P \wedge \delta Q,$$

and it is nilpotent, i.e. $\delta^2 = 0$. We refer to Appendix D for a more detailed discussion within the standard approach to the canonical formulation which is based on the view-point (8).

If we apply the differential $\delta$ to the 1-form (83), we get the so-called

$$\text{(pre-) symplectic current density:} \qquad \boxed{J^\mu \equiv -\delta j^\mu = \delta \varphi^a \wedge \delta \pi_a^\mu.} \qquad (84)$$

By definition, $J^\mu$ is $\delta$-exact, hence $\delta$-*closed*, i.e.

$$\boxed{\delta J^\mu = 0.} \qquad (85)$$

An important property [159] of the current densities $(j^\mu)$ and $(J^\mu)$ is their independence of the choice of fields $\varphi^a$. More precisely, the current densities $(j^\mu)$ and $(J^\mu)$ are *invariant under point transformations in covariant phase space,* i.e. under invertible transformations $\varphi^a \rightsquigarrow \varphi'^a = \Phi^a(\varphi)$ which do not involve derivatives of fields:

$$\boxed{\varphi \rightsquigarrow \varphi' = \Phi(\varphi) \qquad \Longrightarrow \qquad j'_\mu = j_\mu, \quad J'_\mu = J_\mu.} \qquad (86)$$

Indeed, under such a transformation, the differential $\delta \varphi^a$ changes contravariantly with the Jacobian,

$$(\delta \varphi^a)' = \delta \varphi'^a = \mathcal{J}^a{}_b \, \delta \varphi^b, \qquad \text{with} \quad \mathcal{J}^a{}_b \equiv \frac{\partial \Phi^a}{\partial \varphi^b},$$

and so does $\partial_\mu \varphi^a$: $(\partial_\mu \varphi^a)' = \partial_\mu \Phi^a(\varphi) = \mathcal{J}^a{}_b \, \partial_\mu \varphi^b$. Hence, the field $\pi_a^\mu \equiv \frac{\partial \mathcal{L}}{\partial(\partial_\mu \varphi^a)}$ transforms covariantly (i.e. $(\pi_a^\mu)' = \frac{\partial \varphi^b}{\partial \Phi^a} \pi_b^\mu$) which implies that the current densities $j^\mu \equiv \pi_a^\mu \, \delta \varphi^a$ and $J^\mu \equiv -\delta j^\mu$ are invariant under the considered transformations.

**Going on-shell:** By applying $\delta$ to the covariant Hamiltonian equations (4), we obtain the *linearized equations of motion,*

$$\partial_\mu \varphi^a = \frac{\partial \mathcal{H}}{\partial \pi_a^\mu} \implies \partial_\mu(\delta\varphi^a) = \frac{\partial^2 \mathcal{H}}{\partial \varphi^b \partial \pi_a^\mu}\delta\varphi^b + \frac{\partial^2 \mathcal{H}}{\partial \pi_b^\nu \partial \pi_a^\mu}\delta\pi_b^\nu,$$

$$\partial_\mu \pi_a^\mu = -\frac{\partial \mathcal{H}}{\partial \varphi^a} \implies \partial_\mu(\delta\pi_a^\mu) = -\frac{\partial^2 \mathcal{H}}{\partial \varphi^b \partial \varphi^a}\delta\varphi^b - \frac{\partial^2 \mathcal{H}}{\partial \pi_b^\nu \partial \varphi^a}\delta\pi_b^\nu. \tag{87}$$

As a function of $x$, the (pre-)symplectic current density ($J^\mu$) given by (84) is *conserved* by virtue of the linearized equations of motion (87) and the fact that the monomials $\delta\varphi^a$ and $\delta\pi_a^\mu$ are anticommuting entities:

$$\partial_\mu J^\mu = \partial_\mu(\delta\varphi^a) \wedge \delta\pi_a^\mu + \delta\varphi^a \wedge \partial_\mu(\delta\pi_a^\mu) = 0, \qquad \text{on-shell.} \tag{88}$$

Thus, we have the

$$\text{structural conservation law:} \qquad \boxed{\partial_\mu J^\mu = 0, \quad \text{on-shell.}} \tag{89}$$

The qualification of this relation is chosen by analogy [160] to the corresponding relation $\partial_t(dq^a \wedge dp_a) = 0$ which holds for the solutions of the Hamiltonian equations of motion in classical mechanics and which states that the symplectic form $dq^a \wedge dp_a$ on phase space $T^*\mathcal{Q}$ is conserved in time for any Hamiltonian function $H(\vec{q},\vec{p})$.

We note [161] that the result (89) can also be derived directly from the first variational formula (82), i.e. $\delta\mathcal{L} = \partial_\mu j^\mu$ on-shell, by using the nilpotency of $\delta$:

$$\partial_\mu J^\mu = \partial_\mu(-\delta j^\mu) = -\delta(\partial_\mu j^\mu) = -\delta(\delta\mathcal{L}), \qquad \text{on-shell.}$$

Finally, one defines the

$$\text{(pre-) symplectic 2-form:} \quad \boxed{\Omega \equiv \int_\Sigma d\Sigma_\mu J^\mu,} \quad \text{with } d\Sigma_\mu \equiv \frac{1}{(n-1)!}\varepsilon_{\mu\mu_2\cdots\mu_n}dx^{\mu_2}\wedge\cdots\wedge dx^{\mu_n}, \tag{90}$$

where $\Sigma \subset M$ is a space-like hypersurface (having dimension $(n-1)$) and $\varepsilon_{\mu_1\cdots\mu_n}$ are the components of the Levi-Civita symbol in flat space normalized by $\varepsilon_{01\cdots(n-1)} = 1$ [154]. For instance, for the particular choice of hypersurface $t = $ constant (particular Lorentz frame), we obtain

$$\Omega \equiv \int_\Sigma d\Sigma_\mu J^\mu = \int_\Sigma dx^1 \cdots dx^{n-1} J^0 = \int_{\mathbb{R}^{n-1}} d^{n-1}x\, J^0. \tag{91}$$

Substitution of (84) shows that the 2-form $\Omega$ then takes the *canonical expression*

$$\Omega = \int_{\mathbb{R}^{n-1}} d^{n-1}x\, \delta\varphi^a \wedge \delta\pi_a, \tag{92}$$

where $\pi_a \equiv \pi_a^0$ is the usual conjugate momentum associated to the field $\varphi^a$. If there are no constraints (relations for $\varphi^a, \pi_a$ of the form $\Phi(\varphi,\pi) = 0$), then (92) leads to the canonical expression for the Poisson brackets, see equations (D.7)-(D.17) of Appendix D. In the presence of local symmetries, i.e. first class constraints (e.g. in electrodynamics, $(\varphi^a)$ is given by the gauge potentials $(A^\mu)$ and we have the first class constraints $\pi_0 = 0$ and $\partial^i \pi_i = 0$), one has to reduce the phase space and introduce the induced symplectic form $\Omega_{\text{phys}}$ in order to obtain Poisson brackets, see equations (105)-(106) below.

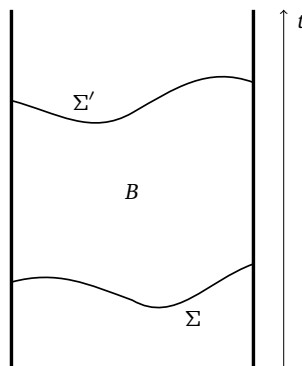

Figure 3: Two space-like hypersurfaces extending to spatial infinity.

By construction, the 2-form $\Omega$ (associated to the Lagrangian field theory under consideration) is *closed,* i.e. $\delta\Omega = 0$. Moreover [58], it follows from the on-shell relation $\partial_\mu J^\mu = 0$ and Stokes theorem that *the given expression for $\Omega$ is independent of the space-like hypersurface $\Sigma$* which is considered for its definition (assuming that fields fall off sufficiently fast at space-like infinity): For the proof (see Figure 3), one considers two space-like hypersurfaces $\Sigma, \Sigma'$ and integrates $\partial_\mu J^\mu$ over the volume $B \subset \mathbb{R}^n$ bounded by $\Sigma, \Sigma'$ and space-like infinity (where $J^\mu$ vanishes), i.e.

$$\text{on-shell}: \qquad 0 = \int_B d^n x\, \partial_\mu J^\mu = \oint_{\partial B} d\Sigma_\mu J^\mu = \int_{\Sigma'} d\Sigma_\mu J^\mu - \int_{\Sigma} d\Sigma_\mu J^\mu + 0\,. \tag{93}$$

Thus, the 2-form $\Omega$ is independent of the choice of $\Sigma$ and thus *Poincaré invariant.*

As we discuss in Appendix D (see equations (D.17) and (D.26)), the 2-form given by expression (92) also has the property of being (weakly) non-degenerate in the absence of constraints. Thus, it defines a *(weak) symplectic structure on covariant phase space* and it is *Poincaré invariant.* The case of constrained systems is discussed below for the example of Yang-Mills theory.

**Ambiguities:** The equations of motion $\delta S / \delta\varphi^a = 0$ are not modified if one adds to the Lagrangian density $\mathcal{L}(\varphi^a, \partial_\mu\varphi^a, x)$ the divergence of a vector field $\Lambda^\mu$ which only depends on $\varphi^a$ and $x$. This addition modifies the symplectic potential current $j^\mu$ by an additive term $\delta\Lambda^\mu$, but, by virtue of $\delta^2 = 0$, it does not change the symplectic current $J^\mu = -\delta j^\mu$ (and thereby not the (pre-)symplectic form $\Omega$ either):

$$\boxed{\mathcal{L}' = \mathcal{L} + \partial_\mu\Lambda^\mu\,, \qquad j'^\mu = j^\mu + \delta\Lambda^\mu\,, \qquad J'^\mu = J^\mu\,.} \tag{94}$$

We will discuss these ambiguities (and another one) further in terms of differential forms in Subsection 6.4 (see equations (194)-(196)).

We note that a given equation of motion may eventually be obtained from Lagrangians which are not related by a gauge transformation (94) (or by a rescaling): such Lagrangians then lead to different symplectic structures. A simple illustration concerning mechanics is given by the two-dimensional isotropic oscillator where the Lagrangians $L_{\text{stand}} = \frac{1}{2} m(\dot{x}^2 + \dot{y}^2) - \frac{1}{2} m\omega^2(x^2 + y^2)$ and $L = m(\dot{x}\dot{y} - \omega^2 xy)$ are not related by a total derivative $\frac{dF}{dt}(x, y, t)$ and yield distinct symplectic structures [162]. We also remark that the addition of a topological term to a Lagrangian density modifies the symplectic potential [162–164], see Subsection 5.1.2 and Subsection 7.8 below for the case of gravity.

**Example of a real scalar field:** A simple example [58] is given by a single real scalar field $\varphi^1 \equiv \varphi$ with a mass/self-interaction potential $V(\varphi)$, i.e. the Lagrangian density

$$\mathcal{L}(\varphi, \partial_\mu \varphi) = \frac{1}{2} \partial^\mu \varphi \, \partial_\mu \varphi - V(\varphi). \tag{95}$$

In this case, the equation of motion reads $0 = \Box \varphi + V'(\varphi)$ (with $\Box \equiv \partial^\mu \partial_\mu$) and the linearized equation of motion takes the form $0 = \Box \delta\varphi + V'' \delta\varphi$.

The quantization of the theory based on the symplectic 2-form $\Omega$ on $Z$ (as restricted to a given class of solutions of the field equations) has been referred to as *on-shell quantization* by the authors of reference [159]: we will come back to this issue in Subsection 5.1.3 below.

**Example of pure Yang-Mills theory 1 - Symplectic formulation:** Let $G$ be a compact matrix Lie group (e.g. $G = SU(N)$) with Lie algebra $\mathfrak{g} \equiv \text{Lie}\, G$. We consider a basis $\{T^a\}$ of $\mathfrak{g}$ given by anti-Hermitian matrices satisfying $[T^a, T^b] = f^{abc} T^c$ and $\text{Tr}(T^a T^b) = \delta^{ab}$. The *Yang-Mills potential* is a Lie algebra-valued vector field $A_\mu(x) \equiv A_\mu^a(x) T^a$ on space-time and the associated field strength is given by

$$F_{\mu\nu} \equiv \partial_\mu A_\nu - \partial_\nu A_\mu + [A_\mu, A_\nu].$$

The latter satisfies the Bianchi identity $0 = D_\lambda F_{\mu\nu} +$ cyclic permutations of the indices, where the covariant derivative $D_\mu$ of a Lie algebra-valued field $Q$ is defined by $D_\mu Q \equiv \partial_\mu Q + [A_\mu, Q]$.

The *Lagrangian density* of pure YM-theory reads

$$\mathcal{L} = -\frac{1}{4} \text{Tr}(F^{\mu\nu} F_{\mu\nu}).$$

Thus, the *equation of motion* of the gauge potential $(A^\mu)$ is given by $0 = \frac{\delta S}{\delta A_\nu} = D_\mu F^{\mu\nu}$ and the *linearized equation of motion* has the form

$$0 = D_\mu(\delta F^{\mu\nu}) + [\delta A_\mu, F^{\mu\nu}], \qquad \text{where} \quad \delta F_{\mu\nu} = D_\mu(\delta A_\nu) - D_\nu(\delta A_\mu). \tag{96}$$

The argumentation (82)-(83) presently yields $j^\mu = \text{Tr}\left(\frac{\partial \mathcal{L}}{\partial(\partial_\mu A_\nu)} \delta A_\nu\right)$. Since

$$\frac{\partial \mathcal{L}}{\partial(\partial_\mu A_\nu)} = \frac{\partial \mathcal{L}}{\partial F_{\rho\sigma}} \frac{\partial F_{\rho\sigma}}{\partial(\partial_\mu A_\nu)} = -F^{\mu\nu}, \qquad \text{i.e.} \quad j^\mu = -\text{Tr}(F^{\mu\nu} \delta A_\nu),$$

we obtain the

$$\boxed{\text{(pre-) symplectic current density:} \qquad J^\mu \equiv -\delta j^\mu = \text{Tr}(\delta A_\nu \wedge \delta F^{\nu\mu}).} \tag{97}$$

As a function of $x$, this expression is *conserved* by virtue of the linearized equation of motion (96), the cyclicity of the trace and the fact that the monomials $\delta A_\mu$ and $\delta F_{\mu\nu}$ are anti-commuting:

$$\partial_\mu J^\mu = \text{Tr}\left[D_\mu(\delta A_\nu) \wedge \delta F^{\nu\mu}\right] + \text{Tr}\left[\delta A_\nu \wedge D_\mu(\delta F^{\nu\mu})\right]$$
$$= \frac{1}{2} \text{Tr}\left[\delta F_{\mu\nu} \wedge \delta F^{\nu\mu}\right] + \text{Tr}\left[\delta A_\nu \wedge [\delta A_\mu, F^{\mu\nu}]\right] = 0 + 0, \qquad \text{on-shell.} \tag{98}$$

By way of consequence, the

$$\boxed{\text{(pre-) symplectic 2-form:} \qquad \Omega \equiv \int_\Sigma d\Sigma_\mu J^\mu = \int_\Sigma d\Sigma_\mu \text{Tr}(\delta A_\nu \wedge \delta F^{\nu\mu}),} \tag{99}$$

is *Poincaré invariant* (independent of the space-like hypersurface $\Sigma$ that is considered for the definition of $\Omega$). Moreover, it is $\delta$-closed due to the nilpotency of $\delta$. In addition, the 2-form $\Omega$ is presently *invariant under (local) gauge transformations* which are given at the infinitesimal level by $\delta_g A_\mu = D_\mu \omega$ where $\omega(x) \equiv \omega^a(x) T^a$ is a Lie algebra-valued parameter: indeed, the induced transformations of the monomials $\delta A_\mu$ and $\delta F_{\mu\nu}$ write

$$
\begin{aligned}
\delta_g(\delta A_\mu) &= \delta(\delta_g A_\mu) = [\delta A_\mu, \omega], \\
\delta_g(\delta F_{\mu\nu}) &= \delta(\delta_g F_{\mu\nu}) = [\delta F_{\mu\nu}, \omega],
\end{aligned}
\tag{100}
$$

hence $\delta_g J^\mu = 0$ by virtue of the cyclicity of the trace.

*The 2-form* (99) *is weakly degenerate* (see Eqn. (D.20)) *due to its gauge invariance* as can be seen as follows [1]. Let $X^\mu \equiv \delta A^\mu$ be a solution of the linearized equations of motion (96), i.e.

$$
0 = D_\mu(\delta F^{\mu\nu}) + [\delta A_\mu, F^{\mu\nu}] = 2 D_\mu D^{[\mu} X^{\nu]} + [X_\mu, F^{\mu\nu}],
\tag{101}
$$

where $D^{[\mu} X^{\nu]} \equiv \frac{1}{2}(D^\mu X^\nu - D^\nu X^\mu)$. A vector field in covariant phase space $Z$ has the form (see equations (D.2) and (D.25))

$$
X = \int_M d^n x \, \mathrm{Tr}\left[\delta A^\mu \frac{\delta}{\delta A^\mu} + \delta F^{\mu\nu} \frac{\delta}{\delta F^{\mu\nu}}\right] = \int_M d^n x \, \mathrm{Tr}\left[X^\mu \frac{\delta}{\delta A^\mu} + 2 D^{[\mu} X^{\nu]} \frac{\delta}{\delta F^{\mu\nu}}\right],
$$

where $X^\mu$ is a solution of the linearized YM equation (101). Application of the 2-form (99) to any two vector fields $X, Y$ yields

$$
\Omega(X, Y) = 2 \int_\Sigma d\Sigma_\mu \, \mathrm{Tr}\left[X_\nu D^{[\nu} Y^{\mu]} - Y_\nu D^{[\nu} X^{\mu]}\right].
\tag{102}
$$

Let us now assume that $X^\mu$ is pure gauge, i.e. $X^\mu = D^\mu f$ for some Lie algebra-valued function $f$. Then, we have $D^{[\nu} X^{\mu]} = \frac{1}{2}[D^\nu, D^\mu]f = \frac{1}{2}[F^{\nu\mu}, f]$ and thereby

$$
\Omega(X, Y) = 2 \int_\Sigma d\Sigma_\mu \, \mathrm{Tr}\left[(D_\nu f) D^{[\nu} Y^{\mu]}\right] - \int_\Sigma d\Sigma_\mu \, \mathrm{Tr}\left[Y_\nu [F^{\nu\mu}, f]\right].
\tag{103}
$$

Application of the Leibniz rule to the first term yields a contribution $2 \int_\Sigma d\Sigma_\mu \, \partial_\nu \mathrm{Tr}\left[f D^{[\nu} Y^{\mu]}\right]$ (which vanishes with the assumption of appropriate boundary conditions at spatial infinity) and a contribution which can be simplified by virtue of the linearized equation of motion (101):

$$
-2 \int_\Sigma d\Sigma_\mu \, \mathrm{Tr}\left[f D_\nu D^{[\nu} Y^{\mu]}\right] = \int_\Sigma d\Sigma_\mu \, \mathrm{Tr}\left[f [Y_\nu, F^{\nu\mu}]\right] = \int_\Sigma d\Sigma_\mu \, \mathrm{Tr}\left[Y_\nu [F^{\nu\mu}, f]\right].
\tag{104}
$$

*In summary,* for $X^\mu = D^\mu f$, we have $\Omega(X, Y) = 0$ for all vector fields $Y$. Conversely [165], $\Omega(X, Y) = 0$ for all $Y$ implies that $X^\mu = D^\mu f$ for some $f$. Henceforth, the 2-form $\Omega$ defined on the space $Z$ of all solutions $(A^\mu)$ of the YM equations is degenerate due to gauge symmetry. Incidentally [1], this fact is closely related to the issue that the Cauchy problem for the YM equation is not well posed due to gauge symmetry. Since the degeneracy of the 2-form $\Omega$ is due to gauge symmetry, a non-degenerate 2-form $\Omega_{\text{phys}}$ can be obtained from $\Omega$ by identifying all gauge field configurations $(A^\mu) \in Z$ which are related by a gauge transformation, i.e. one removes the gauge freedom by factoring out the gauge group from $Z$. We will now describe this procedure in more detail [152].

Due to the gauge symmetry, the *physical* or *reduced phase space* of YM-theory is not given by $Z$ (space of all solutions of the equations of motion for $(A^\mu)$), but rather by the quotient

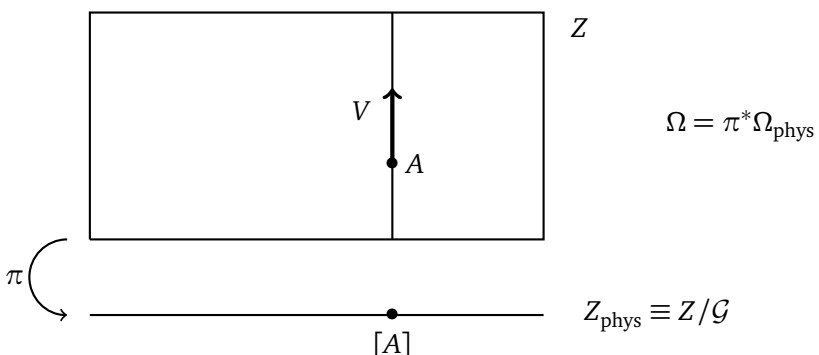

Figure 4: Physical phase space and symplectic 2-form of YM-theory.

space $Z_{\text{phys}} \equiv Z/\mathcal{G}$ where $\mathcal{G}$ is the (infinite-dimensional) group of all gauge transformations.[8] The

$$\text{physical phase space of YM-theory} \qquad \boxed{Z_{\text{phys}} \equiv Z/\mathcal{G}\,,} \qquad (105)$$

is parametrized by the equivalence classes $[A]$ of gauge potentials $A$ (the class $[A]$ being the set of all potentials which are gauge equivalent to $A$ and solve of the equation of motion), see Figure 4.

We would like to show [152] that *the (pre-)symplectic 2-form $\Omega$ on $Z$ defined by* (99) *corresponds to a*

$$\text{symplectic 2-form } \Omega_{\text{phys}} \text{ on } Z_{\text{phys}} : \qquad \boxed{\Omega = \pi^*\Omega_{\text{phys}}} \qquad (106)$$

(pullback of the 2-form $\Omega_{\text{phys}}$ by means of the projection map $\pi : Z \to Z_{\text{phys}}$). This requires to show that the 2-form $\Omega$ on $Z$ has no components in the gauge directions, i.e. that $i_V\Omega = 0$ where $i_V$ denotes the contraction of the differential form $\Omega$ with a vector field $V$ on $Z$ that is tangent to the gauge orbits (fibres) on $Z$. Thus, the vector field $V$ generates gauge transformations on $Z$. In this respect,[9] we note that an infinitesimal gauge transformation (parametrized by $\varepsilon(x) \equiv \varepsilon^a(x)T^a$) of a functional (i.e. 0-form) $\mathcal{F}$ on $Z$ is given by

$$\delta_g\mathcal{F} = \int_M d^n x\, \text{Tr}\left[(D_\mu\varepsilon)\frac{\delta\mathcal{F}}{\delta A_\mu} + [F_{\nu\mu}, \varepsilon]\frac{\delta\mathcal{F}}{\delta F_{\nu\mu}}\right] = L_V\mathcal{F}\,.$$

Thus, we have

$$D_\mu\varepsilon = L_V A_\mu = i_V(\delta A_\mu)\,, \qquad (107)$$
$$[F_{\nu\mu}, \varepsilon] = L_V F_{\nu\mu} = i_V(\delta F_{\nu\mu})\,.$$

From the expression (99) of $\Omega$ and the fact that $i_V$ acts as a graded derivation on forms, it

---

[8]The space $Z_{\text{phys}}$ is also referred to as proper, true, genuine or "gauge invariant" phase space or, in some contexts, as moduli space.

[9]The following proof essentially represents a more geometric formulation of the line of arguments (101)-(104) presented above and thus contributes to a better acquaintance with covariant phase space.

follows that

$$
\begin{aligned}
i_V \Omega &= \int_\Sigma d\Sigma_\mu \operatorname{Tr}\Big[ i_V(\delta A_\nu) \wedge \delta F^{\nu\mu} - \delta A_\nu \wedge i_V(\delta F^{\nu\mu}) \Big] \\
&= \int_\Sigma d\Sigma_\mu \operatorname{Tr}\Big[ (D_\nu \varepsilon)\,\delta F^{\nu\mu} - \delta A_\nu\,[F^{\nu\mu}, \varepsilon] \Big] \\
&= \int_\Sigma d\Sigma_\mu\, \partial_\nu \operatorname{Tr}\Big[ \varepsilon\,\delta F^{\nu\mu} \Big],
\end{aligned}
\tag{108}
$$

where we used the Leibniz rule for covariant derivatives as well as the linearized equation of motion (96) to pass to the last line. The integral (108) over a total derivative vanishes if we assume that $\delta F^{\nu\mu}$ vanishes at space-like infinity: e.g. by choosing $\Sigma$ to be the hypersurface $t = $ constant, the integral (108) reads

$$
\begin{aligned}
\int_\Sigma d\Sigma_\mu\, \partial_\nu \operatorname{Tr}\Big[ \varepsilon\,\delta F^{\nu\mu} \Big] &= \int_{\mathbb{R}^{n-1}} d^{n-1}x\, \partial_\nu \operatorname{Tr}\Big[ \varepsilon\,\delta F^{\nu 0} \Big] \\
&= \int_{\mathbb{R}^{n-1}} d^{n-1}x\, \partial_i \operatorname{Tr}\Big[ \varepsilon\,\delta F^{i0} \Big] = \oint_{\partial \mathbb{R}^{n-1}} dS^i \operatorname{Tr}\Big[ \varepsilon\,\delta F^{i0} \Big] = 0\,.
\end{aligned}
\tag{109}
$$

This completes the proof that $i_V \Omega = 0$ for vector fields $V$ in $Z$ which are tangent to the gauge orbits. The latter result can be rephrased by saying that *the 2-form $\Omega$ has a non-trivial kernel*[10] *given by the tangent vectors to the gauge orbits in $Z$:*

$$
\ker \Omega \equiv \{\text{vector fields } V \text{ on } Z \,|\, i_V \Omega = 0\}\,.
\tag{110}
$$

Since $\Omega(V, W) = i_W i_V \Omega$, we cannot conclude from $\Omega(V, W) = 0$ for all $W$ that $V = 0$; thus $\Omega$ is weakly degenerate. The procedure (105)-(106) amounts to factoring out the gauge group from $Z$ and thus removing the gauge symmetry which is at the origin of the degeneracy of $\Omega$. By virtue of (106), we have (see Appendix C.6 for a detailed discussion of the finite-dimensional case)

$$
\Omega(V, W) = \big( \pi^* \Omega_{\text{phys}} \big)(V, W) = \Omega_{\text{phys}}\big( (T\pi)(V), (T\pi)(W) \big),
$$

where $(T\pi)(V)$ denotes the image of the vector field $V$ (on $Z$) to a vector field on $Z_{\text{phys}}$. Accordingly, $\Omega_{\text{phys}}$ only depends on the gauge potential $A$ by virtue of its equivalence class $[A]$. The gauge freedom having been removed, we have a closed non-degenerate 2-form $\Omega_{\text{phys}}$, i.e. a *symplectic form* on the physical phase space $Z_{\text{phys}}$. We note that the reduction of the phase space $Z$ to $Z_{\text{phys}}$ can also be discussed or rephrased in terms of a foliation by leaves [1, 72], see Appendix C.6.

**Example of pure Yang-Mills theory 2 - Geometric symmetries and energy-momentum tensor:** For concreteness, we consider translations in space-time [166]. The latter are parametrized by a constant space-time vector $(\varepsilon^\mu)$ and act on the coordinates $(A^\mu, F^{\mu\nu})$ of covariant phase space $Z$ by virtue of a vector field $V$ associated to $(\varepsilon^\mu)$. Instead of ordinary infinitesimal translations $(A_\mu \rightsquigarrow A_\mu + \varepsilon^\beta \partial_\beta A_\mu)$, we can consider [167] gauge covariant translations[11] $(A_\mu \rightsquigarrow A_\mu + \varepsilon^\beta F_{\beta\mu})$. With (96) we then have

$$
\begin{aligned}
i_V(\delta F^{\mu\alpha}) = i_V \Big[ D^\mu(\delta A^\alpha) - D^\alpha(\delta A^\mu) \Big] &= D^\mu(i_V \delta A^\alpha) - D^\alpha(i_V \delta A^\mu) \\
&= \varepsilon_\beta \Big[ D^\mu F^{\beta\alpha} - D^\alpha F^{\beta\mu} \Big] = \varepsilon_\beta\, D^\beta F^{\mu\alpha}\,,
\end{aligned}
\tag{111}
$$

---

[10]We note that in the finite-dimensional case (classical mechanics), the kernel of $\Omega$ is given by the eigenvectors of the symplectic matrix which are associated to the eigenvalue zero, see Eqn. (C.41).

[11]Ordinary and gauge covariant translations differ by a local gauge transformation which does not contribute to the conservation law.

where we used the Bianchi identity to pass to the last line.

Let us now substitute

$$i_V(\delta A_\nu) = \varepsilon^\alpha F_{\alpha\nu}, \qquad \text{and} \qquad i_V(\delta F^{\nu\mu}) = \varepsilon^\alpha D_\alpha F^{\nu\mu},$$

into $i_V\Omega$ with $\Omega$ given by (99):

$$
\begin{aligned}
i_V\Omega &= \int_\Sigma d\Sigma_\mu \, \text{Tr}\big[(i_V\delta A_\nu)\,\delta F^{\nu\mu} - \delta A_\nu(i_V\delta F^{\nu\mu})\big] \\
&= \int_\Sigma d\Sigma_\mu \, \varepsilon^\alpha \, \text{Tr}\big[F_{\alpha\nu}\,\delta F^{\nu\mu}\big] - \int_\Sigma d\Sigma_\mu \, \varepsilon^\alpha \, \text{Tr}\big[\delta A_\nu(D_\alpha F^{\nu\mu})\big] \\
&\equiv A + B.
\end{aligned}
\tag{112}
$$

For the $B$-term, we can apply the Leibniz rule for covariant derivatives and then use the expression for $\delta F_{\alpha\nu}$ given in (96):

$$
B = -\int_\Sigma d\Sigma_\mu \, \varepsilon^\alpha \, \partial_\alpha \text{Tr}\big[(\delta A_\nu) F^{\nu\mu}\big] + \int_\Sigma d\Sigma_\mu \, \varepsilon^\alpha \, \text{Tr}\big[(D_\alpha \delta A_\nu) F^{\nu\mu}\big]
\tag{113}
$$

$$
= -\int_\Sigma d\Sigma_\mu \, \varepsilon^\alpha \partial_\alpha \text{Tr}\big[(\delta A_\nu) F^{\nu\mu}\big] + \int_\Sigma d\Sigma_\mu \, \varepsilon^\alpha \, \text{Tr}\big[(\delta F_{\alpha\nu}) F^{\nu\mu}\big] + \int_\Sigma d\Sigma_\mu \, \varepsilon^\alpha \, \text{Tr}\big[(D_\nu \delta A_\alpha) F^{\nu\mu}\big].
$$

The second term of this expression combines with the $A$-term in (112) to yield the total differential $\delta \int_\Sigma d\Sigma_\mu \, \varepsilon^\alpha \, \text{Tr}\big[F_{\alpha\nu} F^{\nu\mu}\big]$.

By virtue of the YM equation $D_\nu F^{\nu\mu} = 0$, the last term in (113) reads $\int_\Sigma d\Sigma_\mu \varepsilon^\alpha \partial_\nu \text{Tr}\big[(\delta A_\alpha) F^{\nu\mu}\big]$. This integral of a total derivative vanishes if one assumes again that the variation $\delta A_\alpha$ decreases fast enough at space-like infinity. The first term in (113) can be rewritten as a total differential:

$$
-\int_\Sigma d\Sigma_\mu \, \varepsilon^\alpha \, \partial_\alpha \text{Tr}\big[(\delta A_\nu) F^{\nu\mu}\big] = \delta \int_\Sigma d\Sigma_\mu \, \varepsilon^\alpha \, \frac{1}{4} \, \eta^\mu_\alpha \, \text{Tr}\big[F^{\rho\sigma} F_{\rho\sigma}\big].
$$

Hence, we end up with the result [166]

$$
\boxed{i_V\Omega = \delta \int_\Sigma d\Sigma_\mu \, \varepsilon_\alpha \, T^{\mu\alpha},} \quad \text{with} \quad \boxed{T^{\mu\alpha} \equiv \text{Tr}\big[F^{\mu\nu} F_\nu{}^\alpha + \frac{1}{4} \eta^{\mu\alpha} F^{\rho\sigma} F_{\rho\sigma}\big].}
\tag{114}
$$

Here, we recognize the *physical (gauge invariant) EMT of the YM field,* e.g. see reference [168]. We note that the result (114) is analogous to the results which hold for translational invariant dynamical systems in classical mechanics, see equations (C.24)-(C.25). In this respect we remark that for the particular choice of a hypersurface $t = $ constant, the integral in (114) writes

$$
H \equiv \int_\Sigma d\Sigma_\mu \, \varepsilon_\alpha \, T^{\mu\alpha} = \int_{\mathbb{R}^{n-1}} d^{n-1}x \, \varepsilon_\alpha \, T^{0\alpha}.
\tag{115}
$$

More specifically, for an infinitesimal time translation (i.e. $\varepsilon_\alpha = \epsilon \, \delta^0_\alpha$ with $\epsilon$ constant), this functional has the form

$$
H = \epsilon \int_{\mathbb{R}^{n-1}} d^{n-1}x \, \mathcal{H}, \qquad \text{with} \quad \mathcal{H} \equiv T^{00},
\tag{116}
$$

i.e. $H$ represents the *Hamiltonian function* and its density $\mathcal{H}$ the energy density of the YM field. Thus, the result (114) amounts to the relation $i_X\omega = dH$ in classical mechanics (see Eqn. (C.16)) which relates a Hamiltonian $H$ and the corresponding vector field $X \equiv X_H$ which generates time translations of the dynamical system. Accordingly, equation (114) may be viewed as the field theoretical generalization of relation $i_X\omega = dH$ to covariant phase space $Z$ endowed with the symplectic 2-form $\Omega$. We will come back to this result in Subsection 7.7 below.

### 5.1.2  General relativity

Before considering field theories coupled to the gravitational field, we discuss *pure Einstein gravity with a vanishing cosmological constant in n dimensions* while using the metric formulation. Other formulations of gravity will also be mentioned along with the relevant references. For manifolds with a boundary, we refer to Subsection 6.7 and Subsection 7.8 below.

We note that the elementary considerations of the present subsection will be reformulated and generalized in Subsection 7.7 for generic diffeomorphism invariant Lagrangian field theories on $n$-dimensional space-time manifolds: following the work of R. Wald and his collaborators [60–65], we will then consider differential forms and address the general construction of conserved quantities for given asymptotic conditions "at infinity". The expressions and results established in the present subsection will then serve as a prototype example.

**Generalities on gravity:**  Let $M$ be a *n-dimensional Lorentzian manifold* endowed with the *metric tensor field* $\left(g_{\mu\nu}\right)$. We denote the covariant derivative of a tensor field with respect to the Levi-Civita-connection by $\nabla_\mu$, e.g. $\nabla_\mu V^\rho = \partial_\mu V^\rho + \Gamma^\rho_{\mu\nu} V^\nu$ where the coefficients $\Gamma^\rho_{\mu\nu}$ are the Christoffel symbols: we have

$$\Gamma^\rho_{\mu\nu} \equiv \frac{1}{2} g^{\rho\sigma}\left(\partial_\mu g_{\nu\sigma} + \partial_\nu g_{\mu\sigma} - \partial_\sigma g_{\mu\nu}\right) = \Gamma^\rho_{\nu\mu}\,, \qquad \text{hence} \quad \Gamma^\lambda_{\mu\lambda} = \partial_\mu \ln\sqrt{|g|}\,, \qquad (117)$$

where $g \equiv \det\left(g_{\mu\nu}\right)$. The commutator of covariant derivatives defines the *Riemann curvature tensor,* i.e.

$$[\nabla_\mu, \nabla_\nu] V^\rho = R^\rho{}_{\sigma\mu\nu} V^\sigma\,,$$

and by a contraction of indices the latter gives rise to the *Ricci tensor* $R_{\mu\nu} \equiv R^\rho{}_{\mu\nu\rho}$, which yields the *curvature scalar* $R \equiv R^\mu{}_\mu$.

The *Einstein field equations for the gravitational field in vacuum* read

$$0 = G_{\mu\nu} \equiv R_{\mu\nu} - \frac{1}{2} g_{\mu\nu} R\,, \tag{118}$$

the *Einstein tensor* $G_{\mu\nu}$ being covariantly conserved, i.e. $\nabla^\mu G_{\mu\nu} = 0$, as a consequence of its definition. By contracting the indices of the field equation, one finds that (for $n \neq 2$) the vacuum field equations are equivalent to $R_{\mu\nu} = 0$.

An infinitesimal variation of the metric, $g_{\mu\nu} \rightsquigarrow g_{\mu\nu} + \delta g_{\mu\nu}$, induces the following variations of fields:

$$\delta g^{\mu\nu} = -g^{\mu\rho}\delta g_{\rho\sigma} g^{\sigma\nu}\,, \tag{119a}$$

$$\delta\sqrt{|g|} = \frac{1}{2}\sqrt{|g|}\,\delta\ln\sqrt{|g|} = \frac{1}{2}\sqrt{|g|}\,g^{\mu\nu}\delta g_{\mu\nu}\,, \tag{119b}$$

$$\delta\Gamma^\alpha_{\mu\nu} = \frac{1}{2} g^{\alpha\beta}\left[\nabla_\mu(\delta g_{\beta\nu}) + \nabla_\nu(\delta g_{\mu\beta}) - \nabla_\beta(\delta g_{\mu\nu})\right]\,, \tag{119c}$$

$$\delta R_{\mu\nu} = \nabla_\alpha(\delta\Gamma^\alpha_{\mu\nu}) - \nabla_\nu(\delta\Gamma^\alpha_{\alpha\mu}) \qquad \text{(Palatini identity)}\,. \tag{119d}$$

Here, the first relation follows from $g^{\mu\nu}g_{\nu\lambda} = \delta^\mu_\lambda$ and the second from the variation of $g \equiv \det e^A = e^{\mathrm{tr}A}$. To calculate $\delta\Gamma^\alpha_{\mu\nu}$ one uses the metricity condition $0 = \nabla_\lambda g_{\mu\nu} = \partial_\lambda g_{\mu\nu} - \Gamma^\rho_{\lambda\mu} g_{\rho\nu} - \Gamma^\rho_{\lambda\nu} g_{\mu\rho}$. The derivation of the Palatini identity is worked out for instance in reference [169]. This identity yields the *linearized equations of motion*, the latter being tantamount to $\delta R_{\mu\nu} = 0$.

**Covariant phase space approach:** Einstein's field equations (118) follow from the Einstein-Hilbert action, i.e. $S \propto \int_M d^n x \, \mathcal{L}$ with $\mathcal{L} \equiv \sqrt{|g|} R$, by varying the variables $g_{\mu\nu}$ while supposing that the variations $\delta g_{\mu\nu}(x)$ and their first derivatives vanish at infinity. (If one only admits that the variations $\delta g_{\mu\nu}(x)$ vanish (on the boundary $\partial \mathcal{V}$ of a finite space-time region $\mathcal{V}$), then one has to include a boundary term into the action, see next paragraph.) Quite generally, a variation of the Einstein-Hilbert Lagrangian density $\mathcal{L} = \sqrt{|g|} R$ (with $R = g^{\mu\nu} R_{\mu\nu}$) yields

$$\delta \mathcal{L} = (\delta \sqrt{|g|}) R + \sqrt{|g|} (\delta g^{\mu\nu}) R_{\mu\nu} + \sqrt{|g|} \, g^{\mu\nu} \delta R_{\mu\nu}. \tag{120}$$

For the first and second terms, we can substitute equations (119a)-(119b). The sum of these two terms leads to the equation of motion function (involving the action functional $S \equiv \int_M d^n x \mathcal{L}$)

$$\frac{\delta S}{\delta g_{\mu\nu}} \delta g_{\mu\nu}, \qquad \text{with} \quad \frac{\delta S}{\delta g_{\mu\nu}} = \sqrt{|g|} \left[ \frac{1}{2} g^{\mu\nu} R - R^{\mu\nu} \right].$$

The last term of (120) can be determined [169] by contracting the Palatini identity (119d) with $g^{\mu\nu}$: this leads to the relation

$$\sqrt{|g|} \, g^{\mu\nu} \delta R_{\mu\nu} = \partial_\alpha \hat{j}^\alpha, \tag{121}$$

which involves the

$$\text{symplectic potential current density}: \qquad \hat{j}^\alpha \equiv \sqrt{|g|} \, j^\alpha, \quad \boxed{j^\alpha \equiv g^{\mu\nu} \delta \Gamma^\alpha_{\nu\mu} - g^{\mu\alpha} \delta \Gamma^\nu_{\nu\mu}.} \tag{122}$$

In summary, we have $\delta \mathcal{L} = \frac{\delta S}{\delta g_{\mu\nu}} \delta g_{\mu\nu} + \partial_\alpha \hat{j}^\alpha$ with $\hat{j}^\alpha$ given by the last equation.

We now regard $\delta$ as a differential in field space and thereby $\delta g_{\mu\nu}$ as Grassmann odd variables. By applying the differential $\delta$ to the 1-form (in field space) $\hat{j}^\alpha$ and using again the formula for $\delta \sqrt{|g|}$, we obtain the

$$\text{symplectic current density}: \qquad \boxed{\hat{J}^\alpha \equiv -\delta \hat{j}^\alpha = \sqrt{|g|} J^\alpha,} \tag{123}$$

with

$$\boxed{J^\alpha \equiv \delta \Gamma^\alpha_{\nu\mu} \wedge \left[ \delta g^{\mu\nu} + \frac{1}{2} g^{\mu\nu} \delta \ln|g| \right] - \delta \Gamma^\nu_{\nu\mu} \wedge \left[ \delta g^{\mu\alpha} + \frac{1}{2} g^{\mu\alpha} \delta \ln|g| \right],} \tag{124}$$

i.e.

$$\hat{j}^\alpha \equiv \delta \Gamma^\alpha_{\nu\mu} \wedge \delta \left( \sqrt{|g|} \, g^{\mu\nu} \right) - \delta \Gamma^\nu_{\nu\mu} \wedge \delta \left( \sqrt{|g|} \, g^{\mu\alpha} \right).$$

Expression (124) is the one defined by Crnkovic and Witten [152] as a starting point for their investigation of general relativity (see also [161, 170]). The *symplectic current density* $J^\alpha$ leads to the

$$\text{(pre-) symplectic 2-form} \qquad \boxed{\Omega \equiv \int_\Sigma d\Sigma_\alpha \sqrt{|g|} J^\alpha = \int_\Sigma d\Sigma_\alpha \hat{J}^\alpha,} \tag{125}$$

where $\Sigma \subset M$ denotes a space-like hypersurface.

We have

$$\boxed{\partial_\alpha \hat{j}^\alpha = \sqrt{|g|} \, \nabla_\alpha J^\alpha = 0,} \qquad \text{on-shell,}$$

since $\partial_\alpha \hat{J}^\alpha = \partial_\alpha(-\delta\hat{j}^\alpha) = -\delta(\partial_\alpha\hat{j}^\alpha) = -\delta(\delta\mathcal{L})$ on-shell. This can also be checked by a direct calculation while using the metricity condition $\nabla_\lambda g_{\mu\nu} = 0$, the linearized equations of motion $\delta R_{\mu\nu} = 0$ and the fact that monomials like $\delta g_{\mu\nu}$, $\delta\Gamma^\lambda_{\mu\nu}$ represent anticommuting variables. More explicitly [152], the terms with $\delta \ln|g|$ in $\nabla_\alpha J^\alpha$ yield $g^{\mu\nu}\delta\Gamma^\alpha_{\mu\nu} \wedge \delta\Gamma^\lambda_{\alpha\lambda}$ and the other terms combine to the opposite of this expression. Thus, we can again apply the argument (93) to conclude that *the expression $\Omega$ given by* (125) *is independent of the choice of hypersurface* $\Sigma$ ("Poincaré-invariance").

*The 2-form* $\Omega = \int_\Sigma d\Sigma_\alpha \hat{J}^\alpha$ *is $\delta$-closed* (i.e. $\delta\Omega = 0$) by virtue of $\hat{J}^\alpha \equiv -\delta\hat{j}^\alpha$ and the nilpotency of $\delta$. (Equivalently, one uses the variation of $\sqrt{|g|}$ and evaluates the one of $J^\alpha$ while using the fact that $\delta$ is a nilpotent, graded derivation: this implies that $\delta J^\alpha = -\frac{1}{2}J^\alpha \wedge \delta \ln|g|$ and thus readily leads to $\delta\Omega = 0$ [152].)

The quantity (125) is invariant under diffeomorphisms on the manifold $M$ due to the fact that all involved quantities transform tensorially, see equations (119a)-(119c). Thus, the *physical phase space of general relativity* is given by $Z_{\text{phys}} \equiv Z/\mathcal{G}$ where $Z$ is the space of all solutions of Einstein's equations and $\mathcal{G}$ represents the (infinite-dimensional) group of diffeomorphisms on $M$. At the infinitesimal level, the latter transformations act on the basic variables according to

$$\delta_\xi x^\mu = -\xi^\mu, \qquad \delta_\xi g_{\mu\nu} = \nabla_\mu \xi_\nu + \nabla_\nu \xi_\mu.$$

By assuming that $(\xi^\mu)$ is asymptotic at infinity to a Killing vector field (so that $\delta_\xi g_{\mu\nu} = 0$ at infinity), the argumentation (106)-(109) for YM-theory can be generalized [152] and allows us to conclude that the presymplectic 2-form $\Omega$ on $Z$ defined by (125) corresponds to a

$$\text{symplectic 2-form } \Omega_{\text{phys}} \text{ on } Z_{\text{phys}}: \qquad \Omega = \pi^*\Omega_{\text{phys}}, \tag{126}$$

where $\pi : Z \to Z_{\text{phys}} \equiv Z/\mathcal{G}$ denotes the projection map.

In summary, the symplectic geometry of general relativity can be formulated along the lines of YM-theory, though the resulting expressions are somewhat more involved in this case. We note that an equivalent expression for the symplectic 2-form $\Omega$ of general relativity has been obtained earlier by J. L. Friedman [171] by a different approach. For a nice and concise treatment of Einstein-Maxwell theory, i.e. electromagnetic fields coupled to gravity, we refer to the work [62].

**Gravitational action, boundary conditions and boundary actions:** Consider pure Einstein gravity with a vanishing cosmological constant in four space-time dimensions as described by the

$$\text{Einstein-Hilbert action:} \qquad \boxed{S_{EH}[g] \equiv \int_\mathcal{V} d^4x \sqrt{|g|}R.} \tag{127}$$

Here, $\mathcal{V}$ represents an arbitrary space-time region which is bounded by a closed hypersurface $\partial\mathcal{V}$. To determine the field equations from the variational principle one assumes that the variations of fields vanish on the boundary $\partial\mathcal{V}$. For the action functional $S_{EH}[g]$ this means that we have the so-called

$$\text{Dirichlet boundary conditions:} \qquad (\delta g_{\mu\nu})\big|_{\partial\mathcal{V}} = 0 \tag{128}$$

(which imposes that the tangential derivatives of $\delta g_{\mu\nu}$ also vanish on $\partial\mathcal{V}$ while the normal derivatives do not need to do so). The fact that the Lagrangian density presently depends on the second order derivatives of the metric tensor field $g_{\mu\nu}$ implies that the variational principle for the Einstein-Hilbert action is not well defined for the boundary conditions (128). Indeed,

the lines of arguments (120)-(122) yields (upon integration over the space-time region $\mathcal{V}$) the Einstein field equation functional $\int_{\mathcal{V}} d^4x \, \delta S_{EH}/\delta g_{\mu\nu} \delta g_{\mu\nu} = -\int_{\mathcal{V}} d^4x \, \sqrt{|g|} \, G^{\mu\nu} \delta g_{\mu\nu}$ plus a non-vanishing boundary contribution which is given by the integral of expression (121). The latter can be eliminated (in the present instance where the field equations are of second order rather than of fourth order as we would naively expect [16]) by adding an appropriate boundary action to $S_{EH}[g]$, following notably the work of G. Gibbons, S. Hawking and J. York, see [15, 17]. (A simple analogy of this mechanism is given by the second order Lagrangian density $-\varphi \Box \varphi$ for a scalar field $\varphi$ and the relation

$$-\varphi \Box \varphi + \partial_\mu(\varphi \, \partial^\mu \varphi) = (\partial_\mu \varphi)(\partial^\mu \varphi),$$

where the second order term $\partial\partial\varphi$ is transformed into a first order term $(\partial\varphi)^2$ by the addition of a total derivative [142]: in the present context, the term $\partial\partial g$ in the Einstein-Hilbert Lagrangian becomes a term $(\partial g)^2$ [16].) To spell out the required boundary term [17], we parametrize the boundary $\partial\mathcal{V}$ by local coordinates $(y^a)_{a=1,2,3}$ and we denote the *induced 3-metric* on $\partial\mathcal{V}$ by $(h_{ab})$ and its determinant by $h$. Let $n = (n^\mu)$ be a *unit normal vector field* on $\partial\mathcal{V}$ and $\epsilon \equiv n^\mu n_\mu$. The scalar function $\epsilon$ takes the values $\pm1$ depending on whether one considers a space-like or a time-like part of $\partial\mathcal{V}$ (see reference [172] for the case of a null, i.e. light-like parts of $\partial\mathcal{V}$). Furthermore, let $(K_{ab})$ denote the *extrinsic curvature* (or *first fundamental form*) of the hypersurface $\partial\mathcal{V}$ and let $K$ denote its trace, i.e. $K \equiv h^{ab}K_{ab} = \nabla_\mu n^\mu$. Then, we have the

Gibbons-Hawking-York boundary action:
$$\boxed{S_{GHY}[g] \equiv 2 \oint_{\partial\mathcal{V}} d^3y \, \sqrt{|h|} \, \epsilon K \, .} \tag{129}$$

The variation of this action functional (subject to the boundary conditions (128), i.e. a fixed induced metric $(h_{ab})$ on $\partial\mathcal{V}$) yields a contribution which cancels exactly the problematic boundary term (121) which follows from the variation of the Einstein-Hilbert action. We note that quite generally the addition of a boundary action induces an extra term in the symplectic potential current density, see Subsection 6.7 and, in particular, Subsection 7.7 below.

To conclude our discussion of the *general form of the gravitational action,* we note that other boundary conditions for the variations (and thereby other boundary actions) can be and have been studied in the literature, e.g. see [16, 18, 173] and references therein. To describe the latter briefly, we remark that the so-called *gravitational momentum,* i.e. the variable which is canonically conjugate to the field $h_{ab}$ (viz. $\Pi^{ab} \equiv \partial/\partial\dot{h}_{ab}(\sqrt{|g|}\mathcal{L})$) is closely related to the extrinsic curvature $(K_{ab})$:

gravitational momentum:
$$\Pi^{ab} = \sqrt{|h|}(K^{ab} - Kh^{ab}) \, . \tag{130}$$

The Einstein-Hilbert action (without an extra boundary term) provides a well-defined variational principle in four-dimensional space-time if one fixes the gravitational momentum on the boundary $\partial\mathcal{V}$, i.e. for the so-called

Neumann boundary conditions:
$$\left.(\delta\Pi^{ab})\right|_{\partial\mathcal{V}} = 0 \, . \tag{131}$$

Another choice is given by *York's mixed boundary conditions* [173] which consist in fixing the conformal metric $\hat{h}_{ab} \equiv |h|^{-1/3} h_{ab}$ and $K$: the latter again require the introduction of the boundary term (129) which has to be multiplied by a factor $1/3$ in the present case, see reference [174] for a unified treatment of all of these boundary conditions and actions. The symplectic potential current associated to the gravitational action $S \equiv S_{EH} + S_{GHY}$ will be addressed in Eqn. (138) as well as in Subsection 7.7 along with the general treatment of boundary actions.

**Canonical (ADM) approach:** The canonical formulation of general relativity in four space-time dimensions has been elaborated by ADM (R. Arnowitt, S. Deser and C. W. Misner) around 1960 [175], e.g. see references [12, 17, 176–178] for textbook treatments and [179] for further elaborations. In this framework, one considers a $(3+1)$-splitting of space-time (foliation by nonintersecting space-like hypersurfaces $\Sigma_t$) and a space-time region $\mathcal{V}$ with boundary $\partial \mathcal{V} = \Sigma_{t_1} \cup \Sigma_{t_2} \cup \mathcal{B}$ (see Figure 3). For each (three-dimensional) Cauchy hypersurface $\Sigma_t$, one can then consider the notation introduced in the previous paragraph.

More precisely, let $(y^a)_{a=1,2,3}$ be local coordinates parametrizing $\Sigma_t$ and let $(h_{ab})$ denote the induced 3-metric on $\Sigma_t$ with $h \equiv \det(h_{ab})$. The components of a unit normal and future-directed vector field $n = (n^\mu)$ to the hypersurface $\Sigma_t$ are given by $n_\mu \equiv N \partial_\mu t$ (with $t$ the time coordinate) where the so-called *lapse* function $N$ ensures the proper normalization of $n$. A congruence of curves intersecting the hypersurfaces $\Sigma_t$ has a tangent vector field $(t^\mu)$ with components $t^\mu \equiv (\frac{\partial x^\mu}{\partial t})_{y^a}$. At each point of $\Sigma_t$, this vector field can be decomposed with respect to a basis given by the unit normal vector field $(n^\mu)$ and the tangent vectors $e_a \equiv (e_a^\mu)$ (with $a \in \{1,2,3\}$) on $\Sigma_t$:

$$t^\mu = N n^\mu + N^a e_a^\mu, \qquad \text{with} \quad e_a^\mu \equiv \left( \frac{\partial x^\mu}{\partial y^a} \right)_t.$$

The 3-vector $(N^a)$ appearing in this expansion is referred to as the *shift* vector. By combining the previous expressions, we have

$$dx^\mu = t^\mu dt + e_a^\mu dy^a = (N\, dt)\, n^\mu + (dy^a + N^a dt)\, e_a^\mu.$$

Thereby, the *line element* in $\mathcal{V}$ corresponding to the metric tensor field $(g_{\mu\nu})$ takes the form

$$ds^2 = N^2 dt^2 + h_{ab}(dy^a + N^a dt)(dy^b + N^b dt), \qquad \text{with} \quad h_{ab} \equiv g_{\mu\nu} e_a^\mu e_b^\nu. \tag{132}$$

The induced decomposition of the integration measure reads $\sqrt{|g|}\, d^4 x = N \sqrt{|h|}\, dt\, d^3 y$. Similarly, the four-dimensional Ricci scalar $R$ admits the following expansion on the space-like hypersurface $\Sigma_t$:

$$R = {}^3R + K^{ab} K_{ab} - K^2 - 2 \nabla_\mu \left( n^\nu \nabla_\nu n^\mu - n^\mu \nabla_\nu n^\nu \right).$$

Here, ${}^3R$ represents the Ricci scalar associated to the 3-metric $(h_{ab})$, the symmetric tensor $(K_{ab})$ denotes the extrinsic curvature of the hypersurface $\Sigma_t$ and $K \equiv h^{ab} K_{ab}$ the trace of the latter. Thus, the Einstein-Hilbert action $\int_\mathcal{V} d^4 x \sqrt{|g|}\, R$ for a space-time region $\mathcal{V}$ with boundary $\partial \mathcal{V} = \Sigma_{t_1} \cup \Sigma_{t_2} \cup \mathcal{B}$ writes

$$S_{EH}[g] \equiv \int_\mathcal{V} d^4 x \sqrt{|g|}\, R = S_{GR}[h,n] + S_{EH/GR}[h,n], \tag{133}$$

with

$$S_{GR}[h,n] = \int_{t_1}^{t_2} dt \int_{\Sigma_t} d^3 y\, N \sqrt{|h|} \left( {}^3R + K^{ab} K_{ab} - K^2 \right), \tag{134}$$

$$S_{EH/GR}[h,n] = -2 \oint_{\partial \mathcal{V}} d\Sigma_\mu \left( n^\nu \nabla_\nu n^\mu - n^\mu \nabla_\nu n^\nu \right),$$

where the last expression follows from the Gauss-Ostrogradski theorem (A.8). The Lagrangian $\mathcal{L}_{GR}$ defining $S_{GR}$ is referred to as *ADM Lagrangian* (or as *bulk Lagrangian* describing the canonical degrees of freedom which are common to any formulation of gravity) and the Lagrangian $\mathcal{L}_{EH/GR}$ defining the action $S_{EH/GR}$ as *boundary Lagrangian* (or "relative Lagrangian") [180].

For the moment being, we focus on $S_{GR}$ (which yields the Einstein field equations in the bulk) and we will come back to the boundary Lagrangian $\mathcal{L}_{EH/GR}$ in Subsection 7.8.

The phase space is parametrized by the fields $h_{ab}$ and by the associated momenta $\Pi^{ab}$ which are given in terms of the extrinsic curvature ($K^{ab}$) of the hypersurface $\Sigma$, see Eqn. (130). On $\Sigma$, the *symplectic potential current* ($j_{GR}^\mu$) associated to the Lagrangian density $\mathcal{L}_{GR}$ has a normal component $j_{GR}^\mu n_\mu$ which is given [60] by

$$j_{GR}^\mu n_\mu = \Pi^{ab} \delta h_{ab}. \tag{135}$$

Henceforth, the *symplectic 2-form expressed in terms of the canonical variables* ($h_{ab}, \Pi^{ab}$) has the standard form (92), i.e. [93]

$$\boxed{\Omega_{GR} = \int_\Sigma d^3 y \, \delta h_{ab} \wedge \delta \Pi^{ab}.} \tag{136}$$

From the invariance of the (pre-) symplectic current under point transformations (see Eqn. (86)) it follows that *the canonical (ADM) symplectic form* (136) *and the covariant expression* (125) *(with* ($J^\alpha$) *given by* (124)) *are equivalent* [60, 159]. Indeed, the integrand of the ADM expression (136) (which is integrated over a constant time slice $\Sigma$) is simply the time component of the current $(\hat{J}^\mu) \equiv (\sqrt{|g|} J^\mu)$ with the canonical variables $N, N_a, h_{ab}$ chosen as the set of fields which parametrize the space-time metric ($g_{\mu\nu}$).

Due to the local symmetries (general coordinate invariance), pure gravity represents a constrained dynamical system involving non-linear constraints for the canonical variables (very much like Yang-Mills theories). While the proper treatment of these constraints represents a major issue for the quantization of the full theory, it does not raise problems for the quantization of particular families of solutions of the field equations since the latter automatically satisfy the constraints [159]. Yet, the restriction of the symplectic form (136) to the subset $\mathcal{Z}$ of phase space consisting of these solutions (the so-called *moduli space* of solutions) requires to cast the solutions (metrics) into the particular form (132) and to evaluate the associated momenta $\Pi^{ab}$. In general, this represents a cumbersome task which is avoided by considering the approach of covariant phase space [59, 152] and subsequently restricting the symplectic form to the moduli space $\mathcal{Z}$ of solutions that one wishes to quantize, see reference [159] as well as Subsection 5.1.3 below.

To conclude, we note that for a space-time region $\mathcal{V}$ with boundary $\partial \mathcal{V} = \Sigma_{t_1} \cup \Sigma_{t_2} \cup \mathcal{B}$, the different actions (127),(129) and (134) are related by

$$\boxed{S_{EH} + S_{GHY} = S_{GR} + S_c,} \qquad \text{with} \quad S_c = 2 \int_{S_t} d^2\theta \, \sqrt{|\sigma|} \, N k. \tag{137}$$

Here, the corner $S_t \equiv \partial \Sigma_t$ represents a closed 2-surface and the collection of these surfaces provides a foliation of $\mathcal{B}$ (see Figure 8 of Section 7.8). The 2-surface $S_t$ is parametrized by local coordinates $\theta \equiv (\theta^A)_{A=1,2}$ and the induced metric ($\sigma_{AB}$) on $S_t$ has components $\sigma_{AB} \equiv h_{ab} e_A^a e_b^B$ with $e_A^a \equiv \frac{\partial y^a}{\partial \theta^A}$. Finally, $k \equiv \sigma^{AB} k_{AB}$ denotes the trace of the extrinsic curvature ($k_{AB}$) of $S_t$ embedded into $\Sigma_t$.

**Pullback of the symplectic current form to a hypersurface:** On a four-dimensional space-time manifold, the 1-form $j_\mu dx^\mu$ (corresponding to the symplectic potential current density ($j^\mu$)) is the dual of a 3-form $j \equiv \frac{1}{3!} j_{\nu\rho\sigma} dx^\nu \wedge dx^\rho \wedge dx^\sigma$ which is referred to as the *symplectic potential 3-form* (cf. Appendix A.1 for the definition of differential forms and their duals). Similarly the symplectic current density with components $J^\mu \equiv -\delta j^\mu$ (satisfying the structural

conservation law $\nabla_\mu J^\mu \approx 0$) can be viewed as the dual of a 3-form $J \equiv -\delta j$ (satisfying $dJ \approx 0$), the latter form being referred to as the *symplectic current* 3-*form*. This notation will be further discussed and applied in sections 6 and 7 below.

For a given space-time $M$, one is not necessarily interested in the explicit expression of the symplectic current 3-form $J$, but only in its flux through a given 3-dimensional hypersurface $\Sigma$ [62]. This flux only requires the knowledge of the components of the current ($J^\mu$) which are normal to the hypersurface or, equivalently, the evaluation of the pullback (restriction) of the 3-form $J$ to the hypersurface. Following reference [62], we put a bar over geometric quantities which are intrinsically defined on $\Sigma$.

More precisely, the inclusion (embedding) map $\iota : \Sigma \hookrightarrow M$ of the 3-dimensional hypersurface $\Sigma$ into $M$ allows us to pull back the 3-form $j$ from $M$ to $\Sigma$ (cf. Appendix A.4): in terms of local coordinates $(y^a)_{a=1,2,3}$ on $\Sigma$, the resulting 3-form $\iota^* j \equiv \underset{\leftarrow}{j}$ then admits the expansion

$$\underset{\leftarrow}{j} = \frac{1}{3!}\, \bar{j}_{abc}\, dy^a \wedge dy^b \wedge dy^c\,.$$

The induced volume element on $\Sigma$ reads $\frac{1}{3!}\, \varepsilon_{abc}\, dy^a \wedge dy^b \wedge dy^c = \sqrt{|h|}\, d^3 y$.

For vacuum general relativity described by the Einstein-Hilbert action, it was shown by G. Burnett and R. Wald [62] that the *pullback* $\underset{\leftarrow}{j}$ *of the symplectic potential* 3-*form* $j$ *given by* (122) *to a (nowhere null) hypersurface* $\Sigma$ *writes*

$$\boxed{\underset{\leftarrow}{j} = \bar{\Pi}^{ab}\, \delta \bar{h}_{ab} + \delta \bar{\alpha} + d\bar{\beta}\,.} \tag{138}$$

Here, $(\bar{\Pi}^{ab})_{cde} \equiv \bar{\Pi}^{ab}\, \bar{\varepsilon}_{cde}$ corresponds to the gravitational momentum (130) in the case of a space-like hypersurface, i.e. the first term in the expansion (138) corresponds to the normal component of the symplectic potential current ($j_{GR}^\mu$) associated to $\mathcal{L}_{GR}$, see equations (134)-(135). The second contribution involves the 3-form

$$\bar{\alpha} \equiv 2\epsilon K \sqrt{|h|}\, d^3 y\,, \tag{139}$$

where we recognize the Lagrangian 3-form defining the Gibbons-Hawking-York boundary action (129) which is associated to the Dirichlet boundary conditions (128) (i.e. fixed metric ($h_{ab}$) on the hypersurface $\Sigma$). The last term in (138) involves a 2-form $\bar{\beta}$ with components given by

$$\beta_{\rho\sigma} = n^\mu\, \delta n^\nu\, \varepsilon_{\mu\nu\rho\sigma}\,. \tag{140}$$

This contribution $d\bar{\beta}$ to $\underset{\leftarrow}{j}$ (which is an exact form) is referred to as a "corner term" and the 2-form $\bar{\beta}$ can be integrated over a 2-surface. We note that the expansion (138) reflects the decomposition (137) of $S_{EH}$, i.e.

$$S_{EH} = S_{GR} - S_{GHY} + S_c\,,$$

where the *boundary Lagrangian* $\mathcal{L}_{GHY}$ yields the contribution $\delta \bar{\alpha}$ to the symplectic potential on $\Sigma$, cf. Eqn. (94).

More specifically, if the hypersurface $\Sigma$ is the lateral boundary $\mathcal{B}$ of space-time as given by $r = $ constant (i.e. a time-like hypersurface) and if we consider space-time slices that are space-like hypersurfaces, then we have "orthogonal corners" given by 2-spheres (corresponding to $t = $ constant and $r = $ constant): for this choice, the corner potential $\bar{\beta}$ vanishes (see [181] and references therein).

Although the result (138) dates back to 1990, it was only realized fairly recently (e.g. see references [180–182]) that such an expansion of the pullback of the symplectic potential form (of vacuum general relativity) has a great interest in more general contexts (other theories, other boundary conditions, presence of anomalies,...) to which we will come back in Subsection 7.8 below.

**Conservation of energy and momentum:** A sensible definition of total energy and momentum for the space-time manifold $M$ exists in the case where the latter manifold is *asymptotically flat*. Then, we can (asymptotically) restrict the general coordinate transformations to Poincaré transformations and construct the corresponding conserved charges. The restriction to an asymptotically flat manifold also avoids the occurrence of singular points in the space of solutions of Einstein's field equations [166, 183]. Our presentation follows the lines of arguments described in reference [166].

From (122)-(125), it follows that

$$\Omega = -\delta\Theta, \qquad \text{with} \quad \Theta \equiv \int_\Sigma d\Sigma_\alpha \sqrt{|g|}\, j^\alpha, \tag{141}$$

where

$$j^\alpha = g^{\mu\nu}\delta\Gamma^\alpha_{\nu\mu} - g^{\mu\alpha}\delta\Gamma^\nu_{\nu\mu}, \qquad \text{i.e.} \quad \boxed{j^\alpha = -\Big[g^{\mu\nu}\nabla^\alpha(\delta g_{\mu\nu}) - g^{\alpha\mu}\nabla^\nu(\delta g_{\mu\nu})\Big].} \tag{142}$$

The linearized equations of motion imply that $\nabla_\alpha j^\alpha = 0$, hence the 1-form $\Theta$ is Poincaré-invariant. If $V$ denotes a vector field in $Z$ that is tangent to the gauge orbits, then the invariance of $\Theta$ is expressed by

$$0 = L_V\Theta = (i_V\delta + \delta i_V)\Theta,$$

i.e., by virtue of (141),

$$\boxed{i_V\Omega = \delta H,} \qquad \text{with} \qquad H \equiv i_V\Theta. \tag{143}$$

(We note that these relations are completely analogous to the ones encountered in classical mechanics, see Eqn. (C.24).)

To evaluate $H$, we use

$$i_V(\delta g_{\mu\nu}) = \Big(L_{\varepsilon^\lambda\partial_\lambda}g\Big)_{\mu\nu} = \nabla_\mu\varepsilon_\nu + \nabla_\nu\varepsilon_\mu,$$

where $L_{\varepsilon^\lambda\partial_\lambda}$ denotes the Lie derivative of the metric tensor field $(g_{\mu\nu})$ with respect to the vector field $\varepsilon^\lambda\partial_\lambda$ on $M$. By substituting this expression into $H \equiv i_V\Theta$ with $\Theta$ given by (141) and by using the fact that $[\nabla_\alpha, \nabla_\beta]\varepsilon^\beta = 0$ due to the equations of motion, one obtains

$$H = -\int_\Sigma d\Sigma_\alpha \sqrt{|g|}\Big[2\nabla^\alpha\nabla^\mu\varepsilon_\mu - \nabla^\mu(\nabla_\mu\varepsilon^\alpha + \nabla^\alpha\varepsilon_\mu)\Big]$$

$$= -\int_\Sigma d\Sigma_\alpha \sqrt{|g|}\,\nabla_\mu B^{\alpha\mu}, \qquad \text{with} \quad B^{\alpha\mu} \equiv \nabla^\alpha\varepsilon^\mu - \nabla^\mu\varepsilon^\alpha. \tag{144}$$

Applications of Stokes' theorem to the $(n-1)$-dimensional hypersurface $\Sigma \subset M$ (see (A.10)) now yields the *surface integral*

$$\boxed{H = -\oint_{\partial\Sigma} d\Sigma_{\alpha\beta} \sqrt{|g|}(\nabla^\alpha\varepsilon^\beta - \nabla^\beta\varepsilon^\alpha).} \tag{145}$$

The integrand of this expression is nothing but the *Komar integrand* [184]. Thus, for a time-like vector field $\varepsilon^\alpha \partial_\alpha$, the surface integral (145) yields the *Komar* or *Noether energy* [177]. Though this quantity enjoys various useful properties, it also suffers from several shortcomings [185].

In order to recover the familiar [176] (so-called ADM) expressions for the conserved charges, one uses the fact that the quantity $H$ satisfying $i_V \Omega = \delta H$ is only defined up to the addition of a term $H' = \delta \mathfrak{H}$ where $\mathfrak{H}$ represents a 0-form, i.e. a functional on $Z$. A natural such addition to $H$ is given by the surface integral

$$\tilde{H} \equiv \oint_{\partial \Sigma} d\Sigma_{\alpha\beta} \left( j^\alpha \varepsilon^\beta - j^\beta \varepsilon^\alpha \right), \tag{146}$$

where the symplectic potential current $j^\alpha$ is given by (142), i.e. $j^\alpha = -\left[ \nabla^\alpha (\delta \ln|g|) - g^{\alpha\mu} \nabla^\nu (\delta g_{\mu\nu}) \right]$ and $\nabla_\alpha j^\alpha = 0$. More precisely, for the definition of the *total energy-momentum on the asymptotically flat manifold M,* we assume that the metric

$$h_{\mu\nu}(x) \equiv g_{\mu\nu}(x) - \eta_{\mu\nu}$$

(and thereby also the variations $\delta g_{\mu\nu}(x)$) fall off at least as fast as $1/r$ for $r \to \infty$ on the hypersurface $\Sigma$ (e.g. constant time hypersurface). As usual, one raises the indices of the metric $h_{\mu\nu}$ with the flat space metric $\eta^{\mu\nu}$, i.e. $h^\mu{}_\mu \equiv \eta^{\mu\nu} h_{\mu\nu}$. While taking into account the asymptotic behavior of $h_{\mu\nu}$ on $\Sigma$, one concludes that $\tilde{H}$ is given by

$$\tilde{H} = \delta \mathfrak{H}, \qquad \text{with} \qquad \mathfrak{H} = \oint_{\partial\Sigma} d\Sigma_{\alpha\beta} \left[ \varepsilon^\beta (\partial^\alpha h^\mu{}_\mu - \partial_\mu h^{\mu\alpha}) - (\alpha \leftrightarrow \beta) \right].$$

In summary, we have

$$\boxed{i_V \Omega = \delta(H + \tilde{H}),} \qquad \text{with} \qquad \boxed{H + \tilde{H} = \varepsilon^\delta P_\delta} \qquad (\varepsilon^\delta \in \mathbb{R}), \tag{147}$$

and

$$\boxed{P_\delta \equiv \oint_{\partial\Sigma} d\Sigma_{\alpha\beta} \left[ -\partial^\alpha h^\beta{}_\delta + (\partial^\alpha h^\mu{}_\mu - \partial_\mu h^{\mu\alpha}) \eta^\beta{}_\delta - (\alpha \leftrightarrow \beta) \right].}$$

For the choice of a constant time hypersurface $\Sigma$ in four-dimensional space-time $M$, we recover [166] the familiar **ADM expressions for the total energy $E$ and momentum $\vec{P}$**:

$$E \equiv P_0 = \oint_{\partial\Sigma} dS_i \left[ \partial_j h_{ij} - \partial_i h_{jj} \right], \tag{148}$$

$$P_k = \oint_{\partial\Sigma} dS_i \left[ \partial^i h^0{}_k - \partial^0 h^i{}_k + (\partial^0 h^i{}_j - \partial_j h^{0j}) \eta^i{}_k \right].$$

**Different formulations of gravity:** For completeness we will briefly outline several equivalent formulations of general relativity for which the covariant phase space approach (symplectic structure, conservation laws for gravitational charges,...) has been addressed in recent years.

In gravity [186], the metric tensor field $(g_{\mu\nu})$ can be decomposed with respect to *tetrad (vielbein) fields* $(e^a{}_\mu)$, i.e.

$$g_{\mu\nu} = \eta_{ab} e^a{}_\mu e^b{}_\nu. \tag{149}$$

Here, the matrix $(e^a{}_\mu(x))$, which is labeled by a curved space index $\mu$ and a flat (tangent) space index $a$, is assumed to be invertible at each space-time point $x$. The inverse of this

matrix is denoted by $(e^\mu{}_a(x))$. The tetrad fields may be gathered in the so-called *vielbein* 1-*forms* $e^a \equiv e^a{}_\mu dx^\mu$.

The decomposition (149) induces a *local Lorentz symmetry*, $e^a{}_\mu \mapsto (e^a{}_\mu)' = \Omega^a{}_b e^b{}_\mu$, where the matrix $(\Omega^a{}_b(x))$ belongs to the Lorentz group $SO(1, n - 1)$. The gauging of this symmetry is realized by the introduction of a $so(1, n - 1)$-valued *connection* 1-*form* $\omega_\mu dx^\mu \equiv \omega \equiv (\omega^a{}_b)$ (often referred to as Lorentz or spin connection) whose curvature 2-form is given by $R \equiv d\omega + \omega \wedge \omega$. For concreteness, we will focus on *four space-time dimensions* in the following.

The action for pure gravity represents a functional of the tetrad fields and of the components of the Lorentz connection (to be considered as independent variables). It is referred to as the *Einstein-Cartan action* or *Palatini-Cartan action* (see references [187, 188] for the underlying history) and reads

$$S[e, \omega] \equiv \int_M \varepsilon_{abcd} e^a \wedge e^b \wedge R^{cd}. \tag{150}$$

Einstein's *cosmological term* then writes $\Lambda \int_M \varepsilon_{abcd} e^a \wedge e^b \wedge e^c \wedge e^d$. Another additional term was put forward more recently by S. Holst [189]: it is given by

$$\int_M \eta_{ac} \eta_{bd} e^a \wedge e^b \wedge R^{cd}. \tag{151}$$

This contribution comes with an overall coefficient that is related to the so-called *Barbero-Immirzi parameter* which was introduced in Ashtekar's canonical formulation of gravity. Even more generally, different topological terms can be added to the action (e.g. see references [190, 192]), namely the *Pontryagin* and *Euler* terms which depend on the curvature as well as the *Nieh-Yan* term which also depends on the torsion 2-forms $T^a \equiv de^a + \omega^a{}_b e^b$. The resulting total action is discussed in references [190, 191], see also [188]. We note that the covariant phase space approach to gravity based on tetrad variables is addressed in particular in references [188, 193–195].

Pure gravity in four dimensions with a cosmological term can equivalently be described by starting from the action for a *BF model*,

$$S[B, F] \equiv \int_M \text{Tr}\left(B \wedge F + \frac{\lambda}{2} B \wedge B\right). \tag{152}$$

Here, $B \equiv (B_{ab})$ represents a 2-form and $F = dA + A \wedge A$ the curvature of a connection 1-form $A \equiv (A_{ab})$. Different variants of this approach exist, some of them involving an extra auxiliary vector field, thereby generalizing the approach of MacDowell and Mansouri [196], see references [191, 197].

### 5.1.3 Moduli spaces of solutions and quantization

For a given classical field theory, one is eventually interested in certain classes of solutions of the field equations: these represent a subspace $\mathcal{Z}$ of the phase space of all solutions, this subspace being often referred to as *moduli space* in the physics literature, e.g. see reference [198] for a general discussion. In this instance, one restricts the symplectic 2-form $\Omega$ on covariant phase space $Z$ to this subspace $\mathcal{Z}$ (i.e. one considers $\Omega|_{\mathcal{Z}}$), then determines the associated Poisson brackets $\{\cdot, \cdot\}_{\mathcal{Z}}$ and finally quantizes the moduli space by replacing the bracket $\{F, G\}_{\mathcal{Z}}$ by $\frac{1}{i\hbar}$ times the commutator of the Hilbert space operators $\hat{F}$ and $\hat{G}$ which are associated to

the real-valued functions $F$ and $G$, respectively. The authors of [159] refer to this procedure as *on-shell quantization*.

By way of illustration [159], we consider the quantization of the left chiral sector of the free massless scalar field in two space-time dimensions: for this field, the action $S[\varphi] \equiv \frac{1}{2} \int_{\mathbb{R}^2} d^2x \, \partial^\mu \varphi \, \partial_\mu \varphi$ yields the equation of motion $0 = \partial^\mu \partial_\mu \varphi$ whose general solution reads

$$\varphi(t,x) = f(t+x) + g(t-x),$$

where $f$ and $g$ represent arbitrary smooth real-valued functions on $\mathbb{R}$. The latter describe left and right moving traveling waves, respectively, which are related by the parity transformation $x \rightsquigarrow -x$. The symplectic 2-form on covariant phase space, as written on a hypersurface $t = $ constant, is again given by (92), i.e.

$$\Omega = \int_{\mathbb{R}} dx \, \delta \varphi \wedge \delta \pi, \qquad \text{with } \pi \equiv \dot{\varphi}.$$

The moduli space $\mathcal{Z}$ of left movers writes

$$\mathcal{Z} \equiv \{\varphi : (t,x) \mapsto f(t+x) \,|\, f \in C^\infty(\mathbb{R})\},$$

and the symplectic 2-form $\Omega$ restricted to $\mathcal{Z}$ reads

$$\Omega|_{\mathcal{Z}} = \int_{\mathbb{R}} d\xi \, \delta f(\xi) \wedge \delta f'(\xi).$$

Thus, it is determined by the variations of the functions parametrizing the moduli space. The quantization of the moduli space $\mathcal{Z}$ now amounts to replacing the Poisson brackets

$$\{f(\xi_1), f'(\xi_2)\}_{\mathcal{Z}} = \delta(\xi_1 - \xi_2),$$

by the commutation relation $[\hat{f}(\xi_1), \hat{f}'(\xi_2)] = i\hbar \, \delta(\xi_1 - \xi_2) \, \mathbb{1}$. The resulting quantum field theory is equivalent to the one obtained by canonical quantization of the scalar field $\varphi$ and then projecting the Fock space of states onto the subspace for which all right movers are in the vacuum state [159]. The terminology on-shell quantization for the above procedure is justified by the fact that all considerations are on-shell apart from the symplectic 2-form $\Omega$ which represents the starting point.

The outlined procedure has been successfully applied to specific moduli spaces in supergravity (with all fluxes fixed) [159, 170, 199] and appears to be the only practical way to quantize in this type of applications.

### 5.1.4 Further examples

For matter and/or gauge fields $\varphi$ coupled to the gravitational field described by a given metric tensor field $\mathfrak{g} \equiv (g_{\mu\nu})$, the action reads $S_M[\varphi, \mathfrak{g}] = \int_M d^n x \sqrt{|g|} \, \mathcal{L}(\varphi, \nabla_\mu \varphi)$. The covariant phase space approach in curved or flat space described above has been applied to numerous field theoretical models in diverse dimensions. A (probably incomplete) list is as follows:

- Complex scalar fields [1, 71]

- Conformally invariant wave equation in four dimensional curved space [1]

- Abelian gauge fields on a Riemannian manifold [200]

- Quantum chromodynamics [201]

- $(2 + 1)$-dimensional gravity [202]

- General relativity in tetrad variables [188, 203]

- Ashtekar's canonical gravity [204]

- Massive spin-2 field [97]

- Chern-Simons theory in arbitrary odd dimension [71, 205]

- Abelian $p$-form theories [206]

- *BF* model [162]

- Topological massive gravity [207]

- Dirac equation for spinor fields [1, 71]

- Fronsdal theory for massless fields of arbitrary integer spin [97]

- Supergravity and related geometries or configurations [159, 170, 199, 208]

- Eleven-dimensional supergravity [205]

- Sigma models [71]

- Massive particle on the $AdS_3$ manifold [209]

- A spinning particle in an electromagnetic and a gravitational field [210]

- String interacting with a scalar field (Lund-Regge equations) [210]

- Inclusion of topological terms for the Lagrangians of gravity or string theory [163, 164]

- String field theory [58, 153, 166]

- Various relativistic and non-relativistic (integrable) field theories in two dimensions [36] (namely the sine-Gordon model, the non-linear Schrödinger equation, the (modified) KdV equation [211]), different versions of the Monge-Ampère equation in two dimensions [72, 212], the WZW model [101, 122, 213]

- Generalization to *higher-derivative field theories* or *non-local theories* [161, 214] with applications to general relativity (considered as a second-order field theory depending on the metric field and its first and second order derivatives) as well as induced gravity, i.e. the gravitational WZW model [161], this treatment being closely related to the one of the Jackiw-Teitelboim model for 2-dimensional gravity with a cosmological constant [214]

- The so-called parametrized field theories on curved space-time [215]

- Edge modes in gauge field theories in the presence of boundaries [216–218]

We also note that the covariant phase space *quantization* has been carried out in detail in reference [214] for some non-trivial two-dimensional models (see also [101, 122]).

## 5.2 Covariant phase space and Peierls bracket

In this section, we show that the symplectic 2-form $\Omega$ on covariant phase space $Z$ can be obtained from the multisymplectic $(n+1)$-form $\omega$ introduced in the multisymplectic approach. Moreover, we show that the *bracket introduced by R. E. Peierls* in his attempt to construct a covariant canonical formulation of field theory *is nothing but the Poisson bracket associated to the symplectic 2-form $\Omega$ on covariant phase space:* this result was first established in reference [51] by considering the standard Hamiltonian approach and more recently [52, 53] within the covariant canonical formulation of field theory.

### 5.2.1 Derivation of the symplectic form $\Omega$ from the multisymplectic form $\omega$

*Covariant phase space $Z$* is defined to be the infinite-dimensional space of all solutions $\phi \equiv (\varphi^a, \pi_a^\mu)$ of the covariant Hamiltonian equations of motion (4) involving the Hamiltonian $\mathcal{H}(\varphi^a, \pi_a^\mu)$, the latter equations being equivalent to the Lagrangian equations of motion. We recall from Section 3.1 that a field $\phi$ represents, from the geometric point of view, a smooth section of *ordinary multiphase space $\tilde{P}$*, see Eqn. (14).

As we discussed in Section 5.1, the space $Z$ is an infinite-dimensional weak symplectic manifold with symplectic 2-form $\Omega$ on $Z$. This 2-form is the differential of a 1-form (see equations (84),(90)) which we denote by $\Theta$:

$$\Omega = -\delta\Theta, \qquad \text{with} \quad \Theta \equiv \int_\Sigma d\Sigma_\mu j^\mu, \quad \text{and} \quad \Omega \equiv \int_\Sigma d\Sigma_\mu J^\mu = -\int_\Sigma d\Sigma_\mu \delta j^\mu. \tag{153}$$

*The forms $\Theta$ and $\Omega$ on $Z$ can be derived* [3,4,52,59] *from the multicanonical $n$-form $\theta$ (defined on extended multiphase space $P$) and the multisymplectic $(n+1)$-form $\omega = -d\theta$*, respectively: the latter forms have been introduced within the multisymplectic approach to field theory in Subsection 3.1, see equations (10)-(11). Here, we outline the derivation following reference [52] to which we refer for the mathematical underpinnings.

Since $\theta$ is a $n$-form on extended multiphase space $P$ and since the integrand of $\Theta$ is a $(n-1)$-form $d\Sigma_\mu j^\mu$ on the Cauchy hypersurface $\Sigma \subset M$, we use pull-back maps to pass from $P$ to $M$ and a contraction with a vector field in order to lower the form degree of $\theta$ by one unit. Similarly, for the $(n+1)$-form $\omega$ on $P$, we use pull-back maps and two contractions in order to obtain the $(n-1)$-form $d\Sigma_\mu J^\mu$ on $M$.

First, one considers the Hamiltonian $\mathcal{H} : \tilde{P} \to P$ to pull back the forms $\theta$ and $\omega$ from $P$ to $\tilde{P}$, thereby defining the forms $\theta_{\mathcal{H}}$ and $\omega_{\mathcal{H}}$, see equations (12)-(13). For the contraction one uses vertical vector fields $X, Y$ on $\tilde{P}$, i.e. vector fields which only have components in the fibre direction (this direction being labeled by $(q^a, p_a^\mu)$):

$$\text{Vertical vector field on } \tilde{P}: \qquad X = X^a(q,p)\frac{\partial}{\partial q^a} + X_a^\mu(q,p)\frac{\partial}{\partial p_a^\mu}. \tag{154}$$

The *canonical 1-form $\Theta$ on covariant phase space $Z$* is now given, at the point $\phi$, by

$$\Theta_\phi(X) \equiv \int_\Sigma \phi^*(i_X \theta_{\mathcal{H}}), \tag{155}$$

i.e. its integrand is a $(n-1)$-form on $M$. For the latter we have

$$\phi^*\big(i_X \theta_{\mathcal{H}}\big) = \phi^*\big(i_X[p_a^\mu dq^a \wedge d^{n-1}x_\mu - \mathcal{H}\, d^n x]\big) = \phi^*\big(p_a^\mu X^a(q,p)\, d^{n-1}x_\mu\big) = \pi_a^\mu X^a(\phi)\, d\Sigma_\mu, \tag{156}$$

with $d\Sigma_\mu = \phi^*(d^{n-1}x_\mu)$. By substituting this expression into (155), we get the result

$$\Theta = \int_\Sigma d\Sigma_\mu j^\mu, \qquad \text{with} \quad j^\mu \equiv \pi_a^\mu \delta \varphi^a, \tag{157}$$

in agreement with (83).

Analogously, the *symplectic 2-form $\Omega$ on covariant phase space $Z$* is given, at the point $\phi$, by

$$\boxed{\Omega_\phi(X,Y) \equiv \int_\Sigma \phi^*(i_Y i_X \omega_\mathcal{H}),} \tag{158}$$

its integrand being a $(n-1)$-form on $M$. From

$$\begin{aligned}
\phi^*\big(i_Y i_X \omega_\mathcal{H}\big) &= \phi^*\big(i_Y i_X[dq^a \wedge dp_a^\mu \wedge d^{n-1}x_\mu + d\mathcal{H} \wedge d^n x]\big) \\
&= [X^a(\phi)Y_a^\mu(\phi) - Y^a(\phi)X_a^\mu(\phi)]d\Sigma_\mu,
\end{aligned}$$

and from (158), we conclude that

$$\Omega = \int_\Sigma d\Sigma_\mu J^\mu, \qquad \text{with} \quad J^\mu \equiv \delta\varphi^a \wedge \delta\pi_a^\mu = -\delta j^\mu, \tag{159}$$

in agreement with (84),(90). From this construction it follows that the symplectic 2-form $\Omega$ depends on the dynamics by virtue of the Hamiltonian $\mathcal{H}$ though it is independent of the Cauchy surface appearing in its definition (according to the argumentation outlined in Section 5.1, see (93)). We also remark that the expression for $\Omega$ given in Eqn. (158) is coordinate independent.

### 5.2.2 The Poisson bracket associated to the symplectic form $\Omega$ on $Z$ coincides with the Peierls bracket

Since the covariant phase space $Z$ endowed with the symplectic 2-form $\Omega$ defined by (158) represents an infinite-dimensional weak symplectic manifold, it induces a non-degenerate Poisson bracket $\{F, G\}$ of smooth functionals $F, G$ on $Z$, see Eqn. (D.27) of Appendix D. Following [55], we will now show that this Poisson bracket coincides with the Peierls bracket.

As discussed in Appendix D, *the Poisson bracket $\{\cdot, \cdot\}$ on an (infinite-dimensional) manifold $Z$ endowed with a (weak) symplectic form $\Omega$ is defined by Eqn. (D.27)*, i.e. for any two smooth functionals $F, G$ on $Z$ we have

$$\boxed{\{F, G\} \equiv \Omega(X_F, X_G).} \tag{160}$$

Here, $X_F$ is the Hamiltonian vector field associated to the functional $F$ with respect to the symplectic form $\Omega$. In this respect, it is judicious to recall the definition of the Hamiltonian vector field $X_f$ on a finite-dimensional symplectic manifold $(M, \omega)$ (see Eqn. (C.16)):

$$i_{X_f} \omega = df. \tag{161}$$

Here, $f : M \to \mathbb{R}$ is a given smooth function and $\omega$ the symplectic 2-form on $M$. By applying this relation to a vector field $Y \equiv Y^i \frac{\partial}{\partial q^i}$ on $M$ (where $(q^i)$ denote local coordinates on $M$), we obtain

$$\big(i_{X_f} \omega\big)(Y) = df(Y).$$

From $df(Y) = (\partial_i f) Y^i \equiv (\vec{\nabla} f) \cdot \vec{Y}$ and the definition of the interior product $i_{X_f}$ we conclude that the definition (161) of $X_f$ is equivalent to

$$\omega(X_f, Y) = (\vec{\nabla} f) \cdot \vec{Y}. \tag{162}$$

Thereby, the Poisson bracket $\{f, g\} \equiv \omega(X_f, X_g)$ associated to the symplectic form $\omega$ on $M$ can be written as

$$\{f, g\} = (\vec{\nabla} f) \cdot \vec{X}_g. \tag{163}$$

The generalization of relations (162) and (163) to the infinite-dimensional weak symplectic manifold $(Z, \Omega)$ obviously reads

$$\Omega(X_F, Y) = \int_M d^n x \, \frac{\delta F}{\delta \phi^I(x)} Y^I[\phi](x), \tag{164}$$

and

$$\{F, G\} = \int_M d^n x \, \frac{\delta F}{\delta \phi^I(x)} X_G^I[\phi](x). \tag{165}$$

In the covariant Hamiltonian approach which we continue to consider here, we have $(\phi^I) = (\varphi^a, \pi_a^\mu)$.

We now come to the fundamental result [55]. As starting point we consider the symplectic form $\Omega$ on $Z$ defined by Eqn. (158), i.e. $\Omega(X, Y) \equiv \int_\Sigma \phi^*(i_Y i_X \omega_{\mathcal{H}})$, and a non-degenerate covariant Hamiltonian $\mathcal{H}$ (i.e. $\det\left(\frac{\partial^2 \mathcal{H}}{\partial p_a^0 \partial p_b^0}\right) \neq 0$). To simplify the notation, we consider the case of a single real-valued field $\varphi$ on the space-time manifold $M$, i.e. $(\varphi^a, \pi_a^\mu) = (\varphi, \pi^\mu)$. The linearized De Donder-Weyl equations (87) can be written in operatorial form as $\mathcal{J}[\phi]\delta\phi = 0$ where $\delta\phi \equiv (\delta\phi^I) = [\delta\varphi, \delta\pi^\mu]^t$:

$$0 = \begin{bmatrix} -\partial_\mu(\delta\pi^\mu) - \frac{\partial^2 \mathcal{H}}{\partial\varphi^2}\delta\varphi - \frac{\partial^2 \mathcal{H}}{\partial\pi^\nu\partial\varphi}\delta\pi^\nu \\ \partial_\mu(\delta\varphi) - \frac{\partial^2 \mathcal{H}}{\partial\varphi\partial\pi^\mu}\delta\varphi - \frac{\partial^2 \mathcal{H}}{\partial\pi^\nu\partial\pi^\mu}\delta\pi^\nu \end{bmatrix} = \begin{bmatrix} -\frac{\partial^2 \mathcal{H}}{\partial\varphi^2} & -\partial_\nu - \frac{\partial^2 \mathcal{H}}{\partial\pi^\nu\partial\varphi} \\ \partial_\mu - \frac{\partial^2 \mathcal{H}}{\partial\varphi\partial\pi^\mu} & -\frac{\partial^2 \mathcal{H}}{\partial\pi^\nu\partial\pi^\mu} \end{bmatrix} \begin{bmatrix} \delta\varphi \\ \delta\pi^\nu \end{bmatrix}$$
$$\equiv \left(\mathcal{J}[\phi]_{IJ}\,\delta\phi^J\right) \equiv \mathcal{J}[\phi]\delta\phi. \tag{166}$$

The linearized field equation (166) is also referred to as *Jacobi equation* for the so-called *Jacobi fields* $\delta\phi$ and the operator $\mathcal{J}[\phi]$ is then referred to as *Jacobi operator* for the dynamical system [55] (see Section 4.2.1). If the space-time manifold $M$ is *globally hyperbolic* and if the linearized equation of motion (166) represents a *hyperbolic system* of PDE's, then the operator $\mathcal{J}[\phi]$ admits uniquely defined retarded and advanced Green functions $G^- \equiv G_{\text{ret}}$ and $G^+ \equiv G_{\text{adv}}$. Thereby it also admits a uniquely defined *causal Green function* $\tilde{G} \equiv G^- - G^+$ (all of these functions being, strictly speaking, distributions). For $M = \mathbb{R}^n$ (which we consider hereafter), we have translation invariance and thereby the functions $(x, y) \mapsto G^\pm(x, y)$ only depend on the difference $x - y$. For the Jacobi operator $\mathcal{J}$, we thus have

$$\mathcal{J}G^\pm = \delta, \qquad \text{and} \qquad \mathcal{J}\tilde{G} = 0,$$

or, more explicitly,

$$\mathcal{J}_x[\phi]_{IJ}\, G^\pm[\phi]^{JK}(x - y) = \delta_I^K\,\delta(x - y),$$

where the notation $\mathcal{J}_x$ means that the operator $\mathcal{J}$ acts on the variable $x$.

For a given functional $F$ on covariant phase space $Z$, solutions $X_F^\pm$ of the inhomogeneous Jacobi equation $\mathcal{J}X_F^\pm = \frac{\delta F}{\delta\phi}$ are thus given by the convolution of the Green functions $G^\pm$ with

the inhomogeneous term $\frac{\delta F}{\delta \phi}$, i.e. $X_F^{\pm} \equiv G^{\pm} * \frac{\delta F}{\delta \phi}$: the difference of these solutions represents a solution of the homogeneous Jacobi equation:

$$\mathcal{J} X_F = 0, \qquad \text{for} \quad X_F \equiv X_F^- - X_F^+ = \tilde{G} * \frac{\delta F}{\delta \phi}, \tag{167}$$

i.e. $X_F[\phi]^I(x) = \int_M d^n y \, \tilde{G}[\phi]^{IJ}(x-y) \, \frac{\delta F}{\delta \phi^J(y)}$.

The fundamental result is that *the formal vector field* $(X_F^I) \equiv X_F \equiv \tilde{G} * \frac{\delta F}{\delta \phi}$ *associated to the functional F on Z is the Hamiltonian vector field associated to F with respect to the symplectic form* $\Omega$ *on Z, i.e. satisfies relation* (164). This result can be established as follows. According to Eqn. (158), we have

$$\Omega_{\phi}(\delta\phi_1, \delta\phi_2) = \int_{\Sigma} \phi^* \omega_{\mathcal{H}}(\delta\phi_1, \delta\phi_2) = \int_{\Sigma} d\Sigma_{\mu} J_{\phi}^{\mu}(\delta\phi_1, \delta\phi_2),$$

with $\delta\phi_r = (\delta\varphi_r, \delta\pi_r^{\mu})$ for $r \in \{1, 2\}$ and (see Eqn. (159))

$$J_{\phi}^{\mu}(\delta\phi_1, \delta\phi_2) = \delta\varphi_1 \delta\pi_2^{\mu} - \delta\varphi_2 \delta\pi_1^{\mu}. \tag{168}$$

For simplicity, we again consider the case of a free scalar field $\varphi$ to illustrate the general arguments put forward in reference [55]. For this case, we have $\mathcal{H}(\phi) = \frac{1}{2} \pi^{\mu} \pi_{\mu} + \frac{m^2}{2} \varphi^2$, hence the linearized equation of motion (166) reads

$$0 = \mathcal{J}[\phi]\delta\phi = \begin{bmatrix} -m^2 & -\partial_{\nu} \\ \partial_{\mu} & -\eta_{\mu\nu} \end{bmatrix} \begin{bmatrix} \delta\varphi \\ \delta\pi^{\nu} \end{bmatrix} = \begin{bmatrix} -\partial_{\nu}(\delta\pi^{\nu}) - m^2 \delta\varphi \\ \partial_{\mu}(\delta\varphi) - \delta\pi_{\mu} \end{bmatrix}, \tag{169}$$

and we have

$$\mathcal{J}[\phi] X_F^{\pm}[\phi] = \mathcal{J} \begin{bmatrix} X_F^{\pm} \\ X_F^{\pm\mu} \end{bmatrix} = \begin{bmatrix} -\partial_{\mu} X_F^{\pm\mu} - m^2 X_F^{\pm} \\ \partial_{\mu} X_F^{\pm} - X_{F\mu}^{\pm} \end{bmatrix}.$$

By using this relation (involving the Jacobi operator $\mathcal{J}$) and expression (168) for the symplectic current density $(J_{\phi}^{\mu})$, we can easily evaluate the divergence of the latter current while taking into account the linearized equation of motion (169): a short calculation yields

$$\partial_{\mu}\left(J_{\phi}^{\mu}(X_F^{\pm}[\phi], \delta\phi)\right) = (\mathcal{J} X_F^{\pm})\delta\varphi + (\mathcal{J} X_F^{\pm})_{\mu} \delta\pi^{\mu} = \frac{\delta F}{\delta\phi} \cdot \delta\phi, \tag{170}$$

where we used the matricial relation $\mathcal{J} X_F^{\pm} = \mathcal{J}(G^{\pm} * \frac{\delta F}{\delta\phi}) = \frac{\delta F}{\delta\phi}$.

To conclude, one assumes that $\frac{\delta F}{\delta\phi}$ has support in a finite time interval $I$ and considers the past and future light-cones of $I$. Moreover, one considers the region $S_+$ between the Cauchy surface $\Sigma$ and a Cauchy surface $\Sigma^+$ in the future as well as the region $S_-$ between the Cauchy surface $\Sigma$ and a Cauchy surface $\Sigma^-$ in the past, see Figure 5.

By virtue of Gauss's theorem we have

$$\int_{S_-} d^n x \, \partial_{\mu} J^{\mu}(X_F^-, \delta\phi) = \int_{\Sigma} d\Sigma_{\mu} J^{\mu}(X_F^-, \delta\phi) - \int_{\Sigma^-} d\Sigma_{\mu} J^{\mu}(X_F^-, \delta\phi)$$

$$\int_{S_+} d^n x \, \partial_{\mu} J^{\mu}(X_F^+, \delta\phi) = -\int_{\Sigma} d\Sigma_{\mu} J^{\mu}(X_F^+, \delta\phi) + \int_{\Sigma^+} d\Sigma_{\mu} J^{\mu}(X_F^+, \delta\phi). \tag{171}$$

Here, the second integral on the right-hand-side of each equation vanishes since $X_F^-|_{\Sigma^-} = 0$ due to the support property of the retarded Green function $G^-$ and similarly $X_F^+|_{\Sigma^+} = 0$. According

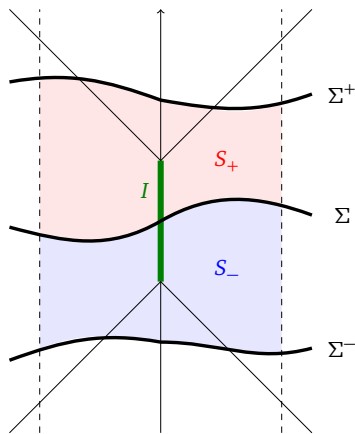

Figure 5: Time-support $I$ of $\frac{\delta F}{\delta \phi}$.

to (170), the integrands on the left-hand-side of the two equations in (171) coincide with $\frac{\delta F}{\delta \phi} \cdot \delta \phi$. We now add the two relations in Eqn. (171) and use the fact that $X_F^- - X_F^+ = X_F$:

$$\int_{S_- \cup S_+} d^n x \, \frac{\delta F}{\delta \phi} \cdot \delta \phi = \int_\Sigma d\Sigma_\mu J^\mu(X_F, \delta \phi) = \Omega(X_F, \delta \phi) \,.$$

Since $\frac{\delta F}{\delta \phi}$ vanishes in $M$ outside of the region $S_- \cup S_+$ (by virtue of the support properties which have been assumed for $\frac{\delta F}{\delta \phi}$), we obtain the result (164). Thus, $X_F \equiv \tilde{G} * \frac{\delta F}{\delta \phi}$ is the Hamiltonian vector field associated to the functional $F$ (with respect to the symplectic form $\Omega$) on $Z$.

From (160) we then conclude that, for two smooth functionals $F, G$ on covariant phase space $Z$ with temporally compact support, *the Poisson bracket associated to the symplectic form $\Omega$ reads* [55]

$$\{F, G\} = \int_M d^n x \int_M d^n y \, \frac{\delta F}{\delta \phi^I(x)} \tilde{G}^{IJ}(x, y) \frac{\delta G}{\delta \phi^J(y)} \,. \tag{172}$$

For the free Klein-Gordon field $\varphi$, the linearized equation of motion for $\delta \varphi$ coincides with the equation of motion for $\varphi$ and thereby *the bracket* (172) *on covariant phase space coincides with the Peierls bracket* (33). This fact is actually true quite generally [51, 52, 55] for non-degenerate Lagrangians and even for degenerate ones if some particular conditions are satisfied, see reference [53].

By construction, the bracket (172) has all of the properties that are required to hold for a Poisson bracket. Yet, in contrast to the standard Poisson bracket of classical field theory (see Appendix D), *this bracket depends on the dynamics (as given by the Hamiltonian $\mathcal{H}$).*

## 5.3 Relationship between the symplectic form $\Omega$ and the Dirac bracket in gauge theories

For gauge field theories we found that the closed 2-form $\Omega_{\text{phys}}$ defined by (106) on the reduced phase space $Z_{\text{phys}}$ is non-degenerate, i.e. represents a symplectic form. Thus, it has the canonical expression (92) in terms of Darboux coordinates if one chooses a hypersurface $\Sigma$ given by $t = \text{constant}$. The associated Poisson bracket then also has the canonical expression on $Z_{\text{phys}}$. As discussed (for the finite-dimensional case) in Appendix C.6, *this Poisson bracket coincides for observables (i.e. gauge invariant functionals) with the Dirac bracket* which is considered in the

standard Hamiltonian approach to gauge field theories [51]. Similarly, *for gauge field theories, it can also be argued that the Peierls bracket of observables coincides with the Dirac bracket of these observables.* Indeed, the object of reference [51] was to establish the equivalence of the different Poisson structures for observables in gauge theories, namely the Dirac bracket of the standard Hamiltonian approach, the Peierls bracket and the Poisson structure associated to the symplectic form $\Omega_{\text{phys}}$ which appears in the covariant phase space approach. We note that the latter approach has the *advantage that one does not have to tackle the constraints originating from gauge invariance* since these are automatically satisfied for the solutions of the field equations [159].

The case of (parametrized) field theories on a globally hyperbolic manifold is discussed in reference [215] which concludes that the standard and covariant phase space approaches both lead to the same reduced phase space in the case of generally covariant theories (see also [60]).

### 5.4 Surface (flux) charges in theories with local symmetries

The derivation of local conservation laws and the definition of conserved charges associated to local symmetries in general relativity that we briefly addressed in Subsection 5.1.2 have a long history going back to the ground-breaking work of A. Einstein [219], e.g. see [220] for the early history and [22,158,185,221–225] as well as references therein for more recent work. In particular, it has been pointed out in the nineties by R. Wald and his collaborators [63–65] that the conserved charges in field theories with local symmetries can be derived in a systematic way by using the covariant space approach: we will discuss this construction in a more general setting in Subsection 7.6 and in Subsection 7.8 along with alternative procedures. The case of Yang-Mills theories will then also be addressed in some detail.

### 5.5 On space-times with boundary

The case of a space-time manifold with a boundary will be discussed towards the end of the next section since it naturally fits into the set-up of the variational bicomplex. A different approach to it will be presented in Subsection 7.8 for the case of diffeomorphism invariant Lagrangian field theories. Here, we only note that the relationship between the Poisson bracket (associated to the symplectic form on covariant phase space) and the Peierls bracket was extended to the case of manifolds with a boundary by the authors of reference [182].

## 6 Approach of variational bicomplex

At the end of the Seventies, the variational bicomplex was introduced independently by I. M. Gel'fand and his collaborators [67] (in his program of making topological invariants local) as well as by A. M. Vinogradov [68], W. M. Tulczyjew [69] and F. Takens [70] in their study of the inverse problem of the calculus of variations. It has been elaborated in particular by I. M. Anderson [226] and applied to the study of PDE's [31] and to the perturbative quantization of field theory, e.g. see [74] and references therein. It may be viewed as a modern geometrical setting for the theory of differential equations and thereby also for classical field theory [71,74,157]. The notion of variational bicomplex (dealing with differential forms on an infinite jet bundle) is closely related to the multisymplectic formulation of field theory (dealing with differential forms on a 1-jet bundle and on its affine dual). In fact, it yields an elegant and powerful reformulation of the latter approach. Accordingly we will introduce it here in terms of the notation and notions that we already considered for multisymplectic geometry and we will illustrate its physical application by virtue of classical mechanics and field

theory [73]. As a matter of fact, the approach to field theory based on the variational bicomplex represents a mathematical framework which encompasses and unifies to some extend all covariant approaches that we have discussed so far including the covariant phase space formulation. It allows in particular for a quite general characterization of the symmetries that appear for ordinary or partial differential equations and in particular for those of Lagrangian or Hamiltonian field theory. We will outline this characterization of symmetries while relating our presentation to the examples discussed in the previous sections.

## 6.1 Definition

The starting point is a given system of differential equations (field equations) on a base manifold which we again assume to be given by Minkowski space-time $M = \mathbb{R}^n$ parametrized by $x \equiv (x^\mu) \equiv (x^0, x^1, \ldots, x^{n-1})$. The dependent variables are fields $x \mapsto \varphi(x) \equiv (\varphi^a(x))_{a=1,\ldots,N}$ solving the field equations under consideration. From the mathematical point of view, a field is viewed as a section $s : M \to E$ of a bundle $E$ over $M$, e.g. the trivial bundle $E = M \times U$, i.e. $s(x) = (x, \varphi(x))$ with $\varphi(x) \in U$. Instead of the 1-*jet bundle* $JE$ over $M$, we now consider the *infinite jet bundle* $J^\infty E$ whose sections $\mathcal{J}^\infty s$ are obtained by *infinite prolongation of the sections $s$ of $E$* according to[12]

$$(\mathcal{J}^\infty s)(x) = \big(x, \varphi(x), \partial_\mu \varphi(x), \partial_\mu \partial_\nu \varphi(x), \ldots \big). \tag{173}$$

Thus, $J^\infty E$ is parametrized by local coordinates

$$\big(x, q \equiv q_{\{0\}}, q_{\{1\}}, q_{\{2\}}, \ldots \big), \qquad \text{with } q \equiv (q^a), \quad q_{\{1\}} \equiv (q^a_\mu), \quad q_{\{2\}} \equiv (q^a_{\mu\nu}), \ldots,$$

and

$$q^a \equiv \varphi^a(x), \quad q^a_\mu \equiv \frac{\partial \varphi^a}{\partial x^\mu}(x), \quad q^a_{\mu\nu} \equiv \frac{\partial^2 \varphi^a}{\partial x^\mu \partial x^\nu}(x) = q^a_{\nu\mu}, \ldots \tag{174}$$

Here, the $q$-variables $q^a_{\mathbf{J}}$ are labeled by an index $a$ and an unordered multi-index $\mathbf{J} = (\mu_1, \ldots, \mu_k)$ with $k \geq 0$ and with the convention that $q^a_\emptyset \equiv q^a$. Even if the given field equations are only of finite order (e.g. of order 2 as it is typically the case in field theory), the consideration of an arbitrarily high order of derivatives is useful for the exploration of general properties of the dynamical system like symmetries and conservation laws (which may depend on higher order derivatives) or of the existence of a Lagrangian (inverse problem of variational calculus).

The algebra $\Omega^\bullet(J^\infty E)$ of *differential forms on $J^\infty E$* is generated (by means of sums and wedge products) by the 1-forms $\{dx^\mu\}$ together with the basis of so-called *contact 1-forms* (whose pullback from $J^\infty E$ to $M$ vanishes),

$$\theta^a \equiv dq^a - q^a_\mu dx^\mu, \quad \theta^a_\mu \equiv dq^a_\mu - q^a_{\mu\nu} dx^\nu, \ldots \tag{175}$$

By definition, a $(k, l)$-**form** $\alpha$ **on** $J^\infty E$ (denoted as $\alpha \in \Omega^{k,l} \equiv \Omega^{k,l}(J^\infty E)$) is a $(k+l)$-form $\alpha$ on $J^\infty E$ which is a finite sum of terms of the form

$$f[q] dx^{\mu_1} \wedge \cdots \wedge dx^{\mu_k} \wedge \theta^{a_1}_{\mathbf{J}_1} \wedge \cdots \wedge \theta^{a_l}_{\mathbf{J}_l}. \tag{176}$$

Here, $f[q]$ denotes a smooth real-valued function of the variables $x, q, q_{\{1\}}, \ldots, q_{\{K\}}$ for some $K \in \mathbb{N}$. The fact that $f$ depends on $x$ and only finitely many variables $q, q^a_\mu, q^a_{\mu\nu}, \ldots$ (corresponding to the values of a field and its derivatives at the same space-time point) means that a $(k, l)$-form is actually a **local form** in the sense of local field theory; e.g. a $(n, 0)$-form

---

[12]Strictly speaking, $(\mathcal{J}^\infty s)(x)$ is defined as an equivalence class, two local sections in $E$ being considered to be equivalent at $x$ if all of their derivatives agree at $x$.

$\mathcal{L}\, d^n x \in \Omega^{n,0}$ (with $d^n x \equiv dx^0 \wedge dx^1 \wedge \cdots \wedge dx^{n-1}$ and $\mathcal{L}$ depending on $x, q^a$ and $q^a_\mu$) describes a first order Lagrangian $\mathcal{L}$ of a local field theory. This locality property of $(k,l)$-forms is also crucial for the validity of various mathematical results to be discussed below and it should be kept in mind.

The exterior derivative of a function $f[q]$ is given by

$$df = \left( dx^\mu \frac{\partial}{\partial x^\mu} + dq^a_{\mathbf{J}} \frac{\partial}{\partial q^a_{\mathbf{J}}} \right) f \,,$$

with summation over all indices $\mu, a$ and multi-indices[13] $\mathbf{J}$. If we reexpress the differential $d$ in terms of the contact forms (175), we have

$$\boxed{d = d_{\mathrm{h}} + d_{\mathrm{v}} \,,} \qquad \text{with} \quad \begin{cases} \boxed{d_{\mathrm{h}} \equiv dx^\mu \partial_\mu : \Omega^{k,l} \to \Omega^{k+1,l} \,,} \\[2mm] \boxed{d_{\mathrm{v}} \equiv \theta^a_{\mathbf{J}} \frac{\partial}{\partial q^a_{\mathbf{J}}} : \Omega^{k,l} \to \Omega^{k,l+1} \,,} \end{cases} \tag{177}$$

and

$$\boxed{\partial_\mu \equiv \frac{\partial}{\partial x^\mu} + q^a_\mu \frac{\partial}{\partial q^a} + q^a_{\mu\nu} \frac{\partial}{\partial q^a_\nu} + \cdots}$$

Here, $\partial_\mu f$ represents the *total derivative* of $f[q]$ with respect to $x^\mu$: it involves the partial derivative $\frac{\partial f}{\partial x^\mu}$ (that measures the explicit dependence of $f$ on $x^\mu$) and the derivatives with respect to the variables $q^a, q^a_\mu, \ldots$ which reflect the $x$-dependence coming from the $x$-dependence of fields $\varphi^a$ and their derivatives,[14] see equations (173)-(174). This total derivative determines the so-called *horizontal differential* $d_{\mathrm{h}} = dx^\mu \partial_\mu$ (acting on the total space of the jet bundle $J^\infty E$ over $M$) which differs from the exterior differential $d = dx^\mu \frac{\partial}{\partial x^\mu}$ that acts on forms on the base space $M$ and does not involve any field dependence. The so-called *vertical differential*

$$\boxed{d_{\mathrm{v}} = \theta^a \frac{\partial}{\partial q^a} + \theta^a_\mu \frac{\partial}{\partial q^a_\mu} + \cdots = d_{\mathrm{v}} q^a \frac{\partial}{\partial q^a} + d_{\mathrm{v}} q^a_\mu \frac{\partial}{\partial q^a_\mu} + \cdots \,,} \tag{178}$$

amounts to an *infinitesimal field variation*. The exterior derivatives $d_{\mathrm{h}}$ and $d_{\mathrm{v}}$ are nilpotent and anticommute with each other in accordance with $d^2 = 0$. Thus, the collection (or, more precisely, direct sum) of spaces $\Omega^{k,l}$ endowed with the differentials $d_{\mathrm{h}}$ and $d_{\mathrm{v}}$ defines a double complex which is referred to as the **variational bicomplex** $\left( \Omega^{\bullet,\bullet}(J^\infty E), d_{\mathrm{h}}, d_{\mathrm{v}} \right)$ for the fibre bundle $E$ over $M$, see Figure 6.

The terminology 'variational' refers to the fact that $d_{\mathrm{v}} \alpha = d_{\mathrm{v}} q^a \frac{\partial \alpha}{\partial q^a} + d_{\mathrm{v}} q^a_\mu \frac{\partial \alpha}{\partial q^a_\mu} + \cdots$ represents the variation of the form $\alpha \in \Omega^{k,l}(J^\infty E)$ induced by the field variation $q^a \rightsquigarrow q^a + d_{\mathrm{v}} q^a$. Since the so-defined bicomplex is only determined by the field content (total space $E$ over $M$), it is also referred to as the *free* variational bicomplex. In Section 6.4 below, we will consider the variational bicomplex $\left( \Omega^{\bullet,\bullet}(\mathcal{R}^\infty), d_{\mathrm{h}}, d_{\mathrm{v}} \right)$ obtained by reducing the bundle $J^\infty E$ to the subbundle $\mathcal{R}^\infty$ describing the solutions of given field equations, i.e. the bicomplex associated to a dynamical system. For the description of local symmetries it may also be convenient to introduce antifields carrying a ghost-number (as in the Batalin-Vilkovisky approach to the quantization of gauge field theories): this leads to the extension of the variational bicomplex to a tricomplex, see Subsection 7.6.2.

---

[13] For $\mathbf{J} = (\mu_1, \ldots, \mu_k)$, the derivative $\partial/\partial q^a_{\mathbf{J}}$ is always understood to be the symmetrized derivative so as to avoid combinatorial factors, e.g. the appendices of references [154, 157].

[14] Here, we have adopted the notation which is standard in physics: in the mathematics literature, the total derivative with respect to $x^\mu$ is usually denoted by $D_\mu$ and the partial derivative by $\partial_\mu$.

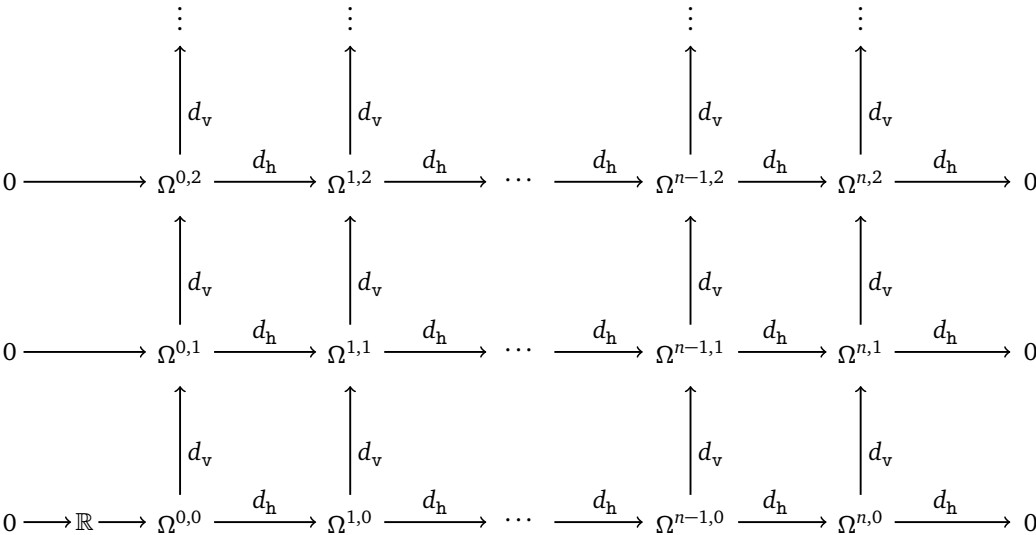

Figure 6: Variational bicomplex $\left(\Omega^{\bullet,\bullet}(J^{\infty}E), d_{\mathrm{h}}, d_{\mathrm{v}}\right)$.

We note that in the case of a trivial fibre bundle $\mathbb{R}^n \times \mathbb{R}^N \to \mathbb{R}^n$ over space-time $\mathbb{R}^n$, both the horizontal and the vertical sequences of the variational bicomplex are exact (except for the bottom and top horizontal degree, see next paragraph), i.e. one can apply the Poincaré lemma for each space and either $d_{\mathrm{h}}$ or $d_{\mathrm{v}}$. Below we will repeatedly use this fundamental result (going back to I. M. Anderson, L. Dickey, F. Takens, T. Tsujishita [227], W. M. Tulczyjew and A. M. Vinogradov) and refer to the literature [228] for further results on the cohomology, e.g. for non-trivial fibre bundles and/or an enlarged bicomplex.

We remark that for the calculus with differential forms, it is important to know the action of the exterior derivative on the monomials $dx^{\mu}$ and $\theta_{\mathbf{J}}^a$:

$$d_{\mathrm{h}}(dx^{\mu}) = 0 = d_{\mathrm{v}}(dx^{\mu}), \qquad d_{\mathrm{h}}\theta_{\mathbf{J}}^a = dx^{\mu} \wedge \theta_{\mathbf{J}\mu}^a, \qquad d_{\mathrm{v}}\theta_{\mathbf{J}}^a = 0. \tag{179}$$

**(Algebraic) Poincaré lemma:** In view of its importance for field theory, we spell out the Poincaré lemma for the horizontal complex which is also referred to as the 'algebraic Poincaré lemma' [229, 230] following R. Stora. In this respect, we first recall the ordinary Poincaré lemma (see Appendix A) which concerns the *de Rham complex* $\left(\Omega^{\bullet}(\mathbb{R}^n), d = dx^{\mu}\frac{\partial}{\partial x^{\mu}}\right)$ on the base manifold $M = \mathbb{R}^n$: for the $k$-th cohomology group $H^k(d, \Omega^{\bullet}(\mathbb{R}^n))$ (i.e. the quotient $Z^k/B^k$ of the vector space $Z^k$ of closed $k$-forms on $\mathbb{R}^n$ by the vector space $B^k$ of exact $k$-forms on $\mathbb{R}^n$), we have:

de Rham cohomology groups on $\mathbb{R}^n$: $\qquad H^k\left(d, \Omega^{\bullet}(\mathbb{R}^n)\right) = \delta_0^k \mathbb{R}, \qquad$ for $0 \le k \le n$. (180)

Thus, the cohomology groups are trivial for $k \ge 1$ (i.e. closed $k$-forms on $\mathbb{R}^n$ are exact) and the closed 0-forms (i.e. smooth functions $f : \mathbb{R}^n \to \mathbb{R}$ with $df = 0$) are real constants. The proof of this lemma requires to show that, for any closed $k$-form $\alpha$ on $\mathbb{R}^n$ (with $k \ge 1$), one can find a $(k-1)$-form $\beta$ on $\mathbb{R}^n$ such $\alpha = d\beta$. Such a $(k-1)$-form can be obtained from $\alpha$ by applying a so-called *contracting homotopy operator* $I$, i.e. $\beta = I(\alpha)$, see equation (A.19) of Appendix A.5.

The **algebraic Poincaré lemma** [229, 230] concerns the horizontal differential acting on forms defined on the bundle $E$ over $M = \mathbb{R}^n$ and states that

$$H^k\big(d_{\mathrm{h}}, \Omega^\bullet(E)\big) = \begin{cases} \mathbb{R}, & \text{for } k = 0, \\ 0, & \text{for } 0 < k < n, \\ \{[\alpha]\}, & \text{for } k = n. \end{cases} \tag{181}$$

Here $[\alpha]$ represents an equivalence class of top forms, i.e. $\alpha \in \Omega^n(E)$, the equivalence of forms being defined by

$$\alpha \sim \alpha + d_{\mathrm{h}} j, \qquad \text{with } j \in \Omega^{n-1}(E).$$

The proof proceeds again by the consideration of a contracting homotopy operator which generalizes the one introduced for the de Rham complex, see equation (273) below.

**Interpretation and relationship with traditional notation:** A comprehensive presentation of the Lagrangian and (in particular) the Hamiltonian formulation of classical dynamical systems (including gauge symmetries and thus constraints) for a finite or an infinite number of degrees of freedom, as well as of the quantization of these systems, has been given in reference [23]. Following loosely this classic work and using the language of jet bundles, a nice concise introduction to classical Lagrangian field theories (notably gauge field theories) and to their perturbative quantization has recently been presented by G. Barnich [74]. Since the corresponding notations are more familiar for physicists, we relate the ones that we introduced above to the latter ones.

The total derivative with respect to $x^\mu$ of a function

$$x \mapsto f(x, \varphi^a(x), \partial_\mu \varphi^a(x), \dots),$$

is given by $\partial_\mu = \frac{\partial}{\partial x^\mu} + \partial_\mu \varphi^a \frac{\partial}{\partial \varphi^a} + \cdots$ where $\frac{\partial f}{\partial x^\mu}$ reflects the explicit $x$-dependence of the function $f$. The horizontal differential writes as in Eqn. (177), i.e. $d_{\mathrm{h}} = dx^\mu \partial_\mu$.

An infinitesimal transformation of a section $x \mapsto s(x) = \big(x, \varphi(x)\big)$ is described by

$$\big(x^\mu, \varphi^a(x)\big) \rightsquigarrow \big(x'^\mu, \varphi'^a(x')\big) = \big(x^\mu + \epsilon\, \xi^\mu(x), \varphi^a(x) + \epsilon\, \psi^a(x)\big),$$

with $\epsilon \in \mathbb{R}$ infinitesimal. It corresponds to a vector field on the bundle $E$ given by

$$v \equiv a^\mu \frac{\partial}{\partial x^\mu} + b^a \frac{\partial}{\partial q^a},$$

where $a^\mu$ and $b^a$ correspond to the values of, respectively, $\xi^\mu$ and of $\psi^a$ at $x$. At first order in $\epsilon$, we have

$$\varphi'^a(x + \epsilon\, \xi) = \varphi'^a(x') = \varphi^a(x) + \epsilon\, \psi^a(x), \qquad \varphi'^a(x + \epsilon\, \xi) = \varphi'^a(x) + \epsilon\, \xi^\mu \partial_\mu \varphi^a(x),$$

henceforth the *variation of the field* $\varphi^a$ takes the form

$$\epsilon\, \delta_Q \varphi^a(x) \equiv \delta \varphi^a(x) \equiv \varphi'^a(x) - \varphi^a(x), \qquad \text{with} \quad \delta_Q \varphi^a = \psi^a - \xi^\mu \partial_\mu \varphi^a.$$

Thus, the variation $\delta_Q \varphi^a$ of the field $\varphi^a$ involves both an intrinsic term $\psi^a$ and a drag term involving the derivative of the field [74]. The value of $\delta_Q \varphi^a$ at the point $x$ (i.e. $b^a - a^\mu q^a_\mu$) is referred to [31, 74] as the *characteristic (representative) of the infinitesimal transformation of fields* (or *of the vector field $v$*). In terms of fibre bundle coordinates $(x^\mu, q^a)$ and of differentials, the last equation corresponds to the infinitesimal field variation as considered above (see equations (175), (177)):

$$d_{\mathrm{v}} q^a = dq^a - q^a_\mu dx^\mu \equiv \theta^a \qquad \text{(contact 1-form)}.$$

In summary, we have the following *relation between the infinitesimal field variations in traditional notation and in the variational bicomplex approach:*

$$\delta_Q \varphi^a(x) = \psi^a(x) - \xi^\mu(x)(\partial_\mu \varphi^a)(x) \qquad \longleftrightarrow \qquad d_v q^a = dq^a - q^a_\mu \, dx^\mu \equiv \theta^a \,.$$

## 6.2 Lagrangians and Euler-Lagrange equations

**Generalities:** Suppose the field equations are the Euler-Lagrange equations following from a first order Lagrangian $n$-form $\mathcal{L} \, d^n x \in \Omega^{n,0}$. In the setting described above, the Euler-Lagrange equations read $0 = \mathcal{E}(\mathcal{L} \, d^n x)$ where the *Euler-Lagrange operator* $\mathcal{E} \equiv \mathcal{I} \, d_v$ is defined in terms of the so-called *interior Euler operator* $\mathcal{I}$ which acts on the $(n, 1)$-form $\alpha \equiv d_v(\mathcal{L} \, d^n x)$ according to

$$\mathcal{I}(\alpha) = \theta^a \wedge \left[ i_{\partial/\partial q^a}\alpha - \partial_\mu \big( i_{\partial/\partial q^a_\mu}\alpha \big) + \partial_\mu \partial_\nu \big( i_{\partial/\partial q^a_{\mu\nu}}\alpha \big) \mp \cdots \right], \tag{182}$$

where the operator $i_X$ denotes the interior product with the vector field $X$ on $J^\infty E$. With $\mathcal{L}$ depending only on $x, q^a, q^a_\mu$, we thus obtain

$$\mathcal{E}(\mathcal{L} \, d^n x) = E_a(\mathcal{L}) \, \theta^a \wedge d^n x \,, \qquad \text{with} \qquad E_a(\mathcal{L}) \equiv \frac{\partial \mathcal{L}}{\partial q^a} - \partial_\mu \left( \frac{\partial \mathcal{L}}{\partial q^a_\mu} \right), \qquad \theta^a \equiv d_v q^a \,. \tag{183}$$

If the Lagrangian $\mathcal{L}$ is of higher order, then $E_a(\mathcal{L})$ involves higher order terms of obvious form.

By way of example, we consider $M = \mathbb{R}^n$ and $\mathcal{L}[q] = \frac{1}{2} \eta^{\mu\nu} q^a_\mu q^a_\nu - \frac{m^2}{2} q^a q^a$ corresponding to $N$ free scalar fields $\varphi^a$ of equal mass $m$. Then, we have the Euler-Lagrange equation $0 = E_a(\mathcal{L}) = -(q^{a\,\mu}_\mu + m^2 q^a)$, i.e. (upon pull-back to $M$) the free Klein-Gordon equation $0 = (\partial^\mu \partial_\mu + m^2)\varphi^a(x)$.

For a Lorentzian manifold $(M, (g_{\mu\nu}))$, explicit expressions of the Lagrangian $n$-form can be defined by using the Hodge star operator $\star$ which is recalled in Appendix A. For instance, for a free massless scalar field $\varphi$ or for the Maxwell potential 1-form $A = A_\mu dx^\mu$ (with field strength $F \equiv dA$), the pull-back of the Lagrangian $n$-form to $M$ is given by

$$(\mathcal{J}^\infty s)^*(\mathcal{L} \, d^n x) = \begin{cases} \frac{1}{2} d\varphi \wedge \star d\varphi = \frac{1}{2}(\partial^\mu \varphi)(\partial_\mu \varphi)\sqrt{|g|} \, d^n x \,, \\ \frac{1}{2} F \wedge \star F = -\frac{1}{4} F^{\mu\nu} F_{\mu\nu} \sqrt{|g|} \, d^n x \,. \end{cases} \tag{184}$$

The appearance of the factor $\sqrt{|g|}$ (with $g \equiv \det(g_{\mu\nu})$) in the volume element can be avoided by the choice of appropriate coordinates [120]. (The latter choice of coordinates is convenient if one wants to use the standard form of the De Donder-Weyl equations for the canonical formulation of field theory [120].)

Let us mention a quite useful property [73, 226] of the interior Euler operator $\mathcal{I}$ which will be repeatedly used in the sequel:

$$\mathcal{I}(d_h \alpha) = 0 \,, \qquad \text{for} \quad \alpha \in \Omega^{n-1,l} \quad (l \geq 1) \,. \tag{185}$$

**Ambiguities:** For a given first order Lagrangian $n$-form $\mathcal{L} \, d^n x \in \Omega^{n,0}$, an *equivalent first order Lagrangian $n$-form* $\mathcal{L}' \, d^n x \in \Omega^{n,0}$ is given by

$$\mathcal{L}' \, d^n x = \mathcal{L} \, d^n x + d_h \Lambda \,, \qquad \text{with} \quad \Lambda \in \Omega^{n-1,0} \,. \tag{186}$$

Here, $\Lambda = \Lambda^\mu[q]\,d^{n-1}x_\mu$ with $d^{n-1}x_\mu \equiv i_{\partial_\mu}d^n x$ and $\Lambda^\mu[q]$ is a function of $x$ and $q^a$ only. Indeed, we have

$$d_{\mathrm{h}}\Lambda = (d_{\mathrm{h}}\Lambda^\mu)\wedge d^{n-1}x_\mu = (\partial_\mu\Lambda^\mu)\,d^n x\,, \qquad \text{with} \quad \partial_\mu\Lambda^\mu = \frac{\partial\Lambda^\mu}{\partial x^\mu} + q^a_\mu\frac{\partial\Lambda^\mu}{\partial q^a}\,,$$

hence $d_{\mathrm{h}}\Lambda$ represents a divergence term depending at most on $x, q^a, q^a_\mu$; moreover, the Euler-Lagrange operator $\mathcal{E}$ annihilates $d_{\mathrm{h}}\Lambda$ since $\mathcal{E}(d_{\mathrm{h}}\Lambda) = \mathcal{I}d_{\mathrm{v}}(d_{\mathrm{h}}\Lambda) = -\mathcal{I}d_{\mathrm{h}}(d_{\mathrm{v}}\Lambda) = 0$ by virtue of relation (185) applied to $\alpha = d_{\mathrm{v}}\Lambda \in \Omega^{n-1,1}$.

## 6.3 Application to classical mechanics (and relation with multisymplectic approach)

For $n = 1$, the multisymplectic approach to field theory corresponds to the symplectic description of time-dependent mechanics (i.e. of non-autonomous systems) on the doubly extended phase space parametrized by $(t, q^a, p_a, E)$ and a Hamiltonian flow involving the extra variables $t$ and $E$, see Appendix C, equations (C.26)-(C.30). The latter approach is also referred to as the "autonomization trick" [73]. The extension of ordinary phase space can be avoided as follows [73] by considering the variational bicomplex.

For the sake of notational simplicity, we choose a system with one degree of freedom, the coordinate and momentum of the particle being denoted by $Q(t)$ and $P(t)$, respectively, and the Hamiltonian function by $H(Q, P, t)$. According to Table 1, we consider the trivial vector bundle $E \equiv M \times U \equiv \mathbb{R} \times \mathbb{R}^2$ over the time axis $\mathbb{R}$, the fibre $\mathbb{R}^2$ being parametrized by the phase space coordinates $(q^a)_{a=1,2} \equiv (Q(t), P(t))$. We now have $\partial_t Q \equiv (\frac{\partial}{\partial t} + \dot{Q}\frac{\partial}{\partial Q} + \dot{P}\frac{\partial}{\partial P} + \cdots)Q = \dot{Q}$ and similarly $\partial_t P = \dot{P}$. The contact 1-forms $\theta^a$ are given by $\theta^1 = dQ - \dot{Q}dt$ and $\theta^2 = dP - \dot{P}dt$. Let us presently define the *generalized symplectic form* $\underline{\omega} \in \Omega^{0,2}$ on $J^\infty E$ as the following $d_{\mathrm{v}}$-closed $(0,2)$-form:

$$\text{symplectic form on } J^\infty E: \qquad \underline{\omega} \equiv \sum_{a<b} d_{\mathrm{v}}q^a \wedge d_{\mathrm{v}}q^b = d_{\mathrm{v}}Q \wedge d_{\mathrm{v}}P\,. \tag{187}$$

By virtue of the definition of the vertical derivative $d_{\mathrm{v}}$, we have

$$\underline{\omega} = \left(\theta^a\frac{\partial Q}{\partial q^a}\right)\wedge\left(\theta^b\frac{\partial P}{\partial q^b}\right) = \left(dQ - \partial_t Q\,dt\right)\wedge\left(dP - \partial_t P\,dt\right)$$
$$= dQ\wedge dP + \left[(\partial_t Q)\,dP - (\partial_t P)\,dQ\right]\wedge dt\,.$$

For the solutions of the

$$\text{Hamiltonian equations of motion} \qquad \partial_t Q = \frac{\partial H}{\partial P}\,, \qquad \partial_t P = -\frac{\partial H}{\partial Q}\,, \tag{188}$$

we thus have

$$\boxed{\underline{\omega} \equiv d_{\mathrm{v}}Q \wedge d_{\mathrm{v}}P = dQ\wedge dP + dH\wedge dt\,,} \tag{189}$$

i.e. the expression of the symplectic 2-form on doubly extended phase space. Thus, the splitting of the exterior derivative on the variational bicomplex into horizontal and vertical parts allows us to give a symplectic description of time-dependent mechanical systems which avoids the extension of phase space. Moreover [73], in this approach the *Hamiltonian flow* takes the simple coordinate free expression

$$\boxed{d_{\mathrm{h}}\underline{\omega} = 0\,,} \qquad \text{on solutions (of the Hamiltonian equations)}\,. \tag{190}$$

This can be seen as follows. By substituting the explicit expression (187) of $\underline{\omega}$ into (190) and using the fact that $d_h$ and $d_v$ anticommute, we obtain

$$0 = d_h \underline{\omega} = (d_h d_v Q) \wedge d_v P - d_v Q \wedge (d_h d_v P) = -d_v \big[ \underbrace{d_h Q \wedge d_v P + d_v Q \wedge d_h P}_{\in \Omega^{1,1}} \big] .$$

According to the Poincaré lemma (applied to $d_v$-closed forms), there exists a $(1,0)$-form $S \equiv H\,dt$ such that we have (on solutions of the Hamiltonian equations)

$$-d_h Q \wedge d_v P - d_v Q \wedge d_h P = d_v S = d_v H \wedge dt .$$

From $d_h Q = (\partial_t Q)\,dt$ and $d_h P = (\partial_t P)\,dt$, we infer that

$$\big[ (\partial_t Q)\,d_v P - (\partial_t P)\,d_v Q \big] \wedge dt = d_v H \wedge dt ,$$

i.e. $H$ only depends on $Q, P, t$ and we have the Hamiltonian equations of motion (188). Conversely, by starting from the set of equations (188) and reversing the previous line of arguments one deduces the validity of (190).

## 6.4 Field theory: Relation with covariant phase space

In Subsection 6.5 we will generalize the previous line of arguments from classical mechanics to field theory. Presently, we relate the covariant phase space formulation of field theory to the variational bicomplex approach.

**General expressions:** The first variational formula (82) for Lagrangian field theory admits the following global formulation in terms of the variational bicomplex [71, 226]. Suppose the fields are sections of a fibre bundle $E$ over the $n$-dimensional space-time manifold $M$ and suppose we are given a Lagrangian $n$-form $\mathcal{L}\,d^n x \in \Omega^{n,0}$. Then, one can prove (using techniques of global analysis) that there exists a form $j \in \Omega^{n-1,1}$ such that we have the

$$\text{global first variational formula :} \qquad \boxed{ d_v(\mathcal{L}\,d^n x) = \mathcal{E}(\mathcal{L}\,d^n x) + d_h j . } \qquad (191)$$

(As a matter of fact, the field variation $\delta\varphi^a$ appearing in (82) can be included into the last relation by contracting the relation with a vertical vector field $X = \delta\varphi^a \frac{\partial}{\partial q^a}$ on $E$ prolonged to $J^\infty E$.) The fact that $j \in \Omega^{n-1,1}$ means that $j$ represents a $(n-1)$-form with respect to the horizontal degree (i.e. the dual of a 1-form $j_\mu\,dx^\mu$ for $n$-dimensional space-time) and a 1-form with respect to the vertical degree (which reflects the linear dependence on the monomials $\delta\varphi^a$ in the local expressions (82),(83)).

By construction, the so-called [59]

$$\text{universal current (for } \mathcal{L}) \qquad \boxed{ J \equiv -d_v j , } \qquad (192)$$

is a $d_v$-closed $(n-1, 2)$-form. The authors of [71] refer to $j$ as the **variational 1-form** and to $J$ as the **local symplectic form** which appears to be a quite pertinent terminology.

In order to compare with the results obtained within the covariant phase space approach, we consider a first order Lagrangian $n$-form $\mathcal{L}\,d^n x$. By explicitly evaluating $d_v(\mathcal{L}\,d^n x)$ and comparing with the general relation (191) and with (183), we obtain the following *explicit expressions [1, 72] for the variational 1-form $j$ and for the local symplectic form $J$:*

$$\boxed{ j = -\frac{\partial \mathcal{L}}{\partial q^a_\mu}\,\theta^a \wedge d^{n-1} x_\mu , } \qquad \boxed{ J = -\theta^a \wedge \Big[ \frac{\partial^2 \mathcal{L}}{\partial q^a_\mu \partial q^b}\,\theta^b + \frac{\partial^2 \mathcal{L}}{\partial q^a_\mu \partial q^b_\nu}\,\theta^b_\nu \Big] \wedge d^{n-1} x_\mu . }$$

$$(193)$$

These expressions correspond precisely to those obtained in Section 5.1 for the symplectic potential and the (pre-)symplectic current density, see equations (83) and (84), respectively. (We note that the global signs depend on the conventions, e.g. our definition of $d^{n-1}x_\mu$.)

**Ambiguities:** The equations of motion are not modified if one adds to the Lagrangian $n$-form a term $d_h\Lambda$ with $\Lambda \equiv \Lambda^\mu[q]\,d^{n-1}x_\mu \in \Omega^{n-1,0}$ (where $\Lambda^\mu[q]$ is a function of $x$ and $q^a$ only), see Eqn. (186). The induced changes of the forms $j$ and $J$ are easily determined:

$$\boxed{\mathcal{L}'\,d^nx = \mathcal{L}\,d^nx + d_h\Lambda, \qquad j' = j - d_v\Lambda, \qquad J' = J,} \qquad \text{with} \quad \Lambda \in \Omega^{n-1,0}. \qquad (194)$$

These transformations correspond to the expressions discussed in covariant phase space, see equation (94).

If one considers *first order* Lagrangian $n$-forms $\mathcal{L}\,d^nx$ and makes the additional *assumption* that the variational 1-form $j$ is "linear over functions" $f \in C^\infty(M)$, then this 1-form is uniquely defined [71] and locally given by expression (193). Yet, in general the variational 1-form $j$ as defined by relation (191) is only determined up to a $d_h$-exact term [63, 64, 71]. The modification $j \rightsquigarrow j' = j + d_h\vartheta$ (with $\vartheta \in \Omega^{n-2,1}$) and the relation $d_v(d_h\vartheta) = -d_h(d_v\vartheta)$ lead to $J' = -d_v j' = J + d_h\beta$ with $\beta \equiv d_v\vartheta$. Such a change of $j$ and $J$ is notably induced by adding a trivial term $d_h\Lambda$ to the Lagrangian $n$-form. Indeed, if we write the variational 1-form $j$ associated (by virtue of (191)) to the Lagrangian $n$-form $\mathcal{L}\,d^nx$ as $j_{\mathcal{L}\,d^nx}$, then relation (191) applied to $\mathcal{L}\,d^nx = d_h\Lambda$ reads

$$d_v(d_h\Lambda) = \mathcal{E}(d_h\Lambda) + d_h j_{d_h\Lambda}.$$

From $d_v(d_h\Lambda) = -d_h(d_v\Lambda)$ and $\mathcal{E}(d_h\Lambda) = 0$, it then follows that $d_h(j_{d_h\Lambda} + d_v\Lambda) = 0$. By virtue of Poincaré's lemma for the variational bicomplex we thus have the following result [180]:

$$\boxed{j_{d_h\Lambda} = -d_v\Lambda + d_h\vartheta_\Lambda,} \qquad \text{for some} \quad \vartheta_\Lambda \in \Omega^{n-2,1}. \qquad (195)$$

As a matter of fact, we already encountered these contributions to the form $j$ (or rather to its pullback on a $(n-1)$-dimensional hypersurface $\Sigma \subset M$) in the case of general relativity in Eqn. (138). The term $d_h\vartheta_\Lambda$ (representing a superpotential term, see equations (240)-(241) below) is known as *corner term* and an explicit expression for $\vartheta_\Lambda$ will be given in Eqn. (277) below. For the corresponding local symplectic form, i.e. $J_{d_h\Lambda} \equiv -d_v j_{d_h\Lambda}$, we thereby obtain the expression

$$J_{d_h\Lambda} = d_h\beta_\Lambda, \qquad \text{with} \quad \beta_\Lambda \equiv d_v\vartheta_\Lambda \in \Omega^{n-2,2}. \qquad (196)$$

As we will see below (cf. Eqn. (201)), the (pre-)symplectic 2-form $\Omega_s$ on the space of solutions $s$ of the Euler-Lagrange equations is defined as an integral of the local symplectic 2-form $J \in \Omega^{n-1,2}$ over a $(n-1)$-dimensional submanifold $\Sigma \subset M$. This integral does not depend on the choice of the variational 1-form $j$ if $\Sigma$ is boundaryless. Yet, for hypersurfaces $\Sigma$ with a non-trivial boundary $\partial\Sigma$, the modification (196) of $J$ generally changes the 2-form $\Omega_s$ (unless the fields satisfy appropriate boundary conditions on $\partial\Sigma$), see Subsection 7.7 and Subsection 7.8 which address corner terms.

If $\Sigma$ is an unbounded hypersurface, then the integral $\oint_{\partial\Sigma}\dots$ can be interpreted [64,65] as a limit of $\oint_{\partial K}\dots$ where the compact region $K \subset \Sigma$ approaches all of $\Sigma$ in a suitable manner:

$$\oint_{\partial\Sigma}\cdots \equiv \lim_{K\to\Sigma}\oint_{\partial K}\cdots \qquad (197)$$

This definition makes sense if the limit exists and is independent of the choice of the compact region $K$ and of the manner that it approaches $\Sigma$.

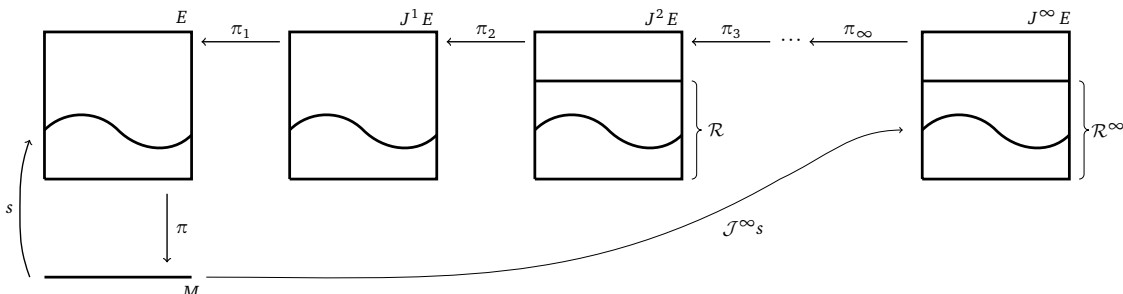

Figure 7: Subbundle $\mathcal{R}^\infty$ of $J^\infty E$ associated to field equations.

**Going on-shell:** Now suppose the Euler-Lagrange equation $\mathcal{E}(\mathcal{L}\,d^n x) = 0$ associated to the Lagrangian $\mathcal{L}$ represents a system of PDE's satisfying the hypotheses of the Cauchy-Kowalewski theorem, e.g. of order 2. In this case, the solutions of the equations of motion are sections of a subbundle $\mathcal{R}$ of the 2-jet bundle $J^2 E$. Moreover, one can then restrict the infinite jet bundle $J^\infty E$ to the infinite prolongation $\mathcal{R}^\infty$ of $\mathcal{R}$ obtained by differentiating the system of equations defining $\mathcal{R}$. Thus, one can consider the **variational bicomplex $(\Omega^{\bullet,\bullet}(\mathcal{R}^\infty), d_{\mathrm{h}}, d_{\mathrm{v}})$ of the differential equations** $\mathcal{R}$, see Figure 7.

In particular, we can now pull back the first variational formula (191) from $J^\infty E$ to $\mathcal{R}^\infty$ (i.e. impose the equations of motion) which yields

$$\boxed{d_{\mathrm{v}}(\mathcal{L}\,d^n x) = d_{\mathrm{h}}j\,,} \qquad \text{on } \mathcal{R}^\infty\,, \tag{198}$$

with $j \in \Omega^{n-1,1}(\mathcal{R}^\infty)$.

On $\mathcal{R}^\infty$, the local symplectic form (192) is $d_{\mathrm{h}}$-closed on the solutions of the field equations by virtue of $d_{\mathrm{h}}(d_{\mathrm{v}}j) = -d_{\mathrm{v}}(d_{\mathrm{h}}j)$ and of relation (198):

$$\text{Structural conservation law :} \qquad \boxed{d_{\mathrm{h}}J = 0\,,} \qquad \text{on } \mathcal{R}^\infty\,. \tag{199}$$

As we already noted, the form $j \in \Omega^{n-1,1}(\mathcal{R}^\infty)$ of the variational bicomplex approach corresponds to the *symplectic potential current density* $(j^\mu)$ given by Eqn. (83) while $J$ corresponds to the *(pre-)symplectic current density* $(J^\mu)$ which is given by (84) and satisfies the structural conservation law $\partial_\mu J^\mu = 0$ (on-shell), see Eqn. (89). This correspondence is also consistent with the interpretation of $d_{\mathrm{v}}$ as the infinitesimal field variation. Quite generally, the forms $\alpha \in \Omega^{n-1,q}$ with $q \geq 0$ satisfying $d_{\mathrm{h}}\alpha = 0$ on solutions of the field equations are referred to as *higher degree* or *form-valued conserved currents* [72, 226], the case $q = 0$ corresponding to the usual instance of conserved currents.

The tangent bundle of $\mathcal{R}^\infty \subset J^\infty E$ can be identified with the set of vertical vector fields

$$X = X^a \frac{\partial}{\partial q^a} + \partial_\mu X^a \frac{\partial}{\partial q_\mu^a} + \partial_\mu \partial_\nu X^a \frac{\partial}{\partial q_{\mu\nu}^a} + \cdots\,, \tag{200}$$

where $(\mathcal{J}^\infty s)^* X^a$ satisfies the Jacobi equations (i.e. the linearized Euler-Lagrange equations), see equations (37)-(38).

Following the work of G. Zuckerman [59], we now consider a $(n-1)$-dimensional submanifold $\Sigma \subset M$ which is compact and oriented. The *canonical 1-form* $\Theta$ on $\mathcal{R}^\infty \subset J^\infty E$ and the associated *(pre-)symplectic 2-form* $\Omega$ on $\mathcal{R}^\infty$ are defined in analogy to (155) and (158) in terms of the form $j \in \Omega^{n-1,1}(\mathcal{R}^\infty)$ and of the form $J \equiv -d_{\mathrm{v}}j \in \Omega^{n-1,2}(\mathcal{R}^\infty)$, respectively: in terms of vertical vector fields $X, Y$ on $\mathcal{R}^\infty$, one introduces (at any solution $s$ of the Euler-Lagrange

equations) the

$$\text{Canonical 1-form on } \mathcal{R}^\infty : \qquad \boxed{\Theta_s(X) \equiv \int_\Sigma (\mathcal{J}^\infty s)^*(i_X j),}$$

$$\text{(Pre-) symplectic 2-form on } \mathcal{R}^\infty : \qquad \boxed{\Omega_s(X,Y) \equiv \int_\Sigma (\mathcal{J}^\infty s)^*(i_Y i_X J).} \tag{201}$$

One then has [59] the results

$$\boxed{\Omega = -d\Theta,} \qquad \boxed{d\Omega = 0,} \tag{202}$$

and the 2-form $\Omega$ does not depend on the manifold $\Sigma$ which is chosen for its definition.[15]

The fact that the previous results hold on $\mathcal{R}^\infty$ means that they hold *on-shell*, i.e. the field equations are taken into account [59, 72]. These results correspond to those found in the seventies [3, 4] in the framework of multisymplectic geometry (see Eqn. (158) and the next subsection) and those found by E. Witten [58] (and by C. Crnkovic [152]) in their study of different classes of models, see Section 6.6. The subtle question whether $\Omega$ actually represents a symplectic form, i.e. is non-degenerate, will be addressed in Section 6.6 following a discussion of local symmetries of the field equations.

To conclude, we note that the Poincaré lemma still applies to the differential $d_v$ of the bicomplex $\left(\Omega^{\bullet,\bullet}(\mathcal{R}^\infty), d_h, d_v\right)$ associated to the field equations, i.e. to the columns of this bicomplex. However, it does not apply to the differential $d_h$ (by contrast to the case of the free variational bicomplex). The non-trivial cohomology groups of the differential $d_h$ acting on $\Omega^{\bullet,\bullet}(\mathcal{R}^\infty)$ are referred to as *characteristic cohomology groups:* they can be related to the so-called Koszul-Tate differential (see Appendix B.3) and we will see in Section 7 that they provide important information on the solutions of the field equations. In particular, they describe non-trivial conservation laws in field theory, see equations (245)-(246) and (270) below.

## 6.5 Field theory: Relation with multisymplectic approach

The field theoretical generalization of the considerations in Section 6.3 on mechanics as formulated on the variational bicomplex is tantamount to considering the base manifold $M = \mathbb{R}^n$ and a multisymplectic dynamical system on $J^\infty E$ with $E = \mathbb{R}^n \times \mathbb{R}^N$. More precisely, following references [73, 160], we consider a *Lagrangian multisymplectic system of PDE's,* i.e. we assume that the system of PDE's represents the Euler-Lagrangian equation $\mathcal{E}(\mathcal{L} d^n x) = 0$ following from a

$$\text{Lagrangian } n\text{-form} \qquad \boxed{\mathcal{L} d^n x = p_a^\mu(x, q_J) d_h q^a \wedge d^{n-1} x_\mu - \mathcal{H} d^n x,} \tag{203}$$

on $J^\infty E$. (Note that this expression corresponds to the Poincaré-Cartan $n$-form on ordinary multiphase space, see Eqn. (12). It is tantamount to the basic relationship $\mathcal{L} = p_a^\mu q_\mu^a - \mathcal{H}$, see Eqn. (3).) In this setting, it is natural to introduce a symplectic 2-form $d_v q^a \wedge d_v p_a^\mu$ for each space-time coordinate $x^\mu$ (in analogy to the symplectic 2-form (187) for mechanical systems). One then gathers all of these forms in the so-called *multisymplectic $(n+1)$-form* on $J^\infty E$, i.e. the following $d_v$-closed $(n-1, 2)$-form:

$$\text{multisymplectic form on } J^\infty E : \qquad \boxed{\underline{\omega} \equiv d_v q^a \wedge d_v p_a^\mu \wedge d^{n-1} x_\mu.} \tag{204}$$

---

[15]For a *non*compact hypersurface $\Sigma$, the convergence of the integral (201) is ensured by evaluating $\Omega_s$ only on vector fields $X, Y$ with compact support in spatial directions or with sufficient decay properties at spatial infinity [71].

The generalization of the calculation (187)-(189) to field theory, which now relies on the De Donder-Weyl equations $q_\mu^a = \frac{\partial \mathcal{H}}{\partial p_a^\mu}$, $\partial_\mu p_a^\mu = -\frac{\partial \mathcal{H}}{\partial q^a}$, yields

$$\underline{\omega} \equiv d_{\mathrm{v}} q^a \wedge d_{\mathrm{v}} p_a^\mu \wedge d^{n-1} x_\mu = dq^a \wedge dp_a^\mu \wedge d^{n-1} x_\mu + d\mathcal{H} \wedge d^n x = \omega_{\mathcal{H}},$$

i.e. we have the "dynamical" multisymplectic $(n+1)$-form (13) on multiphase space, but the present approach does not rely on an extension of phase space.

As for mechanics (see Eqn. (190)), we have the following general result for the multisymplectic form given by (204): *the relation*

$$\boxed{d_{\mathrm{h}} \underline{\omega} = 0,} \qquad \text{on solutions of the field equations}, \tag{205}$$

*is tantamount to the existence of a Lagrangian of the form* (203) *which yields these field equations.* Indeed, by virtue of the Poincaré lemma, the relation $d_{\mathrm{v}} \underline{\omega} = 0$ for the form $\underline{\omega} \in \Omega^{n-1,2}$ given by (204) implies that there (locally) exists a form $\underline{\theta} \in \Omega^{n-1,1}$ such that

$$\underline{\omega} = -d_{\mathrm{v}} \underline{\theta}. \tag{206}$$

The fact that the differentials $d_{\mathrm{v}}$ and $d_{\mathrm{h}}$ anticommute now implies that

$$d_{\mathrm{v}} d_{\mathrm{h}} \underline{\theta} = -d_{\mathrm{h}} d_{\mathrm{v}} \underline{\theta} = d_{\mathrm{h}} \underline{\omega} = 0, \qquad \text{on solutions.}$$

Since $d_{\mathrm{h}} \underline{\theta}$ is $d_{\mathrm{v}}$-closed (on solutions), we again apply the Poincaré lemma to infer the (local) existence of a $n$-form $\mathcal{L} d^n x \in \Omega^{n,0}$ such that

$$d_{\mathrm{h}} \underline{\theta} = d_{\mathrm{v}}(\mathcal{L} d^n x), \qquad \text{on solutions.} \tag{207}$$

The $n$-form $\mathcal{L} d^n x$ actually represents a Lagrangian for the system of PDE's under consideration since application of the interior Euler operator $\mathcal{I}$ defined by Eqn. (182) and enjoying the property (185), yields

$$\mathcal{E}(\mathcal{L} d^n x) \equiv \mathcal{I} d_{\mathrm{v}}(\mathcal{L} d^n x) = \mathcal{I}(d_{\mathrm{h}} \underline{\theta}) = 0, \qquad \text{on solutions,}$$

i.e. the Euler-Lagrange equation. According to (204) and (206), we presently have the following explicit expression for the form $\underline{\theta} \in \Omega^{n-1,1}$:

$$\underline{\theta} = p_a^\mu(x, q_{\mathbf{J}}) d_{\mathrm{v}} q^a \wedge d^{n-1} x_\mu. \tag{208}$$

This expression (involving the vertical differential) corresponds to the kinematical multicanonical $n$-form (10). As for mechanics, *the consideration of the variational bicomplex avoids the extension of multiphase space by an energy-type variable and it allows for the simple coordinate free expression* (205) *for the Hamiltonian flow.*

We note that relation (207) coincides with equation (198) and that (206) coincides with (192), i.e.

$$\text{On } \mathcal{R}^\infty: \qquad j \longleftrightarrow \underline{\theta}, \qquad J \longleftrightarrow \underline{\omega}.$$

In fact, (208) and (204) represent explicit expressions for the forms $j$ and $J$ encountered in the previous subsection: by virtue of $p_a^\mu = \frac{\partial \mathcal{L}}{\partial q_\mu^a}$, these expressions precisely correspond to those encountered in Eqn. (193) (for our choice of a first order Lagrangian).

We refer to the work [73] for a discussion of conservation laws (the so-called multimomentum map) in this geometric set-up of field theory. It turns out that this approach to PDE's not only provides new mathematical insights, but also allows for the derivation of novel predictions concerning various dynamical systems of physical interest, e.g. in fluid mechanics, see references [73, 160].

## 6.6 Symmetries and gauge field theories

We only discuss variational symmetries (i.e. symmetries of the action) as well as local (gauge) symmetries and we refer to the literature [31, 74] for a comprehensive treatment of symmetries and their consequences. We note that in the literature the notation of Lagrangians or Lagrangian densities is far from uniform, e.g. see references [17, 22, 154] for different choices. We assume that *on a space-time manifold* $(M, (g_{\mu\nu}))$, *the Lagrangian $\mathcal{L}$ has the expression* $\sqrt{|g|}\,\mathcal{L}$ *which is often denoted by L in the literature,* e.g. [154, 158]. For the sake of clarity, we recall (cf. text after Eqn. (176)) that the notation $f[q]$ means that $f$ is a smooth real-valued function of $x, q, q_\mu, \ldots, q_{\mu_1 \cdots \mu_K}$ for some $K \in \mathbb{N}$.

**(Global) variational symmetries and Noether's first theorem:** Suppose we are given a (first order) Lagrangian $n$-form $\mathcal{L}\,d^n x \in \Omega^{n,0}$. By definition [72], a *(global) variational symmetry* of the Lagrangian $\mathcal{L}\,d^n x$ is a so-called *evolutionary vector field* $X^a \frac{\partial}{\partial q^a}$ with characteristic $X^a[q]$ and prolongation on $J^\infty E$ given by

$$X = X^a \frac{\partial}{\partial q^a} + \partial_\mu X^a \frac{\partial}{\partial q^a_\mu} + \partial_\mu \partial_\nu X^a \frac{\partial}{\partial q^a_{\mu\nu}} + \cdots, \tag{209}$$

for which there exists a form $K \in \Omega^{n-1,0}$ (depending in general on $X$, i.e. on the local functions $X^a$) such that

$$\boxed{i_X d_{\mathrm{v}}(\mathcal{L}\,d^n x) = d_{\mathrm{h}} K.} \tag{210}$$

We note that the variation of the field variables under $X$ is given by

$$\delta_X q^a = X(q^a) = X^a, \tag{211}$$

and that the variation $\delta_X$ commutes with the total derivative $\partial_\mu$:

$$[\partial_\mu, \delta_X] = 0. \tag{212}$$

From $K \equiv k^\mu[q]\,d^{n-1}x_\mu$, it follows that $d_{\mathrm{h}}K = (\partial_\mu k^\mu)\,d^n x$. Evaluation of the left-hand side of (210) yields $i_X d_{\mathrm{v}}(\mathcal{L}\,d^n x) = (\delta_X \mathcal{L})\,d^n x$, hence relation (210) is equivalent to the

quasi-invariance of $\mathcal{L}$: $\quad \boxed{\delta_X \mathcal{L} = \partial_\mu k^\mu,} \quad$ with $\quad \delta_X \mathcal{L} = X^a \frac{\partial \mathcal{L}}{\partial q^a} + \partial_\mu X^a \frac{\partial \mathcal{L}}{\partial q^a_\mu}. \tag{213}$

We can rewrite $\delta_X \mathcal{L}$ by using the Euler-Lagrange derivative $E_a(\mathcal{L})$ given by (183) as well as the Leibniz rule:

$$\delta_X \mathcal{L} = X^a \frac{\partial \mathcal{L}}{\partial q^a} + \partial_\mu X^a \frac{\partial \mathcal{L}}{\partial q^a_\mu} = X^a E_a(\mathcal{L}) + \partial_\mu \left( X^a \frac{\partial \mathcal{L}}{\partial q^a_\mu} \right).$$

Substitution of this expression into (213) leads to the relation

$$\boxed{0 = E_a(\mathcal{L}) X^a + \partial_\mu \mathcal{J}^\mu,} \qquad \text{with} \qquad \boxed{\mathcal{J}^\mu \equiv \frac{\partial \mathcal{L}}{\partial q^a_\mu} X^a - k^\mu.} \tag{214}$$

This result is nothing but *Noether's first theorem* which states that a symmetry of the action $S \equiv \int_M \mathcal{L}\,d^n x$ implies a local conservation law $\partial_\mu \mathcal{J}^\mu = 0$ for the solutions of the equations of motion $E_a(\mathcal{L}) = 0$.

**Example 1 (Geometric transformations):** By way of example, we consider $M = \mathbb{R}^n$ and *space-time translations* parametrized by constant vectors $(\varepsilon^\mu) \in \mathbb{R}^n$, i.e. a rigid geometric transformation. Then $\delta_X q^a = X^a = \varepsilon^\mu q_\mu^a$ and $\partial_\nu X^a = \varepsilon^\mu q_{\mu\nu}^a$, which yields

$$\delta_X \mathcal{L} = \varepsilon^\mu \left( q_\mu^a \frac{\partial \mathcal{L}}{\partial q^a} + q_{\mu\nu}^a \frac{\partial \mathcal{L}}{\partial q_\nu^a} \right) = \varepsilon^\mu \left( \partial_\mu \mathcal{L} - \frac{\partial \mathcal{L}}{\partial x^\mu} \right).$$

If $\mathcal{L}$ does not explicitly depend on $x$ (i.e. $\frac{\partial \mathcal{L}}{\partial x^\mu} = 0$ for all $\mu$), then $\delta_X \mathcal{L} = \varepsilon^\mu \partial_\mu \mathcal{L} = \partial_\mu(\varepsilon^\mu \mathcal{L})$, i.e. $\delta_X \mathcal{L} = \partial_\mu k^\mu$ with $k^\mu = \varepsilon^\mu \mathcal{L}$. By virtue of Noether's first theorem (214) we now have the following local conservation law:

$$\text{On-shell:} \quad \partial_\mu T^{\mu\nu} = 0, \qquad \text{with} \qquad \boxed{T^\mu_{\ \nu} \equiv \frac{\partial \mathcal{L}}{\partial q_\mu^a} q_\nu^a - \delta_\nu^\mu \mathcal{L}.} \tag{215}$$

Here, $T^{\mu\nu}$ is the familiar expression for the *canonical energy-momentum tensor* as written on the jet bundle. If $\mathcal{L}$ explicitly depends on $x$, then the relations above imply the

$$\text{balance equation:} \qquad \partial_\mu T^{\mu\nu} = -\frac{\partial \mathcal{L}}{\partial x_\nu}. \tag{216}$$

We note that the conserved current density $\mathcal{J}^\mu$ given by (214) and thereby the canonical energy-momentum tensor (215) involve the canonical momentum vector field $\partial \mathcal{L}/\partial q_\mu^a$ appearing in the De Donder-Weyl formulation of dynamics, cf. equations (2)-(3).

We also remark that there exist other derivations of Noether's first theorem. One of them is based on a variation of the region $U$ of space-time in which the fields are defined, e.g. see [231] and in particular [232] for a pedagogical account. If the infinitesimal variations are considered to be generic rather than symmetry variations of an invariant action functional, then this approach to Noether's first theorem amounts to an application of the *Weiss variational principle* [233] (see reference [234]): this principle, which allows for variations of end-points in classical mechanics and for variations of the space-time domain in field theory, provides an alternative derivation of field equations. It has recently been applied to vary and interpret the gravitational action including the GHY term [234].

**Example 2 (Internal transformations):** For later reference, we consider a simple example for a rigid internal symmetry, i.e. a symmetry transformation which acts in the space of fields, but not on space-time coordinates. In *Gell-Mann's model of free quarks* in $\mathbb{R}^4$, one has the triplet $\varphi \equiv (\varphi^a)_{a=1,2,3}$ of quarks $u, d, c$. Each of the quark fields $\varphi^a$ represents a Dirac spinor field for which the adjoint is denoted by $\overline{\varphi^a} \equiv (\varphi^a)^\dagger \gamma^0$. In this respect, we recall that the Dirac matrices $\gamma^\mu$ are $4 \times 4$ matrices satisfying the Clifford algebra relation associated to the Minkowski metric, i.e. $\{\gamma_\mu, \gamma_\nu\} = 2\eta_{\mu\nu} \mathbb{1}$ where $\{\cdot, \cdot\}$ denotes the anticommutator of matrices. The Lagrangian density describing the dynamics of *free massless quarks* reads[16]

$$\mathcal{L} \equiv \mathrm{i}\, \bar\varphi \gamma^\mu \partial_\mu \varphi. \tag{217}$$

This Lagrangian is invariant under the global gauge transformations $\varphi \rightsquigarrow \varphi' = U\varphi$ with $U \in SU(3)_{\text{flavor}}$. In terms of a basis $\{T_r\}_{r=1,\dots,8}$ of the Lie algebra $su(3)$ with $T_r^\dagger = T_r$ and

$$[T_r, T_s] = \mathrm{i} f_{rst} T_t, \qquad \text{for } r, s \in \{1, \dots, 8\},$$

---

[16]In this example, we use the standard physics notation for fields rather than the notation $q_\mu^a \equiv (\partial_\mu \varphi^a)(x)$ of the variational bicomplex approach. We do not spell out the tensor product symbols, e.g. in the product $\bar\varphi \otimes \gamma^\mu$.

we have $U = \mathrm{e}^{-\mathrm{i} f^r T_r}$ with real parameters $f^r$. Thus, the infinitesimal symmetry transformations of fields write

$$\delta_{X_r} \varphi^a = X_r(\varphi^a) = -\mathrm{i}(T_r \varphi)^a = -\mathrm{i}(T_r)^{ab} \varphi^b = X_r^a, \qquad \text{for } r \in \{1, \dots, 8\}, \ a \in \{1, 2, 3\}, \tag{218}$$

and the corresponding Noether currents (214) have the form

$$\boxed{\mathcal{J}_r^\mu = \bar{\varphi} \gamma^\mu T_r \varphi,} \qquad \text{for } r \in \{1, \dots, 8\}. \tag{219}$$

**Local (gauge) symmetries:** A *local symmetry* of the Lagrangian $\mathcal{L} \, d^n x$ is a variational symmetry $X_f$ depending on local functions $f^r[q]$, i.e. [31, 74]

$$\delta_f \mathcal{L} = \partial_\mu k^\mu, \qquad \text{and} \qquad \delta_f q^a = Q^a(f) \equiv Q_r^a(f^r) \equiv Q_r^a f^r + Q_r^{a\mu} \partial_\mu f^r + \cdots \tag{220}$$

The latter relation means that the variation $\delta_f q^a$ depends linearly on the functions $f^r$ and their derivatives. Following P. Bergmann [235], the coefficients $Q_r^a, Q_r^{a\mu}, \dots$ are also referred to as *descriptors* (of the symmetry transformation). E.g. for the gauge potentials $A_\mu$ of electromagnetism, we have $\delta_f A_\mu = \partial_\mu f$ and for the YM potentials $A_\mu^r$ of non-Abelian gauge field theory, we have $\delta_f A_\mu^r = \partial_\mu f^r + \mathrm{i} f^r{}_{st} A_\mu^s f^t$ where $f \equiv f^r T_r$ represents a Lie algebra-valued function. In general relativity, the action of the diffeomorphisms generated by a vector $\xi \equiv \xi^\mu \partial_\mu$ on the metric tensor field $(g_{\mu\nu})$ is given by $\delta_\xi g_{\mu\nu} = (L_\xi g)_{\mu\nu} = \xi^\rho \partial_\rho g_{\mu\nu} + g_{\mu\rho} \partial_\nu \xi^\rho + g_{\rho\nu} \partial_\mu \xi^\rho$. Thus, for many field theories of physical interest, one has at most first order derivatives of the parameters $f^r$ in expansion (220). (As a matter of fact, the expression (220) of $Q_r^a$ is equivalent to the one that we considered in our discussion of the Peierls bracket, see equations (49)-(50).)

**Symplectic 2-form in gauge field theories:** From the discussion after Eqn. (109) we recall that a presymplectic 2-form $\Omega$ is (weakly) degenerate if it admits a non-trivial kernel, i.e. if there are non-trivial vector fields $V$ for which $i_V \Omega = 0$. For covariant phase space $Z$, we noted that such vector fields do exist in gauge field theories where they correspond to the gauge orbits in $Z$. Furthermore, we saw that by factoring out the gauge group $\mathcal{G}$ from $Z$ following equations (105)-(106), i.e.

$$Z_{\text{phys}} \equiv Z/\mathcal{G}, \qquad \text{and} \qquad \Omega = \pi^* \Omega_{\text{phys}}, \quad \text{where} \quad \pi : Z \longrightarrow Z_{\text{phys}},$$

one can get rid of the kernel of $\Omega$ and thereby obtain a *symplectic form* $\Omega_{\text{phys}}$ on the quotient space $Z_{\text{phys}}$. This procedure works for all field theoretic models which have been investigated [59, 60, 152, 236].

Quite generally, for any Lagrangian field theory formulated in terms of the variational bicomplex, one expects [72] that the presymplectic 2-form $\Omega$ is only degenerate if the Lagrangian $n$-form admits some non-trivial local symmetries and that the latter span the kernel of $\Omega$. (Then, a non-degenerate 2-form $\Omega_{\text{phys}}$ can be obtained by factoring out the group of local symmetries from the bundle $\mathcal{R}^\infty$, i.e.

$$\mathcal{R}_{\text{phys}} \equiv \mathcal{R}^\infty/\mathcal{G}, \qquad \text{and} \qquad \Omega = \pi^* \Omega_{\text{phys}}, \quad \text{where} \quad \pi : \mathcal{R}^\infty \longrightarrow \mathcal{R}_{\text{phys}},$$

$\pi$ denoting the projection map.) However, a general proof of this statement and a proper mathematical characterization of the quotient space $\mathcal{R}^\infty/\mathcal{G}$ appear to be missing – see however [237] for some recent work concerning these issues.

## 6.7 On space-times with boundary

The present subsection is based on references [66,238] (see also [122,182] for related earlier work).

A mathematical tool to deal with differential forms on a smooth manifold $M$ along with the differential forms on its boundary $\partial M$ has been introduced by R. Bott and L. Tu and dubbed the *relative de Rham complex* [239]. Recently an extension of this framework to the variational bicomplex (which is referred to as the *relative variational bicomplex*) has been put forward by the authors of reference [66]. (We note that other applications of the relative de Rham complex have been considered in the context of covariant phase space [149].) This generalization is devised in such a way that it also allows us to take into account the *boundary terms* which are added to action functionals (e.g. to the Einstein-Hilbert action for the gravitational field) in order to obtain a well-defined action principle [16,17]. In the following, we will outline these mathematical notions and illustrate the general set-up by considering a scalar field which is subject to different boundary conditions.

### 6.7.1 Relative de Rham complex

The definition given by Bott and Tu [239] is quite general since it applies to a generic pair $(M,N)$ of manifolds which are related by a smooth map. In view of the application that we have in mind, namely a manifold $M$ with a boundary $\partial M$, we focus on a pair $(M,N)$ of manifolds where the manifold $N \subset \partial M$ is of codimension 1 (with respect to $M$) and on the inclusion map[17] $\iota : N \hookrightarrow M$. The set of *relative $k$-forms on* $(M,N)$ is then given by

$$\Omega^k(M,N) \equiv \Omega^k(M) \oplus \Omega^{k-1}(N), \qquad \text{for } k \in \{0,\dots,n \equiv \dim M\}, \tag{221}$$

and the *differential* $\underline{d} : \Omega^k(M,N) \to \Omega^{k+1}(M,N)$ is defined for $(\alpha,\beta) \in \Omega^k(M,N)$ by

$$\underline{d}(\alpha,\beta) \equiv (d\alpha, \iota^*\alpha - d\beta) \tag{222}$$

(where $d\alpha$ and $d\beta$ involve the exterior derivatives on $M$ and $N$, respectively). The so-defined operator $\underline{d}$ is nilpotent so that relative cohomology groups can be defined. In this respect we note that $\underline{d}(\alpha,\beta) = 0$ is tantamount to saying that the form $\alpha$ on $M$ is closed and that the form $\iota^*\alpha$ on $N$ is exact.

The authors of reference [66] specified various geometric operations for the relative de Rham complex like the exterior product of relative forms as well as the interior product and Lie derivative of relative $k$-forms with respect to vector fields on $M$. Here, we only note the following points. For an $n$-dimensional manifold $M$ and $N \subset \partial M$, the *integral of a relative $n$-form $(\alpha,\beta) \in \Omega^n(M,N)$ over the pair* $(M,N)$ is defined by

$$\int_{(M,N)} (\alpha,\beta) \equiv \int_M \alpha - \int_N \beta . \tag{223}$$

The *boundary $\underline{\partial}(M,N)$ of the pair* $(M,N)$ is given by

$$\underline{\partial}(M,N) \equiv (\partial M \setminus N, \partial N), \tag{224}$$

and its inclusion map (induced by the inclusion $\iota : \partial M \hookrightarrow M$ is denoted by $\underline{\iota} : \underline{\partial}(M,N) \hookrightarrow (M,N)$. *Stokes theorem* then generalizes to

$$\int_{(M,N)} \underline{d}(\alpha,\beta) = \int_{\underline{\partial}(M,N)} \underline{\iota}^*(\alpha,\beta), \qquad \text{for } (\alpha,\beta) \in \Omega^{n-1}(M,N). \tag{225}$$

---

[17]Here, we adapt some of the notations of references [66,238] so as to be consistent with our previous notation.

### 6.7.2 Scalar fields on a globally hyperbolic manifold with boundary

A globally hyperbolic manifold $(M, g)$ with boundary $\partial M$ (see reference [146] for the mathematical aspects) is homeomorphic to $\mathbb{R} \times \Sigma$ where $\Sigma$ denotes a Cauchy hypersurface with boundary $\partial \Sigma$. Thus, the boundary of $M$ is given by the "lateral boundary" $\partial_L M$, i.e. $\partial M \cong \mathbb{R} \times \partial \Sigma = \partial_L M$. In the following, we consider the dynamics of a real scalar field $\phi$ on $M$, i.e. $\phi \in C^\infty(M) = \Omega^0(M)$. Quite generally, the quantities on $\partial M$ are denoted by a bar; for instance, the field $\bar{\phi}$ on $\partial M$ is defined by pulling back $\phi$ from $M$ to $\partial M$ by the inclusion map $\iota : \partial M \hookrightarrow M$, i.e.

$$\iota^* : \Omega^0(M) \longrightarrow \Omega^0(\partial M)$$
$$\phi \longmapsto \iota^* \phi = \phi \circ \iota \equiv \bar{\phi} \, .$$

The volume $n$-form $\sqrt{|g|} \, d^n x$ on $M$ is denoted by $\mathrm{vol}_M$ and similarly the induced volume form on the boundary $\partial M$ is denoted by $\mathrm{vol}_{\partial M}$.

We assume that the

$$\text{bulk action} \qquad S_M \equiv \int_M L \equiv \int_M \mathcal{L} \, \mathrm{vol}_M \, ,$$

is supplemented by a

$$\text{boundary action} \qquad S_{\partial M} \equiv \int_{\partial M} \bar{\ell} \, .$$

By way of *example,* we consider

$$\boxed{\mathcal{L} = \frac{1}{2} g^{\mu\nu} (\nabla_\mu \phi)(\nabla_\nu \phi) - V(\phi) \, ,} \qquad \boxed{\bar{\ell} = \frac{1}{2} f \, \bar{\phi}^2 \, \mathrm{vol}_{\partial M} \, ,} \qquad (226)$$

where $V : \mathbb{R} \to \mathbb{R}$ represents a mass/self-interaction function and $f : \partial M \to \mathbb{R}$ a smooth function which allows us to encode different boundary conditions for the field $\phi$. The *total action functional* (for a real scalar field $\phi$ coupled to the gravitational field described by the fixed metric $(g_{\mu\nu})$) thus reads

$$\boxed{\mathbb{S}[\phi] \equiv \int_{(M, \partial M)} (L, \bar{\ell}) = \int_M L - \int_{\partial M} \bar{\ell} = S_M - S_{\partial M} \, .} \qquad (227)$$

Let us recall the first variational formula on $M$, i.e. Eqn. (191): with the notation $\delta \equiv d_{\mathrm{v}}$ and $d j_M \equiv d_{\mathrm{h}} j$, this relation writes as follows for the bulk:

$$\delta L = \mathcal{E}(L) + d j_M \, . \qquad (228)$$

By imposing the condition that $\iota^* j_M$ is decomposable over the lateral boundary of $M$, we have another relation for the boundary (recall Eqn. (222) for the definition of the differential on $\partial M$)

$$\delta \bar{\ell} - \iota^* j_M = \bar{b} \wedge \delta \phi - d j_{\partial M} \, . \qquad (229)$$

For our example (226), the variational formula (228) takes the following expression (involving the vector field $V \equiv V^\mu \partial_\mu$ with $V^\mu \equiv \delta \phi \, \nabla^\mu \phi$ as well as the operator $\Box \equiv g^{\mu\nu} \nabla_\mu \nabla_\nu$),

$$\delta L = -[\Box \phi + V'(\phi)] \delta \phi \, \mathrm{vol}_M + d j_M \, , \qquad (230)$$

with

$$j_M \equiv i_V \mathrm{vol}_M = \sqrt{|g|}\, V^\mu\, d^{n-1}x_\mu = \sqrt{|g|}\, (\nabla^\mu \phi)\, \delta\phi\, d^{n-1}x_\mu\,.$$

The variational relation (229) then yields

$$\bar{b} = -\left[\, \iota^*(\nabla_n \phi) - f\, \bar{\phi}\, \right] \mathrm{vol}_{\partial M}\,, \qquad j_{\partial M} = 0\,, \tag{231}$$

where $n \equiv (n^\mu)$ denotes an outward pointing *unit normal vector field* on the lateral boundary $\partial M$.

The *covariant phase space on* $(M, \partial M)$, i.e. the space of solutions of the field equations resulting from the total action functional (227) is presently determined by the conditions $\mathcal{E}(L) = 0 = \bar{b}(\phi)$, i.e. by virtue of equations (230)-(231),

$$\Box\phi = -V'(\phi)\,, \qquad \iota^*(\nabla_n \phi) = f\, \bar{\phi}\,.$$

For $f \neq 0$, the latter relation represents *Robin (i.e. mixed) boundary conditions* for $\phi$ on $\partial M$. For the choice $f = 0$, this relation defines *Neumann boundary conditions*.

The *Dirichlet boundary condition* $\bar{\phi} = 0$ can be implemented by starting from $\bar{\ell} = 0$ and $\bar{\phi} = 0$, i.e. fields $\phi$ on $M$ which vanish on $\partial M$. The equations of motion of $\phi$ and the symplectic potential $j_M$ on the bulk are then the same as above and there are no boundary terms (i.e. the quantities $\bar{\ell}, j_{\partial M}$ and $\bar{b}$ vanish identically).

For a discussion of the symplectic form and of symmetries as well as the treatment of other field theoretic models (Yang-Mills, Chern-Simons, gravity,...) within the presented framework we refer to the work [66, 195, 238] (see also [19] and references therein). Yet, we note that this approach and the involved interpretation of boundary variations (as equations of motion) differ from the view-point adopted in various other works, see footnote 6 of reference [174] for the case of general relativity. Another approach to boundaries along with the investigation of *corners* and related corner terms will be addressed in Subsection 7.8 below.

# 7  Noether's theorems, conserved currents, forms and charges in gauge field theories

Gauge field (-type) theories like Maxwell's theory of electrodynamics, non-Abelian Yang-Mills theories or Einstein's general relativity are characterized by local symmetries (invariance of the action under gauge transformations or diffeomorphisms). Natural questions which arise in this context are the ones **whether and how these local symmetries may acquire an observable status,** i.e. the issue of the definition and derivation of conserved local and integral quantities associated to the local invariances. This subject is obviously related to Noether's theorems and over the last decades it has been an impetus for formulating these theorems in a more general manner and to look for several generalizations thereof.

The following presentation relies on the work of the Brussels' school (M. Henneaux, G. Barnich and their collaborators, see references [23, 74, 154, 229, 230, 240] as well as references therein for earlier contributions, especially [31]). In Subsection 7.1.1, we consider the basic example of free Maxwell theory to conclude that *the Noether current associated to gauge invariance vanishes on-shell up to a superpotential term.* In Subsection 7.1.2, we point out that *the associated charge represents a surface (flux) integral over the superpotential,* the existence and properties of this integral depending on the properties of fields and symmetry parameters of the underlying theory. The general definition and construction of these so-called **lower degree conservation laws** in gauge field theories are then outlined in the subsequent sections where we again make contact with the (pre-)symplectic current density encountered in the approaches of covariant phase space and of the variational bicomplex. More precisely, for

non-linear gauge field type theories like YM-theories or general relativity, generic field configurations do not have any exact symmetries in general: in this case, the decomposition of the gauge field ($A_\mu$) or metric field ($g_{\mu\nu}$) with respect to a given background ($\bar{A}_\mu$) or ($\bar{g}_{\mu\nu}$) leads to the study of the *linearized theory (around these background fields)* or to the exploration of **asymptotic fields and symmetries.** In all of these cases, the definition of charges in terms of surface integrals as well as their properties are related to the symmetries of the linearized theory [154, 157, 241]. Different approaches to these topics and the relationships between them are briefly surveyed in Subsection 7.6.

In this context, it is worthwhile recalling that the notion of *asymptotic fields* is an important one in classical and quantum field theory, e.g. for the description of scattering phenomena and their interpretation in terms of elementary particles. The underlying idea is that the non-linear interaction of fields in Minkowski space-time $\mathbb{R}^n$ is localized in a bounded region $V \subset \mathbb{R}^{n-1}$ of space and that the fields outside the region $V$ can be approximated arbitrarily well (though in a prescribed manner) by almost free fields (background field theory), e.g. see [242] and references therein. For this instance where the interacting field theory admits a reasonable linearization (with respect to background fields), one can associate well-defined conserved charges to the solutions of the interacting theory, these solutions differing arbitrarily little from the solutions of the background theory outside the interaction region: each of these so-called *asymptotic charges* is given by an integral over a surface that lies outside the interaction region $V$ and their presence thereby depends on the existence of a well-defined linearization (admitting symmetries) of the full theory. The first asymptotic conservation laws which have been constructed in this spirit are the ADM charges in general relativity in asymptotically flat space-time [175], the analogous Abbott-Deser charges in asymptotically anti-de Sitter space-times [243] as well as the color charges in YM-theory [244].

Concerning the *symmetries,* we note that the local symmetries (local gauge symmetry or diffeomorphism invariance in gravity) concern the so-called bulk of space-time whereas the associated charges have to do with the boundary: in gravity, the latter charges become the conserved charges associated to Poincaré invariance (energy-momentum and angular momentum) or to the (larger) BMS symmetry [245]. We note that a similar distinction can be made [197] for a particle for which the world line is reparametrization invariant (gauge symmetry) and the ends (i.e. the boundary) correspond to vertices in the interacting theory: at the latter, the energy-momentum conservation law (related to Poincaré symmetry) and the associated physical conservation laws have to be taken into account. The ongoing interests in field theory include the physical significance of symmetries and conservation laws as well the relationship with infrared problems in quantum field theory and with memory effects, e.g. see the monograph [225] as well as the recent works [246–248] for detailed discussions of symmetries.

In this section, we will use Dirac's notation $F \approx 0$ (*"F vanishes weakly"* or *"F vanishes on-shell"*) for local functions $F$ which vanish on the so-called *stationary surface* $\mathcal{R}^\infty$ (where $0 = \frac{\partial S}{\partial q^a} = \partial_\mu \frac{\partial S}{\partial q^a} = \cdots$). Moreover, we will mostly use the traditional notation of field theory rather than the corresponding notation introduced for the variational bicomplex in Subsection 6.1: the dictionary between both notations has been outlined in Subsection 6.1 and Subsection 6.6.

## 7.1 Motivation (Pedestrian approach)

### 7.1.1 Gauge symmetry and associated Noether current

Consider a pure gauge theory like free Maxwell theory in $n$-dimensional Minkowski space-time $M = \mathbb{R}^n$. The corresponding action functional $S[A] = -\frac{1}{4} \int_M d^n x \, F^{\mu\nu} F_{\mu\nu}$ yields the Euler-Lagrange derivative $\frac{\delta S}{\delta A_\mu} = \partial_\nu F^{\nu\mu}$. The Lagrangian and thereby the action are invariant under

the local gauge transformation $\delta_f A_\mu = \partial_\mu f$ where $x \mapsto f(x)$ is a smooth real-valued function.

Before determining the conserved current associated to the local gauge invariance of the action, we make the following observations (based on reference [154]). The product

$$\frac{\delta S}{\delta A_\mu} \delta_f A_\mu = (\partial_\nu F^{\nu\mu}) \partial_\mu f , \tag{232}$$

which occurs in Noether's first theorem can be rewritten by applying the Leibniz rule to the derivative $\partial_\mu$:

$$\frac{\delta S}{\delta A_\mu} \delta_f A_\mu = (\partial_\nu F^{\nu\mu}) \partial_\mu f = \partial_\mu \Big[ \underbrace{(\partial_\nu F^{\nu\mu})}_{= \frac{\delta S}{\delta A_\mu}} f \Big] - \underbrace{(\partial_\mu \partial_\nu F^{\nu\mu})}_{= 0} f ,$$

$$\text{i.e.} \quad \frac{\delta S}{\delta A_\mu} \delta_f A_\mu = \partial_\mu S_f^\mu , \qquad \text{with } S_f^\mu \equiv S_f^{\mu\nu}\Big(\frac{\delta S}{\delta A_\nu}\Big) \equiv \frac{\delta S}{\delta A_\mu} f = (\partial_\nu F^{\nu\mu}) f . \tag{233}$$

Here, the identity $\partial_\mu \partial_\nu F^{\nu\mu} = 0$ follows from the antisymmetry of the Faraday tensor and actually represents *Noether's identity* $\partial_\mu\big(\frac{\delta S}{\delta A_\mu}\big) = 0$ (see Eqn. (51)) following from the local gauge invariance of the action functional by virtue of Noether's second theorem. The current $S_f^\mu$ represents a linear combination of Euler-Lagrange derivatives and thus vanishes on-shell.

Alternatively, we can rewrite the product (232) by applying the Leibniz rule to the derivative $\partial_\nu$:

$$\frac{\delta S}{\delta A_\mu} \delta_f A_\mu = (\partial_\nu F^{\nu\mu}) \partial_\mu f = \partial_\nu \Big[ F^{\nu\mu} \partial_\mu f \Big] - \underbrace{(F^{\nu\mu} \partial_\nu \partial_\mu f)}_{= 0},$$

$$\text{i.e.} \quad \frac{\delta S}{\delta A_\mu} \delta_f A_\mu = \partial_\mu j_f^\mu , \qquad \text{with } j_f^\mu \equiv F^{\mu\nu} \partial_\nu f . \tag{234}$$

By subtracting the current densities appearing in equations (233),(234), we obtain

$$j_f^\mu - S_f^\mu = F^{\mu\nu} \partial_\nu f - \underbrace{\partial_\nu F^{\nu\mu}}_{= -\partial_\nu F^{\mu\nu}} f = \partial_\nu k_f^{\mu\nu} , \qquad \text{with } k_f^{\mu\nu} \equiv f F^{\mu\nu}.$$

Here, the current density $\partial_\nu k_f^{\mu\nu}$ represents a so-called *superpotential term,* the antisymmetric tensor $(k_f^{\mu\nu})$ being referred to as a *superpotential:* due to the antisymmetry of $k_f^{\mu\nu}$, the divergence $\partial_\mu(\partial_\nu k_f^{\mu\nu})$ vanishes identically, i.e. such a current is identically (off-shell) conserved.

In summary, we have the relation [154]

$$j_f^\mu = \underbrace{S_f^\mu}_{\approx 0} - \underbrace{\partial_\nu k_f^{\nu\mu}}_{\text{superpot. term}} . \tag{235}$$

Finally, we apply Noether's first theorem (see equations (213)-(214)) to the gauge invariance of Maxwell's Lagrangian $\mathcal{L}$ "as if we were dealing with a global symmetry transformation":

$$0 = \delta_f \mathcal{L} = \frac{\delta S}{\delta A_\mu} \delta_f A_\mu + \partial_\mu \mathcal{J}_f^\mu , \qquad \text{with } \mathcal{J}_f^\mu \equiv \frac{\partial \mathcal{L}}{\partial(\partial_\mu A_\nu)} \delta_f A_\nu = -F^{\mu\nu} \partial_\nu f = -j_f^\mu . \tag{236}$$

By virtue of (235), we conclude that *the Noether current* $(\mathcal{J}_f^\mu)$ *associated to a local gauge symmetry vanishes on-shell up to a superpotential term* (which is conserved identically). From the point of view of Noether's first theorem, a current (associated to a variational symmetry) which is on-shell a superpotential term is considered to be a *trivial current.* Accordingly, a non-trivial

*gauge symmetry* of an action (as well as a trivial one, i.e. a linear combination of equations of motion and/or of their derivatives) is to be viewed as a *trivial (global) symmetry* from the point of view of Noether's first theorem. These notions will be formalized mathematically in Subsection 7.2 by defining equivalence relations for both symmetries and currents and by formulating Noether's first theorem in a general form.

We note that the line of arguments presented above for Maxwell's theory generalizes to a generic gauge theory in a space-time of dimension $n \geq 2$ and to possibly field-dependent gauge parameters $x \mapsto f^r(x)$ [154, 249].

### 7.1.2 Definition of the conserved charge associated to gauge symmetry

**Maxwell theory:** As we just noted, the local gauge invariance of the Lagrangian gives rise to a current $(\mathcal{J}_f^\mu)$ which represents on-shell a superpotential term, i.e. on-shell we have $\mathcal{J}_f^\mu = \partial_\nu k_f^{\nu\mu}$ with $k_f^{\mu\nu} = -k_f^{\nu\mu}$. (For the case of Maxwell's theory, we found the explicit expression $k_f^{\mu\nu} = f F^{\mu\nu}$.) The superpotential $k_f^{\mu\nu}$ determining the current $\mathcal{J}_f^\mu$ depends on the arbitrary gauge parameter $f$ and is thus arbitrary itself. Yet, the charge contained in $V \subset \mathbb{R}^3$ (for a four dimensional space-time) can be expressed as a surface integral over the superpotential $(k_f^{i0})$,

$$Q \equiv \int_V d^3x \, \mathcal{J}_f^0 = \int_V d^3x \, \partial_i k_f^{i0} = \oint_{S \equiv \partial V} k_f^{i0} \, dS^i \,, \tag{237}$$

and one can attribute a well-defined mathematical and physical meaning to this integral for certain field theories (isolation of specific superpotentials): *the existence and properties of the flux integral* (237) *depend on the properties of the integrand (i.e. the assumptions made for fields and symmetry parameters) in the vicinity of the surface $S = \partial V$.* E.g. for Maxwell's theory in $\mathbb{R}^4$, we have

$$Q \equiv \int_V d^3x \, \mathcal{J}_f^0 = \int_V d^3x \, \partial_i(f F^{i0}) = \int_V d^3x \, \partial_i(f F_{0i}) = \oint_{S \equiv \partial V} f F_{0i} \, dS^i = \oint_{S \equiv \partial V} f \vec{E} \cdot \overrightarrow{dS} \,, \tag{238}$$

where $\vec{E}$ denotes the electric field strength. In this example, trivial gauge transformations, i.e. $0 \approx \delta_f A_\mu = \partial_\mu f$ correspond to a *constant parameter* $f \equiv \varepsilon$ and the integral (238) then yields a charge $q \equiv \frac{1}{\varepsilon} Q = \oint_{S \equiv \partial V} \vec{E} \cdot \overrightarrow{dS}$: this flux integral coincides with the usual expression for the electric charge in electrodynamics.

**Non-Abelian gauge field theories (and general relativity):** In non-Abelian gauge theories like YM (Yang-Mills) theory, the infinitesimal gauge transformation $\delta_f A_\mu = D_\mu f$ is field dependent (for non-trivial gauge parameters, i.e. for $f \neq 0$). Similarly, in general relativity, the transformation law of the metric tensor field $(g_{\mu\nu})$ under diffeomorphisms generated by a vector field $\xi \equiv \xi^\mu \partial_\mu$, i.e. $\delta_\xi g_{\mu\nu} = \nabla_\mu \xi_\nu + \nabla_\nu \xi_\mu$ depends on the metric field. Generic gauge field configurations $(A_\mu)$ or metrics $(g_{\mu\nu})$ do not admit any symmetries and thereby the relations $\delta_f A_\mu = 0$ and $\delta_\xi g_{\mu\nu} = 0$ do not admit non-trivial solutions. The situation is different if a background field $(\bar{A}_\mu$ or $\bar{g}_{\mu\nu})$ is given which admits symmetries. In this case, the theory may be *linearized* around the background field configuration or *asymptotic symmetries* may be explored if the fields $A_\mu$ (or $g_{\mu\nu}$) tend to $\bar{A}_\mu$ (or $\bar{g}_{\mu\nu}$) in an asymptotic region, e.g. for $|\vec{x}| \to \infty$. In these cases, conserved flux integrals that are analogous to the ones of Maxwell's theory may be introduced as has been pointed out by L. F. Abbott and S. Deser [243, 244] (see also [154, 157] for the underlying general construction and for references to the earlier

literature which includes in particular the work of ADM [175] on general relativity): we will address these issues at the end of Subsection 7.4 and in Subsection 7.5.

The gauge parameters $f$ such that $\delta_f A_\mu$ vanishes on-shell are referred to as *global reducibility parameters* or as *(gauge) Killing vectors* in reference to general relativity where the corresponding relation $0 \approx \delta_\xi g_{\mu\nu} = \nabla_\mu \xi_\nu + \nabla_\nu \xi_\mu$ for the metric field $(g_{\mu\nu})$ describes Killing vector fields $\xi \equiv \xi^\mu \partial_\mu$. Likewise the gauge parameters $f$ such that $\delta_f A_\mu$ vanishes (on-shell) asymptotically are referred to as *asymptotic reducibility parameters* [154]. As a matter of fact, by using general cohomological methods, the authors of references [154,229] have established a *generalized version of Noether's first theorem* (which we will spell out in more detail in Subsection 7.4): this result states that in $n \geq 2$ space-time dimensions, there is a 1-1-correspondence between (global) reducibility parameters and $(n-2)$-forms[18] $k \equiv k^{[\mu\nu]}(d^{n-2}x)_{\mu\nu}$ which are closed on-shell (e.g. for $n = 4$ we have $\varepsilon^{\rho\sigma\mu\nu}\partial_\rho k_{\mu\nu} \approx 0$), but not exact (i.e. there is no $(n-3)$-form $l$ such that $k \approx dl$). This result admits an analog for the case of asymptotic reducibility parameters and asymptotically conserved $(n-2)$-forms. The surface charges are then given in terms of the superpotential $k^{[\mu\nu]}$ (cohomological approach of Barnich and Brandt [154]). For a given gauge field-type theory like YM-theory or general relativity, the superpotential can be (and has also been) constructed by other methods, in particular from the (pre-)symplectic potential that we discussed in the approaches of covariant phase space and of the variational bicomplex, see Subsection 7.6 below.

## 7.2 Noether's first theorem

The two fundamental theorems that E. Noether established in her celebrated article of 1918 [250] have been generalized during the last decades, e.g. see the monograph [251] for an historical account of these accomplishments. In the following, we will outline the generalizations of Noether's first theorem [74,154,252].

Consider a continuous infinitesimal symmetry transformation of the field, i.e. (cf. (220))

$$\delta_f q^a \equiv Q_r^a(f^r) \equiv Q_r^a f^r + Q_r^{a\mu}\partial_\mu f^r + \cdots,$$

where the functions $x \mapsto f^r(x)$ parametrize a local gauge symmetry. For global symmetry transformations (see equations (209),(211)), one introduces the following *equivalence relation*

$$X^a \sim X^a + \underbrace{Q_r^a(f^r)}_{\text{gauge transf.}} + \underbrace{M^{[ba]}\frac{\delta S}{\delta q^b} - \partial_\mu\left(M^{[b(\nu)a(\mu)]}\partial_\nu\frac{\delta S}{\delta q^b}\right) \pm \cdots}_{\text{e.o.m. symmetry transf.}} \tag{239}$$

Here, the coefficients $M^{[ba]}$ are local functions of the fields: the corresponding terms in the previous relation vanish on-shell. The equivalence class of $X^a$ is denoted by $[X^a]$.

For the (on-shell) conserved current densities, one introduces the following *equivalence relation* (which amounts to identify conserved currents differing by trivial currents[19])

$$j^\mu \sim j^\mu + \underbrace{\partial_\nu k^{[\nu\mu]}}_{\text{superpot. term}} + \underbrace{t^\mu}_{\approx 0}. \tag{240}$$

Here, the antisymmetric tensor field $k^{[\nu\mu]}$ is a local function of the fields that is twice differentiable whence the identity $\partial_\mu(\partial_\nu k^{[\nu\mu]}) = 0$. For two equivalent currents $(j_1^\mu)$ and $(j_2^\mu)$, we thus have $\partial_\mu j_1^\mu \approx \partial_\mu j_2^\mu$.

---

[18]Here and in the following, we use the standard notation $k^{[\mu\nu]} \equiv \frac{1}{2}(k^{\mu\nu} - k^{\nu\mu})$ for the antisymmetrization.

[19]For $n = 1$, i.e. for classical mechanics, the current $(j^\mu)$ has a single component $j^0$ to be interpreted as a charge: in this case, the equivalence relation also involves an additional real constant reflecting the fact that charges differing by such a constant are to be identified in mechanics.

In this respect we recall that, in the language of differential forms (see Appendix A), a current density $(j^\mu)$ corresponds to an $(n-1)$-form $j \equiv j^\mu\, d^{n-1}x_\mu$ and an antisymmetric tensor field $(k^{[\mu\nu]})$ corresponds to an $(n-2)$-form $k \equiv k^{[\mu\nu]}(d^{n-2}x)_{\mu\nu}$. For a $d_{\mathrm{h}}$-closed $(n-1)$-form $j$ one has the equivalence

$$d_{\mathrm{h}} j = 0 \iff \partial_\mu j^\mu = 0\,,$$

and for an $d_{\mathrm{h}}$-exact $(n-1)$-form $j$ one has

$$j = d_{\mathrm{h}} k \iff j^\mu = -\partial_\nu k^{[\nu\mu]}\,.$$

With $t \equiv t^\mu\, d^{n-1}x_\mu$, the equivalence relation (240) then reads

$$j \sim j + \underbrace{d_{\mathrm{h}} k}_{\text{superpot. term}} + \underbrace{t}_{\approx 0}\,. \tag{241}$$

The equivalence class of $j$ will be denoted by $[j]$. We will now use these notions to spell out generalized formulations of Noether's first theorem.

### 7.2.1 Complete form of Noether's first theorem

**Complete form of Noether's first theorem:** Consider a local field theory which is defined on a space-time manifold of dimension $n \geq 2$ and whose dynamics is described by an action functional $S[\varphi]$. Suppose this action admits continuous global symmetries and possibly, in addition, local symmetries. Then there is a *one-to-one correspondence between non-trivial Noether (global variational) symmetries and non-trivial Noether (on-shell conserved) currents:*

$$\boxed{[X^a] \longleftrightarrow [j]\,.} \tag{242}$$

The correspondence $[X^a] \mapsto [j]$ is realized by (214), i.e.

$$\boxed{X^a \longmapsto j^\mu \equiv \frac{\partial \mathcal{L}}{\partial q^a_\mu} X^a - k^\mu_X\,,} \qquad \text{for} \quad \delta_X \mathcal{L} = \partial_\mu k^\mu_X\,. \tag{243}$$

Here, the first term of $j^\mu$, i.e.

$$V^\mu_a(X^a, \mathcal{L}) \equiv \frac{\partial \mathcal{L}}{\partial q^a_\mu} X^a = \frac{\partial \mathcal{L}}{\partial q^a_\mu} \delta_X q^a\,, \tag{244}$$

coincides with the symplectic potential (83), viz. the components of the variational 1-form (193).

We note that *a local gauge symmetry represents the trivial class $[X^a] \equiv [0]$ of global symmetries and corresponds to the trivial class $[j] = [0]$ of Noether currents,* see equations (235)-(236).

**Outline of proof:** The proof [31,229] of the given version of Noether's first theorem consists of determining the so-called *characteristic cohomology group (associated with the stationary surface) in form degree $n-1$*, i.e. the $(n-1)$-th cohomology group of the exterior space-time differential $d_{\mathrm{h}}$ pulled back to the space of solutions of the field equations: this group is denoted by

$$H^{n-1}\big(d_{\mathrm{h}}, \Omega^\bullet(S)\big)\,, \tag{245}$$

where $\Omega^\bullet(S) = \Omega^\bullet(\mathcal{R}_\infty)$ represents the space of differential forms pull-backed to the stationary surface $S$, (i.e. solutions of the equations of motion). This means that, for the $(n-1)$-forms

$j \in [j]$ with $[j] \in H^{n-1}\big(d_{\mathrm{h}}, \Omega^{\bullet}(S)\big)$, one does not simply consider $d_{\mathrm{h}}j = 0$ and $j \sim j + d_{\mathrm{h}}k$ (as for the standard cohomology groups), but one rather considers $d_{\mathrm{h}}j \approx 0$ and identifies the $(n-1)$-forms $j$ according to relation (241). For the characteristic cohomology group (245), one has the general result

$$H^{n-1}\big(d_{\mathrm{h}}, \Omega^{\bullet}(S)\big) \cong H^{n-1}(d_{\mathrm{h}}|\delta), \tag{246}$$

i.e. it is equivalent to the $(n-1)$-th cohomology group of $d_{\mathrm{h}}$ modulo the *Koszul-Tate differential* $\delta$, this differential (discussed in Appendix B.3) depending on the action and on its symmetries, see references [23, 74, 229, 240] for the details.

### 7.2.2 Full form of Noether's first theorem

The vector space of equivalence classes $[X^a]$ of non-trivial global symmetries as well as the vector space of equivalence classes $[j]$ of non-trivial conserved currents can be endowed with Lie brackets which are preserved by the correspondence (242). Thus, there is a Lie algebra homomorphism between inequivalent variational symmetries and inequivalent conserved currents. To present this so-called *full form of Noether's first theorem* [240] (also referred to as *Noether representation theorem* [249]), we first introduce the different brackets.

Consider the commutator of two evolutionary vector fields $X_1, X_2$ as defined by (209), (211): from the fundamental property $[\partial_{\mu}, \delta_X] = 0$, it follows that

$$\big[\delta_{X_1}, \delta_{X_2}\big] = \delta_{[X_1, X_2]_{\mathrm{L}}}, \qquad \text{with} \qquad \boxed{[X_1, X_2]_{\mathrm{L}}^a \equiv \delta_{X_1} X_2^a - \delta_{X_2} X_1^a.} \tag{247}$$

Thus, the vector space of evolutionary vector fields endowed with this bracket represents an infinite dimensional Lie algebra. The commutator (247) is also qualified as the *Lie bracket of characteristics* [157].

Now suppose the evolutionary vector fields $X_1, X_2$ represent global symmetries for a field theory described by a Lagrangian $\mathcal{L}$, i.e. $\delta_{X_1}\mathcal{L} = \partial_{\mu}k_{X_1}^{\mu}$ and $\delta_{X_2}\mathcal{L} = \partial_{\mu}k_{X_2}^{\mu}$. Then, we have

$$\delta_{[X_1, X_2]_{\mathrm{L}}}\mathcal{L} = \big[\delta_{X_1}, \delta_{X_2}\big]\mathcal{L} = \delta_{X_1}(\partial_{\mu}k_{X_2}^{\mu}) - (1 \leftrightarrow 2) = \partial_{\mu}k_{12}^{\mu}, \qquad \text{where} \quad k_{12}^{\mu} \equiv \delta_{X_1}k_{X_2}^{\mu} - \delta_{X_2}k_{X_1}^{\mu},$$

and where we again used the relation $[\partial_{\mu}, \delta_X] = 0$. Accordingly, the vector space of global symmetries of a given field theory represents an (infinite dimensional) Lie subalgebra of the Lie algebra of evolutionary vector fields. It can be shown [154] that the bracket of global symmetries induces a well-defined bracket on the vector space of equivalence classes of these symmetries:

$$\big[[X_1], [X_2]\big]_{\mathrm{L}}^a \equiv \big[[X_1, X_2]_{\mathrm{L}}^a\big]. \tag{248}$$

For the Noether currents $j_{X_1}, j_{X_2}$ associated to global symmetries $X_1, X_2$ by virtue of (243), one can introduce the so-called **Dickey bracket** [253] defined by

$$\boxed{\{j_{X_1}, j_{X_2}\}_{\mathrm{D}} \equiv \delta_{X_1}j_{X_2} = -\delta_{X_2}j_{X_1} = \frac{1}{2}\big(\delta_{X_1}j_{X_2} - \delta_{X_2}j_{X_1}\big).} \tag{249}$$

By applying the variation $\delta_{X_1}$ to the local conservation law (214) written in terms of differential forms, i.e. $d_{\mathrm{h}}j_{X_2} = -E_a(\mathcal{L})X_2^a \, d^n x$, and by taking into account the relation $[\partial_{\mu}, \delta_X] = 0$ as well as $\delta_{X_1}\frac{\delta S}{\delta \phi^i} \approx 0$, one finds that [74, 249]

$$\{j_{X_1}, j_{X_2}\}_{\mathrm{D}} = j_{[X_1, X_2]_{\mathrm{L}}} + \text{trivial current}. \tag{250}$$

The trivial current can be eliminated by going over to equivalence classes of currents: thereby we obtain a Lie bracket on the cohomology groups $H^{n-1}(d_{\mathrm{h}}, \Omega^\bullet(S))$, i.e. on the vector space of non-trivial conserved currents:

$$\{[j_{X_1}], [j_{X_2}]\}_{\mathrm{D}}^\mu = \left[\frac{1}{2}\left(\delta_{X_1} j_{X_2} - \delta_{X_2} j_{X_1}\right)\right] = [\,\omega^\mu(X_1, X_2)\,]. \tag{251}$$

Here, $\omega^\mu(X_1, X_2)$ denotes the contraction of the local symplectic form (see equations (84) and (244)), i.e. of $\omega^\mu = d_{\mathrm{v}}\left(V_a^\mu(d_{\mathrm{v}} q^a, \mathcal{L})\right)$, with the evolutionary vector fields $X_1, X_2$ (see references [74, 154]).

From (249)-(251) we infer that

$$\boxed{\{[j_{X_1}], [j_{X_2}]\}_{\mathrm{D}}^\mu = \left[\, j_{[X_1, X_2]_{\mathrm{L}}}^\mu \,\right].} \tag{252}$$

Together with the correspondence (242), this result represents [240] the

**Full form of Noether's first theorem:** We have a *Lie algebra homomorphism between the Lie algebra of non-trivial global symmetries (endowed with the Lie bracket* (248)*) and the Lie algebra of non-trivial Noether currents (endowed with the Dickey bracket* (251)*).*

We refer to the work [240] for the correspondence (isomorphism in the case of non-degenerate Lagrangian field theories) between the Dickey bracket of conserved currents and the standard Hamiltonian Poisson bracket of conserved charges as well as the Batalin-Vilkovisky anti-bracket for the local BRST cohomology classes. The first of these correspondences is illustrated by the following example.

**Example:** Let us consider the free massless quark model in $\mathbb{R}^4$, see equations (217)-(219). For the evolutionary vector fields $X_r, X_s$ (with $r, s \in \{1, \ldots, 8\}$) determined by (218), it then follows from (247) that

$$[X_r, X_s]_{\mathrm{L}} = -f_{rst} X_t, \tag{253}$$

where $f_{rst}$ are the structure constants of the Lie algebra $su(3)$. For the corresponding conserved currents (219), the Dickey bracket (249) is given by

$$\{j_{X_r}, j_{X_s}\}_{\mathrm{D}} = -f_{rst} j_{X_t}. \tag{254}$$

Obviously, relations (253),(254) reflect the Lie algebra homomorphism which is the subject of the full form of Noether's first theorem.

The Hamiltonian Poisson brackets of the classical quark fields $\varphi^a \equiv (\varphi_\alpha^a)_{\alpha=1,\ldots,4}$ have the form

$$\{\varphi_\alpha^a(t, \vec{x}), \varphi_\beta^{\dagger b}(t, \vec{y})\} = -\mathrm{i}\,\delta_{\alpha\beta}\,\delta^{ab}\,\delta(\vec{x} - \vec{y}), \qquad \text{for } a, b \in \{1, 2, 3\}, \;\; \alpha, \beta \in \{1, \ldots, 4\}.$$

This implies that the Noether charges associated to the conserved currents (219), i.e. expressions

$$Q_r \equiv \int_{\mathbb{R}^3} d^3x\, j_r^0 = \int_{\mathbb{R}^3} d^3x\, \varphi^\dagger T_r \varphi,$$

satisfy the relation

$$\{Q_r, Q_s\} = f_{rst} Q_t. \tag{255}$$

The latter equation reflects the homomorphism between the algebra of inequivalent conserved charges (endowed with the Poisson bracket) and the algebra of inequivalent conserved currents (equipped with the Dickey bracket), *two conserved charges being identified if they coincide on-shell.*

## 7.3   Noether identities and Noether's second theorem

The following considerations provide the general underpinning for the results that we obtained in Subsection 7.1.1 for the particular case of gauge invariance of Maxwell's theory. Our presentation is based on references [74, 249].

### 7.3.1   Noether identities

Suppose we have a local symmetry of a Lagrangian $\mathcal{L}(\varphi, \partial_\mu \varphi)$ as described by Eqn. (220), i.e. we have $\delta_f \mathcal{L} = \partial_\mu k_f^\mu$ for the variation of $\mathcal{L}$ induced by the local symmetry transformation $\delta_f \varphi^a = Q^a(f)$. By using $\delta_f(\partial_\mu \varphi) = \partial_\mu(\delta_f \varphi)$ and the Leibniz rule for $\partial_\mu$, we find that

$$\delta_f \mathcal{L} = \frac{\partial \mathcal{L}}{\partial \varphi} \delta_f \varphi + \frac{\partial \mathcal{L}}{\partial(\partial_\mu \varphi)} \delta_f(\partial_\mu \varphi) = \underbrace{\left[ \frac{\partial \mathcal{L}}{\partial \varphi} - \partial_\mu\left(\frac{\partial \mathcal{L}}{\partial(\partial_\mu \varphi)}\right) \right]}_{= \frac{\delta S}{\delta \varphi}} \delta_f \varphi + \partial_\mu\left[ \frac{\partial \mathcal{L}}{\partial(\partial_\mu \varphi)} \delta_f \varphi \right], \quad (256)$$

i.e. the "first variational formula" (82):

$$0 = \delta_f \varphi \frac{\delta S}{\delta \varphi} + \partial_\mu \mathcal{J}_f^\mu, \qquad \text{with } \mathcal{J}_f^\mu = \frac{\partial \mathcal{L}}{\partial(\partial_\mu \varphi)} \delta_f \varphi - k_f^\mu. \quad (257)$$

(For Maxwell's theory, this is our result (236) with $k_f^\mu \equiv 0$.) If we consider the general form of $\delta_f \varphi$ as given by (220), i.e. $\delta_f \varphi^a = Q^a(f) \equiv Q_r^a(f^r) \equiv Q_r^a f^r + Q_r^{a\mu} \partial_\mu f^r$ (assuming for simplicity that there are no higher derivatives of $f^r$ in this transformation law), then we obtain another expression for $\delta_f \varphi \frac{\delta S}{\delta \varphi}$ (using again the Leibniz rule for $\partial_\mu$):

$$\delta_f \varphi \frac{\delta S}{\delta \varphi} = \left[ Q_r^a f^r + Q_r^{a\mu} \partial_\mu f^r \right] \frac{\delta S}{\delta \varphi^a} = f^r \underbrace{\left[ Q_r^a \frac{\delta S}{\delta \varphi^a} - \partial_\mu\left( Q_r^{a\mu} \frac{\delta S}{\delta \varphi^a} \right) \right]}_{= (Q_r^a)^\dagger\left(\frac{\delta S}{\delta \varphi^a}\right)} + \partial_\mu\underbrace{\left[ \frac{\delta S}{\delta \varphi^a} Q_r^{a\mu} f^r \right]}_{\equiv S_f^\mu}. \quad (258)$$

Here, $(Q_r^a)^\dagger$ denotes the adjoint operator (with respect to the $L^2$-inner product) of the differential operator $Q_r^a$ acting on smooth functions.[20] (For Maxwell's theory, equation (258) coincides with the first line of Eqn. (233).)

Combination of equations (257) and (258) now leads to the equality

$$f^r (Q_r^a)^\dagger\left( \frac{\delta S}{\delta \varphi^a} \right) = -\partial_\mu\left( \mathcal{J}_f^\mu + S_f^\mu \right). \quad (259)$$

If we apply the Euler-Lagrange derivative with respect to the arbitrary functions $f^r$ (i.e. the derivative $\frac{\delta \cdot}{\delta f^r} \equiv \frac{\partial \cdot}{\partial f^r} - \partial_\mu\left(\frac{\partial \cdot}{\partial(\partial_\mu f^r)}\right)$) to this relation, then the right-hand side yields a vanishing result since the equations of motion are trivial for a total derivative.

**Summary:** For an action functional $S[\varphi]$ which is invariant under local symmetry transformations $\delta_f \varphi^a = Q_r^a(f^r)$, we have the so-called

Noether identities :
$$\boxed{0 = (Q_r)^\dagger\left( \frac{\delta S}{\delta \varphi} \right) \equiv (Q_r^a)^\dagger\left( \frac{\delta S}{\delta \varphi^a} \right),} \quad \text{for all } r. \quad (260)$$

If $r$ takes $m$ values, then these relations represent $m$ identities relating the functional derivatives $\delta S/\delta \varphi^1, \ldots, \delta S/\delta \varphi^N$: they show that *the equations of motion are not all independent in a theory admitting local symmetries. These identities hold off-shell and are trivially satisfied for the solutions of the equations of motion.* They reflect the fact that if $\varphi$ is a solution of the equations of motion, then the symmetry transformed solution $\varphi'$ is another solution involving $m$ arbitrary functions.

---

[20]By definition, we have $\langle \psi^a, Q_r^a(f^r) \rangle = \int d^n x \, \psi^a Q_r^a(f^r) = \langle (Q_r^a)^\dagger \psi^a, f^r \rangle$.

### 7.3.2  Noether's second theorem

By substituting the Noether identities (260) into (258) we obtain the following general result.

**Noether's second theorem:**  For an action functional $S[\varphi]$ which is invariant under local symmetry transformations $\delta_f \varphi^a = Q_r^a(f^r) \equiv Q_r^a f^r + Q_r^{a\mu} \partial_\mu f^r$, we have

$$\boxed{\delta_f \varphi^a \frac{\delta S}{\delta \varphi^a} = \partial_\mu S_f^\mu,} \qquad \text{with } S_f^\mu \equiv S_r^{\mu a}\left(\frac{\delta S}{\delta \varphi^a}, f^r\right) \equiv \frac{\delta S}{\delta \varphi^a} Q_r^{a\mu} f^r \approx 0. \tag{261}$$

For the solutions of the equations of motion, we thus have a representative $(S_f^\mu)$ of the conserved Noether current which vanishes weakly. (For Maxwell's theory, equation (261) co-incides with (233).)

It is instructive (and useful for the sequel) to spell out the explicit expressions that one obtains for the basic examples. In each case the descriptors $Q_r^a$ and $Q_r^{a\mu}$ of local symmetry transformations can simply be read of from the transformation laws and the current $(S_f^\mu)$ is conveniently determined by evaluating $\delta_f \varphi^a \frac{\delta S}{\delta \varphi^a}$.

**Example of Maxwell theory:**  As we already mentioned, this is the particular case discussed in Subsection 7.1.1, see equations (232)-(233):

| | |
|---:|:---|
| Symmetry transformation of fields $A_\mu$ : | $\delta_f A_\mu = \partial_\mu f$ , |
| Noether identities : | $\partial_\mu \partial_\nu F^{\mu\nu} = 0$, |
| Current $(S_f^\mu)$ : | $S_f^\mu = (\partial_\nu F^{\nu\mu})f$ . |

**Example of YM-theory:**  With the notation $f \equiv f^r T_r$ for the Lie algebra valued symmetry parameters, we have:

| | |
|---:|:---|
| Symmetry transformation of fields $A_\mu^r$ : | $\delta_f A_\mu^r = (D_\mu f)^r \equiv \partial_\mu f^r + ig\,[A_\mu, f]^r$ , |
| Noether identities : | $D_\mu D_\nu F^{\mu\nu} = 0$, |
| Current $(S_f^\mu)$ : | $S_f^\mu = \mathrm{Tr}\left[(D_\nu F^{\nu\mu})f\right]$. |

**Example of general relativity with a cosmological constant:**  In the presence of a cosmo-logical constant $\Lambda$, the action (55) for the metric field $\mathfrak{g} \equiv (g_{\mu\nu})$ becomes

$$S_{\text{grav}}[\mathfrak{g}] \equiv \frac{1}{2\kappa} \int_M d^n x \sqrt{|g|}\,(R - 2\Lambda), \qquad \text{with } \kappa \equiv 8\pi G. \tag{262}$$

Variation with respect to the metric field yields the cosmological Einstein tensor $G_c^{\mu\nu}$:

$$\frac{\delta S_{\text{grav}}}{\delta g_{\mu\nu}} = -\frac{\sqrt{|g|}}{2\kappa} G_c^{\mu\nu}, \qquad \text{with } G_c^{\mu\nu} \equiv R^{\mu\nu} - \frac{1}{2} g^{\mu\nu} R + \Lambda g^{\mu\nu}. \tag{263}$$

The action $S_{\text{grav}}[\mathfrak{g}]$ is invariant under diffeomorphisms of the space-time manifold $M$ which are given at the infinitesimal level by $\delta_\xi g_{\mu\nu} = \nabla_\mu \xi_\nu + \nabla_\nu \xi_\mu$ where $\xi \equiv \xi^\mu \partial_\mu$ is the vector field generating the diffeomorphisms. By using the symmetry of the tensor $(G_c^{\mu\nu})$, the Leibniz rule and the metricity condition $\nabla_\lambda g_{\mu\nu} = 0$, we have

$$\delta_\xi g_{\mu\nu} \frac{\delta S_{\text{grav}}}{\delta g_{\mu\nu}} = -\frac{\sqrt{|g|}}{2\kappa} 2\,(\nabla_\mu \xi_\nu)\,G_c^{\mu\nu} = \partial_\mu \underbrace{\left(-\frac{\sqrt{|g|}}{\kappa} \xi_\nu G_c^{\mu\nu}\right)}_{\equiv S_f^\mu} + \frac{\sqrt{|g|}}{\kappa} \xi_\nu \underbrace{\nabla_\mu G_c^{\mu\nu}}_{=\,\nabla_\mu G^{\mu\nu}}. \tag{264}$$

To summarize [157]:

| | |
|---|---|
| Symmetry transformation of fields $g_{\mu\nu}$ : | $\delta_\xi g_{\mu\nu} = \nabla_\mu \xi_\nu + \nabla_\nu \xi_\mu$ , |
| Noether identities : | $\nabla_\mu G^{\mu\nu} = 0$ , |
| Current $(S_\xi^\mu)$ : | $S_\xi^\mu = -\frac{\sqrt{|g|}}{\kappa} \xi_\nu G_c^{\nu\mu}$ . |

The case of Einstein-Maxwell theory is discussed in reference [254]. In view of the particular form of the Noether identities in the previous examples, these relations are occasionally referred to as *generalized Bianchi identities*.

**Relationship with Dirac's approach to constrained dynamical systems :** For concreteness, we focus on free Maxwell theory in four space-time dimensions. Integration of the conserved Noether current density $S_f^\mu = (\partial_\nu F^{\nu\mu})f$ over a three-dimensional space-like hypersurface $\Sigma$ to be chosen as the hypersurface $t = $ constant (see Eqn. (91)) yields the expression

$$G_f \equiv -\int_\Sigma d\Sigma_\mu S_f^\mu = -\int_{\mathbb{R}^3} d^3x \, S_f^0 = -\int_{\mathbb{R}^3} d^3x \, (\partial_\nu F^{\nu 0})f \, . \tag{265}$$

Partial integration and substitution of $F^{\nu 0} = \pi^\nu$ (canonical momentum associated to the gauge field $A_\nu$) leads to the result

$$G_f \equiv -\int_\Sigma d\Sigma_\mu S_f^\mu = \int_{\mathbb{R}^3} d^3x \, \pi^\nu(\partial_\nu f) \, . \tag{266}$$

This quantity represents the *generator of Lagrangian gauge transformations in the Hamiltonian formulation of Maxwell theory* (within Dirac's approach to constrained Hamiltonian systems), e.g. see [43] and references therein:

$$\delta_f A_\mu \equiv \{A_\mu, G_f\} = \partial_\mu f \, , \qquad \delta_f \pi^\mu \equiv \{\pi^\mu, G_f\} = 0 \, .$$

**Superpotentials associated to a local symmetry :** Substitution of the Noether identities (260) into relation (259) yields the local conservation law $\partial_\mu(j_f^\mu - S_f^\mu) = 0$ (with $j_f^\mu \equiv -\mathcal{J}_f^\mu$, cf. Eqn. (236)) or $d_{\mathrm{h}}(j_f - S_f) = 0$ for the $(n-1)$-forms $j_f \equiv j_f^\mu d^{n-1}x_\mu$ and $S_f \equiv S_f^\mu d^{n-1}x_\mu$. The algebraic Poincaré lemma (181) thus implies the existence of a $(n-2)$-form $k_f \equiv k_f^{[\mu\nu]}(d^{n-2}x)_{\mu\nu}$ such that

$$j_f = S_f + d_{\mathrm{h}} k_f \tag{267}$$

(In the particular case of Maxwell's theory, this is our equation (235)). This relation states that *the conserved current $(j_f^\mu)$ associated to a local symmetry vanishes weakly up to a superpotential term* and is thus trivial in the sense of Noether's first theorem (as we already noted above in that context).

   The corresponding (on-shell) conserved charge is obtained by integration over a $(n-1)$-dimensional space-like hypersurface $\Sigma$ with boundary $\partial\Sigma$ and by applying Stokes theorem:

$$Q_f \equiv \int_\Sigma j_f = \int_\Sigma \underbrace{S_f}_{\approx 0} + \int_\Sigma d_{\mathrm{h}} k_f \approx \oint_{\partial\Sigma} k_f \, . \tag{268}$$

Sometimes one also says that $\Sigma$ has *codimension* 1 and that $\partial\Sigma$ has codimension 2.

*In summary,* in a pure gauge theory where one does not have any non-trivial global symmetries, one obtains on-shell a surface charge by integration of the current $(j_f^\mu)$ over a Cauchy surface $\Sigma$:

$$Q_f \equiv \int_\Sigma j_f^\mu \, d^{n-1}x_\mu \approx \oint_{\partial\Sigma} k_f^{[\mu\nu]} (d^{n-2}x)_{\mu\nu}. \tag{269}$$

The existence and properties of this flux integral (which may be viewed as a *"lower degree conservation law"*) is determined by the properties of the tensor field $(k_f^{[\mu\nu]})$ (which is a local function of the fields and symmetry parameters) in the vicinity of the hypersurface $\partial\Sigma$. *This flux integral provides an appropriate notion of conserved charge in field theories with local symmetries like YM-theories or general relativity.* Yet, this charge is arbitrary due to the fact that the superpotential $k_f^{[\mu\nu]}$ depending on the arbitrary functions $f$ is arbitrary. Thus, one has to make more precise the relationship between the symmetry parameters $f$ and the $(n-2)$-forms $k_f$ (next subsection) and in particular to *isolate specific parameters $f$ so as to define the charges* (269)*: an appropriate choice is given by the Killing vector fields associated to as fixed background gauge field,* see next subsections.

**Converse of Noether's second theorem and derivation of symmetry transformations:**
Before proceeding further, we note that Eqn. (258), i.e.

$$\delta_f \varphi^a \frac{\delta S}{\delta\varphi^a} = f^r (Q_r^a)^\dagger \left( \frac{\delta S}{\delta\varphi^a} \right) + \partial_\mu S_f^\mu,$$

can be read and applied the other way around. More precisely, suppose we can find some differential operators $(Q_r^a)^\dagger$ (with $Q_r^a(f^r) \equiv Q_r^a f^r + Q_r^{a\mu} \partial_\mu f^r$) which annihilate the equation of motion functions $\frac{\delta S}{\delta\varphi^a}$, i.e. the Noether identities (260) are satisfied. The contraction of $(Q_r^a)^\dagger \left( \frac{\delta S}{\delta\varphi^a} \right)$ with arbitrary functions $x \mapsto f^r(x)$ can then be rewritten under the form $\delta_f \varphi^a \frac{\delta S}{\delta\varphi^a} - \partial_\mu S_f^\mu$: from this expression (which vanishes by virtue of the Noether identities) one can read of the local symmetry transformations $\delta_f \varphi^a$ leaving the action functional $S$ invariant as well as the associated (weakly vanishing) Noether current $(S_f^\mu)$. This line of reasoning has been used by the authors of reference [255] to determine some novel local symmetries of the first order action functional describing gravity as well as of some related action functionals (Holst action and non-minimal coupling of matter fields to gravity) that we outlined at the end of Subsection 5.1.2 above.

Here, we simply illustrate the line of arguments with the example of Maxwell's theory described by the equation of motion function $\partial_\nu F^{\nu\mu}$. Application of $\partial_\mu$ yields the Noether identity $\partial_\mu \partial_\nu F^{\nu\mu} = 0$ by virtue of the antisymmetry of $F^{\nu\mu}$. By applying the Leibniz rule to $\partial_\mu$, the product of $\partial_\mu \partial_\nu F^{\mu\nu}$ with an arbitrary function $x \mapsto f(x)$ can be rewritten as follows:

$$f \, \partial_\mu \partial_\nu F^{\mu\nu} = -f \, \partial_\mu \partial_\nu F^{\nu\mu} = (\partial_\mu f) \underbrace{\partial_\nu F^{\nu\mu}}_{= \frac{\delta S}{\delta A_\mu}} - \partial_\mu (f \underbrace{\partial_\nu F^{\nu\mu}}_{= \frac{\delta S}{\delta A_\mu}}) \equiv \delta_f A_\mu \frac{\delta S}{\delta A_\mu} - \partial_\mu S_f^\mu.$$

Thus one recovers the local gauge transformation of $A_\mu$ from the obvious Noether identity for the dynamical system under consideration.

## 7.4 Conserved forms of lower degree and corresponding Noether theorem

### 7.4.1 Generalized form of Noether's first theorem

In view of relation (268), *the non-trivial conservation laws for forms of lower form degree are determined by the characteristic cohomology group in degree $n-2$:* for the latter, one has a result

of the form Eqn. (246), i.e.

$$H^{n-2}\Big(d_{\mathrm{h}}, \Omega^\bullet(S)\Big) \cong H^{n-2}(d_{\mathrm{h}}|\delta).\tag{270}$$

For $n \geq 3$, the calculation of the cohomology group $H^{n-2}(d_{\mathrm{h}}|\delta)$ leads [74] to the determination of physically distinct **(global) reducibility parameters** which we have mentioned at the end of Subsection 7.1.2, i.e. *equivalence classes* $[f^r]$ *of local gauge parameters:* the equivalence relation is defined by

$$f^r \sim f^r + \underbrace{t^r}_{\approx 0}, \qquad \text{where} \quad \boxed{Q_r^a(f^r) \approx 0\,.}\tag{271}$$

Here, $Q_r^a(f^r) \approx 0$ means that all gauge variations $Q_r^a(f^r)$ of the fields $\varphi^a$ vanish weakly. (These gauge transformations which leave the solutions of the equations of motion invariant are also qualified as *"ineffective gauge transformations"* [154].) The upshot of this line of arguments is known as the

**Generalized form of Noether's first theorem:**   Consider a local field theory which is defined on a space-time manifold of dimension $n \geq 3$ and whose dynamics is described by an action functional $S[\varphi]$. Suppose this action admits continuous global symmetries and possibly, in addition, local symmetries. Then there is a *one-to-one correspondence between non-trivial global reducibility parameters* (i.e. gauge parameters $f^r \not\approx 0$ such that $Q_r^a(f^r) \approx 0$ for all fields $\varphi^a$ on $M$) and $(n-2)$-*forms* $k \not\approx 0$ *which are* $d_{\mathrm{h}}$-*closed on-shell* $(d_{\mathrm{h}}k \approx 0)$, *but not* $d_{\mathrm{h}}$-*exact:*

$$\boxed{[f^r] \longleftrightarrow [k]\,,} \qquad \text{with} \quad Q_r^a(f^r) \approx 0\,, \ \text{and} \ d_{\mathrm{h}}k \approx 0\,.\tag{272}$$

Here, the correspondence $[f^r] \mapsto [k]$ is given by the so-called [158, 249] *Barnich-Brandt procedure* [154, 254] which yields representatives $k_f$ associated to global reducibility parameters $(f^r)$ which are constructed from the Euler-Lagrange derivatives of the Lagrangian describing the dynamics of the theory. To formulate the latter correspondence, we first introduce a convenient mathematical device (see [254] for references to the related mathematical and physical literature).

### 7.4.2   Contracting homotopy operators

As in the case of the ordinary Poincaré lemma for the differential $d = dx^\mu \frac{\partial}{\partial x^\mu}$ acting on $\Omega^\bullet(\mathbb{R}^n)$, a form $\beta$ satisfying $d_{\mathrm{h}}\beta = \alpha$ can be obtained from the $d_{\mathrm{h}}$-closed $(n-1)$-form $\alpha$ by applying the *(contracting) homotopy operator* [31, 226] which generalizes the one of the de Rham complex $(\Omega^\bullet(\mathbb{R}^n), d)$, see Eqn. (A.19) of Appendix A.5 for the definition of the latter. Explicit expressions for the homotopy operators and applications thereof are discussed for instance in appendix A of reference [157] and in [249]. In the following, we summarize the notions that we will need in the sequel.

**General expression:**   Consider an infinitesimal field variation $\delta_Q \varphi^a = Q^a$ or (in terms of the variational bicomplex notation (178)) $d_{\mathrm{v}}q^a = Q^a \in \Omega^{0,1}$. The *contracting homotopy operator of the horizontal bicomplex (with respect to the characteristic* $Q^a)$ has been introduced by I. M. Anderson [226] and is used for instance in the proof of the algebraic Poincaré lemma [74, 154, 249]: one defines the map

$$I_Q^k : \Omega^{k,l} \longrightarrow \Omega^{k-1,l+1}$$
$$\alpha \longmapsto I_Q^k \alpha\,,$$

by

$$I_Q^k \alpha \equiv \sum_{p \geq 0} \frac{p+1}{n-k+p+1} \partial_{\mu_1} \cdots \partial_{\mu_p} \left[ Q^a \frac{\delta}{\delta \varphi_{\mu_1 \cdots \mu_p \nu}^a} \left( \frac{\partial \alpha}{\partial (dx^\nu)} \right) \right], \qquad (273)$$

i.e.

$$I_Q^k \alpha = \frac{1}{n-k+1} Q^a \frac{\delta}{\delta \varphi_\nu^a} \left( \frac{\partial \alpha}{\partial (dx^\nu)} \right) + \frac{2}{n-k+2} \partial_\mu \left[ Q^a \frac{\delta}{\delta \varphi_{\mu\nu}^a} \left( \frac{\partial \alpha}{\partial (dx^\nu)} \right) \right] + \cdots$$

Here, $\varphi_\nu^a \equiv \partial_\nu \varphi^a$, $\varphi_{\mu\nu}^a \equiv \partial_\mu \partial_\nu \varphi^a$, ... Furthermore, the derivatives $\frac{\delta}{\delta \varphi_{\mu_1 \cdots \mu_p}^a}$ represent *(higher order) Euler-Lagrange operators:* the action of the latter on functions $F$ depending at most on second order derivatives of fields is given by

$$\frac{\delta F}{\delta \varphi^a} = \frac{\partial F}{\partial \varphi^a} - \partial_\mu \left( \frac{\partial F}{\partial \varphi_\mu^a} \right) + \partial_\mu \partial_\nu \left( \frac{\partial F}{\partial \varphi_{\mu\nu}^a} \right), \qquad \frac{\delta F}{\delta \varphi_\mu^a} = \frac{\partial F}{\partial \varphi_\mu^a} - 2\partial_\nu \left( \frac{\partial F}{\partial \varphi_{\nu\mu}^a} \right), \qquad \frac{\delta F}{\delta \varphi_{\mu\nu}^a} = \frac{\partial F}{\partial \varphi_{\mu\nu}^a}, \qquad (274)$$

where $\partial / \partial \varphi_{\mu\nu}^a$ represents the symmetrized derivative, i.e. $\frac{\partial \varphi_{\alpha\beta}^b}{\partial \varphi_{\mu\nu}^a} = \frac{1}{2} \delta_a^b (\delta_\alpha^\mu \delta_\beta^\nu + \delta_\alpha^\nu \delta_\beta^\mu)$.

**Application 1 (Derivation of (pre-)symplectic potential):** As a simple application [157], we consider the action of $I_{\delta\varphi}^n$ on a first order Lagrangian $n$-form $\mathcal{L} d^n x \in \Omega^{n,0}$: from $\frac{\partial (d^n x)}{\partial (dx^\nu)} = i_{\partial_\nu}(d^n x) = d^{n-1} x_\nu$ and (274) we conclude that

$$I_{\delta\varphi}^n(\mathcal{L} d^n x) = \delta \varphi^a \frac{\partial \mathcal{L}}{\partial (\partial_\mu \varphi^a)} d^{n-1} x_\mu \equiv j^\mu[\varphi; \delta\varphi] d^{n-1} x_\mu \equiv j[\varphi; \delta\varphi] \in \Omega^{n-1,1}. \qquad (275)$$

Here, we recognize the *(pre-)symplectic potential*, see equation (193).

**Application 2 (Corner term):** An explicit expression for the corner term $\vartheta_\Lambda \in \Omega^{n-2,1}$ appearing in the (pre-) symplectic potential (195) is given by

$$\vartheta_\Lambda = -I_{\delta\varphi}^{n-1} \Lambda, \qquad \text{with} \quad \Lambda \equiv \Lambda^\mu[q] d^{n-1} x_\mu \in \Omega^{n-1,0}. \qquad (276)$$

Indeed, from this expression and (275), i.e. $j_{\mathcal{L} d^n x} = I_{\delta\varphi}^n(\mathcal{L} d^n x)$, as well as the properties of the homotopy operators, it follows that

$$j_{d_h \Lambda} - d_h \vartheta_\Lambda = I_{\delta\varphi}^n d_h \Lambda + d_h I_{\delta\varphi}^{n-1} \Lambda = \delta \Lambda.$$

More precisely, the relative signs appearing in relation (195) are obtained by choosing signs according to $j_{\mathcal{L} d^n x} \equiv -I_{\delta\varphi}^n(\mathcal{L} d^n x)$ and

$$\vartheta_\Lambda \equiv I_{\delta\varphi}^{n-1} \Lambda = \sum_{p \geq 0} \frac{p+1}{p+2} \partial_{\mu_1} \cdots \partial_{\mu_p} \left[ \delta \varphi^a \frac{\delta}{\delta \varphi_{\mu_1 \cdots \mu_p \nu}^a} \left( \frac{\partial \Lambda}{\partial (dx^\nu)} \right) \right]. \qquad (277)$$

With $\frac{\partial \Lambda}{\partial (dx^\nu)} = i_{\partial_\nu} \Lambda$, this result coincides with the one given in appendix A of the first of references [180].

**Application 3 (Derivation of superpotential):**  As a second example [154,157], we consider the action of $I_{\delta\varphi}^{n-1}$ on a $d_{\mathrm{h}}$-closed current form $J_f \in \Omega^{n-1,0}$: the operator (273) is defined so as to ensure that $d_{\mathrm{h}}J_f = 0$ implies that

$$J_f = d_{\mathrm{h}}k_f \,, \tag{278}$$

with

$$\text{superpotential} \qquad \boxed{k_f[\varphi;\delta\varphi] = I_{\delta\varphi}^{n-1}J_f \ \in \Omega^{n-2,1}\,,} \tag{279}$$

this form being defined up to an $d_{\mathrm{h}}$-exact $(n-2)$-form. More specifically, *for $J_f^\mu = S_f^\mu = \frac{\delta S}{\delta\varphi^a}Q_r^{a\mu}f^r$ (see Eqn. (261)) and a first order Lagrangian, the definition (273) yields a simple expression for the superpotential (279):*

$$\boxed{k_f[\varphi;\delta\varphi] = \frac{1}{2}\delta\varphi^a \frac{\delta}{\delta\varphi_\mu^a}\left(\frac{\partial S_f}{\partial(dx^\mu)}\right).} \tag{280}$$

The expressions (279)-(280) depend on the fields $\delta\varphi^a$ of linearized field theory. *An expression depending on the fields of the full interacting theory can be obtained [249,254] by integrating the form (279) along a path $\gamma$ in the space $\mathcal{R}_\infty$* (of solutions of the field equations) which goes from a given field configuration $\bar\varphi$ satisfying $J_f[\bar\varphi] = 0$ to a generic solution $\varphi$:

$$\boxed{K_f[\varphi] \equiv \int_\gamma k_f[\varphi;\delta\varphi].} \tag{281}$$

By using $d_{\mathrm{h}}J_f = 0$ and the properties of the operators $I_Q^k$, we then have

$$d_{\mathrm{h}}K_f[\varphi] = \int_\gamma d_{\mathrm{h}}k_f = \int_\gamma \underbrace{(d_{\mathrm{h}}I_{\delta\varphi}^{n-1} + I_{\delta\varphi}^n d_{\mathrm{h}})J_f}_{=\delta J_f} = J_f[\varphi] - \underbrace{J_f[\bar\varphi]}_{=0} = J_f[\varphi],$$

in accordance with Eqn. (278). Due to the fact that the $(n-2)$-form (281) generically depends on the path which interpolates between the solutions $\bar\varphi$ and $\varphi$ of the field equations [254], the $(n-2)$-form (279) is considered to be more fundamental [249].

Since $k_f[\varphi;\delta\varphi] \in \Omega^{n-2,1}$ is a 1-form in field space, its integral over a closed $(n-2)$-dimensional surface $\partial\Sigma$ (typically a sphere), $\mathchar'26\mkern-12mu d Q_f[\delta\varphi] \equiv \oint_{\partial\Sigma} k_f[\varphi;\delta\varphi]$ yields a *surface charge 1-form:* the properties and algebra of these surface charges are investigated in detail in reference [157]. Here, we only note that they are $d_{\mathrm{h}}$-*closed on-shell* and that they do not depend on the homology class of the closed surface $\partial\Sigma$.

**Homotopy contraction with respect to gauge parameters:**  In the context of gauge field type theories, one also needs the contracting homotopy operator of the horizontal bicomplex with respect to gauge parameters $f \equiv (f^r)$ (e.g. with respect to a diffeomorphism generating vector field $\xi \equiv \xi^\mu\partial_\mu$): it is [157,249] a map

$$I_f^k \ : \ \Omega^{k,l} \ \longrightarrow \ \Omega^{k-1,l}\,,$$

whose expression has the same structure as (273):

$$\boxed{I_f^k\alpha \equiv \sum_{p\geq 0} \frac{p+1}{n-k+p+1}\partial_{\mu_1}\cdots\partial_{\mu_p}\left[f^r\frac{\delta}{\delta f_{\mu_1\cdots\mu_p\nu}^r}\left(\frac{\partial\alpha}{\partial(dx^\nu)}\right)\right].} \tag{282}$$

It enjoys the characteristic property

$$I_f^{k+1} d_{\mathrm{h}} + d_{\mathrm{h}} I_f^k = 1 \,.$$

For later reference, we consider its application to the (pre-)symplectic potential (275): for the case where the gauge parameters are given by a vector field $\xi \equiv \xi^\mu \partial_\mu$ (acting on the fields $\varphi^a$ by virtue of the Lie derivative, $\delta_\xi \varphi^a \equiv L_\xi \varphi^a$), we obtain the

$$\text{Noether-Wald surface charge form}: \qquad Q_\xi[\varphi] \equiv -(I_\xi^{n-1} j)[\varphi; L_\xi \varphi] \in \Omega^{n-2,0} \,. \qquad (283)$$

We will come back to this expression in Subsection 7.6 (Eqn. (306) below).

### 7.4.3 Barnich-Brandt procedure

**Cohomological determination of superpotentials and of surface charges:**  By contracting the *reducibility identity* $\delta_f \varphi^a = Q_r^a(f^r) \approx 0$ (characterizing the global reducibility parameters $f^r$) with $\frac{\delta S}{\delta \varphi^a}$, we get (see Eqn. (261))

$$0 \approx \delta_f \varphi^a \frac{\delta S}{\delta \varphi^a} = \partial_\mu S_f^\mu, \qquad \text{with } S_f^\mu \equiv S_r^{\mu a}\left( \frac{\delta S}{\delta \varphi^a}, f^r \right). \qquad (284)$$

By spelling out the equation of motion term on the right-hand side of relation $Q_r^a(f^r) \approx 0$, one finds that equation (284) can be written as a *divergence identity* [154]:

$$\partial_\mu J_f^\mu = 0, \qquad \text{with } J_f^\mu \equiv S_f^\mu + M^{\mu b a}\left( \frac{\delta S}{\delta \varphi^b}, \frac{\delta S}{\delta \varphi^a} \right). \qquad (285)$$

Since $\partial_\mu J_f^\mu = 0$ is equivalent to the relation $d_{\mathrm{h}} J_f = 0$ for the $(n-1)$-form $J_f \equiv J_f^\mu d^{n-1} x_\mu$ and since the cohomology of the differential $d_{\mathrm{h}}$ is trivial in degree $n-1$ for $n \geq 2$ (due to the algebraic Poincaré lemma (181)), there exists a $(n-2)$-form $k_f \equiv k_f^{[\mu\nu]}(d^{n-2}x)_{\mu\nu}$ such that $J_f = d_{\mathrm{h}} k_f$, i.e. such that

$$\partial_\nu k_f^{[\nu\mu]} = J_f^\mu \approx 0 \,. \qquad (286)$$

Now suppose that the non-trivial gauge transformations $\delta_{\bar f} \varphi^a = Q_r^a(\bar f^r) = Q_r^a \bar f^r + Q_r^{a\mu} \partial_\mu \bar f^r + \dots$ contain only field *in*dependent operators. If the functional derivatives $\delta S/\delta \varphi^a$ are linear and homogenous in the fields, then the reducibility identity $Q_r^a(\bar f^r) = 0$ holds *off*-shell (for which case we put a bar on $f^r$ following [154]) and the expression (285) for $J_{\bar f}^\mu$ reduces to $J_{\bar f}^\mu = S_{\bar f}^\mu$. If, furthermore, $S_{\bar f}^\mu$ contains a most second order derivatives of the fields, then the expression (279) for the superpotential $k_{\bar f}$ given by the homotopy contraction of $J_{\bar f}^\mu$ reduces to a *simple formula which provides an explicit realization of the correspondence* (272):

$$\boxed{\bar f \longmapsto k_{\bar f}^{[\nu\mu]} = \frac{1}{2} \varphi^a \frac{\partial S_{\bar f}^\mu}{\partial \varphi_\nu^a} + \left[ \frac{2}{3} \varphi_\lambda^a - \frac{1}{3} \varphi^a \partial_\lambda \right] \frac{\partial S_{\bar f}^\mu}{\partial \varphi_{\lambda\nu}^a} - (\mu \longleftrightarrow \nu) \,.} \qquad (287)$$

This expression is determined by the Lagrangian and by the on-shell vanishing Noether current $S_{\bar f}^\mu$. Accordingly, the associated surface charges do not depend on total divergences that might be added to the Lagrangian or to the Noether current [157].

**Example of Maxwell's theory:** All assumptions made before equation (287) are satisfied for Maxwell's theory in $\mathbb{R}^n$ (which represents a linear gauge theory). The general solution of $0 = \delta_{\bar{f}} A_\mu \equiv \partial_\mu \bar{f}$ is given by $\bar{f} = \varepsilon = $ const. For $J_{\bar{f}}^\mu = S_{\bar{f}}^\mu = (\partial_\nu F^{\nu\mu})\bar{f} = (\partial_\nu F^{\nu\mu})\varepsilon$ (see Eqn. (233)) we presently have $J_{\bar{f}}^\mu = \partial_\nu k_{\bar{f}}^{[\nu\mu]}$ with a superpotential $k_{\bar{f}}^{[\nu\mu]} = \varepsilon F^{\nu\mu}$ (in agreement with the general expression (287)). For the associated conserved surface charge in $\mathbb{R}^4$ we thus obtain the result (237)-(238) which represents (upon division by $\varepsilon$ or derivation with respect to $\varepsilon$) the usual expression for the electric charge in electrodynamics.

**Examples of YM-theory and of gravity:** For Yang-Mills theory with a non-Abelian structure group and for general relativity (which represent non-linear gauge theories), the identity $Q_r^a(\bar{f}^r) = 0$ defining *exact Killing vectors* (or *exact reducibility parameters*) $(\bar{f}^r)$ reads

$$0 = D_\mu \bar{f} = \partial_\mu \bar{f} + \mathrm{i}g\,[A_\mu, \bar{f}\,], \qquad \text{with} \ \ \bar{f} \equiv \bar{f}^r T_r\,,$$

and

$$0 = (L_{\bar{\xi}}g)_{\mu\nu} = \nabla_\mu \bar{\xi}_\nu + \nabla_\nu \bar{\xi}_\mu\,, \qquad \text{with} \ \ \bar{\xi} \equiv \bar{\xi}^\mu \partial_\mu\,,$$

respectively. Since these equations have to hold for arbitrary gauge fields $(A_\mu)$ and metrics $(g_{\mu\nu})$, the only solutions are the trivial ones $\bar{f}^r = 0$ and $\bar{\xi}^\mu = 0$ [74]. These results reflect the fact that generic gauge field configurations in YM-theories or generic geometries in curved space-time do not admit any symmetries. This obstacle for the definition of surface charges can be overcome in the case where a *background field configuration* having symmetries is given and in the case where the theory is linearized around this configuration (next paragraph) or where the linearized theory describes the full theory asymptotically in the vicinity of some boundary (next subsection): in the latter case, one deals with the corresponding asymptotic symmetries in a general way.

**Examples of linearized YM-theory and gravity:** For YM-theory on $\mathbb{R}^n$ with $n \geq 3$, suppose we are given a background field configuration $(\bar{A}_\mu)$ with which the gauge fields $A_\mu$ coincide for $|\vec{x}| \to \infty$, i.e. the deviation $a_\mu \equiv A_\mu - \bar{A}_\mu$ tends to zero for $|\vec{x}| \to \infty$. Upon *linearization of the theory around this background field,* we then have [74] the gauge variation $\delta_f a_\mu = \bar{D}_\mu f \equiv \partial_\mu f + \mathrm{i}g\,[\bar{A}_\mu, f\,]$ which is independent of $a_\mu$. For a flat background, i.e. $\bar{F}_{\mu\nu} = 0$, the solution of $\bar{D}_\mu f = 0$ and the associated surface charges ("color charges") will be derived below, see equation (301).

Similarly, in general relativity, the *linearization of the metric field around a given geometric background* $(\bar{g}_{\mu\nu})$, i.e. $g_{\mu\nu} = \bar{g}_{\mu\nu} + h_{\mu\nu}$ (where $h_{\mu\nu}$ vanishes asymptotically) leads to the local symmetry transformation $\delta_\xi h_{\mu\nu} = (L_\xi \bar{g})_{\mu\nu}$ which does not depend on $h_{\mu\nu}$. For a flat background $\bar{g}_{\mu\nu} = \eta_{\mu\nu}$, the Killing vector fields $\xi = \xi^\mu \partial_\mu$ (i.e. solutions of $0 = (L_\xi \bar{g})_{\mu\nu} = \partial_\mu \xi_\nu + \partial_\nu \xi_\mu$) are those generating Poincaré transformations, i.e. $\xi^\mu(x) = a^\mu + \varepsilon^{\mu\nu} x_\nu$ with constant real parameters $a^\mu$ and $\varepsilon^{\mu\nu} = -\varepsilon^{\nu\mu}$: the associated surface charges yield the famous ADM charges of linearized gravity [74, 154, 176].

## 7.5 Asymptotic symmetries and asymptotically conserved forms

**Generalities:** Suppose some background fields $\bar{\varphi}^a$ are given and that the deviation $\phi^a \equiv \varphi^a - \bar{\varphi}^a$ of fields $\varphi^a$ from these background fields is small (but not necessarily zero) in a prescribed manner in the *asymptotic region:* For the latter region, one can consider spatial infinity or the boundary of a finite domain like the horizon of a black hole, e.g. see [154] and the review [249] for various examples. (For spatial infinity and a radial variable $r$, one typically assumes an asymptotic behavior of the form $\phi^a \to \mathcal{O}(1/r^{m_a})$ for some number $m_a$ that may

depend on the field under consideration. Analogous assumptions are made for the asymptotic symmetry parameters. In the following we will not spell out these technical details for which we refer to the cited references.) Upon making appropriate general assumptions on the boundary conditions that hold in the asymptotic region, *the linearized theory describes the full theory asymptotically in the vicinity of the boundary.*

Within this general setting, G. Barnich and F. Brandt [154] have extended the correspondence (272) (which is the subject of the generalized form of Noether's first theorem) to the case of asymptotic symmetries and asymptotically conserved $(n-2)$-forms (see also references [155, 157]). The *asymptotic symmetry parameters* $\tilde{f}$ are defined to be field independent gauge parameters $\tilde{f}^r$ for which $\delta_{\tilde{f}} \phi^a$ vanishes asymptotically in a prescribed manner. In this case, the superpotential $\tilde{k}_{\tilde{f}}^{[\nu\mu]}$ associated to the parameters $\tilde{f}$ is still given by relation (287), but with the fields $\varphi^a$ replaced by their deviation $\phi^a$ from the background and with $S_{\tilde{f}}^{\mu}[\varphi]$ replaced by its linearized expression $s_{\tilde{f}}^{\mu}[\phi; \bar{\varphi}]$ (see example below), i.e. we have the *correspondence* [154]

$$\tilde{f} \longmapsto \tilde{k}_{\tilde{f}}^{[\nu\mu]} = \frac{1}{2} \phi^a \frac{\partial s_{\tilde{f}}^{\mu}}{\partial \phi_{\nu}^a} + \left[ \frac{2}{3} \phi_{\lambda}^a - \frac{1}{3} \phi^a \partial_{\lambda} \right] \frac{\partial s_{\tilde{f}}^{\mu}}{\partial \phi_{\lambda\nu}^a} - (\mu \longleftrightarrow \nu). \tag{288}$$

**Summary:** The asymptotic symmetry parameters $\tilde{f}$ represent Killing vector fields associated to a fixed background gauge field and the $(n-2)$-form $\tilde{k}_{\tilde{f}}$ is associated to the free theory obtained by linearization of the full interacting theory around the given background. By construction, $\tilde{k}_{\tilde{f}}^{[\nu\mu]}$ *depends linearly on the deviations* $\delta\varphi^a \equiv \varphi^a - \bar{\varphi}^a = \phi^a$ *(which are small in the asymptotic region) and on their derivatives up to a finite order* (as well as on a finite number of derivatives of the background fields $\bar{\varphi}^a$). As pointed out in reference [254], the integration of $\tilde{k}_{\tilde{f}}$ along a path $\gamma$ in the space of solutions of field equations (see Eqn. (281)) allows us to get, under suitable assumptions, a $(n-2)$-*form* $K_{\tilde{f}}[\varphi]$ *depending on the fields* $\varphi$ *of the full interacting theory.*

**Example of non-Abelian YM-theories:** For YM-theory in $\mathbb{R}^n$ with $n \geq 3$, we can determine [154, 244] *conserved local and integral quantities associated to symmetries of a background gauge field which satisfies the vacuum YM equations.* The conserved quantities are constructed from the gauge Killing vectors of the background gauge field and the conserved charges can be expressed as flux integrals having the same form as those encountered in electrodynamics.

To construct all of these quantities, we start by considering the gauge field configuration $(A^{\mu})$ produced by a source $(j^{\mu})$ which is bounded in space, i.e. $(j^{\mu})$ *vanishes outside of a finite volume.* The YM equation reads $D_{\mu} F^{\mu\nu} = j^{\nu}$ and implies $D_{\nu} j^{\nu} = 0$. Now the gauge field $A_{\mu}$ is decomposed as

$$A_{\mu} = \bar{A}_{\mu} + a_{\mu}, \qquad \text{with } a_{\mu}(t, \vec{x}) \overset{|\vec{x}| \to \infty}{\longrightarrow} 0. \tag{289}$$

Here, $\bar{A}_{\mu}$ is viewed as a *background field corresponding to the source* $j = 0$, i.e.

$$0 = \bar{D}_{\mu} \bar{F}^{\mu\nu} \equiv \partial_{\mu} \bar{F}^{\mu\nu} + \mathrm{i}g[\bar{A}_{\mu}, \bar{F}^{\mu\nu}], \qquad \text{with } \bar{F}_{\mu\nu} \equiv \partial_{\mu} \bar{A}_{\nu} - \partial_{\nu} \bar{A}_{\mu} + \mathrm{i}g[\bar{A}_{\mu}, \bar{A}_{\nu}]. \tag{290}$$

The *deviation* $a_{\mu} = A_{\mu} - \bar{A}_{\mu}$ is small for $|\vec{x}| \to \infty$, but it is not assumed to be small otherwise. For a detailed treatment of the decaying properties of fields and parameters, we refer to the work [154]. Here, we only note that (289) means that the fields $a_{\mu}$ vanish on the spatial $(n-2)$-sphere at infinity, $\partial\mathbb{R}^{n-1} = \lim_{R\to\infty} S_R$: instead of the whole space $\mathbb{R}^{n-1}$ one can consider more

generally an $(n-1)$-dimensional domain $\Sigma \subset \mathbb{R}^{n-1}$ and assume in equation (289) that the fields $a_\mu$ vanish for $\vec{x} \in \partial \Sigma$.

A short calculation shows that we have the expansion

$$F_{\mu\nu} = \bar{F}_{\mu\nu} + f_{\mu\nu} + \mathrm{i} g\,[a_\mu, a_\nu], \qquad \text{with } f_{\mu\nu} \equiv \bar{D}_\mu a_\nu - \bar{D}_\nu a_\mu. \tag{291}$$

The computation that we made for free Maxwell theory in Eqn. (233) can be generalized to the present setting of YM-theory:

$$\mathrm{Tr}\Big[(\delta_f A_\mu)\frac{\delta S}{\delta A_\mu}[A]\Big] = \mathrm{Tr}\big[(D_\mu f)D_\nu F^{\nu\mu}\big] = \partial_\mu \underbrace{\mathrm{Tr}\big[f\,D_\nu F^{\nu\mu}\big]}_{\equiv\, S_f^\mu[A]} - \mathrm{Tr}\big[f\,\underbrace{(D_\mu D_\nu F^{\nu\mu})}_{=\,0}\big]. \tag{292}$$

From $\bar{D}_\nu \bar{F}^{\nu\mu} = 0$ it follows that $D_\nu F^{\nu\mu} = \bar{D}_\nu f^{\nu\mu} + \mathrm{i} g\,[a_\nu, \bar{F}^{\nu\mu}]$ plus terms which are quadratic or of higher order in $a_\nu$. Thus, the linearized expression of $S_{\tilde{f}}^\mu[A] = \mathrm{Tr}\big[\tilde{f}\,D_\nu F^{\nu\mu}\big]$ is given by

$$s_{\tilde{f}}^\mu[a;\bar{A}] = \mathrm{Tr}\Big[\tilde{f}\,\big(\bar{D}_\nu f^{\nu\mu} + \mathrm{i} g\,[a_\nu, \bar{F}^{\nu\mu}]\big)\Big]. \tag{293}$$

Now assume that there are *asymptotic symmetries of the gauge field*, i.e. that there exist infinitesimal gauge transformations parametrized by one or several independent **g**-valued fields $x \mapsto \tilde{f}(x) \equiv \tilde{f}^a(x)T_a \in \mathbf{g}$ with

$$\boxed{0 = \delta_{\tilde{f}}\bar{A}_\mu \equiv \bar{D}_\mu \tilde{f}\,,} \qquad \text{for } \mu \in \{0,1,\dots,n-1\}. \tag{294}$$

In other words, there exist **g**-valued fields $\tilde{f}$ which are *covariantly constant with respect to the background field* $\bar{A}_\mu$. The matrices $\tilde{f}$ (i.e. gauge parameters) are referred to as *gauge Killing vectors* [244, 256] or as *asymptotic reducibility parameters* [154]. (In YM-theories and in general relativity on space-times of dimension $n \geq 3$, they are field-independent and satisfy an off-shell condition [74, 154].)

We note that $\bar{D}_\mu \tilde{f} = 0$ and the general identity $[\bar{D}_\mu, \bar{D}_\nu]\tilde{f} = \mathrm{i} g[\bar{F}_{\mu\nu}, \tilde{f}]$ imply the integrability conditions $[\bar{F}_{\mu\nu}, \tilde{f}] = 0$ for the system of differential equations $\bar{D}_\mu \tilde{f} = 0$. From relation $[\bar{F}_{\mu\nu}, \tilde{f}] = 0$ and the cyclicity of the trace it follows that the second term in expression (293) vanishes. The relation $\bar{D}_\mu \tilde{f} = 0$ and the fact that the trace is a gauge singlet then imply that the current (293) represents a superpotential term:

$$s_{\tilde{f}}^\mu = \partial_\nu \tilde{k}_{\tilde{f}}^{[\nu\mu]}, \qquad \text{with } \boxed{\tilde{k}_{\tilde{f}}^{[\nu\mu]} = \mathrm{Tr}(\tilde{f}\,f^{\nu\mu}).} \tag{295}$$

(This result for $\tilde{k}_{\tilde{f}}^{[\nu\mu]}$ can also [154] be obtained from the general relation (288).)

As a matter of fact, the property $\bar{D}_\mu \tilde{f} = 0$ and the relation $f^{\nu\mu} = \bar{D}^\nu a^\mu - \bar{D}^\mu a^\nu$ can be used to rewrite the superpotential in terms of $a^\mu$:

$$\boxed{\tilde{k}_{\tilde{f}}^{[\nu\mu]} = \partial^\nu \mathcal{A}^\mu - \partial^\mu \mathcal{A}^\nu \equiv \mathcal{F}^{\nu\mu},} \qquad \text{with } \boxed{\mathcal{A}^\mu \equiv \mathrm{Tr}(\tilde{f}\,a^\mu).} \tag{296}$$

This expression for $\mathcal{F}^{\nu\mu}$ in terms of $\mathcal{A}^\mu$ has the same form as the field strength tensor in electrodynamics (i.e. Abelian gauge theory) where $(\mathcal{A}^\mu)$ represents the electromagnetic potentials. In particular, we have $\mathcal{F}_{0i} = \mathrm{Tr}(\tilde{f}\,f_{0i}) \equiv \mathcal{E}_{x^i}$ with $\vec{\mathcal{E}} = -\overrightarrow{\mathrm{grad}}\,\mathcal{A}^0 - \partial_t \vec{\mathcal{A}}$.

To the conserved current $(s_{\tilde{f}}^\mu)$ we can associate an "electric charge", e.g. for $n = 4$, we have the *flux integral* over the 2-sphere at infinity, $\partial \mathbb{R}^3$,

$$Q_{\tilde{f}} \equiv \int_{\mathbb{R}^3} d^3x\, s_{\tilde{f}}^0 = \int_{\mathbb{R}^3} d^3x\, \partial_i \mathrm{Tr}(\tilde{f}\,f^{i0}) = \oint_{\partial \mathbb{R}^3} dS^i\, \mathrm{Tr}(\tilde{f}\,f^{i0}), \tag{297}$$

or

$$Q_{\tilde{f}} = \oint_{\partial \mathbb{R}^3} \vec{\mathcal{E}} \cdot \overrightarrow{dS}, \qquad \text{with} \quad \boxed{\vec{\mathcal{E}} = -\overrightarrow{\text{grad}}\,\mathcal{A}^0 - \partial_t \vec{\mathcal{A}}.} \tag{298}$$

Here, the final result (298) for the conserved charge has the same form as the one in electro-dynamics, see equation (238), but we may presently have several charges $Q_{\tilde{f}}$ namely one for each asymptotic symmetry, i.e. gauge Killing vector $\tilde{f}$ of the background field $\bar{A}$.

**Particular case of an asymptotically flat connection:**   In relation with the structure group $G$ of the theory we use the notation $n_G$ for the dimension of the Lie group $G$ and the symbol $\mathcal{G}$ for the (infinite-dimensional) group of gauge transformations $x \mapsto U(x) \in G$. Furthermore, $\{T_a\}_{a=1,\dots,n_G}$ denotes a basis of the Lie algebra $\mathbf{g}$ of $G$ and $f_{abc}$ the corresponding structure constants, i.e. $[T_a, T_b] = i f_{abc} T_c$.

A particularly important example of background fields in YM-theory is the one where the background field strength vanishes (case of an *asymptotically flat connection*),

$$\bar{F}_{\mu\nu} = 0, \qquad \text{i.e. } \exists U \in \mathcal{G} \,|\, \bar{A}_\mu = -\frac{i}{g} U \partial_\mu U^{-1}. \tag{299}$$

The "gauge Killing equation" $\bar{D}_\mu \tilde{f} = 0$ can then be solved [154, 244] by multiplying it from the left by $U^{-1}$ and from the right by $U$ and by using (299):

$$0 = U^{-1}(\bar{D}_\mu \tilde{f})U = \partial_\mu(U^{-1}\tilde{f}U).$$

Thus, $U^{-1}\tilde{f}U = c^a T_a$ with some constants $c^a$ and thereby we obtain $n_G$ gauge Killing vectors (one for each group generator $T_a$):

$$\forall x \in \mathbb{R}^n : \qquad \tilde{f}_{(a)}(x) \equiv U(x)T_a U^{-1}(x) \in \mathbf{g}. \tag{300}$$

In the present case, the gauge Killing vectors define a representation of the underlying Lie algebra,

$$[\tilde{f}_{(a)}, \tilde{f}_{(b)}] = i f_{abc} \tilde{f}_{(c)},$$

and we have $n_G$ charges $Q_{\tilde{f}_{(a)}} \equiv \oint_{\partial \mathbb{R}^3} \vec{\mathcal{E}}_{(a)} \cdot \overrightarrow{dS}$ with $\vec{\mathcal{E}}_{(a)} = -\overrightarrow{\text{grad}}\,\mathcal{A}^0_{(a)} - \partial_t \vec{\mathcal{A}}_{(a)}$ where $\mathcal{A}^\mu_{(a)} \equiv \text{Tr}(\tilde{f}_{(a)}a^\mu)$.

Using $\bar{F}_{\mu\nu} = 0$ and $a_\mu(t,\vec{x}) \overset{|\vec{x}| \to |\infty}{\longrightarrow} 0$, we conclude from (291) that $F_{\mu\nu} = f_{\mu\nu}$ for $|\vec{x}| \to \infty$. By substituting the latter result and expression (300) into (297) we conclude that the $n_G$ *conserved charges ("color charges")* are given, for $n = 4$, by

$$\boxed{Q_{\tilde{f}_{(a)}} = \oint_{\partial \mathbb{R}^3} dS^i \, \text{Tr}\big[U(x)T_a U^{-1}(x)F^{i0}(x)\big],} \qquad \text{for } a \in \{1,\dots,n_G\}, \tag{301}$$

with $U$ such that (299) holds. The Poisson bracket for the charges (301) again reflects the underlying Lie algebra, $\{Q_{\tilde{f}_{(a)}}, Q_{\tilde{f}_{(b)}}\} = f_{abc} Q_{\tilde{f}_{(c)}}$ and these charges generate gauge transformations [154].

The charges $Q_{\tilde{f}_{(a)}}$ vanish if the components of the field strength tensor decrease faster than $1/r^2$ at spatial infinity (for $n = 4$): this is the case of *instanton* configurations due to the fact that they have a finite action [154]. The electric charge of the so-called *Julia-Zee dyon* is discussed in reference [244]. We note that one can also define *magnetic charges* which are non-zero for non-Abelian *monopole-type* configurations [244], see also reference [257].

**Example of general relativity (with or without a cosmological constant):** The definition and construction of conserved charges in general relativity and the exploration of its asymptotic structure has a long history going back to Einstein's work, e.g. see the recent monograph [22]. The interest in this subject has been revived in the sixties with the work of BMS [245] and in more recent years with the discovery of its connection with the so-called *soft theorems* (related to the infrared structure of quantized gauge theories) and *memory effects* related in particular to gravitational waves and the black hole information paradox, e.g. see [158, 225, 249, 258] for reviews of different aspects. Here, we only mention that G. Barnich and F. Brandt [154] also applied their cohomological procedure (described above for YM-theories) to asymptotically flat and non-flat space-time manifolds while relating the resulting expressions to earlier results (see also [158] and [249] for reviews of this approach). The case of asymptotically flat space-times is of particular interest in that it allows us to make sense of the important notion of an isolated system in curved space [259, 260].

The starting point of the approach of Barnich and Brandt is to decompose the metric $(g_{\mu\nu})$ into a background metric $(\bar{g}_{\mu\nu})$ and a deviation $(h_{\mu\nu})$ from the latter which is asymptotically small:

$$g_{\mu\nu} = \bar{g}_{\mu\nu} + h_{\mu\nu}.$$

The conserved charges are then constructed from the *Killing vector fields of the background metric* $(\bar{g}_{\mu\nu})$ (these vector fields parametrizing the asymptotic symmetries). More precisely, application of (288) to the Noether current $S_\xi^\mu \equiv -\frac{\sqrt{|g|}}{\kappa} \xi_\nu G_c^{\nu\mu}$ determined in Eqn. (264) yields [154, 249] the following superpotential associated to Killing vector fields $\tilde{\xi} \equiv \tilde{\xi}^\mu \partial_\mu$ of the background metric $(\bar{g}_{\mu\nu})$:

$$\tilde{k}_{\tilde{\xi}}^{[\mu\nu]}[g;h] = \frac{\sqrt{|g|}}{\kappa} \Big[ \tilde{\xi}^\nu \nabla^\mu h + \tilde{\xi}^\mu \nabla_\sigma h^{\sigma\nu} + \tilde{\xi}_\sigma \nabla^\nu h^{\sigma\mu} \tag{302}$$
$$+ \frac{1}{2} \big( h \nabla^\nu \tilde{\xi}^\mu + h^{\mu\sigma} \nabla_\sigma \tilde{\xi}^\nu + h^{\nu\sigma} \nabla^\mu \tilde{\xi}_\sigma \big) - (\mu \leftrightarrow \nu) \Big].$$

Here, $h \equiv h^\mu{}_\mu$ and the indices are lowered by $(g_{\mu\nu})$ and raised with $(g^{\mu\nu})$. By integrating the $(n-2)$-form $\tilde{k}_{\tilde{\xi}}[g;h = \delta g]$ along a path in solution space (see Eqn. (281)) one finds [254] the *Komar integrand* (encountered in equation (145) of the covariant phase space approach) plus an extra term which is linear in $\tilde{\xi}^\mu$.

Concerning the choice of the background metric, let us mention an important class of examples involving a cosmological constant. A spherically symmetric matter distribution (of mass $M$ in Newtonian gravity) produces on its outside a gravitational field which is described by the so-called *Schwarzschild-de Sitter metric* [261]: in terms of spherical coordinates (i.e. $(r, \theta, \varphi)$ for $n = 4$), we have

$$ds^2 = \left(1 - \frac{2MG}{r} - \frac{\Lambda}{3} r^2\right) dt^2 - \left(1 - \frac{2MG}{r} - \frac{\Lambda}{3} r^2\right)^{-1} dr^2 - r^2 d\Omega, \tag{303}$$

with $d\Omega \equiv d\theta^2 + \sin^2\theta \, d\varphi^2$ for $n = 4$). This metric represents a solution of the vacuum cosmological Einstein equations $G_c^{\mu\nu} = 0$ (or equivalently $R_{\mu\nu} = \frac{2}{n-2} \Lambda g_{\mu\nu}$). For $\Lambda = 0$, the metric (303) represents the *Schwarzschild solution* describing an asymptotically flat space-time with spherical symmetry. For $M \to 0$ or for large values of $r$, the metric (303) reduces to the *de Sitter metric*

$$ds^2 = \left(1 - \frac{\Lambda}{3} r^2\right) dt^2 - \left(1 - \frac{\Lambda}{3} r^2\right)^{-1} dr^2 - r^2 d\Omega. \tag{304}$$

For $\Lambda > 0$, one speaks about the *de Sitter space* and for $\Lambda < 0$ about the *anti-de Sitter (AdS) space*. These spaces generalize Minkowski space (as a basic solution of the vacuum Einstein equations) to the case where $\Lambda \neq 0$: in the latter instance, the space-time is no longer asymptotically flat. (Anti-) de Sitter space-time has received a lot of attention during the last decades, in particular in cosmology involving a cosmological constant [262] and in relationship with string theory (notably the AdS/CFT correspondence [263]).

## 7.6  Different constructions of lower degree conservation laws

In the following, we survey different methods and their interrelationships. For a chronological presentation and further references to the original literature, we refer to [222] and to the preamble of the memoir [241].

### 7.6.1  Cohomological methods 1: Barnich-Brandt and Wald-Iyer

**Barnich-Brandt procedure:**    This cohomological approach that we described in the previous section is quite general and applies to field theories with internal and/or geometric symmetries like YM-theories or general relativity. The resulting conserved charges are referred to as the *Barnich-Brandt charges* [158]: their explicit expression has been determined for instance for electrodynamics, YM-theories and general relativity [154].

**Wald-Iyer procedure:**    This cohomological method [63–65] which is based on the *covariant phase space approach* does not apply to arbitrary gauge theories, but only to diffeomorphism invariant theories like general relativity. Thus, the gauge parameters $f^r$ are given by the components $\xi^\mu$ of a vector field $\xi \equiv \xi^\mu \partial_\mu$ which is assumed to act on the fields $\varphi^a$ by virtue of the Lie derivative, i.e. $\delta_\xi \varphi^a = L_\xi \varphi^a$, e.g. $\delta_\xi \varphi^a = \xi^\mu \partial_\mu \varphi^a$ for scalar fields $\varphi^a$. Our presentation is based on [249], see also references [157, 158].

The starting point is the first variational formula (82) written in terms of differential forms (see (191) with (183)): for an arbitrary variation of the Lagrangian $n$-form $\mathcal{L} d^n x$ we have[21]

$$\delta(\mathcal{L} d^n x) = \delta \varphi^a \frac{\delta S}{\delta \varphi^a} - d_{\mathrm{h}} j,$$

where the (pre-)symplectic potential form $j[\varphi; \delta \varphi] \in \Omega^{n-1,1}$ is given by (275), i.e. application of the contracting homotopy operator $I^n_{\delta\varphi}$ to $\mathcal{L} d^n x$. The (pre-)symplectic current $J^\mu = \delta j^\mu$ (see equations (84)-(89) and (192)-(193)) is then given by the

(pre-)symplectic current form :      $\boxed{J[\varphi; \delta \varphi; \delta \varphi] \equiv \delta j[\varphi; \delta \varphi; \delta \varphi] \in \Omega^{n-1,2}.}$      (305)

*For asymptotic symmetries generated by a vector field $\xi$,* one uses the contracting homotopy (282) to define (see Eqn. (283)) the

Noether-Wald surface charge form :      $\boxed{Q_\xi[\varphi] \equiv -(I^{n-1}_\xi j)[\varphi; L_\xi \varphi] \in \Omega^{n-2,0}.}$      (306)

Now consider *general relativity* as described by the Einstein-Hilbert action functional[22] $S[g] = \frac{1}{2\kappa} \int_M d^n x \sqrt{|g|} R$ with $\kappa \equiv 8\pi G$ (see Eqn. (262)). For this theory we already de-

---

[21]We recall from (178) that the vertical differential $d_v$ represents an infinitesimal field variation. For convenience, we changed the global (conventional) sign of $j$.

[22]For simplicity, we will write $S[g]$ and $j[g; h]$ for the functional dependence on the fields $g_{\mu\nu}$ and $h_{\mu\nu}$, the distinction with $g \equiv \det(g_{\mu\nu})$ and $h \equiv \det(h_{\mu\nu})$ being clear from the context.

termined the symplectic potential $j$ in our discussion of covariant phase space, see equations (122),(142): from

$$j[g;h] = \frac{\sqrt{|g|}}{2\kappa}\left(\nabla_\nu h^{\nu\mu} - \nabla^\mu h\right)d^{n-1}x_\mu\,, \tag{307}$$

one finds [249] that the Noether-Wald surface charge form (306) writes[23]

$$Q_\xi[g] = \frac{\sqrt{|g|}}{\kappa}(\nabla^\mu\xi^\nu)(d^{n-2}x)_{\mu\nu} = \frac{1}{2}\frac{\sqrt{|g|}}{\kappa}(\nabla^\mu\xi^\nu - \nabla^\nu\xi^\mu)(d^{n-2}x)_{\mu\nu}\,. \tag{308}$$

This expression obviously agrees with the *Komar integrand* which also results from the line of arguments followed in the covariant phase space approach, see equations (144)-(145).

The so-called *Iyer-Wald* $(n-2)$-*superpotential form* associated to asymptotic symmetries in a diffeomorphism invariant field theory is now defined by the following expression [154, 249]:

Iyer-Wald $(n-2)$-superpotential form : $\quad k_\xi^{IW} \equiv -\delta Q_\xi + i_\xi j \in \Omega^{n-2,1}\,, \tag{309}$

or, more explicitly,

$$k_\xi^{IW}[\varphi;\delta\varphi] \equiv -\delta\, Q_\xi[\varphi] + i_\xi j[\varphi;\delta\varphi]\,.$$

Here, $i_\xi j$ is the interior product of the form $j \in \Omega^{n-1,1}$ with respect to the vector field $\xi$ and $\delta$ denotes the variation of forms induced by the field variations $\delta\varphi^a$. The $(n-2)$-form (309) is defined up to a $d_h$-exact term[24] and it involves an extra term $Q_{\delta\xi}$ if the asymptotic Killing vector fields $\xi$ depend on the fields. *For general relativity* (possibly including a cosmological constant), the expressions (307)-(309) lead to [249]

$$k_\xi^{IW}[g;h] = \frac{\sqrt{|g|}}{\kappa}\left[\xi^\mu\nabla_\sigma h^{\sigma\nu} - \xi^\mu\nabla^\nu h + \xi_\sigma\nabla^\nu h^{\mu\sigma} + \frac{1}{2}h\nabla^\nu\xi^\mu - h^{\nu\sigma}\nabla_\sigma\xi^\mu\right](d^{n-2}x)_{\mu\nu}\,. \tag{310}$$

**Comparison of expressions:** The relationship between the superpotentials $k_\xi^{BB}$ of Barnich-Brandt and $k_\xi^{IW}$ of Iyer-Wald has been addressed in references [154, 157, 249]. In this respect one introduces the form [157, 249]

$$E[\varphi;\delta\varphi;\delta\varphi] \equiv -\frac{1}{2}I_{\delta\varphi}^{n-1}j = -\frac{1}{2}I_{\delta\varphi}^{n-1}I_{\delta\varphi}^n(\mathcal{L}\,d^n x) \in \Omega^{n-2,2}\,.$$

Up to an $d_h$-exact $(n-2)$-form, one then has the relation

$$k_\xi^{IW}[\varphi;\delta\varphi] \approx k_\xi^{BB}[\varphi;\delta\varphi] + E[\varphi;\delta\varphi;L_\xi\varphi]\,.$$

E.g. *for general relativity* where $k_\xi^{BB}$ and $k_\xi^{IW}$ are given by (302) and (310), respectively, one has

$$E[g;\delta g;\delta g] = \frac{\sqrt{|g|}}{4\kappa}\delta g^\mu{}_\sigma \wedge \delta g^{\sigma\nu}(d^{n-2}x)_{\mu\nu}\,,$$

hence

$$E[g;\delta g;L_\xi g] = -\frac{\sqrt{|g|}}{2\kappa}\left[\nabla^\mu\xi_\sigma + \nabla_\sigma\xi^\mu\right]\delta g^{\sigma\nu}(d^{n-2}x)_{\mu\nu}\,.$$

---

[23]We will come back to this quantity in equation (350) below. As noted by R. Wald [63], it has been introduced earlier by W. Simon [264] who referred to it as the "gravitational field strength".

[24]Another ambiguity results from the freedom to add a term $d_h\alpha$ to the (pre-)symplectic potential as discussed after Eqn. (194).

We note that the latter expression (introduced in reference [154]) vanishes for Killing vector fields $\xi^\mu \partial_\mu$ as well as for some particular gauge choices for the metric field $(g_{\mu\nu})$, e.g. see reference [249]. The relationships with the covariant phase expressions of B. Julia and S. Silva [205] is discussed in appendix A of [157]. The case of spatially bounded regions was addressed in particular by the authors of reference [265], see also [266, 267]. For the *algebra of charges* (in particular in the presence of gravitational radiation) we refer to [268] (see also [242]).

### 7.6.2 Cohomological methods 2: Antifield formalism

An alternative algebraic approach to the determination of lower-dimensional conservation laws in gauge field-type theories was recently discussed in reference [242]. It relies on the descent equations appearing in the extension of the variational bicomplex [240, 269] which is brought about the introduction of antifields into gauge field theories in view of their quantization along the lines of the Batalin-Vilkovisky (BV) formalism.

**Summary of antifield formalism:** We recall that the BV approach [23, 74, 270–274] amounts to a symplectic-type reformulation and generalization of the usual BRST quantization procedure [275, 276] for which the interplay between Lagrangian symmetries and equations of motion plays a central role. In the following, we outline some of its basic points for the case of an irreducible gauge theory [23] while taking pure YM-theory in four-dimensional Minkowski space-time as an explicit example. Thus, we consider a compact matrix Lie group $G$ and an anti-Hermitian basis $(T_r)$ of the associated Lie algebra $\mathfrak{g}$ satisfying

$$[T_r, T_s] = f_{rst} T_t, \qquad \mathrm{Tr}(T_r T_s) = \delta_{rs}.$$

The basic fields $(\varphi^a)$ are presently given by the components of the connection 1-form $A \equiv A_\mu dx^\mu$ with $A_\mu(x) \equiv A_\mu^r(x) T_r$. The associated curvature 2-form reads $F \equiv dA + \frac{g}{2}[A, A] \equiv \frac{1}{2} F_{\mu\nu} dx^\mu \wedge dx^\nu$ with $F_{\mu\nu}(x) \equiv F_{\mu\nu}^r(x) T_r$ and $g$ denoting the coupling constant of YM-theory. The *action functional* (cf. Eqn. (184))

$$S_{\mathrm{inv}}[A] \equiv \frac{1}{2} \int_M \mathrm{Tr}(F \wedge \star F) = -\frac{1}{4} \int_M d^4x \, \mathrm{Tr}(F^{\mu\nu} F_{\mu\nu}),$$

is invariant under local gauge transformations which are given at the infinitesimal level by $\delta A = Df \equiv df + g[A, f]$ where $x \mapsto f(x) \equiv f^r(x) T_r$ denotes a smooth Lie algebra-valued function.

In the BRST approach to the quantization of gauge field theories, the symmetry parameter $f$ is turned into a so-called *Faddeev-Popov ghost field* $c$ of ghost-number one. For any Lie algebra-valued $p$-form of ghost number $g \in \mathbb{Z}$, i.e. $\alpha_g^p \in \Omega_g^p(M, \mathfrak{g})$, the *total degree* is defined by

$$\widetilde{\alpha_g^p} \equiv p + g.$$

This $\mathbb{Z}$-grading induces a $\mathbb{Z}_2$-*grading* or *Grassmann parity*

$$\mathrm{grading}(\alpha_g^p) \equiv (-1)^{\widetilde{\alpha_g^p}}.$$

The commutator $[\alpha, \beta]$ of two Lie algebra-valued forms $\alpha, \beta$ is now supposed to be graded, i.e.

$$[\alpha, \beta] \equiv \alpha \wedge \beta - (-1)^{\tilde\alpha\tilde\beta} \beta \wedge \alpha.$$

Finally, the *BRST transformations* of $A$ and $c$ are defined by

$$sA = -Dc, \qquad sc = -\frac{1}{2} g[c, c], \tag{311}$$

hence $sA_\mu = D_\mu c$ by virtue of $dc = dx^\mu \partial_\mu c = -(\partial_\mu c) dx^\mu$. The so-defined *BRST operator* $s$ is assumed to anticommute with the exterior derivative $d$ (i.e. their graded commutator $[s, d] = sd + ds$ vanishes) and it acts on the algebra of fields as a graded derivation which increases the ghost-number by one. Its nilpotency, i.e. $s^2 = 0$, reflects the closure of the gauge algebra. Since the action of the $s$-operator on $A$ amounts to an infinitesimal gauge transformation (with symmetry parameter replaced by ghost field), it leaves the action $S_{\text{inv}}[A]$ invariant. In summary, the first transformation law in (311) describes the local symmetries of the action $S_{\text{inv}}[A]$ and the second one reflects the non-trivial (non-Abelian) structure of the gauge algebra.

The gauge fixing procedure (which is considered for obtaining a well-defined propagator for the gauge field) [276] leads to the addition of the *Faddeev-Popov ghost term*

$$\int_M d^4x \operatorname{Tr}(\bar{c} Dc) = -\int_M d^4x \operatorname{Tr}(\bar{c} sA),$$

to the $s$-invariant action $S_{\text{inv}}[A]$. The BV formalism in $n$-dimensional space-time now consist of associating a so-called *antifield* $\Phi^*$ to each field $\Phi \in \{A, c\}$ (i.e. to the basic field $\varphi = A$ as well as to the ghost field $c$) such that $\operatorname{Tr}(\Phi^* s\Phi)$ represents a $n$-form of ghost-number zero. Thus, if $\Phi$ is a $p$-form with ghost-number $g$, i.e. $\operatorname{gh}(\Phi) = g$, then $\Phi^*$ is a $(n-p)$-form with $\operatorname{gh}(\Phi^*) = -(g+1)$. In particular, for YM-theory in four dimensions we have

$$A \in \Omega_0^1(M, \mathfrak{g}), \qquad c \in \Omega_1^0(M, \mathfrak{g}),$$
$$A^* \in \Omega_{-1}^3(M, \mathfrak{g}), \qquad C^* \equiv c^* dx^0 \wedge \cdots \wedge dx^3 \in \Omega_{-2}^4(M, \mathfrak{g}).$$

Accordingly, $A^*$ corresponds (by Hodge duality) to a $\mathfrak{g}$-valued vector field $(A_\mu^*)$ of ghost-number $-1$ and $C^*$ to the $\mathfrak{g}$-valued function $c^*$ of ghost-number $-2$. For gauge field-type theories with *closed* algebras involving structure *constants* (like YM-theories) or field-independent structure operators (like general relativity), the so-called

minimal action
$$\boxed{S[\Phi, \Phi^*] \equiv S_{\text{inv}}[A] + \sum_\Phi \int_M \operatorname{Tr}(\Phi^* s\Phi) \equiv \int_M d^4x \, \mathcal{L},} \tag{312}$$

i.e. for YM-theories

$$\mathcal{L} = \operatorname{Tr}\left[-\frac{1}{4} F^{\mu\nu} F_{\mu\nu} + A_\mu^* D^\mu c + c^*\left(-\frac{g}{2}[c, c]\right)\right], \tag{313}$$

is a solution of the *(non-linear) Slavnov-Taylor identity* or

BV master equation
$$0 = \frac{1}{2}[[S, S]] = \sum_\Phi \int_M \frac{\delta S}{\delta \Phi^*} \frac{\delta S}{\delta \Phi}. \tag{314}$$

Here, the *BV bracket* (or *antibracket*) of two functionals $X, Y$ of the variables $\Phi, \Phi^*$ (viewed as differential forms on $M$) is the graded bracket defined as follows in $n$ dimensions:[25]

BV bracket
$$\boxed{[[X, Y]] = \sum_\Phi \int_M \left[(-1)^{\tilde{X}\widetilde{\Phi^*}} \frac{\delta X}{\delta \Phi^*} \frac{\delta Y}{\delta \Phi} + (-1)^{\tilde{X}\tilde{\Phi} + n(\tilde{\Phi}+1)} \frac{\delta X}{\delta \Phi} \frac{\delta Y}{\delta \Phi^*}\right].} \tag{315}$$

---

[25]All of our functional derivatives are left derivatives [272], but we note that the BV bracket is traditionally expressed in terms of both left and right derivatives which implies the presence of other sign factors.

We remark that the grading of the functional derivative $\frac{\delta X}{\delta \Phi}$ is given by $(-1)^{\tilde{X}+\tilde{\Phi}+n}$ whence the last equality in Eqn. (314). From (315) we deduce in particular that $(\Phi, \Phi^*)$ can be viewed as a conjugate pair with respect to the BV bracket:

$$[\![\Phi^*(x), \Phi(y)]\!] = (-1)^{\widetilde{\Phi^*}} \delta(x-y). \tag{316}$$

The BV bracket $(X, Y) \mapsto [\![X, Y]\!]$ on the graded vector space of functionals $(\Phi, \Phi^*) \mapsto F[\Phi, \Phi^*]$ (graded by the ghost-number) represents a *graded Poisson structure of degree one,* i.e. we have the homogeneity or

$$\text{grading property} \qquad \text{gh}[\![X, Y]\!] = \text{gh}(X) + \text{gh}(Y) + 1,$$

as well as the properties of $\mathbb{R}$-bilinearity, graded antisymmetry, graded Jacobi identity and graded derivation (Leibniz) rule.

From (315) we can also infer that the *"linearized Slavnov-Taylor operator"* $\mathcal{S}_S$ *associated to the minimal action $S$,* i.e. the infinitesimal canonical transformation induced by $S$ and the BV bracket,

$$\mathcal{S}_S \equiv [\![S, \cdot]\!], \tag{317}$$

acts on fields and antifields (in four dimensions) by the

$$\boxed{\text{BRST transformations} \qquad \mathcal{S}_S \Phi = \frac{\delta S}{\delta \Phi^*} = s\Phi, \qquad \mathcal{S}_S \Phi^* = \frac{\delta S}{\delta \Phi} \equiv s\Phi^*.} \tag{318}$$

Here, the last equality of the first relation follows from the expression (312) of the minimal action $S$ whereas the last equality in the second relation amounts to the extension of the BRST operator $s$ from fields (as defined by Eqn. (311)) to antifields. *The operator $\mathcal{S}_S$ raises the ghost-number by one unit, it is nilpotent and it leaves the minimal action $S$ invariant.* (The nilpotency of $\mathcal{S}_S$ readily follows from the graded Jacobi identity satisfied by the BV bracket and from the BV master equation (314).) If all antifields are set to zero (along with their $s$-variations), then $S[\Phi, \Phi^*]$ reduces to $S_{\text{inv}}[A]$ and we recover the initial BRST transformations (311) together with the equation of motion of the basic field $A$.

The ghosts and antifields are incorporated into the variational bicomplex formulation by extending the double complex $(\Omega^{\bullet,\bullet}(J^\infty E), d_{\text{h}}, d_{\text{v}})$ to the

$$\boxed{\text{variational tricomplex} \qquad \left(\Omega^{\bullet,\bullet}_\bullet, d_{\text{h}}, d_{\text{v}}, \delta_Q\right).} \tag{319}$$

Here, $\Omega^{\bullet,\bullet}_\bullet$ is a short-hand notation for $\Omega^{\bullet,\bullet}_\bullet(J^\infty E)$ where the field space $E$ is now assumed to be $\mathbb{Z}$-graded by the ghost-number. The operator $\delta_Q$ represents a further differential (anticommuting with both $d_{\text{h}}$ and $d_{\text{v}}$) to be identified with the BRST operator [240, 242, 269] which increases the ghost-number by one unit.

The BV bracket is a (non-degenerate) graded Poisson bracket and thus corresponds to a *graded symplectic form*

$$\Omega \equiv \int_M \omega, \tag{320}$$

where the local form $\omega \in \Omega^{n,2}_{-1}$ has the expression (cf. Eqn. (316))

$$\boxed{\omega = \sum_\Phi \text{Tr}[\delta\Phi \wedge \delta\Phi^*].} \tag{321}$$

For four-dimensional YM-theory, we obtain (with $C \equiv c$)

$$\omega = \text{Tr}\left[\delta A \wedge \delta A^* + \delta C \wedge \delta C^*\right], \tag{322}$$

i.e.

$$\omega = \hat{\omega}\, d^4 x\,, \qquad \text{with} \quad \hat{\omega} = \text{Tr}\left[\delta A^\mu \wedge \delta A_\mu^* + \delta c \wedge \delta c^*\right]\,, \quad \text{gh}(\hat{\omega}) = -1\,. \tag{323}$$

We note [248] that, from a geometric point of view, the fields $\Phi$ and antifields $\Phi^*$ can be viewed, respectively, as base and fibre coordinates of the *odd cotangent bundle* $T^*[-1]\mathcal{E}$: here, $\mathcal{E}$ denotes the graded (by the ghost-number) vector space of fields $x \mapsto \Phi(x) \in E$ on space-time and $-1$ represents the shift $g \rightsquigarrow -(g+1)$ of ghost-number which occurs upon passage from fields to antifields. With this geometric interpretation, the two-form (320) defined by Eqn. (321) represents the *canonical shifted symplectic structure on* $T^*[-1]\mathcal{E}$.

**Application of antifield formalism:**  We will now make contact with the local symplectic form $J = J^\mu\, d^3 x_\mu$ encountered in the covariant phase space approach. Application of the first variational formula (82) to the minimal Lagrangian (313) yields

$$\delta \mathcal{L} \approx \partial_\mu j^\mu\,, \qquad \text{with} \quad j^\mu = \text{Tr}\left[-F^{\mu\nu}\delta A_\nu + A^{*\mu}\delta c\right]\,, \tag{324}$$

hence

$$\boxed{J^\mu \equiv -\delta j^\mu = \text{Tr}\left[\delta A_\nu \wedge \delta F^{\nu\mu} + \delta c \wedge \delta A^{*\mu}\right]\,.} \tag{325}$$

The fact that this expression extends the one obtained for YM-theory without antifields (see (97)) is due to the "boundary condition" $\mathcal{L}(\Phi, \Phi^* = 0) = \mathcal{L}_{\text{inv}}(A)$. Equivalently [242, 269], the *symplectic current density* ($J^\mu$) can be extracted from the graded symplectic structure (323) corresponding to the BV bracket by application of the BRST differential $\delta_Q$: with (318) we have

$$\delta_Q A_\mu = D_\mu c\,, \qquad\qquad \delta_Q c = -\frac{g}{2}\,[c,c]\,, \tag{326}$$

$$\delta_Q A^{*\mu} = \frac{\delta S}{\delta A_\mu} = D_\nu F^{\nu\mu} - g\,[A^{*\mu},c]\,, \qquad \delta_Q c^* = \frac{\delta S}{\delta c} = D^\mu A_\mu^* + g\,[c^*,c]\,,$$

and a short calculation leads to the result

$$\delta_Q \hat{\omega} = -\partial_\mu J^\mu\,, \qquad \text{with} \quad J^\mu \equiv \text{Tr}\left[\delta A_\nu \wedge \delta F^{\nu\mu} + \delta c \wedge \delta A^{*\mu}\right]\,, \qquad \text{gh}(J^\mu) = 0\,,$$

$$\delta_Q J^\mu = -\partial_\nu J^{\nu\mu}\,, \qquad \text{with} \quad J^{\nu\mu} \equiv \text{Tr}\left[\delta c \wedge \delta F^{\nu\mu}\right]\,, \qquad\qquad \text{gh}(J^{\nu\mu}) = 1\,,$$

$$\delta_Q J^{\nu\mu} = 0\,.$$

These expressions can also be written in terms of differential forms [242] by starting from the 4-form (322):

$$\delta_Q \omega = d_{\text{h}}\omega_1\,, \qquad \text{with} \quad \omega_1 \equiv \text{Tr}\left[\delta A \wedge \delta \star F + \delta C \wedge \delta A^*\right] \in \Omega_0^{3,2}\,,$$

$$\delta_Q \omega_1 = d_{\text{h}}\omega_2\,, \qquad \text{with} \quad \omega_2 \equiv \text{Tr}\left[\delta C \wedge \delta \star F\right] \in \Omega_1^{2,2}\,,$$

$$\delta_Q \omega_2 = 0\,. \tag{327}$$

Thus, we have *descent equations* (analogous to the Stora-Zumino descent equations in the BRST quantization of gauge field theories [276]) and *the local symplectic form $\omega_1$ of the covariant phase space approach* (extended with antifields) *represents the first descendant of the local symplectic form $\omega$ corresponding to the BV bracket.*

To recover the color charges of YM-theory [242], one makes an expansion of the minimal Lagrangian $\mathcal{L}$ and of the BRST operator $\delta_Q$ with respect to the coupling constant $g$ viewed as a deformation parameter of the free (Abelian) field theory: we have

$$\mathcal{L} = \mathcal{L}_0 + g\mathcal{L}_1 + g^2\mathcal{L}_2\,, \qquad \text{with} \quad \mathcal{L}_0 = \text{Tr}\Big[-\frac{1}{4}\,\overset{\circ}{F}{}^{\mu\nu}\overset{\circ}{F}_{\mu\nu} + A^*_\mu\partial^\mu c\Big] \quad (\overset{\circ}{F}_{\mu\nu}\equiv\partial_\mu A_\nu - \partial_\nu A_\mu)\,, \tag{328}$$

and

$$Q = Q_0 + gQ_1 + g^2Q_2\,, \qquad \text{with} \quad \begin{cases} \delta_{Q_0}A_\mu = \partial_\mu c\,, & \delta_{Q_0}A^{*\mu} = \Big(\frac{\delta S}{\delta A_\mu}\Big)_0 = \partial_\nu\overset{\circ}{F}{}^{\nu\mu}\,, \\ \delta_{Q_0}c = 0\,, & \delta_{Q_0}c^* = \partial^\mu A^*_\mu\,. \end{cases} \tag{329}$$

Thus, $\mathcal{L}_0$ describes a collection $(A^r_\mu)$ of *free Abelian gauge fields*. The Lagrangian $\mathcal{L}_0$ is invariant under the

$$\text{shift symmetry} \qquad \delta c = \epsilon f\,, \tag{330}$$

where $\epsilon$ denotes a constant parameter of ghost-number one and $f \in \mathfrak{g}$ does not depend on space-time coordinates. By virtue of Noether's first theorem, we thus have a

$$\text{conserved current density} \qquad \mathcal{J}^f_\mu \equiv \text{Tr}(fA^*_\mu)\,, \quad \text{gh}(\mathcal{J}^f_\mu) = -1\,, \quad \partial^\mu\mathcal{J}^f_\mu \approx 0\,,$$

where the local conservation law holds thanks to the equation of motion following from $\mathcal{L}_0$. The free BRST-variation of the current density $(\mathcal{J}^f_\mu)$ yields a

$$\text{descendant current density:} \qquad \delta_{Q_0}\mathcal{J}^f_\mu = \partial^\nu\mathcal{J}^f_{\nu\mu}\,, \quad \text{with} \quad \mathcal{J}^f_{\nu\mu} \equiv \text{Tr}(f\overset{\circ}{F}_{\nu\mu})\,, \quad \text{gh}(\mathcal{J}^f_{\nu\mu}) = 0\,. \tag{331}$$

The latter is again conserved by virtue of the free equation of motion (i.e. $\partial^\nu\mathcal{J}^f_{\nu\mu} \approx 0$) and it is invariant under the free BRST differential $Q_0$ (which acts on fields and antifields according to Eqn. (329)). Since its ghost-number vanishes, the integral

$$\boxed{\mathcal{Q}^f \equiv \oint_S dS^i\,\mathcal{J}^f_{i0} = \oint_S dS^i\,\text{Tr}(f\overset{\circ}{F}_{i0})\,.} \tag{332}$$

can be interpreted [242] as the *color charge* enclosed by the surface $S = \partial V$ with $V \subset \mathbb{R}^3$. Clearly it has the structure that we encountered in Eqn. (297) for the case of a vanishing background field. The algebra of these charges is investigated in reference [242] which also applies the outlined approach to general relativity as well as unimodular gravity.

To conclude we note that, for an $n$-dimensional space-time manifold $M$ with boundary $\partial M$, the local symplectic form $\omega \in \Omega^{n,2}_{-1}$ given by (321) can be integrated over the manifold $M$ (see Eqn. (320)) and that its descendant $\omega_1 \in \Omega^{n-1,2}_0$ can be integrated over the (codimension 1) boundary $\partial M$. In fact, $\Omega_1 \equiv \oint_{\partial M}\omega_1$ represents the canonical symplectic form on the space of boundary fields [248]. More generally, the second descendant $\omega_2 \in \Omega^{n-2,2}_1$ can be integrated over a (codimension 2) corner $K$ (i.e. the boundary of boundary components) and so on for large values of $n$. As a matter of fact, the Hamiltonian version of the BV approach to field theories which is known as the *BFV (Batalin-Fradkin-Vilkovisky) formalism* is naturally associated to the boundary of $M$ (if $M$ has such a boundary) and the descent equations encountered above play an important role in the *BV-BFV construction* in that they relate the BV data

(associated with the bulk) to the BFV data associated to the boundary and, more generally, they relate the strata (submanifolds) of codimensions $k$ and $k + 1$, see [248] and references therein, notably [121]. As a matter of fact, the authors of reference [248] investigate in detail the case of a complex matter field which is minimally coupled to an Abelian gauge field (i.e. scalar electrodynamics) and they extend the BV-BFV construction to non-compact manifolds and appropriate asymptotic conditions for fields. In this context, the symmetries and more precisely the fall-off properties of gauge parameters also have to be dealt with with care.

### 7.6.3 Abbott-Deser approach

For non-Abelian YM-theory in flat space-time, the starting point of L. F. Abbott and S. Deser [244] is based on the decomposition $A_\mu = \bar{A}_\mu + a_\mu$ of the YM field $A_\mu$, see equations (289)-(291). The relationship between the cohomological method of Barnich and Brandt (that we considered after equation (291)) and the approach of Abbott and Deser is established by introducing an effective current density $j_{\text{eff}}^\mu$ in the decomposition of the YM equation $D_\mu F^{\mu\nu} = j^\nu$:

$$\bar{D}_\mu f^{\mu\nu} + \mathrm{i} g \left[ a_\mu, \bar{F}^{\mu\nu} \right] = j_{\text{eff}}^\nu, \qquad \text{with} \ \ j_{\text{eff}}^\nu \equiv j^\nu - (D_\mu F^{\mu\nu})_{\text{N}} .$$

Here, $(D_\mu F^{\mu\nu})_{\text{N}}$ only involves terms which are non-linear (quadratic or of higher order) in the deviation $a_\mu = A_\mu - \bar{A}_\mu$. The conserved "source current" $V_{\tilde{f}}^\mu[a; \bar{A}] \equiv \mathrm{Tr}(\tilde{f} j_{\text{eff}}^\mu)$ (where $\tilde{f}(x) \equiv \tilde{f}^a(x) T_a \in \mathfrak{g}$ denotes asymptotic reducibility parameters) can then be rewritten as

$$V_{\tilde{f}}^\mu = -\partial_\nu \tilde{k}_{\tilde{f}}^{[\nu\mu]}, \qquad \text{with} \ \ \tilde{k}_{\tilde{f}}^{[\nu\mu]} \equiv \mathrm{Tr}(\tilde{f} f^{\nu\mu}),$$

see equations (295)-(296).

   Thus, as emphasized in reference [154], the approaches of Abbott-Deser and of Barnich-Brandt are closely related. Indeed, the first starts from effective sources from which superpotentials are derived whereas the second concentrates right away on the superpotentials: thereby it allows for a more direct construction and control of the resulting surface charges. Besides considering YM-theories, Abbott and Deser also constructed conserved charges in general relativity for asymptotically anti-de Sitter space-times [243] (actually this was their initial concern and investigation). In the sequel, the procedure was also applied to YM-theory in curved space-time [277] as well as to *higher curvature gravity theories* [278].

### 7.6.4 Hamiltonian formulation

As discussed in references [154, 157], the surface charges $Q_f \equiv \oint_{\partial \Sigma} k_f^{[\mu\nu]}(d^{n-2}x)_{\mu\nu}$ (associated to asymptotic symmetries) constructed by the procedure of Barnich and Brandt can be directly related to the ones obtained from the (non-manifestly covariant) Hamiltonian approach. For the comparison with the Hamiltonian expressions one considers the first order (i.e. Hamiltonian) form of the action functional associated to the Lagrangian. We note that surface charges have originally been introduced in the canonical framework for general relativity on asymptotically flat space-times in the work of ADM [175] and that the systematic construction of asymptotic conservation laws in this setting has been addressed thereafter in reference [279] (see also [280] and [281, 282] for the general theory and algebra of charges).

### 7.6.5 Other approaches

As we already mentioned in Subsection 5.4, the derivation of differential or integral expressions for conserved quantities in field theories with local symmetries dates back to the early days of general relativity. In addition to the reviews listed in Subsection 5.4, we mention the overviews given in references [241, 256, 283]. Besides the approaches discussed so far we

quote a few other ones here that have been considered in the Lagrangian framework: the so-called energy-momentum *pseudo-tensors* (the best known one being the one of Landau and Lifschitz [284]), the *Komar integral* [184] that we encountered in equations (145) and (308), the *Lagrangian Noether method* [206, 285–288] *quasi-local methods* [12, 64, 185, 289] which have been initiated by R. Penrose [290] and in the work of J. D. Brown and J. W. York [291] and which amount to define quantities with respect to a bounded region of space-time, *conformal methods* (exploring the asymptotic structure of space-time at infinity while adding a suitable conformal boundary to physical space-time) which are based on the pioneering work of BMS [245] and of R. Penrose [292] (see also [12, 259, 293–297] as well as the reviews [158, 223, 298, 299]), the *spinorial definition of energy* [12, 300–302], notions of energy related to *radiation* (like the one of the Bondi mass), approaches based on Hamilton-Jacobi analysis, the so-called *dressing field method* [303],...

### 7.6.6 On the relationship between the different methods and results

S. Hollands, A. Ishibashi and D. Marolf [304] compared different methods that apply to AdS space-times. We also mentioned a certain number of comparisons and we will indicate some others for the specific case of gravity in the next subsection, in particular [180, 194]. However, as pointed out for instance in reference [305] a systematic and more complete comparison of methods and the corresponding results is currently lacking.

## 7.7 Gravitational (or diffeomorphism) charges following Wald et al.

In this subsection (which is based on the work of R. Wald and his collaborators [60]- [65]) we discuss the derivation of *conserved charges associated to diffeomorphism invariance* in terms of the symplectic potential form on covariant phase space. Accordingly we adopt some of the notation of reference [65] (which is nowadays used in most of the related literature) while considering the recently given geometric reformulation of this work [181, 216].

Moreover, we will point out the close relationship of some of these considerations with various results presented earlier in these notes, in particular with those of C. Crnkovic [166] which we outlined in equations (141)-(148).

Over the last twenty years, the investigations of R. Wald and his collaborators have been further elaborated and generalized in different respects by numerous authors, e.g. see [19, 154, 157, 174, 180–182, 268, 306–311] and references therein. We will mention some of this work towards the end of this section and in the subsequent ones.

**Notation and general framework:**  We start from an $n$-dimensional space-time manifold $(M, g)$ admitting a foliation by space-like hypersurfaces, e.g. a globally hyperbolic space-time. Following R. Wald and A. Zoupas [65] we consider a collection of fields $\varphi \equiv (\varphi^a)$ which consists of the metric field and possibly some other tensor fields on $M$. By $\mathcal{F}$ we denote the *space of fields $\varphi$ satisfying some given asymptotic conditions,* e.g. asymptotic flatness of the metric and vanishing of the matter fields at spatial infinity (or at null infinity) in general relativity. As a general rule, the decay rates of fields at infinity are assumed to be strong enough to ensure that the considered integrals exist. The *covariant phase space* $\bar{\mathcal{F}}$ is then given by the fields $\varphi \in \mathcal{F}$ which solve the field equations determined by a given Lagrangian $n$-form **L**.

The covariant derivative of tensor fields with respect to the Levi-Civita connection is written as $\nabla_\mu$ and we again assume for simplicity that **L** is of first order. We note that for a set of fields $\varphi^a$ described by a Lagrangian density $\mathcal{L}$, the *Euler-Lagrange equations* locally read [186]

$$\nabla_\mu \left( \frac{\partial \mathcal{L}}{\partial (\nabla_\mu \varphi^a)} \right) - \frac{\partial \mathcal{L}}{\partial \varphi^a} = 0.$$

Differential forms on $M$ (like the Lagrangian $n$-form $\mathbf{L}$) are denoted by boldface letters.

We presently write d for the horizontal differential $d_\mathrm{h}$ and $\delta$ for the vertical differential $d_\mathrm{v}$. More precisely (cf. Appendix A), for a smooth function (0-form) on the space-time manifold $M$,

$$
\begin{aligned}
f : M &\longrightarrow \mathbb{R} \\
x \equiv (x^\mu) &\longmapsto f(x),
\end{aligned}
\tag{333}
$$

the *exterior derivative* d is defined by $df \equiv dx^\mu \partial_\mu f$, i.e. $df|_x \in T_x^* M$ (cotangent space of $M$ at $x$). For a vector field $\xi = \xi^\mu \partial_\mu$ on $M$, we have $\xi|_x \in T_x M$ (tangent space of $M$ at $x$). By definition, the *interior product* $i_\xi$ (with respect to $\xi$) acts on differential forms on $M$ as a graded derivation lowering the form degree by one, its action on a 0-form $f$ and on the 1-form $dx^\mu$ being given by

$$
i_\xi f \equiv 0, \qquad i_\xi dx^\mu \equiv \xi^\mu, \qquad \text{hence} \quad i_\xi df = \xi^\mu \partial_\mu f.
$$

The *Lie derivative* $L_\xi$ (with respect to $\xi$) acts on differential forms on $M$ as a derivation by virtue of Cartan's "magic formula"

$$
L_\xi \equiv [i_\xi, d] \equiv i_\xi d + d i_\xi, \qquad \text{hence} \quad L_\xi f = i_\xi df = \xi^\mu \partial_\mu f.
$$

The generalization of these notions to the (infinite-dimensional) field space $\mathcal{F}$ proceeds as follows, cf. Appendix D.2 and Section 5.1 or references [216, 312]. For a smooth functional (0-form) on $\mathcal{F}$,

$$
\begin{aligned}
F : \mathcal{F} &\longrightarrow \mathbb{R} \\
\varphi \equiv (\varphi^a) &\longmapsto F[\varphi],
\end{aligned}
\tag{334}
$$

the *differential (field variation)* $\delta$ is defined by $\delta F \equiv \int_M d^n x \, \delta \varphi^a(x) \frac{\delta F}{\delta \varphi^a(x)}$, i.e. $\delta F|_\varphi \in T_\varphi^* \mathcal{F}$. Since we are presently concerned with diffeomorphism invariant theories on $M$ and since diffeomorphisms on $M$ are generated by vector fields $\xi$ on $M$, we have to deal with the induced action of $\xi$ on fields and forms as described by the *vector field $X_\xi$ (associated to $\xi$)*. This vector field writes $X_\xi \equiv \int_M d^n x \, X_\xi^a(\varphi(x)) \frac{\delta}{\delta \varphi^a(x)}$ and we have $X_\xi|_\varphi \in T_\varphi \mathcal{F}$. The *interior product* $I_{X_\xi} \equiv I_\xi$ (with respect to the vector field $X_\xi$) acts on differential forms on $\mathcal{F}$ as a graded derivation of degree $-1$, its action on a 0-form $F$ and on the 1-form $\delta \varphi^a$ being defined by

$$
I_\xi F \equiv 0, \qquad I_\xi \delta \varphi^a \equiv X_\xi^a(\varphi) = \delta_\xi \varphi^a, \qquad \text{hence} \quad I_\xi \delta F = \int_M d^n x \, \delta_\xi \varphi^a(x) \frac{\delta F}{\delta \varphi^a(x)} = \delta_\xi F.
\tag{335}
$$

The *Lie derivative* $L_{X_\xi}$ (with respect to $X_\xi$) acts on differential forms on $\mathcal{F}$ by

$$
L_{X_\xi} \equiv [I_\xi, \delta] \equiv I_\xi \delta + \delta I_\xi, \qquad \text{hence} \quad L_{X_\xi} F = I_\xi \delta F = \delta_\xi F,
\tag{336}
$$

for 0-forms $F$ on $\mathcal{F}$.

All of the linear operators that we just introduced act on $(k,l)$-*forms* $\alpha$ on the infinite jet-bundle $J^\infty(E)$ (a section $s : M \to E$ of the field bundle $E$ over $M$ being given by $s(x) = (x, \varphi(x))$) and we write $\alpha \in \Omega^{k,l}$, see Subsection 6.1. Thus, we have for instance $dx^\mu \in \Omega^{1,0}$, $\delta \varphi^a \in \Omega^{0,1}$, $\mathbf{L} \in \Omega^{n,0}$ and

$$
\begin{aligned}
d &: \Omega^{k,l} \longrightarrow \Omega^{k+1,l}, & i_\xi &: \Omega^{k,l} \longrightarrow \Omega^{k-1,l}, \\
\delta &: \Omega^{k,l} \longrightarrow \Omega^{k,l+1}, & I_\xi &: \Omega^{k,l} \longrightarrow \Omega^{k,l-1}.
\end{aligned}
\tag{337}
$$

Following the conventions which are used in most of the literature based on the seminal work of Wald et al., we assume that the operators $d, i_\xi, \dots$ acting on space-time commute with the operators $\delta, I_\xi, \dots$ acting on field space (which is consistent with the fact that the local forms $\alpha$ are characterized by a bidegree $(k,l)$ on which the different operators act separately, see Eqn. (337)).

**Symplectic potential current and symplectic 2-form:** A generic field variation $\varphi \rightsquigarrow \varphi + \delta\varphi$ induces the following variation of $\mathbf{L}$ (*first variational formula*), cf. Eqn. (82) and equations (191), (193):

$$\delta\mathbf{L} = \mathbf{E}_\varphi \, \delta\varphi + \mathrm{d}\boldsymbol{\theta} \,, \tag{338}$$

with

$$\boxed{\boldsymbol{\theta} \equiv \boldsymbol{\theta}(\varphi; \delta\varphi) \equiv \frac{\partial\mathbf{L}}{\partial(\nabla_\mu\varphi^a)} \, \delta\varphi^a \wedge \mathrm{d}^{n-1}x_\mu \,.} \tag{339}$$

As we already pointed out in Section 6.4, the definition of the *symplectic potential current form* $\boldsymbol{\theta} \in \Omega^{n-1,1}$ is only determined by the first variational formula up to the addition of an exact form. This ambiguity can be, and will be fixed for now [174, 181], by considering the customary expression (339) for $\boldsymbol{\theta}$ (which is obtained by using the Leibniz rule to evaluate $\delta\mathbf{L}$): this expression also results from the Lagrangian $n$-form $\mathbf{L}$ by application of the contracting homotopy operator of the variational bicomplex (see Eqn. (275)) and it is referred to as the *'bare' choice.*

The field variation of the $(n-1)$-form $\boldsymbol{\theta}$ defines[26] the

symplectic current $(n-1)$-form $\boldsymbol{\omega} \equiv \delta\boldsymbol{\theta}$: $\boxed{\boldsymbol{\omega}(\varphi; \delta_1\varphi, \delta_2\varphi) = \delta_1\boldsymbol{\theta}(\varphi; \delta_2\varphi) - \delta_2\boldsymbol{\theta}(\varphi; \delta_1\varphi) \,.}$
$$\tag{340}$$

We adopt the convention that

$$\delta\varphi^a \wedge \delta\varphi^b = \delta\varphi^a \otimes \delta\varphi^b - \delta\varphi^b \otimes \delta\varphi^a = \delta_1\varphi^a \, \delta_2\varphi^b - \delta_2\varphi^a \, \delta_1\varphi^b \,,$$

where the indices $1, 2$ label the position of the field differentials.[27] Thereby, the expression (193) for the current components $\omega^\mu$ (which are dual to the components of the $(n-1)$-form $\boldsymbol{\omega}$ [62]) reads as follows [60]:

$$\omega^\mu(\varphi; \delta_1\varphi, \delta_2\varphi) = \frac{\partial^2\mathbf{L}}{\partial\varphi^a \, \partial(\nabla_\mu\varphi^b)} \left[ \delta_1\varphi^a \, \delta_2\varphi^b - \delta_2\varphi^a \, \delta_1\varphi^b \right] \tag{341}$$

$$+ \frac{\partial^2\mathbf{L}}{\partial(\nabla_\nu\varphi^a) \, \partial(\nabla_\mu\varphi^b)} \left[ (\nabla_\nu\delta_1\varphi^a) \, \delta_2\varphi^b - (\nabla_\nu\delta_2\varphi^a) \, \delta_1\varphi^b \right] .$$

From the definitions (338) and (340) of $\boldsymbol{\theta}$ and $\boldsymbol{\omega}$ it follows that $\mathrm{d}\boldsymbol{\omega} = \mathrm{d}\delta\boldsymbol{\theta} = \delta(\mathrm{d}\boldsymbol{\theta}) \approx \delta^2\mathbf{L} = 0$, i.e. $\boldsymbol{\omega}$ satisfies the

structural conservation law : $\boxed{\mathrm{d}\boldsymbol{\omega} \approx 0 \,,} \tag{342}$

or, equivalently [62], the *covariant conservation law* $\nabla_\mu\omega^\mu \approx 0$ (cf. Eqn. (199)). Integration of the $(n-1)$-form $\boldsymbol{\omega} \in \Omega^{n-1,2}$ over a space-like $(n-1)$-dimensional hypersurface $\Sigma$ without boundary yields the *(pre-)symplectic 2-form* $\Omega$ on field space $\mathcal{F}$ (and thereby on covariant phase space $\bar{\mathcal{F}}$ by reduction, cf. equations (105) and (106)); its functional dependence on the field variations is presently denoted by square brackets:

$$\boxed{\Omega[\varphi; \delta_1\varphi, \delta_2\varphi] = \int_\Sigma \omega^\mu(\varphi, \delta_1\varphi; \delta_2\varphi) \, d\Sigma_\mu \,.} \tag{343}$$

---

[26]Cf. action of the exterior derivative on a 1-form defined on $M$: $\mathrm{d}(\alpha_\nu \mathrm{d}x^\nu) = \frac{1}{2}(\partial_\mu\alpha_\nu - \partial_\nu\alpha_\mu)\mathrm{d}x^\mu \wedge \mathrm{d}x^\nu$.

[27]One may also view [60] the variations $\delta_1\varphi^a$ and $\delta_2\varphi^a$ as derivatives of a smooth two-parameter family of field configurations $\varphi^a(\lambda_1, \lambda_2)$ with respect to $\lambda_1$ and $\lambda_2$, respectively: $\delta_1\varphi^a \equiv \partial\varphi^a/\partial\lambda_1$, $\delta_2\varphi^a \equiv \partial\varphi^a/\partial\lambda_2$.

In this context, one assumes that the fields (and their variations) satisfy asymptotic fall-off conditions which are sufficiently strong to ensure that the integral (343) exists and that it does not depend on the slice $\Sigma$ (by virtue of the argumentation in equation (93)).

Although $\Sigma$ does not have a boundary, one can assign a sense to the integral $\oint_{\partial\Sigma}\ldots$ by viewing it as a limit of $\oint_{\partial K}\ldots$ where the compact region $K \subset \Sigma$ approaches all of $\Sigma$ in a suitable manner [64, 65], see equation Eqn. (197) above. (For instance, for an asymptotically flat *four-dimensional* space-time, the integral $\oint_{\partial\Sigma}\ldots$ can then be viewed as the limit of $\oint_{\partial K_R}\ldots$ where the radius $R$ of the 2-sphere $\partial K_R$ tends to infinity.) In the sequel, the integrals $\oint_{\partial\Sigma}\ldots$ are to be interpreted in this sense along with the provisos attached this definition.

**Noether current associated to diffeomorphism invariance:** Diffeomorphisms on the space-time manifold $M$ are generated by vector fields which are locally given by $\xi \equiv \xi^\mu \partial_\mu$. The latter act on the tensor fields $\varphi$ by the Lie derivative with respect to $\xi$, i.e. $\delta_\xi \varphi = L_\xi \varphi$ for the infinitesimal variation of fields $\varphi$. For a *diffeomorphism invariant field theory,* the Lagrangian $n$-form $\mathbf{L}$ is covariant with respect to diffeomorphisms on $M$ and thereby it also varies with the Lie derivative $L_\xi = [i_\xi, \mathrm{d}]$ under infinitesimal transformations. Since $\mathbf{L}$ represents a form of top degree on the space-time manifold $M$, we have

$$\delta_\xi \mathbf{L} = L_\xi \mathbf{L} = \mathrm{d}(i_\xi \mathbf{L}). \tag{344}$$

Let us now apply the interior product $I_\xi$ (defined by (335)) to the relation (338) for $\mathbf{L}$. From $\mathbf{L} \in \Omega^{n,0}$ it follows that we have $I_\xi(\delta\mathbf{L}) = \delta_\xi \mathbf{L}$. Thus, equation (338) yields the following result for the

$$\text{variation of } \mathbf{L} \text{ under diffeomorphisms}: \qquad \delta_\xi \mathbf{L} = \mathbf{E}_\varphi \, L_\xi \varphi + \mathrm{d}(I_\xi \boldsymbol{\theta}), \tag{345}$$

with

$$I_\xi \boldsymbol{\theta} = \boldsymbol{\theta}(\varphi; L_\xi \varphi) = \frac{\partial \mathbf{L}}{\partial(\nabla_\mu \varphi^a)} \, L_\xi \varphi^a \, \mathrm{d}^{n-1} x_\mu \in \Omega^{n-1,0}. \tag{346}$$

Here, the last expression (resulting from expression (339) for the symplectic potential current form $\boldsymbol{\theta} \in \Omega^{n-1,1}$) is again referred to as the *'bare' choice* for $\boldsymbol{\theta}(\varphi; L_\xi \varphi)$. In the sequel, we will systematically use the writing $\boldsymbol{\theta}$ for $\boldsymbol{\theta}(\varphi; \delta\varphi)$ and $I_\xi \boldsymbol{\theta}$ for $\boldsymbol{\theta}(\varphi; L_\xi \varphi)$.

In conclusion, the

$$\text{Noether current } (n-1)\text{-form}: \qquad \boxed{\mathcal{J}_\xi \equiv I_\xi \boldsymbol{\theta} - i_\xi \mathbf{L}} \tag{347}$$

(which is associated to the vector field $\xi$ on $M$), is conserved for all solutions of the field equations since

$$\mathrm{d}\mathcal{J}_\xi = \mathrm{d}I_\xi \boldsymbol{\theta} - \mathrm{d}(i_\xi \mathbf{L}) \overset{(344)}{=} \mathrm{d}I_\xi \boldsymbol{\theta} - \delta_\xi \mathbf{L} \overset{(345)}{=} -\mathbf{E}_\varphi \, L_\xi \varphi \approx 0. \tag{348}$$

Thus, $\mathcal{J}_\xi \in \Omega^{n-1,0}$ represents a closed $(n-1)$-form on covariant phase space $\bar{\mathcal{F}}$. It is associated to the local symmetry given by diffeomorphism invariance described by the vector field $\xi$.

Since the $(n-1)$-form $\mathcal{J}_\xi$ is closed for fields $\varphi \in \bar{\mathcal{F}}$, it is locally exact by Poincaré's lemma, i.e. locally there exists a $(n-2)$-form $\mathbf{Q}_\xi \in \Omega^{n-2,0}$ such that $\mathcal{J}_\xi = \mathrm{d}\mathbf{Q}_\xi$ for the solutions of the field equations, i.e.

$$\boxed{\mathcal{J}_\xi \approx \mathrm{d}\mathbf{Q}_\xi.} \tag{349}$$

As a matter of fact, one can show [64] that

$$\mathcal{J}_\xi = \mathrm{d}\mathbf{Q}_\xi + \xi^\mu \, \mathbf{C}_\mu(\varphi), \qquad \text{with} \quad \mathbf{C}_\mu \approx 0. \tag{350}$$

More precisely, the relations $\mathbf{C}_\mu \approx 0$ correspond to the *constraint equations* which follow from the diffeomorphism invariance of the theory (cf. equations (233),(235) and (265),(266) for the current associated to the local gauge invariance in free Maxwell theory on Minkowski space-time).

The quantity $\mathbf{Q}_\xi$ is referred to [63] as *Noether charge $(n-2)$-form*[28] (or as *Noether-Wald surface charge form*) and its integral over a closed $(n-2)$-dimensional hypersurface $\mathcal{S}$ as *Noether charge of $\mathcal{S}$ relative to $\xi$*. A *general expression for the charge form $\mathbf{Q}_\xi$ in a diffeomorphism invariant theory* (described by a Lagrangian $n$-form which may depend on higher order derivatives) has been determined by V. Iyer and R. Wald [64]. For general relativity (as described by the Einstein-Hilbert Lagrangian), an explicit expression for $\mathbf{Q}_\xi$ was given (and further discussed) in equation (308) above.

**Variation of Noether current and gravitational Noether charge or Hamiltonian:** On a finite dimensional manifold $M$ which is endowed with a symplectic 2-form $\omega$, there is a correspondence between smooth functions $H \in C^\infty(M)$ and symplectic vector fields $X \in \mathfrak{X}(M)$, i.e. vector fields $X$ on $M$ which leave the symplectic form $\omega$ invariant, viz. $0 = L_X\omega = \mathrm{d}(i_X\omega)$. This correspondence $H \mapsto X$ finds its expression in the relation $i_X\omega = \mathrm{d}H$ (where $i_X$ denotes the interior product with $X$), see Eqn. (C.16) and the related discussion in Appendix C.4. The function $H : M \to \mathbb{R}$ is referred to as *Hamiltonian* or *canonical generator* for the infinitesimal transformations on $M$ given by the vector field $X$. The generalization of this result to the covariant phase space $\bar{\mathcal{F}}$ (of a field theory) viewed as an infinite-dimensional symplectic manifold endowed with a symplectic 2-form $\Omega$ has the form

$$I_X\Omega = \delta H.$$

Here, $\delta$ acts as an exterior differential on $H \in C^\infty(\bar{\mathcal{F}})$ (i.e. it is the field variation of the functional $H$) and $X$ represents a vector field on $\bar{\mathcal{F}}$. In the following, we will recover [60,63,64] this relation from the results of the previous paragraphs for the case of a vector field $X_\xi$ which is associated to infinitesimal diffeomorphisms $\xi$ on the space-time manifold $M$.

As in equation (338), we start with a generic field variation $\varphi \rightsquigarrow \varphi + \delta\varphi$ off an arbitrary solution $\varphi$ of the field equations. The latter variation does not act on diffeomorphisms (i.e. $\delta\xi = 0$) and by virtue of Eqn. (338) and $L_\xi = i_\xi\mathrm{d} + \mathrm{d}i_\xi$ we have

$$\delta(i_\xi\mathbf{L}) = i_\xi(\delta\mathbf{L}) \approx i_\xi\mathrm{d}\boldsymbol{\theta} = L_\xi\boldsymbol{\theta} - \mathrm{d}i_\xi\boldsymbol{\theta}.$$

Henceforth, the *variation of the Noether current form* (347) reads

$$\delta\mathcal{J}_\xi \approx \delta I_\xi\boldsymbol{\theta} - L_\xi\boldsymbol{\theta} + \mathrm{d}i_\xi\boldsymbol{\theta}.$$

From relation (340) it thus follows that we have the *general result* [63]

$$\delta\mathcal{J}_\xi \approx \boldsymbol{\omega}(\varphi; \delta\varphi, L_\xi\varphi) + \mathrm{d}i_\xi\boldsymbol{\theta}, \tag{351}$$

or, equivalently,

$$\boxed{\boldsymbol{\omega}(\varphi; \delta\varphi, L_\xi\varphi) \approx \delta\mathcal{J}_\xi - \mathrm{d}i_\xi\boldsymbol{\theta}.} \tag{352}$$

---

[28]As noted by the authors of reference [182], this terminology for $\mathbf{Q}_\xi$ is somewhat misleading since this quantity is not conserved and does not directly generate symmetry transformations: these authors rather suggest the terminology "Noether potential".

Let us now consider a globally hyperbolic space-time which is asymptotically flat and integrate relation (352) over a Cauchy hypersurface $\Sigma$ which has a single asymptotic region:

$$\int_\Sigma \boldsymbol{\omega}(\varphi; \delta\varphi, L_\xi\varphi) \approx \int_\Sigma \delta\mathcal{J}_\xi - \int_\Sigma \mathrm{d}i_\xi\boldsymbol{\theta}\,. \tag{353}$$

On the left hand side of this relation we have the contraction of the symplectic 2-form $\Omega$ on $\mathcal{F}$ with the vector field $X_\xi$ generating diffeomorphisms in field space:

$$I_\xi\Omega[\varphi; \delta_1\varphi, \delta_2\varphi] = \int_\Sigma \boldsymbol{\omega}(\varphi; \delta_1\varphi, L_\xi\varphi)\,. \tag{354}$$

By virtue of Eqn. (350) we can write $\delta\mathcal{J}_\xi \approx \delta\mathrm{d}\mathbf{Q}_\xi = \mathrm{d}\delta\mathbf{Q}_\xi$, hence

$$\int_\Sigma \delta\mathcal{J}_\xi = \int_{\partial\Sigma} \delta\mathbf{Q}_\xi = \delta\int_{\partial\Sigma} \mathbf{Q}_\xi\,, \tag{355}$$

for field variations vanishing on-shell, i.e. satisfying the linearized equations of motion.

Next suppose that we can find a $(n-1)$-form $\mathbf{B}$ such that the functional $\mathcal{B} \equiv \int_{\partial\Sigma} i_\xi\mathbf{B}$ satisfies $\delta\mathcal{B} = \int_{\partial\Sigma} i_\xi\boldsymbol{\theta}$. Then, we can conclude from the previous equations that we have the following *general result (for the solutions of the field equations)*:

$$\boxed{I_\xi\Omega = \delta H_\xi\,,} \qquad \text{with} \qquad \boxed{H_\xi \equiv \int_{\partial\Sigma} (\mathbf{Q}_\xi - i_\xi\mathbf{B})\,,} \tag{356}$$

hence

$$\delta H_\xi = \int_{\partial\Sigma} (\delta\mathbf{Q}_\xi - i_\xi\boldsymbol{\theta})\,. \tag{357}$$

Relation (356) represents the generalization to covariant phase space $(\bar{\mathcal{F}}, \Omega)$ of the relation (C.16) between a *Hamiltonian $H_\xi$ (on phase space)* and the corresponding *Hamiltonian vector field $X_\xi$* (which describes the evolution of the dynamical system on phase space that is generated by $\xi$). The Hamiltonian functional $H_\xi : \mathcal{F} \to \mathbb{R}$ (satisfying $\delta H_\xi = I_\xi\Omega$) is said to be *canonically conjugate to the vector field $\xi$ on the slice $\Sigma$* [65].

More specifically, the result (356) shows that (if it exists) *the Hamiltonian $H_\xi$ associated to a diffeomorphism invariant Lagrangian theory of gravity* (i.e. the Hamiltonian describing the evolution determined by a diffeomorphism generating vector field $\xi$) *represents on-shell a surface term* (cf. the analogous result (269) for the charge associated to the local gauge invariance in a gauge field theory). This introduction of the Hamiltonian represents a natural definition for a *conserved quantity which is associated to $\xi$ at "time" $\Sigma$*. In the particular case of a closed universe (i.e. compact Cauchy hypersurfaces $\Sigma$), the so-defined Hamiltonian vanishes for the solutions of the field equations.

For $\mathbf{B} = 0$, we have $\delta H_\xi = \delta\int_\Sigma \mathcal{J}_\xi$ for the solutions of the field equations, i.e. *the Noether current $\mathcal{J}_\xi$ may be viewed as the Hamiltonian density $\mathcal{H}_\xi$* (i.e. as the canonical generator of symmetry transformations $\delta_\xi$). This instance (which is familiar from Noether's theorem in Minkowski space-time) occurs if $\int_S i_\xi\boldsymbol{\theta} = 0$, i.e. for infinitesimal diffeomorphisms which are tangent to the corner $S \equiv \partial\Sigma$ (viz. for $\xi \in TS$), cf. Figure 8 below for the geometric set-up [65, 174]. For infinitesimal diffeomorphisms $\xi \notin TS$, the mathematical situation and its physical interpretation are fairly different [65, 174]. The diffeomorphisms then move the corners which may be viewed as a sensitivity with respect *degrees of freedom that could enter or escape the causal domain of the hypersurface $\Sigma$*. This instance corresponds to an *open physical system* and is relevant for the investigation of gravitational radiation as well as the study of entanglement and quantum gravity, e.g. see [180, 216–218, 268, 307–309, 313–316] and references therein.

**Example of general relativity:** By choosing appropriately the asymptotic conditions on the fields as well as the $(n-1)$-form **B** in an asymptotically flat region such that all surface integrals exist, the Hamiltonian (356) represents (for the solutions of the field equations) the *canonical energy associated to an asymptotic flat region* if $\xi$ describes an asymptotic time translation.

More specifically, let us now consider general relativity in a space-time which is asymptotically flat at *spatial* infinity. The form **B** can presently be chosen [64] in such a way that the canonical energy that we just defined coincides with the familiar *ADM energy:* the asymptotic time translation is then part of the asymptotic Poincaré group.

As a matter of fact, we have already encountered the covariant phase space relation $I_\xi \Omega = \delta H_\xi$ along with explicit expressions for $\Omega$ and $H_\xi$ in our discussion (based on the investigations of C. Crnkovic [166]) of the translational invariance of pure YM theory in Minkowski space (cf. Eqn. (99) and equations (114)-(116)) and in our discussion of the diffeomorphism invariance of the Einstein-Hilbert action in general relativity (cf. equations (141)-(144) which use the notation $\varepsilon^\mu$ instead of $\xi^\mu$). For the latter case, the expressions for $\Omega$ and for the *Komar energy* $\int_{\partial\Sigma} \mathbf{Q}_\xi$ coincide with those given in reference [64]; furthermore, the addition of a surface term (analogous to $\int_{\partial\Sigma} i_\xi \mathbf{B}$) to the Komar energy was also considered by Crnkovic in order to recover the ADM energy. The canonical momentum and angular momentum in an asymptotic flat region can be discussed along the same lines [64].

**First law of black hole mechanics:** For diffeomorphisms which leave the fields invariant (i.e. $L_\xi \varphi = 0$), the left hand side of equation (353) vanishes. Its right hand side can be rewritten using Stokes' theorem and Eqn. (355). In summary, for fields $\varphi \in \bar{\mathcal{F}}$ such that $\delta\varphi$ solves the linearized field equations and such that $L_\xi \varphi = 0$, we have

$$\int_{\partial\Sigma} \left( \delta\mathbf{Q}_\xi - i_\xi \boldsymbol{\theta} \right) = 0 , \tag{358}$$

where the Noether charge form $\mathbf{Q}_\xi$ depends on $\xi$ by virtue of the relation $\mathcal{J}_\xi \approx \mathrm{d}\mathbf{Q}_\xi$. From equation (358) one can deduce [63, 64] under fairly general assumptions the so-called *first law of black hole mechanics* (e.g. see references [17, 317] for a review of these laws), i.e.

$$\boxed{\frac{\kappa}{2\pi} \delta S = \delta\mathcal{E} - \Omega_H \delta J .} \tag{359}$$

Here, $\mathcal{E}$ denotes the *canonical energy* and $J$ the *canonical angular momentum* mentioned above; furthermore, $\kappa$ represents the *surface gravity* and $\Omega_H$ the *angular velocity* of the black hole horizon while $S$ denotes its *entropy* related to its *area A* by the *area law* $S = \frac{1}{4}A$.

**Equivalence of Noether current and energy-momentum current for matter fields:** Relation (348) allows us to readily show [64] that the Noether current $\mathcal{J}_\xi$ associated to diffeomorphism invariance is equivalent to the conserved current density $j_\xi^\mu \equiv T^{\mu\nu}\xi_\nu$ that we encountered in our discussion of the Peierls bracket for a scalar field coupled to gravity, see equations (68) and (76). To do so, we consider a collection of tensor fields $\varphi \equiv ((g_{\mu\nu}), \psi)$ where $(g_{\mu\nu})$ represents a non-dynamical (background) metric and $\psi$ some tensorial matter fields (e.g. scalar and/or gauge fields). The EMT $(T^{\mu\nu})$ of the matter fields is defined by Eqn. (58), i.e. (with the volume $n$-form $\mathbf{vol} \equiv \sqrt{|g|}\,\mathrm{d}^n x$)

$$T^{\mu\nu}\mathbf{vol} \equiv -2\,\frac{\delta S_M[\psi, g_{\mu\nu}]}{\delta g_{\mu\nu}}\,\mathrm{d}^n x = -2\left(\mathbf{E}_g\right)^{\mu\nu} , \tag{360}$$

where $S_M[\psi, g_{\mu\nu}]$ denotes the diffeomorphism invariant action functional for the matter fields $\psi$ coupled to the background metric $(g_{\mu\nu})$.

Now consider the Noether current $(n-1)$-form (347) which is associated to the diffeomorphism invariant action $S_M$. For the solutions of the matter field equations $\mathbf{E}_\psi = 0$, relation (348) (i.e. $\mathrm{d}\mathcal{J}_\xi = -\mathbf{E}_\varphi \, L_\xi \varphi$) reads

$$\mathrm{d}\mathcal{J}_\xi = -\left(\mathbf{E}_g\right)^{\mu\nu} L_\xi g_{\mu\nu}. \tag{361}$$

With $L_\xi g_{\mu\nu} = \nabla_\mu \xi_\nu + \nabla_\nu \xi_\mu$ and relation (360) we get

$$\begin{aligned}
\mathrm{d}\mathcal{J}_\xi = T^{\mu\nu}(\nabla_\mu \xi_\nu)\mathbf{vol} &= \nabla_\mu(T^{\mu\nu}\xi_\nu)\mathbf{vol} - (\nabla_\mu T^{\mu\nu})\xi_\nu \mathbf{vol} \\
&= \mathrm{d}(i_{j_\xi}\mathbf{vol}) - (\nabla_\mu T^{\mu\nu})\xi_\nu \mathbf{vol},
\end{aligned} \tag{362}$$

where $j_\xi \equiv j_\xi^\mu \partial_\mu$ and where we used the relations

$$i_{j_\xi}\mathbf{vol} = \sqrt{|g|}\, j_\xi^\mu \, \mathrm{d}^{n-1}x_\mu, \qquad \mathrm{d}x^\nu \wedge \mathrm{d}^{n-1}x_\mu = \delta_\mu^\nu \mathrm{d}^n x, \qquad \nabla_\mu j_\xi^\mu = \frac{1}{\sqrt{|g|}}\partial_\mu\left(\sqrt{|g|}\, j_\xi^\mu\right).$$

Equation (362), i.e.

$$\mathrm{d}(\mathcal{J}_\xi - i_{j_\xi}\mathbf{vol}) = -(\nabla_\mu T^{\mu\nu})\xi_\nu \mathbf{vol},$$

states that its right hand side is an exact $n$-form for any vector field $\xi$: this can only hold if $\nabla_\mu T^{\mu\nu} = 0$, i.e. if the EMT ($T^{\mu\nu}$) is covariantly conserved for all solutions of the matter field equations. For these solutions, application of the Poincaré lemma then implies that the closed $(n-1)$-form

$$\mathcal{J}_\xi - i_{j_\xi}\mathbf{vol} = \left(\mathcal{J}_\xi^\mu - j_\xi^\mu\right)\sqrt{|g|}\, \mathrm{d}^{n-1}x_\mu,$$

is an exact $(n-1)$-form, i.e. represents a superpotential term.

In summary, *the Noether current* $(\mathcal{J}_\xi^\mu)$ *associated to the diffeomorphism invariant action of a matter field coupled to a non-dynamical metric field coincides with the EMT-current* $j_\xi^\mu \equiv T^{\mu\nu}\xi_\nu$ *of this matter field up to a superpotential term and an equation of motion term* (cf. Eqn. (240) for the equivalence of conserved currents). As discussed after equation (68), the EMT-current $\sqrt{|g|}\, j_\xi^\mu$ is conserved if $\xi$ represents a Killing vector field of the background metric, i.e. if $L_\xi g_{\mu\nu} = 0$. In this case, we also have $\mathrm{d}\mathcal{J}_\xi = 0$ for the solutions of the matter field equations by virtue of relation (361).

**Further examples, extensions and generalizations:** Instead of a space-time which is asymptotically flat at spatial infinity, one can consider general relativity in a space-time which is asymptotically flat at *null* infinity. This instance was addressed within the covariant phase space approach to diffeomorphism invariant Lagrangian field theories (outlined above) by R. Wald and A. Zoupas [65]. The mathematical complications which arise in this setting are related to each other: at null infinity the symplectic current in vacuum general relativity can be radiated away (i.e. null asymptotic flatness corresponds to radiative or *"leaky" boundary conditions*), a Hamiltonian generating the given asymptotic symmetry does not exist and the "conserved charges" associated to the asymptotic symmetry are actually *not* conserved in general.

In the sequel, the latter investigations have been further generalized in different directions, e.g. see reference [181]. First [268, 310], for the case where one has *field-dependent diffeomorphisms* (i.e. $\delta\xi \neq 0$) and where one has *field-dependent quantities with anomalous transformation laws* meaning that they not transform simply with the Lie derivative under diffeomorphisms. (The latter instance occurs for example in the presence of fixed background structures which are inert under diffeomorphisms.) And second [306, 311], to more general boundaries and boundary conditions, notably null hypersurfaces at finite distance or non-expanding horizons.

All of these investigations take into account (or make explicit use) of the ambiguities in the choice of the Lagrangian $n$-form $\mathbf{L}$ and of the symplectic potential current form $\boldsymbol{\theta}$. As we already discussed (in particular in equations (128)-(131)), a boundary term $\mathrm{d}\ell$ has eventually to be added to $\mathbf{L}$ (notably to the Einstein-Hilbert Lagrangian $\mathbf{L}$) in order to have a well-defined action principle for *given* boundary conditions. Thus, we encounter the

$$\text{cohomological ambiguities :} \qquad \boxed{\mathbf{L} \rightsquigarrow \mathbf{L} + \mathrm{d}\mathbf{Y}, \qquad \boldsymbol{\theta} \rightsquigarrow \boldsymbol{\theta} + \delta\mathbf{Y} + \mathrm{d}\boldsymbol{\alpha}} \tag{363}$$

(cf. equations (194)-(195)), where the term $\mathrm{d}\boldsymbol{\alpha}$ is annihilated by the differential $\mathrm{d}$ in the defining relation (338) for the symplectic potential current form $\boldsymbol{\theta}$ associated to $\mathbf{L}$. Since $\mathbf{L} \in \Omega^{n,0}$ and $\boldsymbol{\theta} \in \Omega^{n-1,1}$, we have $\mathbf{Y} \in \Omega^{n-1,0}$ and $\boldsymbol{\alpha} \in \Omega^{n-2,1}$. The transformation (363) of $\boldsymbol{\theta}$ implies that the associated symplectic current form $\boldsymbol{\omega} = \delta\boldsymbol{\theta}$ changes according to

$$\boldsymbol{\omega} \rightsquigarrow \boldsymbol{\omega} + \mathrm{d}(\delta\boldsymbol{\alpha}), \tag{364}$$

where the last term represents an identically conserved contribution to $(\omega^\mu)$.

Under the transformations (363), the Noether current form $\mathcal{J}_\xi = I_\xi \boldsymbol{\theta} - i_\xi \mathbf{L}$ associated to the pair $(\mathbf{L}, \boldsymbol{\theta})$ changes as follows:

$$\mathcal{J}_\xi \rightsquigarrow I_\xi(\boldsymbol{\theta} + \delta\mathbf{Y} + \mathrm{d}\boldsymbol{\alpha}) - i_\xi(\mathbf{L} + \mathrm{d}\mathbf{Y}) = \underbrace{I_\xi \boldsymbol{\theta} - i_\xi \mathbf{L}}_{=\mathcal{J}_\xi} + \underbrace{I_\xi \delta\mathbf{Y} - i_\xi \mathrm{d}\mathbf{Y}}_{=\delta_\xi \mathbf{Y}} + \underbrace{I_\xi \mathrm{d}\boldsymbol{\alpha}}_{=\mathrm{d}I_\xi \boldsymbol{\alpha}}.$$

With $\mathcal{J}_\xi \approx \mathrm{d}\mathbf{Q}_\xi$ and $-i_\xi \mathrm{d}\mathbf{Y} = -L_\xi \mathbf{Y} + \mathrm{d}i_\xi \mathbf{Y}$ we thus have $\mathcal{J}_\xi \rightsquigarrow \mathrm{d}(\mathbf{Q}_\xi + i_\xi \mathbf{Y} + I_\xi \boldsymbol{\alpha})$. Henceforth, the Noether charge form $\mathbf{Q}_\xi$ transforms according to

$$\boxed{\mathbf{Q}_\xi \rightsquigarrow \mathbf{Q}_\xi + i_\xi \mathbf{Y} + I_\xi \boldsymbol{\alpha}.} \tag{365}$$

In the next subsection, we will discuss how the ambiguities (363)-(365) and in particular the one for the charges can be related to, and fixed by, physical considerations while providing some explicit examples.

## 7.8 Boundaries and corners (and associated charges)

### 7.8.1 Generalities

For concreteness we consider a four-dimensional space-time manifold $M$. We suppose that this manifold is foliated by space-like hypersurfaces $\Sigma$, e.g. hypersurfaces $t = $ constant. If the hypersurface $\Sigma_t$ is given by a ball, then its boundary $S_t \equiv \partial\Sigma_t$ represents a 2-sphere $r = R = $ constant (with a finite value of $R$ in the case of a spatially bounded region $\Sigma_t$). One may consider different hypersurfaces $\Sigma_t, \Sigma_t'$ having the same boundary $S_t$, see Figure 8.

In this context, the set $S_t$ (which has codimension 2) is referred to as a *corner*: it is the boundary of boundary components of $M$. The collection of all corners makes up the (potentially asymptotic) *boundary* of $M$ which has codimension 1 and whose interior is referred to as the *bulk* of space-time.

### 7.8.2 Construction of Noether charges following Wald and Zoupas

**General procedure:** In Subsection 5.1.2, we already noted that, for a given space-time, one is not necessarily interested in the explicit expression of the symplectic current $(n-1)$-form $\boldsymbol{\omega}$, but only in its flux through a given $(n-1)$-dimensional hypersurface [62]. This flux only requires the knowledge of the components of the current $(\omega^\mu)$ which are normal to the hypersurface or, equivalently, the evaluation of the pullback (restriction) of the $(n-1)$-form

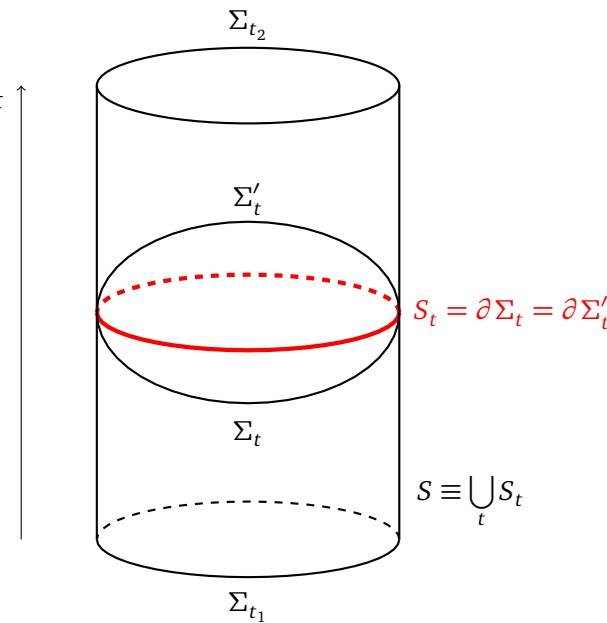

Figure 8: Definition of corners, boundary and bulk of space-time.

$\boldsymbol{\omega}$ to the hypersurface. Following reference [181], we denote the pullback of $\boldsymbol{\theta} \in \Omega^{n-1,1}$ and of $\boldsymbol{\omega} \in \Omega^{n-1,2}$ to the hypersurface by $\underleftarrow{\boldsymbol{\theta}}$ and $\underleftarrow{\boldsymbol{\omega}}$, respectively.

In reference [65], R. Wald and A. Zoupas put forward the idea that, given a generic diffeomorphism invariant Lagrangian theory of gravity and a very general class of asymptotic conditions "at infinity", the ambiguity (365) in the definition of the charges $\mathbf{Q}_\xi$ should be fixed by specifying under which physical requirements the charges are to be conserved. This idea can be implemented [64, 65, 181, 182, 318] by going over from the initial symplectic potential $\boldsymbol{\theta}$ (given by the bare choice or by any other choice) to a symplectic potential such that its pullback $\boldsymbol{\theta}'$ to the lateral boundary $\mathcal{B}$ (which is attached to space-time) vanishes in the subspace of phase space which corresponds to the considered physical requirements (boundary conditions), i.e.

$$\boldsymbol{\theta}' \overset{\mathcal{B}}{=} 0. \tag{366}$$

The physical requirement may for instance be given by a choice of conservative boundary conditions or of stationarity conditions. More precisely, for a given Lagrangian form $\mathbf{L}$, one decomposes the pullback $\underleftarrow{\boldsymbol{\theta}}$ of $\boldsymbol{\theta}$ to $\mathcal{B}$ (notably by application of the Leibniz rule) according to

$$\boxed{\underleftarrow{\boldsymbol{\theta}} = \boldsymbol{\theta}' - \delta\boldsymbol{\ell} + \mathrm{d}\boldsymbol{\vartheta}\,.} \tag{367}$$

Here, the new potential $\boldsymbol{\theta}'$ corresponds to the Lagrangian $\mathbf{L}' = \mathbf{L} + \mathrm{d}\boldsymbol{\ell}$ (which yields the same equations of motion as $\mathbf{L}$) and $\boldsymbol{\theta}'$ *is required to be of the form $p\,\delta q$* (i.e. $\theta'^\mu$ of the form $\pi_a^\mu\,\delta\varphi^a$) for some choice $(q,p)$ of polarization of phase space into "position" coordinates $q$ and canonical momenta $p$. The boundary Lagrangian $\mathrm{d}\boldsymbol{\ell}$ is the one to be added to the initial Lagrangian $\mathbf{L}$ in order to have a well-defined variational principle for the given boundary conditions. According to the freedom (363), the new symplectic potential is equivalent to the initial one.

The expression of $\boldsymbol{\ell}$ is not uniquely defined by condition (367) since this relation is invariant under the following transformations:

$$\text{corner ambiguity}: \qquad (\boldsymbol{\ell}, \boldsymbol{\vartheta}) \rightsquigarrow (\boldsymbol{\ell} + \mathrm{d}\mathbf{c}, \boldsymbol{\vartheta} + \delta\mathbf{c}), \qquad \text{with} \quad \mathbf{c} \in \Omega^{n-2,0}. \tag{368}$$

Indeed, we have $\delta(\mathrm{d}\mathbf{c}) = \mathrm{d}(\delta\mathbf{c})$ (and by virtue of $\mathrm{d}(\mathrm{d}\mathbf{c}) = 0$ the shift $\boldsymbol{\ell} + \mathrm{d}\mathbf{c}$ also leaves invariant $\mathbf{L}' = \mathbf{L} + \mathrm{d}\boldsymbol{\ell}$). Thus, for a given representative $\boldsymbol{\theta}$, one can only find a unique expression for $\boldsymbol{\vartheta}$ (the so-called "corner symplectic potential for $\boldsymbol{\ell}$") once a choice for both $\boldsymbol{\theta}'$ and $\boldsymbol{\ell}$ has been made.[29]

For the case where $\boldsymbol{\vartheta} = 0$ or where $\boldsymbol{\vartheta} = \delta\boldsymbol{\gamma}$ (so that $\boldsymbol{\vartheta}$ can be absorbed in the term $\delta\boldsymbol{\ell}$ in equation (367)), the procedure of Wald and Zoupas (WZ) is tantamount [65, 181, 182] to defining a

$$\text{(boundary-) improved Noether charge :} \qquad \boxed{\mathbf{Q}_\xi^{WZ} \equiv \mathbf{Q}_\xi + i_\xi \boldsymbol{\ell}\,,} \qquad (369)$$

where $\boldsymbol{\ell}$ corresponds to the boundary Lagrangian $\mathrm{d}\boldsymbol{\ell}$ which is chosen on physical grounds (compare expression (369) with Eqn. (365) which exhibits the freedom in choosing $\mathbf{Q}_\xi$). This amounts to a modification of equation (356) that must be satisfied by (the variation of) a Hamiltonian $H_\xi$, i.e. of expression (357). More precisely, it comes up to adding [65] a "correction term" $\int_{\partial\Sigma} i_\xi \boldsymbol{\ell}$ to $H_\xi$.

**Examples:** As a first example, we consider vacuum general relativity with the Dirichlet boundary conditions (128) on the lateral boundary $\mathcal{B}$ corresponding to the Gibbons-Hawking-York boundary Lagrangian $\boldsymbol{\ell}$ given by the action (129). The decomposition (367) then coincides with the expansion (138), i.e. $\boldsymbol{\theta}' = \bar{\boldsymbol{\Pi}}^{ab} \delta\bar{h}_{ab}$ and the boundary conditions $\delta g_{\mu\nu} \overset{\mathcal{B}}{=} 0$ imply $\boldsymbol{\theta}' \overset{\mathcal{B}}{=} 0$ (cf. Eqn. (366)), whence $\boldsymbol{\omega}' \overset{\mathcal{B}}{=} 0$. This means that the symplectic current form is preserved between the initial and final space-like hypersurfaces, i.e. the system is conservative (closed system or no leakage of symplectic flux by gravitational radiation, see references [181, 205, 241, 315]). In the present case, the allowed diffeomorphisms preserve the boundary conditions on $\mathcal{B}$ and do not not move the boundary (corners) [174]. In particular, for orthogonal corners, the corner potential $\boldsymbol{\vartheta}$ can be chosen to vanish [181]. The (boundary-) improved Noether charges presently yield [64, 181, 182] the so-called *Brown-York formulas* at finite distance. The latter provide the ADM charges at spatial infinity. (We note that the Brown-York formulas can also be obtained by considering the so-called Weiss variation of the action functional which involves a variation of the space-time domain [234], cf. our discussion after Eqn. (216).)

As a second example, we mention vacuum general relativity in a space-time which is asymptotically flat at future null infinity. The procedure of Wald and Zoupas then yields [65, 181, 319] the conserved quantities (associated to infinitesimal BMS generators $\xi^\mu$) found earlier by T. Dray and M. Streubel [320] using a quite different approach.

The cases of non-vanishing corner terms and of anomalous transformation laws have been addressed in particular in references [180–182]. We refer to the works [65, 174, 180–182, 268] (and references therein) for the detailed results and technicalities, in particular for the existence and conservation of gravitational charges as well as for their construction and algebra.

### 7.8.3 Corner degrees of freedom and "corner proposal"

For vacuum general relativity in the metric formulation as described by the Einstein-Hilbert Lagrangian (cf. equations (133) and (137)),

$$S_{EH} = S_{GR} + S_{EH/GR}\,, \qquad (370)$$

---

[29]We note [181] that Eqn. (367) is also invariant if one adds an exact form $\mathrm{d}\boldsymbol{\beta}$ to $\boldsymbol{\vartheta}$, but such terms are discarded if one assumes that the corner hypersurfaces $\mathcal{S}$ (having dimension $n-2$) are compact (e.g. 2-spheres for $n = 4$) so that $\int_{\mathcal{S}} \mathrm{d}\boldsymbol{\beta} = \oint_{\partial\mathcal{S}} \boldsymbol{\beta} = 0$.

we saw that the boundary/corner term $S_{EH/GR}$ yields an extra contribution $\boldsymbol{\theta}_{EH/GR}$ to the symplectic potential $\boldsymbol{\theta}_{GR}$ associated to the GR (ADM-) Lagrangian $\mathbf{L}_{GR}$, cf. equations (136),(138) and (367). The corresponding symplectic form $\boldsymbol{\omega}_{EH/GR} = \delta \boldsymbol{\theta}_{EH/GR}$ may be viewed as reflecting the fact that *the corner involves some extra degrees of freedom* as compared to those appearing in the canonical (ADM) formulation of general relativity as defined by the Einstein-Hilbert Lagrangian (see also the discussion after Eqn. (357) above): these degrees of freedom come along with *extra symmetries and charges,* and thereby allow us to "covariantize" the canonical formulation in some way [180]. Over the last two decades, the symmetries and surface charges in gravity have been investigated within different approaches and compared to each other, e.g. see references [174, 180–182, 194]. More specifically, the contribution of charges coming from the bulk (bounded or unbounded spatial regions $\Sigma$) and/or from their boundary $\partial \Sigma$ (corners) have been determined and the imprint of the different terms appearing in the Lagrangian (cosmological, Holst and topological terms mentioned in Subsection 5.1.2) have been discussed. It turns out that different formulations of gravity which are equivalent in the bulk (in that they all yield Einstein's field equations in the bulk) involve different corner terms (symplectic potential, symplectic current,...) and thus lead to different corner symmetries and charges [174, 180, 182].

The authors of reference [180] emphasized that the symmetry algebra of gravity associated to a corner ("corner symmetry algebra") represents an *universal* component of any boundary symmetry algebra in the sense that it is independent of the choice of the boundary conditions. Henceforth, it may be expected to play a fundamental role for the theory, in particular for its quantization (see references [19, 321] for a recent assessment of the so-called *corner proposal*).

# 8 Concluding remarks

In these notes, we have focused for simplicity on first order Lagrangians which are typically encountered in physical applications. We note that the case of higher order Lagrangians has also been addressed for some of the approaches that we described, e.g. see references [154, 161, 180, 322].

As indicated in the introductory overview and described in more detail in the main body of the notes, the variational bicomplex represents a general and versatile framework for classical field theory. Within this framework, we have discussed a few classes of symmetries as well as their consequences (conservation laws). The variational bicomplex indeed represents the appropriate setting for the formulation of partial differential equations and for the investigation of their symmetries, e.g. see [31, 226, 253]. More generally, it may serve as a solid mathematical basis for the perturbative quantization of field theories, e.g. see [50, 74, 145] and references therein.

Throughout our notes, we recalled and put forward various relationships between the different covariant approaches to the canonical formulation of classical field theories and between some of the results obtained within these formulations. A further exploration of these relationships should not only be worthwhile on its own interest, but may also contribute to a better understanding of some aspects of classical and quantum field theory as well as to the derivation of some novel results.

# Acknowledgments

I wish to express my deep gratitude to Stefan Hohenegger for many enlightening discussions and for having contributed the sophisticated figures to the manuscript. I am indebted to Ro-

main Cazali for instructive remarks on some points of the text and to François Delduc, Sucheta Majumdar, Frédéric Hélein and Tilmann Wurzbacher for fruitful discussions. Particular thanks to Marc Geiller for his helpful explanations and indications concerning the covariant phase space approach to gravitational theories. I also acknowledge pleasant exchanges with the late Krzysztof Gawędzki who provided some useful hints on the literature.

The instructive and constructive remarks of the anonymous referees led to an elaboration of several points of the text, thereby enriching (and hopefully improving) the original version: I owe them a great depth of gratitude.

# A   Some mathematical notions

Since differential forms and various operations thereon (like inner product with vector fields, pullback or integration) are frequently used in the main body of the text, we provide a short summary of the relevant points in this appendix. Some useful general references for these topics are [118, 133, 323]; excellent presentations which are oriented towards applications in physics (and in particular field theory) can be found in references [169, 324].

## A.1   Differentials forms

**Differentials forms:**   Having in mind space-time, we consider a smooth real $n$-dimensional manifold $M$ parametrized by local coordinates $(x^\mu)_{\mu=0,1,\dots,n-1}$. The exterior differential $df$ of a smooth function $f : M \to \mathbb{R}$ has the local expression $df = \partial_\mu f \, dx^\mu$. We assign the *form-degree* 0 to a function and the form-degree 1 to the monomials $dx^\mu$. Thus, the differential $df$ is a particular 1-form and a generic 1-form $\alpha = \alpha_\mu dx^\mu$ represents a covariant vector field $(\alpha_\mu(x))$ on $M$.

For the basic 1-forms $dx^\mu$, we consider the exterior (Grassmann) product, i.e. $dx^\mu \wedge dx^\nu = -dx^\nu \wedge dx^\mu$. Moreover, we extend the action of the differential $d = dx^\mu \partial_\mu$ to 1-forms by

$$\beta \equiv d\left(\alpha_\nu dx^\nu\right) \equiv \partial_\mu \alpha_\nu \, dx^\mu \wedge dx^\nu = \frac{1}{2}\beta_{\mu\nu} \, dx^\mu \wedge dx^\nu, \qquad \text{with } \beta_{\mu\nu} \equiv \partial_\mu \alpha_\nu - \partial_\nu \alpha_\mu = -\beta_{\nu\mu}.$$

Henceforth, the 2-form $\beta$ represents a (particular) antisymmetric covariant tensor field of rank 2 on $M$. Quite generally, the consideration of differential forms ensures an index free notation for antisymmetric covariant tensor fields on $M$.

For $\alpha = df$, we conclude from the previous relation that $d^2 f \equiv d(df) = 0$ by virtue of the Schwarz lemma, i.e. the differential $d$ is *nilpotent*. This differential acts on the exterior product of forms $\alpha, \beta$ according to the

$$\text{graded derivation rule}: \qquad d\left(\alpha \wedge \beta\right) = d\alpha \wedge \beta + (-1)^{\deg \alpha} \alpha \wedge d\beta.$$

The previous definitions lead to the *algebra of differential forms* (i.e. antisymmetric, covariant tensor fields) on $M$: $\Omega^\bullet(M) \equiv \oplus_{p \in \mathbb{Z}} \Omega^p(M)$ where we have, $\Omega^p(M) = 0$ for $p < 0$ (by definition) and for $p \geq 0$:

$$\boxed{\alpha = \frac{1}{p!}\alpha_{\mu_1 \cdots \mu_p} \, dx^{\mu_1} \wedge \cdots \wedge dx^{\mu_p},} \qquad \text{for } \alpha \in \Omega^p(M). \tag{A.1}$$

We note that $\Omega^p(M) = 0$ for $p > n$ due to the

$$\text{graded symmetry of the exterior product}: \qquad \alpha \wedge \beta = (-1)^{(\deg \alpha)(\deg \beta)} \beta \wedge \alpha$$

(which includes as a particular case the relation $dx^\mu \wedge dx^\nu = -dx^\nu \wedge dx^\mu$).

In summary, the vector space $\Omega^\bullet(M) \equiv \oplus_{p\in\mathbb{Z}}\Omega^p(M)$ endowed with the exterior product represents an associative, graded commutative algebra. On this graded algebra, the differential $d$ acts as a graded derivation of order 1, i.e. $d$ increases the form degree by one unit, $d : \Omega^\bullet(M) \to \Omega^{\bullet+1}(M)$; moreover, we have $d^2 = 0$. By restricting the coefficients of forms to a submanifold $N$ of $M$, one obtains the algebra of forms on $N$.

A form $\alpha$ is said to *closed* if $d\alpha = 0$ and it is said to be *exact* if $\alpha = d\beta$ for some form $\beta$. The nilpotency of the differential $d$ implies that an exact form is closed. The *Poincaré lemma* states that the converse is true for $p$-forms with $p \geq 1$ on a simply connected manifold $M$, i.e. $d\alpha = 0$ on $M$ implies that there exists a form $\beta$ on $M$ with $\alpha = d\beta$. (For $p = 0$, all constant 0-forms $\alpha$, i.e. constant real-valued functions, are $d$-closed, but not $d$-exact.) In particular, the Poincaré lemma holds locally on any manifold. It can be proven by introducing a so-called *contracting homotopy operator,* see Section A.5 below.

**Hodge dual:** On an $n$-dimensional (pseudo-)Riemannian manifold $(M,(g_{\mu\nu}))$, the Hodge star operator $\star$ maps a $p$-form (as locally given by (A.1)) to a $(n-p)$-form, the *Hodge dual* $\star\alpha$ being defined by [324]

$$\star\alpha = \frac{1}{(n-p)!}\tilde{\alpha}_{\mu_1...\mu_{n-p}}dx^{\mu_1}\wedge\cdots\wedge dx^{\mu_{n-p}}, \qquad \text{where} \quad \tilde{\alpha}_{\mu_1...\mu_{n-p}} = \frac{1}{p!}\varepsilon_{\mu_1...\mu_n}\alpha^{\mu_{n-p+1}...\mu_n}. \tag{A.2}$$

Here, the *Levi-Civita tensor* $\varepsilon$ is normalized by $\varepsilon_{01\cdots(n-1)} = 1$ with respect to flat (i.e. tangent space) indices, hence with respect to curved space indices we have [17]

$$\varepsilon_{\mu_1=0,...,\mu_n=(n-1)} = \sqrt{|g|}. \tag{A.3}$$

It then follows that $\star(\star\alpha) = (-1)^{p(n-p)}\alpha$ for the $p$-form $\alpha$.

## A.2 Integration of differentials forms

**Integration of differentials forms:** The graded symmetry of the exterior product implies that $dx^{\mu_1}\wedge\cdots\wedge dx^{\mu_n} = -\varepsilon^{\mu_1\cdots\mu_n}dx^0\wedge\cdots\wedge dx^{n-1}$ where $\varepsilon^{01\cdots(n-1)} = -1$. A generic $n$-form on $M$ thus writes $f\,dx^0\wedge\cdots\wedge dx^{n-1}$ (with some function $f$) and its integral on $M$ is defined by

$$\int_M f\,dx^0\wedge\cdots\wedge dx^{n-1} \equiv \int_M f\,d^n x.$$

**Stokes theorem:** Differential forms provide the appropriate volume forms for the integrals, e.g. see references [17, 154]. With applications to gravity in mind, we consider here a general $n$-dimensional *Lorentzian manifold*, i.e. a real smooth $n$-dimensional manifold $M$ which is equipped with a metric tensor field $(g_{\mu\nu}(x))$ of signature $(+,-,\cdots,-)$: Minkowski space with standard coordinates $(x^0,\vec{x})$ then represents the particular case where $M = \mathbb{R}^n$ and $g_{\mu\nu} = \eta_{\mu\nu}$, hence $|g| \equiv |\det(g_{\mu\nu})| = 1$. The manifold $M$ is always assumed to be orientable, the orientation being given by the natural order $(x^0, x^1, \ldots, x^{n-1})$.

Let $0 \leq p \leq n$. For a $(n-p)$-dimensional submanifold of the $n$-dimensional Lorentzian manifold $M$, the *volume $(n-p)$-form* is defined by[30]

$$\boxed{\left(d^{n-p}x\right)_{\mu_1\cdots\mu_p} \equiv \frac{1}{p!(n-p)!}\varepsilon_{\mu_1\cdots\mu_n}dx^{\mu_{p+1}}\wedge\cdots\wedge dx^{\mu_n},} \tag{A.4}$$

---

[30]We note that this form may be related by a lowering of indices to the Hodge-dual of the $p$-from $dx^{\mu_1}\wedge\cdots\wedge dx^{\mu_p}$ and that the conventions for the normalizations depend on the reference that is considered.

where the Levi-Civita tensor $\varepsilon$ is normalized with respect to curved space indices by (A.3).

For $p = 0$, we recover the usual *Riemannian volume element*

$$\left(d^n x\right) = \frac{1}{n!} \varepsilon_{\mu_1 \cdots \mu_n} dx^{\mu_1} \wedge \cdots \wedge dx^{\mu_n} = \sqrt{|g|}\, dx^0 \wedge \cdots \wedge dx^{n-1}\,. \tag{A.5}$$

The latter allows us to integrate a scalar field $\phi$:

$$\int_M \left(d^n x\right) \phi = \int_M dx^0 \wedge \cdots \wedge dx^{n-1} \sqrt{|g|}\, \phi \equiv \int_M d^n x \sqrt{|g|}\, \phi\,.$$

(Of course, for the existence of this integral, the manifold $M$ has to be compact or the field $\phi$ has to have an appropriate asymptotic behavior, e.g. to have compact support.)

For $p = 1$ and a vector field $(J^\mu)$, we have the $(n-1)$-form

$$\left(d^{n-1} x\right)_\mu J^\mu = \frac{1}{(n-1)!} J^\mu \varepsilon_{\mu \mu_2 \cdots \mu_n} dx^{\mu_2} \wedge \cdots \wedge dx^{\mu_n}\,. \tag{A.6}$$

Its integral over the $(n-1)$-dimensional space-like hypersurface $\Sigma \subset M$ given by $x^0 = $ constant writes

$$Q \equiv \int_\Sigma \left(d^{n-1} x\right)_\mu J^\mu = \int_\Sigma dx^1 \cdots dx^{n-1} \sqrt{|g|} J^0 = \int_\Sigma dx^1 \cdots dx^{n-1} j^0\,, \qquad \text{with } j^0 \equiv \sqrt{|g|} J^0\,. \tag{A.7}$$

In Minkowski space ($M = \mathbb{R}^n$, $g_{\mu\nu} = \eta_{\mu\nu}$), this integral represents the *total charge* if $(J^\mu)$ is the current density.

The covariant divergence of the vector field $(J^\mu)$, i.e. $\nabla_\mu J^\mu \equiv \frac{1}{\sqrt{|g|}} \partial_\mu(\sqrt{|g|} J^\mu)$, is a scalar field and can therefore be integrated over the manifold $M$ with the volume form (A.5). Now suppose that $M$ is a manifold with boundary $\partial M$ (the latter representing the empty set if $M$ is boundaryless). Then we can apply the general Stokes' theorem (i.e. $\int_M d\omega = \oint_{\partial M} \omega$ for a $(n-1)$-form $\omega$ on $M$) and obtain

$$\boxed{\int_M d^n x \sqrt{|g|} \nabla_\mu J^\mu = \oint_{\partial M} \left(d^{n-1} x\right)_\mu J^\mu\,.} \tag{A.8}$$

This is the *covariant form of the Gauss-Ostrogradski theorem* which corresponds to $n = 3$, $M \equiv V \subset \mathbb{R}^3$ and $(g_{\mu\nu}) = \mathbb{1}$: $\int_V d^3 x \operatorname{div} \vec{J} = \oint_{\partial V} \overrightarrow{dS} \cdot \vec{J}$, where $dS_i = \frac{1}{2} \varepsilon_{ijk} dx^j \wedge dx^k$.

For a current superpotential $B^{\mu\nu} = -B^{\nu\mu}$, i.e. an antisymmetric tensor field on $M$, we can consider the $(n-2)$-form

$$\boxed{\left(d^{n-2} x\right)_{\mu\nu} B^{\mu\nu} = \frac{1}{2(n-2)!} B^{\mu\nu} \varepsilon_{\mu\nu\mu_3 \cdots \mu_n} dx^{\mu_3} \wedge \cdots \wedge dx^{\mu_n}\,.} \tag{A.9}$$

Then application of the general Stokes' theorem to a $(n-1)$-dimensional hypersurface $\Sigma_{n-1} \subset M$ with boundary $\partial \Sigma_{n-1}$ yields

$$\int_{\Sigma_{n-1}} \left(d^{n-1} x\right)_\mu \nabla_\nu B^{\mu\nu} = \oint_{\partial \Sigma_{n-1}} \left(d^{n-2} x\right)_{\mu\nu} B^{\mu\nu}\,. \tag{A.10}$$

For $n = 3$ and a surface $\Sigma_2 \equiv S$, this result represents the *ordinary Stokes' theorem* as applied to the vector field $\vec{B}$ with components $B_i \equiv \frac{1}{2} \varepsilon_{ijk} B^{jk}$: $\int_S \overrightarrow{dS} \cdot \overrightarrow{\operatorname{rot}} \vec{B} = \oint_{\partial S} \overrightarrow{dx} \cdot \vec{B}$. In Minkowski space $\mathbb{R}^n$, the *charge $Q$ associated to a current $(j^\mu)$ given by a superpotential*, i.e. $j^\mu = \partial_\nu B^{\mu\nu}$ with $B^{\mu\nu} = -B^{\nu\mu}$, is given by the integral (A.10): this integral vanishes if $B^{\mu\nu}$ decreases fast enough on the boundary $\partial \Sigma_{n-1}$ or if $\partial \Sigma_{n-1}$ is the empty set.

### A.3   Vector fields

**Vector fields:**   A vector field $X$ on $M$ admits the local expression $X = X^\mu \partial_\mu$ where the coefficients $X^\mu(x)$ are to be viewed as the components of a contravariant vector field. The *Lie bracket* $[X, Y]$ of two vector fields $X$ and $Y$ on $M$ is again a vector field on $M$ with components

$$[X, Y]^\mu = X^\nu \partial_\nu Y^\mu - Y^\nu \partial_\nu X^\mu \,. \tag{A.11}$$

**Interior product and Lie derivative of forms:**   Given a vector field $X = X^\mu \partial_\mu$ on $M$, the *interior product* (or *contraction*) $i_X : \Omega^\bullet(M) \to \Omega^{\bullet-1}(M)$ of differential forms on $M$ with the vector field $X$ is the graded derivation of order $-1$ defined by

$$i_X f \equiv 0 \,, \quad \text{for 0-forms } f \,, \qquad i_X(dx^\mu) \equiv X^\mu \quad (\text{hence } i_X(\alpha_\mu dx^\mu) = X^\mu \alpha_\mu) \,. \tag{A.12}$$

If we view $p$-forms $\alpha(x, dx)$ as quantities depending on $x^\mu$ and $dx^\mu$, then we can write [74, 249] the interior product $i_X$ as

$$i_X = X^\mu \frac{\partial}{\partial(dx^\mu)} \,, \qquad \text{for } X = X^\mu \partial_\mu \,,$$

where $\partial / \partial(dx^\mu)$ acts as an antiderivation.

The *Lie derivative* $L_X : \Omega^\bullet(M) \to \Omega^\bullet(M)$ of forms with respect to the vector field $X = X^\mu \partial_\mu$ on $M$ is the derivation (i.e. graded derivation of order 0) which is defined by virtue of Cartan's 'magic formula' as the graded commutator of the graded derivations $i_X$ and $d$:

$$L_X \equiv [i_X, d] = i_X d + d i_X \,. \tag{A.13}$$

For 0-forms $f$, we thus have $L_X = i_X(df) = X^\mu \partial_\mu f$, i.e. the derivative of the function $f$ in the direction of the vector field $X$. The operators $d, i_X$ and $L_X$ satisfy the graded commutation relations

$$[d, d] = 0 \,, \quad [i_X, i_Y] = 0 \,, \quad [i_X, d] = L_X \,, \quad [L_X, d] = 0 \,, \quad [L_X, i_Y] = i_{[X,Y]} \,. \tag{A.14}$$

For a smooth map $f : M \to N$ between two smooth manifolds $M$ and $N$, the *tangent map (or differential) $Tf$ of $f$* is a linear map between the tangent bundles $TM$ and $TN$ which can be defined in terms of local coordinates $(x^\mu)$ of $M$ and $(y^i)$ of $N$ by

$$Tf \,:\, TM \,\longrightarrow\, TN$$

$$X^\mu \partial_\mu \,\longmapsto\, (Tf)(X^\mu \partial_\mu) = X^\mu \frac{\partial y^i}{\partial x^\mu} \partial_i \,. \tag{A.15}$$

Thus, the map $Tf$ is represented in local coordinates by the Jacobian matrix of the map $x \mapsto y = f(x)$. It has the fundamental properties

$$T(f \circ g) = Tf \circ Tg \,, \qquad T(\mathrm{id}_M) = \mathrm{id}_{TM} \,.$$

For a diffeomorphism $f : M \to N$, the tangent map $Tf$ is also referred to as *push-forward* map and denoted by $f_*$. Relation (A.15) then amounts to the usual change of variables formula for vector fields.

## A.4  Mapping of differential forms

A smooth map $f : M \to N$ between two smooth manifolds $M$ and $N$ induces a so-called *pullback map* $f^* : \Omega^\bullet(N) \to \Omega^\bullet(M)$ of differential forms on $N$ to differential forms on $M$. The latter map is a linear map. In more detail, for a 0-form $g : N \to \mathbb{R}$ on $N$, its pullback to $M$ is the function $g \circ f : M \to \mathbb{R}$, i.e. $f^*g \equiv g \circ f$. For a 1-form $\alpha$ on $N$, its pullback $f^*\alpha$ to $M$ is a 1-form which acts on a vector field $X$ on $M$ by virtue of

$$(f^*\alpha)(X) \equiv \alpha\Big(Tf(X)\Big), \tag{A.16}$$

where $Tf : TM \to TN$ denotes the tangent map of $f$, see Eqn. (A.15). In terms of local coordinates $(x^\mu)$ of $M$ and $(y^i)$ of $N$, we thus have

$$f^* : T^*N \longrightarrow T^*M$$

$$\alpha_i dy^i \longmapsto \boxed{f^*(\alpha_i dy^i) = \alpha_i \frac{\partial y^i}{\partial x^\mu} dx^\mu.} \tag{A.17}$$

Definition (A.16) readily extends to arbitrary $p$-forms on $N$ by viewing the latter as multilinear maps acting on vector fields, e.g. see equations (C.35)-(C.36) below.

If $f : M \to N$ is the inclusion map, then the pullback map is the restriction of forms from $N$ to $M$. If $f : M \to N$ represents a diffeomorphism, then the pullback of differential forms locally amounts to the change of variables formula for covariant tensor fields (involving the Jacobian matrix of the map $x \mapsto y = f(x)$).

The pullback map enjoys several important properties. In particular, it preserves exterior products and it commutes with the exterior derivative $d$ ("naturality of $d$"):

$$f^*(\alpha \wedge \beta) = f^*\alpha \wedge f^*\beta, \qquad f^*(d\alpha) = d(f^*\alpha). \tag{A.18}$$

## A.5  Poincaré lemma on $\mathbb{R}^n$ and de Rham homotopy operator

To prove the Poincaré lemma on $\mathbb{R}^n$ (or on a star-shaped open subset of $\mathbb{R}^n$), one considers an arbitrary closed $p$-form $\alpha$ (with $p \geq 1$) on $\mathbb{R}^n$ and obtains a $(p-1)$-form $\beta$ with $\alpha = d\beta$ by applying a contracting homotopy operator to $\alpha$, i.e. a method going back to the work of J. A. Schouten and his collaborators [31, 74, 325, 326]. More precisely, one introduces the contraction (inner product (A.12)) $\rho \equiv i_V$ of forms by the *scaling vector field* $V \equiv x^\mu \frac{\partial}{\partial x^\mu}$ on $\mathbb{R}^n$: we have $\rho(dx^\mu) = i_V(dx^\mu) = x^\mu$ and thereby we can write $\rho = x^\mu \frac{\partial}{\partial(dx^\mu)}$. For the $p$-form $\alpha \equiv \alpha(x, dx)$, the so-called *contracting homotopy operator* (or *chain homotopy*) of the de Rham complex $\big(\Omega^\bullet(\mathbb{R}^n), d = dx^\mu \frac{\partial}{\partial x^\mu}\big)$ is now defined by

$$\boxed{I(\alpha) \equiv \int_0^1 \frac{d\lambda}{\lambda} (\rho\alpha)(\lambda x, \lambda dx),} \qquad \text{with } \rho = x^\mu \frac{\partial}{\partial(dx^\mu)}, \tag{A.19}$$

and for $p$-forms $\alpha$ satisfying $d\alpha = 0$, we then have $\alpha = d\beta$ with $\beta \equiv I(\alpha)$.

# B  Differentials in field space: Horizontal, vertical, BRST, Koszul-Tate, BV

In this appendix, we present a synthetic introduction to various differentials which are considered in the space of fields over Minkowski space-time $M = \mathbb{R}^n$. For this space denoted by

$E$ we assume a trivial product structure, i.e. $E = M \times \mathbb{R}^N$ where $\mathbb{R}^N$ labels a collection of $N$ real-valued fields $x \mapsto \varphi(x) \equiv (\varphi^a(x))_{a=1,\dots,N}$: from the mathematical point of view, $E$ is to be interpreted as a fibre bundle over $M$ and the fields as smooth sections in this bundle, see Figure 1 of Section 3. With first order Lagrangian densities in mind, we also consider the 1-jet bundle $J^1 E \equiv JE = M \times \mathbb{R}^N \times \mathbb{R}^{nN}$ over $M$ which includes the $nN$ first order derivatives $x \mapsto (\partial_\mu \varphi^a)(x)$, see again Figure 1. In various field theoretical investigations (like the study of symmetries and conservation laws), one encounters derivatives of fields of arbitrary high order and thus it is adequate to consider the infinite jet bundle $J^\infty E$ (see Subsection 6.1 and references given there for technical details).

## B.1 Horizontal differential

A *horizontal k-form* $\alpha$ on $J^\infty E$ (i.e. $\alpha \in \Omega^{k,0} \equiv \Omega^{k,0}(J^\infty E)$ in terms of the notation of Subsection 6.1), has the expression

$$\alpha = \sum_{0 \le \mu_1 < \cdots < \mu_k \le n-1} \alpha_{\mu_1 \cdots \mu_k}[\varphi] \, dx^{\mu_1} \wedge \cdots \wedge dx^{\mu_k} \,, \tag{B.1}$$

where the coefficients $\alpha_{\mu_1 \cdots \mu_k}[\varphi]$ are (smooth) local functions of the fields, i.e. they only depend on $x$, the field $\varphi$ and its partial derivatives (at the same space-time point) up to some finite order. The *horizontal differential* $d_\mathrm{h}$ on $\Omega^{k,0}$ (cf. Eqn. (177)) is now defined by

$$d_\mathrm{h} : \Omega^{k,0} \longrightarrow \Omega^{k+1,0}$$

$$\alpha \longmapsto d_\mathrm{h}\alpha \equiv dx^\mu \partial_\mu \alpha = dx^\mu \left( \frac{\partial \alpha}{\partial x^\mu} + \partial_\mu \varphi^a \frac{\partial \alpha}{\partial \varphi^a} + \partial_\mu \partial_\nu \varphi^a \frac{\partial \alpha}{\partial (\partial_\nu \varphi^a)} + \dots \right). \tag{B.2}$$

Since the linear operator $d_\mathrm{h}$ is nilpotent, we have a (cohomology) complex $\Omega^{\bullet,0} \equiv \oplus_{k \ge 0} \Omega^{k,0}$ (which is referred to as the *horizontal complex* [74]) as well as the associated *cohomology groups* $H^k(d_\mathrm{h})$ which are given by the algebraic Poincaré lemma, see Eqn. (181).

On the jet bundle $J^\infty E$ we may also introduce the differential in "field direction" which is referred to as the *vertical differential* $d_\mathrm{v}$, see Subsection 6.1.

## B.2 BRST differential

With gauge field theories in mind, we will not discuss here the vertical differential $d_\mathrm{v}$ describing generic field variations, but rather introduce the BRST differential $s$ which is associated to local symmetry transformations of fields. The infinitesimal form of these transformations is addressed is various places of the main text (in particular in equations (49)-(51)), in relationship with Eqn. (220) and in Subsection 7.3–Subsection 7.6): for the fields $\varphi = (\varphi^a)$, we consider the local transformation law (parametrized by smooth real-valued functions $x \mapsto f^r(x)$)

$$\delta_f \varphi^a = Q^a(f) \equiv Q^a_r(f^r) \equiv Q^a_r f^r + Q^{a\mu}_r \partial_\mu f^r \,, \tag{B.3}$$

where we limit ourselves to first order derivatives of the parameters $f^r$ since this case covers the standard physical applications like YM (Yang-Mills) theory and general relativity.

For instance, for pure YM-theory (with a structure group given by a compact matrix Lie group $G$), we have the YM potential $x \mapsto A_\mu(x) = A^r_\mu(x) T_r$ (where the anti-Hermitian matrices $T_r$ represent a basis of the Lie algebra $\mathtt{Lie}\, G$ associated to the Lie group $G$) and a local gauge transformation then has the expression

$$\delta_f A_\mu = D_\mu f \equiv \partial_\mu f + [A_\mu, f] \,, \qquad \text{with} \quad f(x) \equiv f^r(x) T_r \,. \tag{B.4}$$

The commutator of two such transformations with parameters $f_1$ and $f_2$, respectively, again represents such a transformation with a parameter which is the Lie commutator of $f_1$ and $f_2$, i.e.

$$[\delta_{f_1}, \delta_{f_2}] = \delta_{[f_1, f_2]}. \tag{B.5}$$

Thus, in YM-theory, the set $\{f : M \rightarrow \mathtt{Lie}\,G\}$ of infinitesimal gauge transformations, endowed with the Lie commutator, represents an (infinite-dimensional) Lie algebra which is referred to as the *gauge symmetry algebra* [74]. It was realized by BRS(T) [275] and put forward in particular by R. Stora [327] that one can associate a differential graded algebra to the gauge symmetry algebra (see also [328]). To do so, one replaces the gauge parameters $x \mapsto f(x) \equiv (f^r(x))$ by fields $x \mapsto c(x) \equiv (c^r(x))$ to which one assigns a *(Faddeev-Popov) ghost-number* $+1$ (and accordingly refers to them as ghost fields), the basic fields $A_\mu$ having a vanishing ghost-number. This means that the field space $E$ is also extended by these variables and thereby becomes graded (by the ghost-number). The local gauge transformation (B.4) then turns into the so-called *BRST transformation* $sA_\mu = D_\mu c$ where the linear operator $s$ is referred to as *BRST differential*. The $s$-variation of the ghost field $c$ is defined so as to reflect the structure (B.5) of the gauge symmetry algebra, the commutator of Lie algebra-valued fields now becoming a graded commutator, e.g. $[c, c] \equiv cc - (-1)^{1 \cdot 1} cc = cc + cc$:

$$sA_\mu = D_\mu c, \qquad sc = -\frac{1}{2}[c, c]. \tag{B.6}$$

The operator $s$ is assumed to commute with the derivation $\partial_\mu$. In summary, *the BRST differential* (as defined on the generators of the field algebra by Eqn. (B.6)) increases the ghost-number by one unit and it is nilpotent.

The line of arguments that we just presented for pure YM-theories generalizes straightforwardly to more general field theories with local symmetries, e.g. see references [23, 74, 230, 276].

## B.3 Koszul-Tate differential

Suppose we have a gauge field-type theory described by an action functional $S[\varphi]$ which is invariant under some local symmetry transformations (B.3). By virtue of Noether's second theorem (see Subsection 7.3), the invariance of the action functional $S[\varphi]$ under non-trivial gauge symmetries (B.3) is equivalent to the validity of non-trivial

$$\text{Noether identities}: \qquad 0 = (Q_r)^\dagger \left( \frac{\delta S}{\delta \varphi} \right) \equiv (Q_r^a)^\dagger \left( \frac{\delta S}{\delta \varphi^a} \right), \quad \text{for all } r. \tag{B.7}$$

(These relations represent identities relating the functional derivatives $\delta S/\delta \varphi^1, \ldots, \delta S/\delta \varphi^N$ and reflect the fact that the equations of motion are not all independent in a theory admitting local symmetries.) E.g. for pure YM-theory, the action functional

$$S[A] = -\frac{1}{4} \int_{\mathbb{R}^n} d^n x \, \mathrm{Tr}(F^{\mu\nu} F_{\mu\nu}), \qquad \text{with} \quad F_{\mu\nu} \equiv \partial_\mu A_\nu - \partial_\nu A_\mu + [A_\mu, A_\nu], \tag{B.8}$$

yields the equations of motion $D_\nu F^{\nu\mu} = 0$ and relations (B.3),(B.4),(B.7) thereby lead to the Noether identities

$$0 = D_\mu D_\nu F^{\nu\mu} = \frac{1}{2}[D_\mu, D_\nu]F^{\nu\mu} = -\frac{1}{2}[F_{\mu\nu}, F^{\mu\nu}], \tag{B.9}$$

which are trivially satisfied.

As was pointed out by J. Fisch and M. Henneaux [329] (following earlier work of J. Stasheff and collaborators [330]) an algebraic tool for dealing with the dynamics of gauge field-type theories (i.e. encoding information related to the equations of motion $\frac{\delta S}{\delta \varphi^a} = 0$) is given by the so-called Koszul-Tate differential [331] which is associated to the gauge symmetry transformations (B.3) or equivalently to the Noether identities (B.7). In the case of an irreducible gauge field theory (like pure YM-theory considered in equations (B.4)-(B.6)), one associates so-called *antifields* to the fields as follows. To the collection of basic fields $\varphi = (\varphi^a)$, one associates a collection of antifields $\varphi^* = (\varphi_a^*)$: to the latter and to their derivatives, one associates an *antifield number* $+1$, the basic fields $\varphi^a$ and their derivatives having a vanishing antifield number. Furthermore, to each non-trivial Noether identity (B.7) (or equivalently to each non-trivial gauge symmetry transformation (B.3) parametrized by $f^r$) one associates an antifield $c_r^*$, this field and its derivatives having antifield number 2. In other words, to the fields $\Phi \equiv (\varphi, c) \equiv (\varphi^a, c^r)$ (where the ghost fields $c^r$ may be thought of as labeling infinitesimal symmetry transformations or Noether identities) one associates antifields $\Phi^* \equiv (\varphi^*, c^*) \equiv (\varphi_a^*, c_r^*)$. This procedure amounts to an extension of the field space $E$ by antifields, the space $E$ thereby becoming graded (by the antifield number $p \in \mathbb{Z}$). We denote the set of $k$-forms of antifield number $p$ on the extended field space by $\Omega_p^{k,0}$. The so-called

$$\text{Koszul-Tate differential}: \qquad \delta : \Omega_p^{\bullet,0} \longrightarrow \Omega_{p-1}^{\bullet,0}, \tag{B.10}$$

is a linear operator that *lowers the antifield number by one unit*. The action of $\delta$ on the generators of the field algebra is defined by $\delta x^\mu = 0 = \delta(dx^\mu)$ and

$$\boxed{\delta \varphi^a = 0, \qquad \delta \varphi_a^* = \frac{\delta S}{\delta \varphi^a}, \qquad \delta c_r^* = (Q_r^a)^\dagger (\varphi_a^*).} \tag{B.11}$$

The assumption that $[\delta, \partial_\mu] = 0$ is tantamount to saying that the graded commutator of $\delta$ and $d_{\mathrm{h}}$ vanishes. We obviously have $\delta(\delta \varphi^a) = 0 = \delta(\delta \varphi_a^*)$ and

$$\delta(\delta c_r^*) = (Q_r^a)^\dagger (\delta \varphi_a^*) = (Q_r^a)^\dagger \left( \frac{\delta S}{\delta \varphi^a} \right) = 0,$$

where the last equality holds by virtue of the Noether identities (B.7). Thus, the Koszul-Tate operator is nilpotent and represents a differential on the complex $\Omega_\bullet^{\bullet,0} \equiv \oplus_{p \in \mathbb{Z}} \Omega_p^{\bullet,0}$. Since it lowers the degree (antifield number) of forms by one unit rather than increasing the degree (as it is the case for the operators $d_{\mathrm{h}}$ and $s$, it represents a *homology operator* rather than a cohomology operator. The differential $\delta$ provides an algebraic control of expressions related to the equations of motion and thereby it appears in the classification of conservation laws in field theory [23, 74, 230], see Subsection 7.2 (in particular Eqn. (246)).

For later reference, we spell out the $\delta$-variations (B.11) for pure YM-theory as described by the action (B.8) and the Noether identities (B.9):

$$\delta A_\mu = 0, \qquad \delta A_\mu^* = D^\nu F_{\nu\mu}, \qquad \delta c^* = D^\mu A_\mu^*. \tag{B.12}$$

## B.4  BV differential

The BV formalism [23, 74, 270–274] represents a symplectic-type formulation and generalization of the BRST formalism in that it introduces a graded bracket $[\![\cdot, \cdot]\!]$ with respect to which fields $\Phi$ and antifields $\Phi^*$ define pairs of conjugate variables, i.e. $[\![\Phi(x), \Phi^*(y)]\!] = \pm \delta(x-y)$. In the sequel, we provide an outline of this formalism which differs from the one given in Subsection 7.6.2. More precisely, we put forward here the fact that the BV differential represents (for an irreducible gauge field theory) the sum of the differentials $s$ and $\delta$ introduced above (the $s$-invariant action functional being extended to the so-called minimal action).

**General setting:** For concreteness, we again focus on the example of pure YM-theory as described by the action functional (B.8) which is invariant under the local symmetry transformations (B.4)-(B.5), the latter giving rise the BRST algebra (B.6) and the Noether identities (B.9) while the combination of the action and its local symmetries yields the $\delta$-variations (B.12). Thus, the starting point is the set $\Phi$ of basic fields and ghost fields as well as the corresponding antifields:

$$\text{Fields}: \qquad (\Phi^A) \equiv \Phi \equiv (A, c), \qquad \text{with} \quad A_\mu \equiv A^r_\mu T_r, \quad c \equiv c^r T_r, \qquad \text{(B.13)}$$

$$\text{Antifields}: \qquad (\Phi^*_A) \equiv \Phi^* \equiv (A^*, c^*), \qquad \text{with} \quad A^{\mu*} \equiv A^{\mu*}_r T^r, \quad c^* \equiv c^*_r T^r.$$

As noted above, the antifields $A^*$ and $c^*$ have antifield numbers 1 and 2, respectively, while the fields $\Phi^A$ have antifield number zero. In the conception of a symplectic formulation one not only introduces conjugate variables (i.e. presently the antifields), but one also extends the initial action by means of these variables: to the $s$-invariant action functional, we add a linear coupling of the $s$-variations of the fields $\Phi^A$ to the corresponding antifields $\Phi^*_A$, thus giving rise the so-called *minimal action*:

$$\boxed{S[\varphi] \rightsquigarrow S_{\min}[\Phi, \Phi^*] \equiv S[\varphi] + \int_M d^n x \, \Phi^*_A s \Phi^A.} \qquad \text{(B.14)}$$

For YM-theories, this expression takes the form

$$S[A] \rightsquigarrow S_{\min}[\Phi, \Phi^*] \equiv S[A] + \int_M d^n x \, \text{Tr}\left(A^*_\mu s A^\mu + c^* s c\right), \qquad \text{(B.15)}$$

with $sA$ and $sc$ given by Eqn. (B.6). The requirement that the minimal action has ghost-number zero, i.e. $\text{gh}(S_{\min}) = 0$, entails that we have $\text{gh}(A^*_\mu) = -1$ and $\text{gh}(c^*) = -2$.

**Combing the differentials $s$ and $\delta$ :** Since the $s$-operator acts on the basic fields like a gauge transformation (with the parameters $f^r$ replaced by the ghost fields $c^r$), it is natural to extend its action to the Lie algebra-valued fields $A^*_\mu$ and $c^*$ by requiring them to transform with the adjoint representation under gauge transformations, i.e. we consider $s$-variations $sA^*_\mu = -[c, A^*_\mu]$ and $sc^* = -[c, c^*]$. Since the field $A_\mu$ is invariant under $\delta$-variations, it is also natural to assume that this holds as well for the field $c$. Altogether, we then have the variations

$$\begin{aligned} sA_\mu &= D_\mu c, & \delta A_\mu &= 0, \\ sc &= -\frac{1}{2}[c, c], & \delta c &= 0, \\ sA^*_\mu &= -[c, A^*_\mu], & \delta A^*_\mu &= D^\nu F_{\nu\mu}, \\ sc^* &= -[c, c^*], & \delta c^* &= D^\mu A^*_\mu, \end{aligned} \qquad \text{(B.16)}$$

and the relations $0 = s^2 = \delta^2 = [s, \delta]$.

We presently assume that the algebra generated by the fields $\Phi^A$ and antifields $\Phi^*_A$ is graded by the ghost-number and we define the

$$\text{BV operator}: \qquad \boxed{\delta_Q \equiv s + \delta.} \qquad \text{(B.17)}$$

This operator is then nilpotent and acts on the complex $\Omega^{\bullet,0}_\bullet$ as a differential which raises the ghost-number by one unit. Moreover, it can be checked that this operator leaves the minimal action (B.15) invariant, i.e. $\delta_Q S_{\min}[\Phi, \Phi^*] = 0$. By construction, the so-defined BV differential describes both the gauge symmetry algebra and the dynamics that this algebra leaves invariant.

It relies on the introduction of ghost fields and of antifields (for all fields) as well as on the extension of the gauge invariant action $S$ to the associated minimal action (as appropriate for a symplectic-type formulation). *The BV operator represents an extension of the BRST operator to antifields* (and thus to the antifield dependent action $S_{\min}$): upon setting to zero all antifields together with their $\delta_Q$-variations, one recovers the gauge invariant action $S[\varphi]$ and the $s$-variations of the fields $\Phi^A$ (together with the equations of motion of the basic fields $\varphi^a$), see equations (B.14) and (B.16).

**Alternative approach:**  The relation $\delta\varphi_a^* = \frac{\delta S}{\delta\varphi^a}$ (see Eqn. (B.11)) motivates us to define the action of the *BV operator* $\delta_Q$ on the generators $\Phi_A^*$ and $\Phi^A$ as follows:

$$\text{BV operator:} \qquad \boxed{\delta_Q\Phi_A^* \equiv \frac{\delta S_{\min}}{\delta\Phi^A}\,, \qquad \delta_Q\Phi^A \equiv \frac{\delta S_{\min}}{\delta\Phi_A^*} = s\Phi^A\,.} \tag{B.18}$$

For pure YM-theory, this definition yields the same variations of fields and antifields as the expression (B.17) and these results coincide with those encountered in Eqn. (326) where we introduced the BV differential by starting from the BV bracket (see equations (317)-(318)). In that context the BV differential is rather denoted by $\mathcal{S}_S$ and the corresponding variations are referred to as BRST transformations.

The case of reducible gauge field theories is addressed for instance in references [23, 230]. In this case, the gauge symmetry algebra has a more complicated structure than (B.5) and this implies that the expression (B.17) of the BV operator then involves some extra contributions $\sum_{p\geq 0} s_p$ (where $s_p$ increases the antifield number by $p$); similarly the minimal action (B.14) (which involves three terms of antifield number zero, one and two, respectively) then also contains some extra terms having an antifield number which is greater than or equal to 2.

# C  Poisson brackets and symplectic forms in classical mechanics

Before discussing the geometric formulation of classical field theory in Appendix D, we consider the corresponding formulation of classical mechanics. In fact, various physical and mathematical notions are much more familiar within this context and the "continuum limit" of mechanics formally yields the expressions of field theory. Moreover, the functional analytic complications related to the infinite number of degrees of freedom of field theory do not have to be dealt with in mechanics.

While the mathematically minded textbooks on classical mechanics like [57, 118, 323, 332–334] generally rely on symplectic geometry, we rather start with the Poisson bracket on phase space which is familiar from physics. For the general mathematical background, we refer in particular to [118, 133].

## C.1  Poisson brackets

**Poisson brackets:**  Consider a Lagrangian system in non-relativistic mechanics defined on a configuration space $\mathcal{Q}$ (i.e. a smooth real manifold parametrized by local coordinates $(q^i)_{i=1,\dots,n}$), the associated momenta being denoted by $(p_i)_{i=1,\dots,n}$. The *canonical expression* for the Poisson bracket of any two smooth real-valued functions $f, g$ on *phase space $M$* parametrized by $(q^i, p_i)_{i=1,\dots,n}$ reads as follows:

$$\text{Canonical expression of Poisson bracket:} \qquad \boxed{\{f, g\} = \sum_{i=1}^n \left( \frac{\partial f}{\partial q^i}\frac{\partial g}{\partial p_i} - \frac{\partial f}{\partial p_i}\frac{\partial g}{\partial q^i} \right).} \tag{C.1}$$

Here, the functions $f$ and $g$ may explicitly depend on time as well. With the notation

$$\vec{Q} \equiv (Q^I)_{I=1,\ldots,N=2n} \equiv (q^1, \ldots, q^n, p_1, \ldots, p_n), \tag{C.2}$$

and $\partial_I \equiv \partial/\partial Q^I$, the expression (C.1) may be rewritten under the following form:

Poisson bracket:
$$\boxed{\{f, g\} = \sum_{I,J=1}^{N} (\partial_I f) \Theta^{IJ} (\partial_J g),} \quad \text{i.e.} \quad \{Q^I, Q^J\} \equiv \Theta^{IJ}. \tag{C.3}$$

For our choice of canonical (Darboux) coordinates, i.e. phase space coordinates $(Q^I) \equiv (q^i, p_i)$ such that $\{f, g\}$ takes the form (C.1), we have explicitly

$$\Theta \equiv (\Theta^{IJ}) = \begin{bmatrix} 0_n & \mathbb{1}_n \\ -\mathbb{1}_n & 0_n \end{bmatrix}. \tag{C.4}$$

The Poisson tensor $(\Theta^{IJ})$ is antisymmetric and non-degenerate (and constant for the choice of canonical coordinates).

If one goes over to other local coordinates of the phase space manifold by an invertible coordinate transformation, then $(\partial_I f)$ transforms like a covariant vector field and $(\Theta^{IJ})$ like a contravariant tensor field.[31] Thus, we still have the expression (C.3) for the Poisson bracket, but with a matrix $(\Theta^{IJ})$ that depends in general on the coordinates $(Q^I)$, which is antisymmetric, non-degenerate on $M$ and satisfies the

Poisson-Jacobi identity: $\quad \Theta^{IL} \partial_L \Theta^{JK} + \text{cyclic permutations of } I, J, K = 0, \tag{C.5}$

the latter relation being equivalent to the

Jacobi identity for the Poisson bracket: $\quad \{f, \{g, h\}\} + \{g, \{h, f\}\} + \{h, \{f, g\}\} = 0. \tag{C.6}$

In Eqn. (C.5) and in the sequel, we use the Einstein summation convention over identical indices. Note that relation (C.5) is trivially satisfied for the choice of canonical coordinates, i.e. for (C.4).

By definition [118, 133], a finite-dimensional *symplectic manifold* $\left(M, (\Theta^{IJ})\right)$ is a manifold $M$ of even dimension $2n$ (for some $n$ with $1 \leq n < \infty$) which is endowed with a tensor field $(\Theta^{IJ})$ (with $I, J \in \{1, \ldots, 2n\}$) that is antisymmetric, non-degenerate and satisfies the Poisson-Jacobi identity (C.5). From the point of view of physics, the most important example is the one that we just considered, i.e. $M$ is the cotangent bundle $T^*\mathcal{Q}$ associated to the configuration space manifold $\mathcal{Q}$ of a mechanical system, $\mathcal{Q}$ having local coordinates $(q^i)$ and $T^*\mathcal{Q}$ having local coordinates $(Q^I) \equiv (q^i, p_i)$. For any finite-dimensional symplectic manifold $M$, the so-called *Darboux theorem* states that there exist local coordinates (C.2) (referred to as *Darboux* or *canonical coordinates*) such that the Poisson bracket defined by (C.3) takes the form (C.1).

## C.2  Poisson structure

**Poisson structure:** A symplectic manifold $\left(M, (\Theta^{IJ})\right)$ represents a particular instance of the more general notion of a *Poisson manifold* for which the tensor field $(\Theta^{IJ})$ on $M$ may be degenerate. While a (finite-dimensional) symplectic manifold is necessarily of even dimension $N = 2n$, this does not have to be the case for a Poisson manifold. An instructive example with $N = 3$ (related to the dynamics of rigid bodies in $\mathbb{R}^3$) is given by $M = so(3)^* \cong \mathbb{R}^3$

---

[31]In the case where phase space is simply a vector space, we can consider invertible *linear* transformations which implies that the associated Jacobian is constant.

(dual of the Lie algebra $so(3)$): in this case, the phase space $M$ is the vector space $\mathbb{R}^3$ (with coordinates $\vec{Q} \equiv (Q^I)_{I=1,2,3} \equiv (x, y, z)$) endowed with the so-called Lie-Poisson structure: For $f, g \in C^\infty(\mathbb{R}^3)$, the *Lie-Poisson bracket* is defined (in terms of the Levi-Civita symbol $\varepsilon^{IJK}$) by

$$\{f, g\} \equiv \varepsilon^{IJK} Q^K \partial_I f \, \partial_J g = \vec{Q} \cdot \left[ (\vec{\nabla} f) \times (\vec{\nabla} g) \right], \qquad \text{hence} \qquad \{Q^I, Q^J\} = \varepsilon^{IJK} Q^K. \quad \text{(C.7)}$$

Thus, the Poisson matrix represents a (non-constant) antisymmetric $(3 \times 3)$-matrix,

$$(\Theta^{IJ}) = \begin{bmatrix} 0 & z & -y \\ -z & 0 & x \\ y & -x & 0 \end{bmatrix}, \quad \text{(C.8)}$$

which implies that its determinant necessarily vanishes, i.e. we have a degenerate Poisson structure. We note that the rank of the matrix (C.8) is not constant on $M = \mathbb{R}^3$: it is two for $\vec{Q} \neq \vec{0}$ and zero for $\vec{Q} = \vec{0}$. The rotationally invariant quantity $\|\vec{Q}\|^2$ represents a *Casimir function* for the Lie-Poisson bracket i.e. $\{f, \|\vec{Q}\|^2\} = 0$ for any function $f \in C^\infty(\mathbb{R}^3)$. The *level surfaces* of the Casimir function $\vec{Q} \mapsto \|\vec{Q}\|^2$ are 2-spheres $S_R^2$ of radius $R \geq 0$ centered at the origin. The collection of these spheres defines a **symplectic foliation** of the Poisson manifold $(\mathbb{R}^3, (\Theta^{IJ}))$, i.e. a *foliation of the manifold $\mathbb{R}^3$ by symplectic leaves* given by the spheres $S_R^2$ with $R \geq 0$. (This foliation is *singular* since $\dim S_R^2 = 2$ for $R > 0$ and $\dim S_0^2 = 0$.) Restriction of the degenerate Lie-Poisson bracket to a symplectic leaf $S_R^2$ with $R > 0$ yields a non-degenerate Poisson structure on this leaf. (In terms of spherical coordinates $(\varphi, \theta)$ we have $\{\theta, \varphi\} = (R \sin \theta)^{-1}$ on $S_R^2$.)

A *Hamiltonian system* on a Poisson manifold $(M, (\Theta^{IJ}))$ is defined by a choice of function $H : M \to \mathbb{R}$ ("energy function") which determines the time evolution of any observable $f$ on $M$ (i.e. of any smooth function $f : M \to \mathbb{R}$) by virtue of the

$$\text{Hamiltonian equation of motion:} \qquad \boxed{\dot{f} = \{f, H\}.} \quad \text{(C.9)}$$

Since there is a one-to-one correspondence between a Poisson tensor field $(\Theta^{IJ})$ on $M$ and a Poisson bracket $\{\cdot, \cdot\}$ on $M$, a Poisson manifold may equivalently be defined in terms of the latter [118, 133]: a *Poisson manifold* is a (not necessarily even-dimensional) smooth real manifold $M$ for which the associative, commutative algebra $C^\infty(M)$ of smooth real-valued functions on $M$ is not only equipped with the ordinary pointwise multiplication of functions, but also with a so-called *Poisson structure*, i.e. a *Poisson bracket* on the vector space $C^\infty(M)$,

$$\begin{aligned} \{\cdot, \cdot\} : C^\infty(M) \times C^\infty(M) &\longrightarrow C^\infty(M) \\ (f, g) &\longmapsto \{f, g\}. \end{aligned} \quad \text{(C.10)}$$

By definition, the latter bracket is supposed to have the properties of $\mathbb{R}$-bilinearity, antisymmetry (also referred to as skew-symmetry or anticommutativity), Jacobi identity (C.6) and

$$\text{Derivation (or Leibniz) rule:} \qquad \{f, gh\} = \{f, g\} h + g \{f, h\}. \quad \text{(C.11)}$$

These properties imply that the Poisson bracket has the local expression (C.3) where $(\Theta^{IJ})$ is an antisymmetric tensor field on $M$ which satisfies the Poisson-Jacobi identity (C.5).

The non-degeneracy condition for $(\Theta^{IJ})$ is equivalent to the

$$\text{Non-degeneracy requirement:} \{f, g\} = 0 \text{ for all } g \in C^\infty(M) \text{ implies that } f \text{ is constant.} \quad \text{(C.12)}$$

If this requirement is fulfilled, we have a symplectic manifold and the Poisson bracket takes the form (C.1) in terms of Darboux coordinates: from (C.1) and $\{f, g\} = 0$ for all $g \in C^\infty(M)$ it then follows that $\partial f / \partial q^i = 0 = \partial f / \partial p_i$ for all $i$, hence $f$ is constant (on each connected component of the manifold $M$).

## C.3 Symplectic form

**Symplectic form:** Consider a symplectic manifold $(M, (\Theta^{IJ}))$. Following the sign conventions of reference [133], we set $(\pi^{IJ}) \equiv (-\Theta^{IJ})$. The matrix $(\Theta^{IJ})$ being invertible, the matrix $(\pi^{IJ})$ also is and its inverse is denoted by $(\omega_{IJ})$, i.e. $(\omega_{IJ}) \equiv ((-\Theta^{IJ}))^{-1}$. The properties of the Poisson tensor field $(\Theta^{IJ})$ imply that the covariant tensor field $(\omega_{IJ})$ defines a closed, non-degenerate 2-form on phase space, i.e. a so-called

$$\text{symplectic form:} \qquad \boxed{\omega \equiv \frac{1}{2}\,\omega_{IJ}\,dQ^I \wedge dQ^J\,.} \qquad \text{(C.13)}$$

Here, $\omega_{IJ} = -\omega_{JI}$ and the matrix $(\omega_{IJ}(\vec{Q}))$ is non-degenerate for all $\vec{Q}$. Moreover, the closedness property $d\omega = 0$ is equivalent to $0 = \partial_I \omega_{JK} +$ cyclic permutations of $I, J, K$, this relation being equivalent to the Poisson-Jacobi identity (C.5) for $(\Theta^{IJ})$. In terms of canonical coordinates $(Q^I) \equiv (q^i, p_i)$, we have the following expression (corresponding to the expression (C.1) for the Poisson bracket on a symplectic manifold):

$$\text{symplectic form in canonical coordinates:} \qquad \omega = dq^i \wedge dp_i\,. \qquad \text{(C.14)}$$

In physics, one usually starts with a dynamical system whose coordinates $(q^i)$ parametrize an $n$-dimensional configuration space manifold $\mathcal{Q}$ and whose dynamics is described by some *Lagrangian* $L : T\mathcal{Q} \to \mathbb{R}$, i.e. a smooth real-valued function on the *tangent bundle* $T\mathcal{Q}$ of $\mathcal{Q}$. If this Lagrangian is non-degenerate (i.e. its Hessian $\det\left(\frac{\partial^2 L}{\partial \dot{q}^i \partial \dot{q}^j}\right)$ does not vanish), then one can go over to the Hamiltonian function $H$ by the Legendre transform, $H \equiv p_i \dot{q}^i - L$ (where $p_i \equiv \frac{\partial L}{\partial \dot{q}^i}$): the latter is a well-defined smooth function on the cotangent bundle $T^*\mathcal{Q}$ which is parametrized by local coordinates $(q^i, p_i)$. On general grounds, the *cotangent bundle* $M = T^*\mathcal{Q}$ is a $(2n)$-dimensional manifold which is *exact symplectic* since it is endowed with a symplectic 2-form $\omega$ which is the exterior derivative of a naturally given and globally defined 1-form $\theta$ on $M$, the so-called *canonical 1-form*:[32]

$$\text{symplectic form on } M = T^*\mathcal{Q}: \qquad \boxed{\omega \equiv -d\theta\,.} \qquad \text{(C.15)}$$

In terms of Darboux coordinates, we have $\theta \equiv p_i dq^i$. The dynamical equations then take a simple geometric expression in terms of the symplectic structure, see next subsection.

While the cotangent bundles are the symplectic manifolds that are of primary interest in mechanics since they originate from its Lagrangian formulation, there exist symplectic manifolds that are neither cotangent bundles nor exact symplectic, e.g. the unit 2-sphere $S^2$ endowed with a volume form $\omega$. As a matter of fact, there exist physical systems (like the spinning massive particle in Minkowski space-time [57]) which do not admit a standard Lagrangian formulation and for which the phase space is neither a cotangent bundle nor exact symplectic. Yet, (by virtue of the Poincaré lemma) the relation $\omega = -d\theta$ holds *locally* on *any* symplectic manifold due to the fact that $d\omega = 0$.

For a Poisson manifold with a degenerate Poisson structure we saw in the previous subsection that the restriction of the Poisson bracket to a symplectic leaf yields a non-degenerate Poisson bracket (and thereby a symplectic structure) on this leaf. E.g. for the Lie-Poisson bracket (C.7) on $M = so(3)^* \cong \mathbb{R}^3$, the inherited symplectic structure on the unit sphere is given by the area form $\omega = \sin\theta\, d\theta \wedge d\varphi$.

In view of its importance for constrained Hamiltonian systems in mechanics (as well as in field theory), we note that a manifold $M$ endowed with a closed 2-form (which is possibly

---

[32]Beware of not mixing up the 1-form $\theta$ on $M = T^*\mathcal{Q}$ and the Poisson matrix $(\Theta^{IJ}) = ((-\omega_{IJ}))^{-1}$ which is associated to the symplectic 2-form $\omega \equiv -d\theta$.

degenerate so that $M$ is not necessarily of even dimension) is called a *presymplectic manifold*; the 2-form is then referred to as a *presymplectic form*. (We note that some authors include in this definition the requirement that the 2-form $\omega$ is of constant rank.)

### C.4 Hamiltonian vector fields and Hamiltonian equations

**Hamiltonian vector fields and Hamiltonian equations:** For a symplectic manifold, the definition of the Poisson bracket and the formulation of the Hamiltonian equations of motion can be described in geometric terms as follows.

We start from a given smooth function $f : M \to \mathbb{R}$ ("Hamiltonian") on the symplectic manifold $(M, \omega)$ and associate to it the so-called *Hamiltonian vector field $X_f$* on $M$ which is uniquely defined by the relation

$$i_{X_f}\omega = df . \tag{C.16}$$

Here, $i_{X_f}\omega$ denotes the interior product of the 2-form $\omega$ with the vector field $X_f \equiv X_f^I \partial_I$: in terms of local coordinates $(Q^I)$ on $M$, the relation (C.16) reads

$$X_f^I \omega_{IJ} dQ^J = (\partial_J f) dQ^J , \qquad \text{i.e.} \quad X_f^I \omega_{IJ} = \partial_J f . \tag{C.17}$$

Since the matrix $(\omega_{IJ})$ is invertible, it admits an inverse $(\pi^{IJ}) \equiv (-\Theta^{IJ})$: multiplication of relation (C.17) with $\Theta^{JK}$ then yields an *explicit expression for the vector field $X_f$ associated to $f$*:

$$X_f = -\Theta^{IJ}(\partial_I f)\partial_J . \tag{C.18}$$

From $\omega(X_f, X_g) = i_{X_g} i_{X_f}\omega = -i_{X_f} i_{X_g}\omega = -i_{X_f}(dg) = \Theta^{IJ} \partial_I f\, \partial_J g$ and Eqn. (C.3), we thus obtain the following expression for the

Poisson bracket on a symplectic manifold $(M, \omega)$: $\quad \boxed{\{f, g\} = \omega(X_f, X_g) .} \qquad$ (C.19)

For a function $H \in C^\infty(M)$, the associated vector field $X_H$ may therefore also be written as follows:

Hamiltonian vector field associated to $H \in C^\infty(M)$: $\quad \boxed{X_H = -\{H, \cdot\} .} \qquad$ (C.20)

For a particle (which moves on the configuration space $\mathcal{Q}$ and whose dynamics is governed by the Hamiltonian function $H \in C^\infty(M = T^*\mathcal{Q})$), the trajectories $t \mapsto (Q^I(t)) \equiv (\vec{q}(t), \vec{p}(t))$ in phase space are solutions of the Hamiltonian equations of motion $\dot{Q}^I = \{Q^I, H\} = -\{H, Q^I\}$: by virtue of (C.20), these equations may be rewritten as

Hamiltonian equations: $\quad \boxed{\dfrac{dQ^I}{dt}(t) = X_H^I(\vec{Q}(t)),} \qquad$ for $I \in \{1, \ldots, N\}$. (C.21)

This means that *the trajectories in phase space are the integral curves of the vector field $X_H$ associated to the Hamiltonian function $H$*. In terms of canonical coordinates $(Q^I) \equiv (q^i, p_i)$ on $M$, one infers from (C.4) that we have the following familiar expressions:

$X_H$ in canonical coordinates: $\quad X_H = \dfrac{\partial H}{\partial p_i}\dfrac{\partial}{\partial q^i} - \dfrac{\partial H}{\partial q^i}\dfrac{\partial}{\partial p_i},$

Hamiltonian equations in canonical coordinates: $\quad \dot{q}^i = \dfrac{\partial H}{\partial p_i}, \quad \dot{p}_i = -\dfrac{\partial H}{\partial q^i} .$ (C.22)

We note that *equations* (C.18) *and* (C.20)-(C.21) *hold on any Poisson manifold* $\left(M, (\Theta^{IJ})\right)$ *with local coordinates* $(Q^I)_{I=1,\ldots,N}$ and not only on symplectic manifolds. If the Poisson structure is degenerate, the restriction to the symplectic leaves allows for the introduction of canonical coordinates (on these leaves) in terms of which one again gets the expressions (C.22).

For later reference, we remark that the Lie bracket (see Eqn. (A.11)) of two Hamiltonian vector fields is again a Hamiltonian vector field and more precisely we have

$$[X_f, X_g] = -X_{\{f,g\}}.$$
(C.23)

**Symmetries and conserved quantities:** Let us discuss the relationship between continuous global symmetries and conserved quantities in the symplectic setting by considering the example of *translation invariance* for a particle moving in the configuration space $\mathcal{Q} \equiv \mathbb{R}^n$. Under an infinitesimal translation parametrized by constants $\varepsilon^k$ (with $k \in \{1, \ldots, n\}$), we have ("lift of the action of the translation group on the configuration space $\mathcal{Q} \equiv \mathbb{R}^n$ to an action on the phase space $M \equiv T^*\mathcal{Q}$" [118, 133])

$$\delta q^i = \varepsilon^k \partial_k q^i = \varepsilon^i, \qquad \delta p_i = 0 \qquad (\text{with } \partial_k \equiv \tfrac{\partial}{\partial q^k}).$$

Thus, the canonical 1-form $\theta = p_i dq^i$ is invariant under the vector field $v \equiv \varepsilon^k \partial_k$ acting on phase space $M$ by the Lie derivative $L_v$, i.e. $L_v q^i = \varepsilon^i$ and $L_v p_i = 0$. From $0 = L_v \theta \equiv (i_v d + d i_v)\theta$ (where $i_v$ denotes the interior product with respect to the vector field $v$) and $\omega = -d\theta$, it follows that

$$i_v \omega = dJ, \qquad \text{with} \qquad J \equiv i_v \theta.$$
(C.24)

Since the Lie derivative commutes with the exterior derivative, the symplectic 2-form $\omega$ is also invariant under translations, i.e. $L_v \omega = 0$. For the vector field $v = \varepsilon^k \partial_k$ describing translations we have the explicit expression

$$J \equiv i_v \theta = i_v (p_i dq^i) = p_i \varepsilon^i.$$
(C.25)

Thus, the function $J$ on phase space represents the momentum of the particle [117, 118]. From the canonical equations of motion $\dot{q}^i = \{q^i, H\}$, $\dot{p}_i = \{p_i, H\}$ (involving a *given* Hamiltonian function $H$), it follows that

$$\dot{J} = \varepsilon^i \dot{p}_i = -\varepsilon^i \partial_i H.$$

In summary, *the momentum is conserved in time for a translation invariant Hamiltonian $H$* (Hamiltonian version of Noether's first theorem applied to the translation group). In the setting of symplectic geometry, conserved quantities like the momentum are formulated in mathematical terms as "momentum maps", see references [117, 118, 133] for details.

### C.5 Explicitly time-dependent and relativistic systems

**Case of explicitly time-dependent Hamiltonians:** In some instances (in particular for time-dependent Hamiltonians) it is convenient to consider time as a dependent variable [1, 334] and to parametrize trajectories by some real parameter $s$. (This view-point is natural in the context of relativistic mechanics, see last paragraph below.) In this case, one considers the so-called *extended configuration space* $\mathbb{R} \times \mathcal{Q}$ parametrized by the time coordinate $t$ and the position coordinates $q^i$. The *"doubly extended phase space"* $\mathcal{P} \equiv T^*(\mathbb{R} \times \mathcal{Q}) = \mathbb{R}^2 \times T^*\mathcal{Q}$ is now parametrized by $(t, q^i, p_i, E)$ where $E$ corresponds to the energy (canonically conjugate

variable associated to $t$). In terms of canonical coordinates, the *canonical* 1-*form* $\theta$ and the associated *symplectic* 2-*form* $\omega$ on $\mathcal{P} = T^*(\mathbb{R} \times \mathcal{Q})$ read

$$\theta = p_i \, dq^i - E \, dt \,, \qquad \boxed{\omega \equiv -d\theta = dq^i \wedge dp_i + dE \wedge dt \,.} \qquad \text{(C.26)}$$

The associated Poisson bracket of any two smooth functions $f, g$ on $\mathcal{P}$ has the expression

$$\{f, g\} = \frac{\partial f}{\partial q^i} \frac{\partial g}{\partial p_i} - \frac{\partial f}{\partial p_i} \frac{\partial g}{\partial q^i} + \frac{\partial f}{\partial E} \frac{\partial g}{\partial t} - \frac{\partial f}{\partial t} \frac{\partial g}{\partial E} \,,$$

which yields $\{q^i, p_j\} = \delta^i_j$ and $\{E, t\} = 1$. The Hamiltonian vector field associated to a function $h$ on $\mathcal{P}$ (with respect to the symplectic form $\omega$) writes

$$X_h = \frac{\partial h}{\partial p_i} \frac{\partial}{\partial q^i} - \frac{\partial h}{\partial q^i} \frac{\partial}{\partial p_i} + \frac{\partial h}{\partial t} \frac{\partial}{\partial E} - \frac{\partial h}{\partial E} \frac{\partial}{\partial t} \,.$$

Henceforth, the *Hamiltonian equations of motion* $\frac{du}{ds} = X_h(u)$ take the form

$$\frac{dq^i}{ds} = \frac{\partial h}{\partial p_i} \,, \qquad \frac{dp_i}{ds} = -\frac{\partial h}{\partial q^i} \,, \qquad \frac{dt}{ds} = -\frac{\partial h}{\partial E} \,, \qquad \frac{dE}{ds} = \frac{\partial h}{\partial t} \,. \qquad \text{(C.27)}$$

Now, suppose we have a dynamical system described by a time-dependent Hamiltonian function[33] $\mathcal{H}(\vec{q}, \vec{p}, t)$ on ordinary phase space (parametrized by $(\vec{q}, \vec{p})$). For the function

$$h(\vec{q}, \vec{p}, t, E) \equiv \mathcal{H}(\vec{q}, \vec{p}, t) - E \,, \qquad \text{(C.28)}$$

on $\mathcal{P}$, the evolution equations (C.27) for $t$ and $E$ then reduce to

$$\frac{dt}{ds} = 1 \,, \qquad \frac{dE}{ds} = \frac{\partial \mathcal{H}}{\partial t} \,. \qquad \text{(C.29)}$$

The first of these relations implies that $t$ and $s$ differ by a constant which we choose to vanish for simplicity. Since $h$ does not explicitly depend on $s$, we have $\frac{dh}{ds} = 0$. Accordingly, we can restrict ourselves to the submanifold of $\mathcal{P}$ where $h \equiv 0$: on this space, the Hamiltonian $\mathcal{H}$ represents the actual energy $E$ and if $\mathcal{H}$ does not explicitly depend on time, then the energy is conserved by virtue of (C.29).

By pulling back the canonical 1-form $\theta$ to configuration space $\mathcal{Q}$, we obtain the

$$\text{Lagrangian 1-form :} \qquad p_i \, dq^i - \mathcal{H} \, dt = (p_i \, \dot{q}^i - \mathcal{H}) \, dt = \mathcal{L} \, dt \,. \qquad \text{(C.30)}$$

For a *time independent Lagrangian* $\mathcal{L}$, we have $L_{\partial_t}(\mathcal{L} \, dt) = 0$ (where $L_{\partial_t}$ denotes the Lie derivative with respect to the vector field $\partial_t$) and the *conserved energy* is given by

$$i_{\partial_t} \theta \Big| = i_{\partial_t}(p_i \, dq^i - E \, dt) \Big| = -E \,,$$

where the bar denotes the restriction to solutions $t \mapsto (\vec{q}(t), \vec{p}(t))$ of the Hamiltonian equations of motion. For further elaboration on (and application of) the extended phase space approach, we refer to the work [335].

---

[33]We use the notation $\mathcal{H}$ and $\mathcal{L}$ instead of $H$ and $L$ so as to stress the similarities with field theory.

**Relativistic mechanics:** Since this subject does not require any extra work or input, we briefly show that the expressions that we just introduced also provide the symplectic formulation of relativistic mechanics in Minkowski space. To do so, we consider the natural system of units where $c \equiv 1$ and the Minkowski metric $(\eta_{\mu\nu}) \equiv \text{diag}(+, -, \ldots, -)$.

In terms of standard coordinates $(x^\mu) \equiv (t, x^i) \equiv (t, \vec{x})$ of Minkowski space $M \equiv \mathbb{R}^n$ and $(p^\mu) \equiv (E, \vec{p})$ of momentum space, the *canonical* 1-*form* $\theta$ and the associated *symplectic* 2-*form* $\omega$ on $T^*M$ read

$$\theta = -p_\mu \, dx^\mu = \vec{p} \cdot d\vec{x} - E \, dt \,, \qquad \boxed{\omega \equiv -d\theta = dp_\mu \wedge dx^\mu = d\vec{x} \wedge d\vec{p} + dE \wedge dt \,.} \quad \text{(C.31)}$$

Thus, we have the same expressions as in Eqn. (C.26), The corresponding Poisson brackets are given by $\{x^\mu, p^\nu\} = -\eta^{\mu\nu}$.

For a *free relativistic particle* of mass $m$, we have $p^i = \gamma m \dot{x}^i$ (with $\dot{x}^i \equiv dx^i/dt$ and $\gamma \equiv (1 - \dot{\vec{x}}^2)^{-1/2}$) and a dynamics determined by the Hamiltonian

$$\mathcal{H}(\vec{p}) = E \equiv \gamma m = \sqrt{m^2 + \vec{p}^2} \,.$$

The Hamiltonian equations of motion $\dot{x}^i = \frac{\partial \mathcal{H}}{\partial p_i}, \dot{p}_i = -\frac{\partial \mathcal{H}}{\partial x^i}$ then provide the equation of motion $\dot{p}^i = 0$ whereas the Lagrangian 1-form takes the well-known expression

$$\mathcal{L} \, dt \equiv -p_\mu dx^\mu = (\vec{p} \cdot \dot{\vec{x}} - \mathcal{H}) \, dt = -m \sqrt{1 - \dot{\vec{x}}^2} \, dt \,.$$

## C.6 Lagrangians with local symmetries and constrained Hamiltonian systems

In a gauge field theory like electrodynamics, the basic dynamical variables are gauge potentials $(A^\mu)$ and the gauge invariance is at the origin of first class constraints (FCC's) for the phase space variables $(A^\mu, \pi_\mu)$ with $\pi_\mu \equiv \partial \mathcal{L}/\partial \dot{A}^\mu$: we have [23, 43] the constraints $0 = \gamma_1(A, \pi) \equiv \pi_0$ and $0 = \gamma_2(A, \pi) \equiv \partial^i \pi_i = \text{div} \vec{E}$ satisfying $\{\gamma_1, \gamma_2\} = 0$. Thus, we have a purely FCC system in the canonical formulation of the theory. With this example in mind, we will describe the geometric formulation of such a dynamical system in classical mechanics (based on chapter 2 of reference [23] while using the mathematical tools that we introduced above). We mostly consider local expressions which actually provide a global description up to mathematical technicalities on which we do not expand here. The latter have been addressed in the sixties and seventies by various authors, in particular J.-M. Souriau, G. Hinds, R. Hermann, J. Śniatycki, W. M. Tulczyjew, A. Lichnerowicz, M. J. Gotay,..., e.g. see references [336, 337] for some more details.

The starting point is a classical dynamical system described by a Lagrangian $L(\vec{q}, \dot{\vec{q}})$ which gives rise to a Hamiltonian system defined on a phase space manifold $(P, \omega)$ parametrized by $(Q^I)_{I=1,\ldots,N=2n} \equiv \vec{Q}$. The symplectic form $\omega$ is given by (C.13), i.e. we have a closed 2-form $\omega \equiv \frac{1}{2} \omega_{IJ} \, dQ^I \wedge dQ^J$ where the $N \times N$ matrix $(\omega_{IJ}(\vec{Q}))$ is invertible for all $\vec{Q}$. The inverse of this matrix, which is again denoted by $(-\Theta^{IJ}(\vec{Q}))$, yields the Poisson bracket $\{f, g\} = \Theta^{IJ} \partial_I f \, \partial_J g$ of real-valued functions on phase space $P$.

Now suppose the Lagrangian $L(\vec{q}, \dot{\vec{q}})$ describing the dynamics is singular, i.e. $\det\left(\frac{\partial^2 L}{\partial \dot{q}_i \partial \dot{q}_j}\right) = 0$, and that this results in the Hamiltonian formulation in a system of

$$\text{regular, independent FCC's:} \qquad \gamma_a(\vec{Q}) = 0 \,, \quad \text{for } a = 1, \ldots, M \,,$$

for the phase space coordinates $Q^I$. For such a FCC system, the functions $\gamma_a$ form a (generally field dependent) Lie algebra with the Poisson bracket as Lie bracket:

$$\{\gamma_a, \gamma_b\} = f_{ab}^c \gamma_c \,, \tag{C.32}$$

where the coefficients $f_{ab}^c$ may depend on the phase space coordinates $Q^I$. The collection of these constraints defines the

$$\text{constraint (hyper-) surface} \qquad \Sigma \equiv \{ \vec{Q} \in P \,|\, \gamma_a(\vec{Q}) = 0 \text{ for } a = 1, \ldots, M \}.$$

Since the constraints are regular and independent, this space represents a $(N-M)$-dimensional submanifold of phase space $P$ and is parametrized by local coordinates $(y^i)_{i=1,\ldots,N-M} \equiv \vec{y}$, see Figure 9 below. The hypersurface $\Sigma$ may be viewed as the physically accessible portion of phase space. In the mathematical literature, it is referred to as a *co-isotropic submanifold* [323] due to the specific properties of the symplectic form on $\Sigma$ that we will now discuss.

Let us denote the *inclusion (embedding) map* of $\Sigma$ into $P$ by

$$\begin{aligned} \mathcal{I} \;:\; \Sigma &\;\longrightarrow\; P \\ \vec{y} \equiv (y^i) &\longmapsto \vec{Q}(\vec{y}) = (Q^I(\vec{y})), \end{aligned} \tag{C.33}$$

where $\partial Q^I / \partial y^i \neq 0$ for a proper coordinate system $(y^i)$ on $\Sigma$. In the following, we use the short-hand notation $\partial_i \equiv \partial/\partial y^i$ and $\partial_I \equiv \partial/\partial Q^I$.

The *tangent* (or *differential*) *map* $T\mathcal{I}$ of the map $\mathcal{I} : \Sigma \to P$ is a linear map between the tangent bundles $T\Sigma$ of $\Sigma$ and $TP$ of $P$ (see Eqn. (A.15)):

$$\begin{aligned} T\mathcal{I} \;:\; T\Sigma &\;\longrightarrow\; TP \\ X \equiv X^i \partial_i &\longmapsto (T\mathcal{I})(X) = \left( X^i \partial_i Q^I \right) \partial_I. \end{aligned} \tag{C.34}$$

We can apply the pull-back map $\mathcal{I}^*$ to the space $\Omega^2(P)$ of 2-forms on $P$ so as to obtain a 2-form $\sigma$ on $\Sigma$ from the symplectic 2-form $\omega$ on $P$:

$$\begin{aligned} \mathcal{I}^* : \Omega^2(P) &\;\longrightarrow\; \Omega^2(\Sigma) \\ \omega &\;\longmapsto\; \boxed{\mathcal{I}^*\omega \equiv \sigma} = \text{induced (pre-) symplectic form on } \Sigma. \end{aligned} \tag{C.35}$$

More explicitly, we have

$$\begin{aligned} \sigma_{ij} = \sigma(\partial_i, \partial_j) &\equiv (\mathcal{I}^*\omega)(\partial_i, \partial_j) \equiv \omega\big((T\mathcal{I})(\partial_i), (T\mathcal{I})(\partial_j)\big) \\ &= \omega\big((\partial_i Q^I)\partial_I, (\partial_j Q^J)\partial_J\big) = (\partial_i Q^I)(\partial_j Q^J) \underbrace{\omega(\partial_I, \partial_J)}_{= \,\omega_{IJ}}, \end{aligned}$$

hence

$$\boxed{\sigma \equiv \frac{1}{2}\,\sigma_{ij}\,dy^i \wedge dy^j,} \qquad \text{with} \qquad \boxed{\sigma_{ij}(\vec{y}) = (\partial_i Q^I)(\partial_j Q^J)\,\omega_{IJ}(\vec{Q}(\vec{y})).} \tag{C.36}$$

Since the exterior derivative $d$ commutes with the pullback map, we have $d\sigma = d(\mathcal{I}^*\omega) = \mathcal{I}^*(d\omega) = 0$ due to the closedness of $\omega$. Accordingly, $\sigma$ represents a presymplectic form on $\Sigma$.

The rank of $\sigma$, i.e. the rank of the matrix $(\sigma_{ij})$ which we assume to be *constant* on $\Sigma$, is at most $N - M$. The fact that all constraints are first class actually implies that the rank of $\sigma$ is $N - 2M$ and thereby even-dimensional. In this respect, we consider, for each $a \in \{1, \ldots, M\}$, the *Hamiltonian vector field* $X_a$ *on* $P$ which is associated to the function $\gamma_a$ (with respect to the symplectic form $\omega$, see Eqn. (C.18)):

$$X_{\gamma_a} \equiv X_a \equiv X_a^J \partial_J, \qquad \text{with} \qquad \boxed{X_a^J \equiv \Theta^{JI}(\partial_I \gamma_a).} \tag{C.37}$$

The FCC's $\gamma_a$ and thereby the vector fields $X_a$ generate *Hamiltonian gauge symmetries:* the Lie derivative of a function $f \in C^\infty(P)$ along the vector field $X_a$ reads

$$L_{X_a} f = X_a^J \partial_J f = \Theta^{JI}(\partial_I \gamma_a)(\partial_J f) = \{f, \gamma_a\}, \tag{C.38}$$

and this expression represents an infinitesimal gauge transformation of $f$ generated by $\gamma_a$ (where we consider the usual postulate that all FCC's generate Hamiltonian gauge symmetries [23]). *The vector fields $X_a$ are tangent to the gauge orbits on the constraint surface $\Sigma$* since

$$X_a \cdot \nabla \gamma_b = X_a^J \partial_J \gamma_b = \Theta^{JI}(\partial_I \gamma_a)(\partial_J \gamma_b) = \{\gamma_b, \gamma_a\} = 0, \qquad \text{on } \Sigma.$$

Moreover, the vector fields $X_a$ with $a \in \{1, \ldots, M\}$ are *linearly independent* at each point of $\Sigma$ due to the fact that the FCC's are regular and independent.

For any vector field $Y$ which is tangent to the hypersurface $\Sigma$ and for the vector fields (C.37), we have the skew-product

$$\omega(X_a, Y) = \omega_{IJ} X_a^I Y^J = \omega_{IJ} \Theta^{IK}(\partial_K \gamma_a) Y^J = (\partial_J \gamma_a) Y^J = (\nabla \gamma_a) \cdot Y = 0. \tag{C.39}$$

Thus, the hypersurface $\Sigma$ contains the directions $X_a$ which are $\omega$-orthogonal to it, i.e.

$$(T\Sigma)^\perp \subset T\Sigma,$$

where $T\Sigma$ denotes the tangent bundle of $\Sigma$ and $(T\Sigma)^\perp$ its symplectic complement: This is the defining property for a *co-isotropic submanifold* $\Sigma$ of a symplectic manifold $(P, \omega)$.

Next, we consider the *kernel of the 2-form $\sigma$* which is defined by

$$\boxed{\ker \sigma \equiv \{\text{vector fields } X \text{ on } \Sigma \,|\, i_X \sigma = 0\},} \tag{C.40}$$

where $i_X \sigma$ denotes the interior product of the 2-form $\sigma$ with respect to the vector field $X$, see Eqn. (A.12). With $X \equiv X^i \partial_i$, we have

$$i_X \sigma = i_X \left( \frac{1}{2} \sigma_{ij} \, dy^i \wedge dy^j \right) = \sigma_{ij} \underbrace{(i_X dy^i)}_{= X^i} dy^j = -dy^j (\sigma_{ji} X^i),$$

i.e.

$$\ker \sigma = \{(X^i) \,|\, \sigma_{ji} X^i = 0 \text{ for all } j\} = \{\text{null eigenvectors } (X^i) \text{ of } (\sigma_{ji})\}, \tag{C.41}$$

where null eigenvector means eigenvector associated to the eigenvalue zero. By using equations (C.36), (C.34) and (C.39), we obtain

$$\sigma_{ji} X_a^i = (\partial_j Q^J) \omega_{JI} (\partial_i Q^I) X_a^i = (\partial_j Q^J) \omega_{JI} X_a^I = \omega(Y_j, X_a) = 0, \qquad \text{with } Y_j \equiv (\partial_j Q^J)_{J=1,\ldots,N}$$

where $Y_j$ represents a tangent vector to $\Sigma$. This means that the tangent vectors $X_a$ to the gauge orbits on $\Sigma$ are null eigenvectors of the induced presymplectic form $\sigma$ on $\Sigma$ and thereby represent its kernel.

By virtue of (C.23), the Lie bracket of the vector fields $X_a, X_b$ reads $[X_a, X_b] = -X_{\{\gamma_a, \gamma_b\}}$. Substitution of (C.32) into this relation yields

$$[X_a, X_b]^J = -X_{f_{ab}^c \gamma_c}^J = \Theta^{IJ} \partial_I (f_{ab}^c \, \gamma_c) = \Theta^{IJ} (\partial_I f_{ab}^c) \gamma_c + f_{ab}^c \underbrace{\Theta^{IJ} (\partial_I \gamma_c)}_{= -X_c^J},$$

hence

$$[X_a, X_b] = -f_{ab}^c X_c, \qquad \text{on } \Sigma.$$

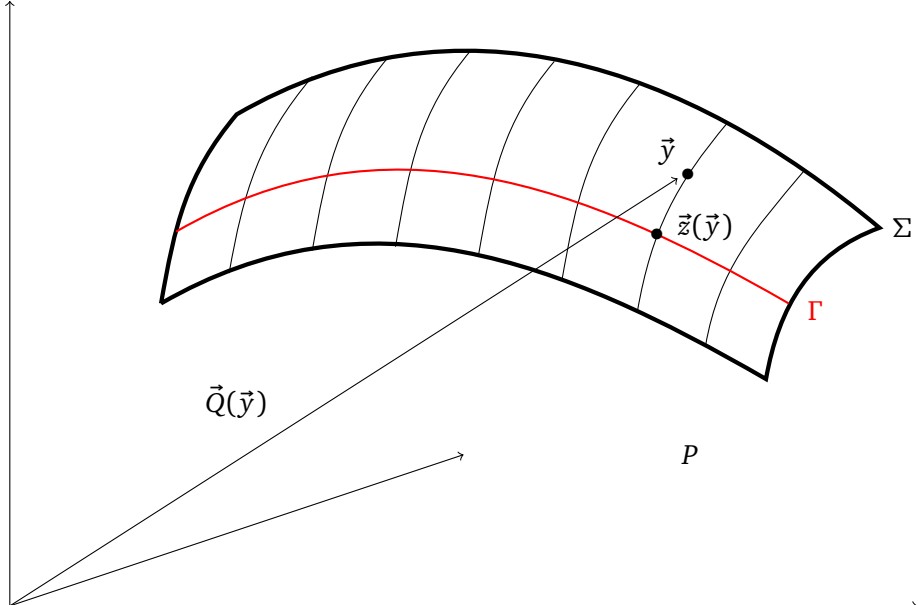

Figure 9: Constraint surface $\Sigma$ in phase space $P$, gauge orbits and reduced phase space $\Gamma$.

This relation represents the *Frobenius integrability condition* [338] which ensures that the collection of vector fields $X_a$ generates $M$-dimensional submanifolds of the $(N-M)$-dimensional hypersurface $\Sigma$.

**In summary,** the $(N-M)$-dimensional presymplectic manifold $(\Sigma, \sigma)$ is defined by the $M$ regular and independent FCC's $\gamma_a = 0$ and is endowed with the presymplectic form $\sigma$ of rank $N-2M$ that is induced from the symplectic form $\omega$ on $N$-dimensional phase space $(P, \omega)$. In the hypersurface $\Sigma$, the constraints $\gamma_a$ generate $M$-dimensional submanifolds which represent the gauge orbits (for Hamiltonian gauge symmetries) and which coincide with the space of null eigenvectors of $\sigma$. This family of submanifolds defines a *foliation* $\mathcal{F}_\sigma$ of the presymplectic manifold $(\Sigma, \sigma)$ whose *leaves* are given by the gauge orbits.

We denote the group of Hamiltonian gauge symmetries by $\mathcal{G}$ and we introduce the so-called

$$\text{reduced phase space:} \qquad \boxed{\Gamma \equiv \Sigma / \mathcal{G}\,,} \tag{C.42}$$

or, equivalently, $\Gamma = \Sigma / \ker \sigma$. By construction, this space represents the *set of leaves* of the foliation $\mathcal{F}_\sigma$ of $\Sigma$ and it has dimension $N - 2M = 2(n - M)$. It may be parametrized by some local coordinates $(z^\alpha)_{\alpha=1,\dots,N-2M} \equiv \vec{z}$ such that $\{z^\alpha(\vec{y}) = \text{const.}\}$ represents the gauge orbits in $\Sigma$, see Figure 9. By introducing $M$ canonical gauge fixing conditions $C_a(\vec{y}) = 0$ which are transversal to the gauge orbits in $\Sigma$ (so that they select one point on each orbit), the reduced phase space can also be represented as

$$\boxed{\Gamma = \{\vec{y} \in \Sigma \mid C_a(\vec{y}) = 0 \text{ for } a = 1, \dots, M\}\,.} \tag{C.43}$$

Though the reduced phase space is globally well defined, the latter characterization of $\Gamma$ may only hold locally due to the Gribov problem (which appears in particular for non-Abelian gauge field theories and a specific asymptotic behavior of gauge fields) [21, 339, 340].

If we denote the projection map from $\Sigma$ onto $\Gamma$ by

$$\Pi : \Sigma \longrightarrow \Gamma \equiv \Sigma/\mathcal{G}$$
$$\vec{y} \equiv (y^i) \longmapsto \vec{z}(\vec{y}) = (z^\alpha(\vec{y})), \tag{C.44}$$

then we have the pullback map

$$\Pi^* : \Omega^2(\Gamma) \longrightarrow \Omega^2(\Sigma)$$
$$\sigma_{\text{phys}} \longmapsto \boxed{\Pi^*\sigma_{\text{phys}} = \sigma\,.} \tag{C.45}$$

Here,

$$\sigma_{\text{phys}} \equiv \frac{1}{2}(\sigma_{\text{phys}})_{\alpha\beta}\,dz^\alpha \wedge dz^\beta\,, \qquad \text{with} \quad \sigma_{ij}(\vec{y}) = (\partial_i z^\alpha)(\partial_j z^\beta)(\sigma_{\text{phys}})_{\alpha\beta}(\vec{z}(\vec{y})), \tag{C.46}$$

represents the uniquely defined 2-form on $\Gamma$ which is determined by the 2-form $\sigma$ on $\Sigma$. This form is closed due to the closedness of $\sigma$ and it is non-degenerate since the kernel of $\sigma$ has been discarded upon passage to $\Gamma = \Sigma/\ker\sigma$. Henceforth, we have a *symplectic 2-form $\sigma_{\text{phys}}$ on the reduced phase space $\Gamma$ which may be viewed as the physical subspace for the constrained dynamical system under consideration.* This symplectic form gives rise to a Poisson bracket $\{z^\alpha, z^\beta\} = (\theta_{\text{phys}})^{\alpha\beta}$ with $(\theta_{\text{phys}})^{\alpha\delta}(\sigma_{\text{phys}})_{\beta\delta} = \delta^\alpha_{\ \beta}$.

In this respect, we recall that for a Hamiltonian system with FCC's $\gamma_a = 0$ and canonical gauge fixing conditions $C_a = 0$, the Dirac bracket is defined as follows [23, 43]. One collects the functions $\gamma_a$ and $C_a$ (the latter being chosen in such a way that $\det \mathcal{A} \neq 0$ for the $(M \times M)$-matrix $\mathcal{A} \equiv (\{C_a, \gamma_b\})$) into a $(2M)$-tuple

$$(\varphi_A)_{A=1,\ldots,2M} \equiv (\gamma_1,\ldots,\gamma_M, C_1,\ldots,C_M),$$

and one introduces the invertible matrix $X \equiv (X_{AB})$ with $X_{AB} \equiv \{\varphi_A, \varphi_B\}$. Since $\{\gamma_a, \gamma_b\} \approx 0$ (where the symbol $\approx$ denotes an equality that holds on the constraint hypersurface $\Sigma$), the matrices $X$ and $X^{-1}$ have the following structure:

$$X \approx \left[\begin{array}{c|c} 0 & -\mathcal{A}^t \\ \hline \mathcal{A} & \mathcal{B} \end{array}\right] \implies X^{-1} \approx \left[\begin{array}{c|c} \mathcal{A}^{-1}\mathcal{B}(\mathcal{A}^{-1})^t & \mathcal{A}^{-1} \\ \hline -(\mathcal{A}^{-1})^t & 0 \end{array}\right]. \tag{C.47}$$

For any two real-valued functions $F, G$ on phase space $P$, one now considers the *Dirac bracket associated to the choice $(C_a)_{a=1,\ldots,M}$ of gauge fixing functions:*

$$\text{Dirac bracket:} \qquad \boxed{\{F, G\}_D \equiv \{F, G\} - \{F, \varphi_A\}(X^{-1})^{AB}\{\varphi_B, G\}\,.} \tag{C.48}$$

Let us presently consider *observables,* i.e. gauge invariant functions: $\{F, \gamma_a\} = 0 = \{G, \gamma_a\}$ on $\Sigma$ for all $a$. These functions are constant on the gauge orbits of $\Sigma$ and, by virtue of (C.47) and (C.48), we have $\{F, G\}_D = \{F, G\}$ on $\Sigma$. Thus, *the Dirac bracket of observables coincides with their Poisson bracket as given by $(\theta_{\text{phys}})^{\alpha\beta}$ on reduced phase space.*

# D  Poisson brackets and symplectic forms in classical field theory

## D.1  Passage from classical mechanics to field theory

We are interested in classical field theory on Minkowski space-time $M \equiv \mathbb{R}^n$ or more generally on an $n$-dimensional pseudo-Riemannian manifold $\left(M, (g_{\mu\nu})\right)$. The passage from the finite

number of degrees of freedom of classical mechanics to the infinite number of field theory proceeds by the replacement

$$q_i(t) \equiv q(t,i) \rightsquigarrow q(t,\vec{x},a) \equiv \varphi(t,\vec{x},a) \equiv \varphi^a(x),$$

i.e. the discrete index $i \in \{1,\dots,n\}$ becomes a continuous index $\vec{x} \in \mathbb{R}^d \equiv \mathbb{R}^{n-1}$ possibly supplemented by a discrete index $a \in \{1,\dots,m\}$. Accordingly the summation over the index $i$ becomes an integration over $\mathbb{R}^d$ together with a summation over the index $a$. Furthermore, derivatives with respect to $q_i$ or $\dot{q}_i$ become functional derivatives, e.g. the momentum which is canonically conjugate to the field $\varphi^a$ is defined by

$$\pi_a(x) \equiv \frac{\delta L}{\delta \dot{\varphi}^a(x)} = \frac{\partial \mathcal{L}}{\partial \dot{\varphi}^a}(x), \qquad \text{with } L[\varphi, \dot{\varphi}] \equiv \int_{\mathbb{R}^d} d^d x \, \mathcal{L}(\varphi^a, \partial_k \varphi^a, \dot{\varphi}^a),$$

where $L$ is the Lagrangian function occurring in the action functional $S[\varphi] = \int_{\mathbb{R}} dt \, L$.

In the present case, the phase space $P$ is parametrized by the collection of functions

$$\vec{\Phi} \equiv \left(\Phi^I\right)_{I=1,\dots,\mathcal{M}=2m} \equiv (\varphi^a, \pi_a)_{a=1,\dots,m}, \tag{D.1}$$

and therefore represents an infinite-dimensional vector space. As pointed out in equations (8)-(9), this phase space can equivalently be parametrized in a manifestly covariant manner and in that approach we have denoted the space by $Z$ in Section 5. We note that by contrast to the finite-dimensional case, the distinction between even and odd dimensions does not make sense anymore for an infinite-dimensional space. Yet, one can still distinguish between Poisson and symplectic structures as in the finite-dimensional setting as we will further discuss below.

From the mathematical point of view, the collection of fields $(\varphi^a)$ represents a section in a fiber bundle over the space-time manifold $M$. Since we focus on Minkowski space-time $M \equiv \mathbb{R}^n$, we can (and will) avoid this terminology and refer to the literature for the related notions, e.g. see references [41, 42, 341]. The underlying ideas are conveyed by the table that we have given in Section 3.1 to describe the multisymplectic approach to field theory.

## D.2 Vector fields and forms on phase space

In Subsection D.10 below, we will shortly elaborate on smooth infinite-dimensional manifolds $P$, i.e. manifolds modeled on a real infinite-dimensional vector space $E$. Here, we consider the simplest instance, the so-called *linear case* where $P = E$. More explicitly, we suppose that the phase space $P$ is a real vector space $E$ of smooth functions $(\Phi^I)_{I=1,\dots,\mathcal{M}}$ defined on $\mathbb{R}^{n-1} \equiv \mathbb{R}^d$ (which functions depend in general also smoothly on the time parameter $t$): for these functions, we thus use the notation $\vec{x} \mapsto \vec{\Phi}(\vec{x}) = (\Phi^I(\vec{x}))_{I=1,\dots,\mathcal{M}}$ with $\vec{x} \in \mathbb{R}^d$. The tangent space $T_{\vec{\Phi}}P$ to $P$ at $\vec{\Phi}$ is spanned by the derivations $\frac{\delta}{\delta \Phi^I(\vec{x})}$ which act on smooth functionals $\vec{\Phi} \mapsto F[\vec{\Phi}]$, and a *vector field* $X$ on $P$ admits the expansion $X = \int_{\mathbb{R}^d} d^d x \, \vec{X} \frac{\delta}{\delta \vec{\Phi}}$. At the point $\vec{\Phi} \in P$, we thereby have the derivation

$$X_{[\vec{\Phi}]} = \int_{\mathbb{R}^d} d^d x \, \vec{X}(\vec{\Phi}(\vec{x})) \frac{\delta}{\delta \vec{\Phi}(\vec{x})}, \tag{D.2}$$

where $\vec{X} \equiv (X^1,\dots,X^{\mathcal{M}})$ and where $\vec{x} \mapsto X^I(\vec{\Phi}(\vec{x}))$ is a smooth function of $\vec{x}$ for each $I \in \{1,\dots,\mathcal{M}\}$. (For a mathematically rigorous description, we refer for instance to the recent work [248].)

The basis of $T_{\vec{\Phi}}^* P \equiv (T_{\vec{\Phi}}P)^*$ which is dual to the basis of derivations $\left\{\frac{\delta}{\delta \Phi^I(\vec{x})}\right\}_{I=1,\dots,\mathcal{M}}$ is given by the infinitesimal variations $\{\delta \Phi^I(\vec{x})\}_{I=1,\dots,\mathcal{M}}$. The exterior derivative $d \equiv dx^\mu \frac{\partial}{\partial x^\mu}$

acting on differential forms on $\mathbb{R}^n$ can be generalized to the infinite-dimensional vector space $P$: we introduce a *differential* $\delta$ (satisfying $\delta^2 = 0$) which acts on forms on $P$ by

$$\delta \equiv \int_{\mathbb{R}^d} d^d x \, \delta \vec{\Phi}(\vec{x}) \frac{\delta}{\delta \vec{\Phi}(\vec{x})} \, . \tag{D.3}$$

This definition amounts to the one of the infinitesimal variation of a smooth functional $\vec{\Phi} \mapsto F[\vec{\Phi}]$ (i.e. a 0-form on $P$) induced by the infinitesimal variations $\delta \Phi^I(\vec{x})$.

Accordingly, a 1-*form* on $P$ has the following expression:

$$\text{1-form on } P: \qquad a_{[\vec{\Phi}]} = \int_{\mathbb{R}^d} d^d x \, \vec{a}(\vec{\Phi}(\vec{x})) \, \delta \vec{\Phi}(\vec{x}),$$

where $\vec{a} \equiv (a_1, \ldots, a_{\mathcal{M}})$ and where $\vec{x} \mapsto a_I(\vec{\Phi}(\vec{x}))$ is a smooth function of $\vec{x}$ for each $I \in \{1, \ldots, \mathcal{M}\}$. As in the finite-dimensional case, we obtain

$$\delta a_{[\vec{\Phi}]} = \delta \int_{\mathbb{R}^d} d^d y \, a_J(\vec{\Phi}(\vec{y})) \, \delta \Phi^J(\vec{y}) = \int_{\mathbb{R}^d} d^d y \int_{\mathbb{R}^d} d^d x \, \frac{\delta a_J(\vec{\Phi}(\vec{y}))}{\delta \Phi^I(\vec{x})} \, \delta \Phi^I(\vec{x}) \wedge \delta \Phi^J(\vec{y})$$

$$= \frac{1}{2} \int_{\mathbb{R}^d} d^d x \int_{\mathbb{R}^d} d^d y \left( \frac{\delta a_J(\vec{\Phi}(\vec{y}))}{\delta \Phi^I(\vec{x})} - \frac{\delta a_I(\vec{\Phi}(\vec{x}))}{\delta \Phi^J(\vec{y})} \right) \delta \Phi^I(\vec{x}) \wedge \delta \Phi^J(\vec{y}). \tag{D.4}$$

Here and in the following, we assume that the monomials $\delta \Phi^I$ represent *anticommuting* variables with respect to the exterior product $\wedge$ of forms on $P$.

The 2-form (D.4) is exact since it is the differential of a 1-form. A generic 2-*form* $\Omega$ on phase space $P$ writes as follows:

$$\text{2-form on } P: \qquad \boxed{\Omega \equiv \frac{1}{2} \int_{\mathbb{R}^d} d^d x \int_{\mathbb{R}^d} d^d y \, \Omega_{IJ}^{\vec{x}, \vec{y}}(\vec{\Phi}) \, \delta \Phi^I(\vec{x}) \wedge \delta \Phi^J(\vec{y}) \, .} \tag{D.5}$$

We note that the pairing between a 1-form $a$ and a vector field $X$ on $P$ is given by

$$\langle \vec{a}, \vec{X} \rangle \equiv \int_{\mathbb{R}^d} d^d x \, a_I(\vec{\Phi}(\vec{x})) X^I(\vec{\Phi}(\vec{x})). \tag{D.6}$$

(Here, the integral may be viewed as a formal expression for the application of the regular distribution defined by the functions $a_I$ on the test functions $X^I$.)

## D.3  Poisson brackets

We start from a Lagrangian field theory with fields $\varphi^a$ (with $a \in \{1, \ldots, m\}$) and associated canonical momenta $\pi_a$ defined on Minkowski space-time $\mathbb{R}^n \equiv \mathbb{R}^{1+d}$. For any two smooth functionals $F, G$ depending on the fields $(\varphi^a, \pi_a)$, we have the *canonical Poisson bracket,* defined as follows at fixed time $t$:

Canonical expression of Poisson bracket:  $\displaystyle \{F, G\} = \sum_{a=1}^{m} \int_{\mathbb{R}^d} d^d x \left( \frac{\delta F}{\delta \varphi^a} \frac{\delta G}{\delta \pi_a} - \frac{\delta F}{\delta \pi_a} \frac{\delta G}{\delta \varphi^a} \right).$
$$\tag{D.7}$$

This is the field theoretical generalization of expression (C.1) which holds for canonical coordinates $(q^i, p_i)$ in classical mechanics. The expression (C.3) of the Poisson bracket with respect to general phase space coordinates $\vec{\Phi} \equiv (\Phi^I)_{I=1,\ldots,\mathcal{M}=2m}$ presently writes as follows:

Poisson bracket:  $\boxed{\displaystyle \{F, G\} \equiv \sum_{I,J=1}^{\mathcal{M}} \int_{\mathbb{R}^d} d^d x \int_{\mathbb{R}^d} d^d y \, \frac{\delta F}{\delta \Phi^I(\vec{x}, t)} \Theta_{\vec{x}, \vec{y}}^{IJ}(\vec{\Phi}) \frac{\delta G}{\delta \Phi^J(\vec{y}, t)},}$ (D.8)

hence

$$\{\Phi^I(\vec{x}, t), \Phi^J(\vec{y}, t)\} = \Theta^{IJ}_{\vec{x}, \vec{y}}(\vec{\Phi}).$$
(D.9)

As in the finite-dimensional case, the Poisson bracket is *antisymmetric* (i.e. $\Theta^{IJ}_{\vec{x}, \vec{y}}$ must be anti-symmetric with respect to the pair of indices $(\vec{x}, I)$ and $(\vec{y}, J)$) and satisfies the *Jacobi identity* as well as the *Leibniz rule* (derivation property). Moreover, for local field theory we assume that the Poisson brackets are *local* [342] in the sense that the generalized function $\Theta^{IJ}_{\vec{x}, \vec{y}}(\vec{\Phi})$ is a linear combination of the delta function $\delta(\vec{x} - \vec{y})$ and its derivatives up to finite order, with coefficients which depend on the values of the fields $\Phi^I$ and their derivatives at the points $\vec{x}, \vec{y}$.

The expression (D.7) is recovered from (D.8) by considering (D.1) and

$$(\Theta^{IJ}_{\vec{x}, \vec{y}}) = \begin{bmatrix} 0_m & \delta(\vec{x} - \vec{y})\mathbb{1}_m \\ -\delta(\vec{x} - \vec{y})\mathbb{1}_m & 0_m \end{bmatrix}.$$
(D.10)

The *non-degeneracy condition* is discussed in the next subsection.

## D.4 Poisson structure and Hamiltonian systems

The definition of a *Poisson structure* (or *Poisson bracket*) which we have given in the finite-dimensional case (see Eqn. (C.10)) generalizes verbatim to the case of an infinite-dimensional manifold $P$, i.e. to the case where the local coordinates on $P$ belong to some real infinite-dimensional vector space $E$. We will briefly discuss the latter vector spaces and manifolds in Subsection D.10. Here, we only note that the mathematical choice for the infinite dimensional space of fields and the subtle notion of locality have quite recently been addressed in a foundational work (motivated in particular by [11]) in relationship with classical and quantum field theory [343].

Whatever the choice of the model vector space $E$, the simplest instance for a manifold $P$ is the one where $P = E$, i.e. the manifold $P$ is a real infinite-dimensional vector space: this case is discussed in some detail in reference [151] for different types of boundary conditions satisfied by the fields at spatial infinity. For rapidly decreasing fields, the admissible functionals on $P$ are assumed to be *real-analytic*. In this respect, it should be stressed that the latter functionals are not local functionals in general and that the restriction to the subspace of local functionals is not closed for the point-wise product since the product of two such functionals is not local: hence the consideration of the Leibniz rule does not make sense if one considers such a restriction [31].

As in the finite-dimensional case (see Eqn. (C.12)), the Poisson structure on $P$ is said to be *non-degenerate* if

$$\{F, G\} = 0 \text{ for all admissible functionals } G \text{ on } P \text{ implies that } F \text{ is constant.}$$
(D.11)

By way of example, on a real infinite-dimensional vector space $P$, the Poisson bracket (D.7) is non-degenerate since $\{F, G\} = 0$ for all $G$ implies that $\delta F/\delta \varphi^a = 0 = \delta F/\delta \pi_a$ for all $a$, hence $F$ is constant. We note that the infinite-dimensional "matrix" (D.10) corresponding to this Poisson bracket is invertible.

The definition of *Hamiltonian systems* also carries over verbatim from a finite-dimensional Poisson manifold $M$ to an infinite-dimensional space $P$: the choice of a Hamiltonian function (i.e. of a functional $H$ on $P$) again determines the time evolution of all admissible functionals $F$ according to $\dot{F} = \{F, H\}$. For instance, for a real scalar field $\varphi$ on $\mathbb{R}^n$ described by the Lagrangian density (95), the canonical Poisson bracket (D.7) yields the same equation of motion for $\varphi$ as the Lagrangian formulation.

A more subtle example for an infinite-dimensional Hamiltonian system (with $d = 1$, $\vec{x} \equiv x$ and $\vec{\Phi} \equiv u$) is given by the Korteweg-de Vries (KdV) equation $\partial_t u = \partial_x^3 u + 6u\partial_x u$: in this case, the so-called first Poisson structure [342, 344] (also known as *Gardner-Zakharov-Faddeev bracket* [345, 346]) is given by the Poisson tensor $\Theta_{x,y}^{11} = \partial_x \delta(x - y)$ (and the Hamiltonian $H[u] \equiv \int_{\mathbb{R}} dx \left[ -\frac{1}{2}(\partial_x u)^2 + u^3 \right]$), hence the Poisson bracket reads

$$\{F, G\} \equiv \int_{\mathbb{R}} dx \, \frac{\delta F}{\delta u} \, \partial_x \, \frac{\delta G}{\delta u} \, . \tag{D.12}$$

Indeed, substitution of the given Hamiltonian $H[u]$ into the Hamiltonian equation of motion $\dot{u} = \{u, H\} = \partial_x \frac{\delta H}{\delta u}$ yields the KdV equation for the field $u$. In the bracket (D.12), the functionals $F$ and $G$ are assumed to have the property that $x \mapsto \delta F/\delta u(x)$ and $x \mapsto \delta G/\delta u(x)$ are smooth functions which decay sufficiently fast for $|x| \to \infty$. Since this excludes the constant functions which represent the kernel of the operator $\partial_x$, the so-defined Poisson bracket (D.12) is non-degenerate. The linear operator $\mathcal{J} \equiv \partial_x$ (*"Poisson operator"*) in (D.12) acting on such functions is skew-self-adjoint with respect to the $L^2$ inner product $\langle u_1, u_2 \rangle_{L^2} \equiv \int_{\mathbb{R}} dx \, u_1(x) u_2(x)$: this fact represents the infinite-dimensional counterpart of the antisymmetry of the Poisson matrix $(\Theta^{IJ})$ in Eqn. (C.3). We note that substitution of $F[u] = u(x)$ and $G[u] = u(y)$ into (D.12) formally yields [151]

$$\{u(x), u(y)\} = \frac{1}{2}(\partial_y - \partial_x)\delta(x - y) \, .$$

As a third example, we adapt the Poisson bracket (D.12) to the periodic case [347]: the consideration of $(2\pi)$-periodic functions $x \mapsto u(x)$ amounts to considering smooth functions $x \mapsto u(x)$ on the unit circle $S^1$, hence the integration in integrals like (D.12) is also limited to $S^1$:

$$\{F, G\} \equiv \int_{S^1} dx \, \frac{\delta F}{\delta u} \, \partial_x \, \frac{\delta G}{\delta u} \, . \tag{D.13}$$

The functionals $F$ and $G$ in (D.13) are only assumed to be smooth, hence $x \mapsto \delta F/\delta u(x)$ and $x \mapsto \delta G/\delta u(x)$ are smooth functions on $S^1$. The skew-self-adjointness of the Poisson operator $\mathcal{J} \equiv \partial_x$ and the antisymmetry of the bracket (D.13) are now ensured by the periodicity (continuity of functions on $S^1$). The smooth functions on $S^1$ include the constant ones, hence the considered Poisson structure is degenerate. As in the finite-dimensional case (see discussion after Eqn. (C.7)), a non-degenerate Poisson structure can be obtained by restricting oneself to functionals defined on the subspace of phase space $P$ given by

$$S_c \equiv \left\{ u \in P \equiv C^\infty(S^1) \, | \, I[u] \equiv \int_{S^1} dx \, u(x) = c \equiv \text{given real constant} \right\}, \tag{D.14}$$

e.g. $c = 0$. On this level surface $S_c$, the Poisson bracket is no longer degenerate. As a matter of fact, one also resorts to this argumentation in the literature for the Poisson bracket (D.12) if one follows an algebraic approach (e.g. see references [31, 151, 344])), i.e. ignores the analytic aspects (nature of functional spaces, domains of definition,...) that we have taken into account above. Indeed, for the functional $I[u] \equiv \int_{\mathbb{R}} dx \, u(x)$ (which is also referred to as the average of $u$), we have $\frac{\delta I}{\delta u} = 1$, hence the Poisson bracket (D.12) of the functional $I$ with any functional $G$ vanishes. Upon restricting oneself to functionals defined on the subspace $S_c$ of functions $x \mapsto u(x)$ belonging to the level surface where $I[u] \equiv \int_{\mathbb{R}} dx \, u(x) = $ given real constant, one then obtains a non-degenerate Poisson structure.

## D.5 Symplectic structure

Following the line of arguments of classical mechanics, we introduce the inverse $(\Omega^{\vec{x},\vec{y}}_{IJ}(\vec{\Phi}))$ of the "matrix" $(-\Theta^{IJ}_{\vec{x},\vec{y}}(\vec{\Phi}))$, i.e.

$$\int_{\mathbb{R}^d} d^d y\, \Theta^{IJ}_{\vec{x},\vec{y}}\, \Omega^{\vec{y},\vec{z}}_{JK} = -\delta^I_K\, \delta(\vec{x}-\vec{z})\,. \tag{D.15}$$

This inverse yields the

symplectic 2-form: $\qquad \boxed{\Omega \equiv \dfrac{1}{2}\int_{\mathbb{R}^d} d^d x \int_{\mathbb{R}^d} d^d y\, \Omega^{\vec{x},\vec{y}}_{IJ}(\vec{\Phi})\, \delta\Phi^I(\vec{x})\wedge\delta\Phi^J(\vec{y})\,.} \tag{D.16}$

Like $(\Theta^{IJ}_{\vec{x},\vec{y}})$ the "matrix" $(\Omega^{\vec{x},\vec{y}}_{IJ})$ is *antisymmetric* in its indices. Moreover, the *closedness* of $\Omega$ (i.e. the relation $\delta\Omega = 0$) reflects the Poisson-Jacobi identity for the Poisson bracket (D.8). Before discussing the non-degeneracy of $\Omega$, we consider two examples of symplectic forms.

First, we consider the Poisson bracket (D.7) corresponding to the Poisson "matrix" (D.10). Substitution of its inverse (which has the same form) and of (D.1) into (D.16) yields the

canonical expression of the symplectic 2-form: $\qquad \boxed{\Omega = \int_{\mathbb{R}^d} d^d x\, \delta\varphi^a \wedge \delta\pi_a\,.} \tag{D.17}$

Obviously, this result is the infinite-dimensional generalization of expression (C.14) appearing in classical mechanics. We will see in Eqn. (D.26) below that the symplectic structure (D.17) is weakly non-degenerate.

As a second example [118, 345, 346], we consider the Poisson bracket (D.12) that we introduced for the KdV equation and for which we only considered functionals $F$ having the property that $x \mapsto \frac{\delta F}{\delta u(x)}$ is a smooth function which tends sufficiently fast to zero for $|x| \to \infty$. The space of these functions does not include the constant functions, hence the Poisson operator $\mathcal{J} \equiv \partial_x$ acting on this space has a trivial kernel and is thus invertible, its inverse being given by

$$(\mathcal{J}^{-1}f)(x) = \frac{1}{2}\left(\int_{-\infty}^x d\sigma\, f(\sigma) - \int_x^\infty d\sigma\, f(\sigma)\right)\,. \tag{D.18}$$

Thereby, *the symplectic form associated to the Poisson bracket* (D.12) reads

$$\Omega = \frac{1}{2}\int_{\mathbb{R}} dx \int_{-\infty}^x dy\, \delta u(x)\wedge\delta u(y)\,. \tag{D.19}$$

This symplectic structure is also weakly non-degenerate, see discussion after Eqn. (D.30) below. A systematic derivation of this symplectic form from multisymplectic geometric is presented in reference [100].

## D.6 Non-degeneracy conditions in infinite dimensions

Let us again consider a real infinite-dimensional vector space $P = E$. By definition, a *weak symplectic structure* on $E$ is given by a skew-symmetric bilinear form $\Omega : E \times E \to \mathbb{R}$ which is weakly non-degenerate in the following sense:

weak non-degeneracy: $\quad \Omega(u,v) = 0$ for all $v \in E$ implies that $u = 0$. $\tag{D.20}$

In the case of a Banach space $E$ (i.e. a complete normed vector space), we also assume that the form $\Omega : E \times E \to \mathbb{R}$ is continuous.

We note that the definition (D.20) also makes sense in the case of a finite-dimensional vector space $E$: upon choosing a basis $\{e_I\}_{I=1,\dots,N}$ of $E$, we then have $u = u^I e_I$ and $v = v^J e_J$ (with real numbers $u^I$ and $v^J$), henceforth $\Omega(u,v) = u^I \Omega_{IJ} v^J$ with real constants $\Omega_{IJ} \equiv \Omega(e_I, e_J)$: the weak non-degeneracy condition for $\Omega$ is now equivalent to the non-degeneracy of the matrix $(\Omega_{IJ})$, i.e. the definition which has been considered in the finite-dimensional case. Accordingly, a weak symplectic structure on a finite-dimensional vector space $E$ is a symplectic structure in the sense defined before in the context of classical mechanics, see Eqn. (C.13).

To get a better understanding of condition (D.20) in the general case, it is judicious to reformulate it first in different terms. Let us for the moment being consider the notation

$$\Omega(u,v) \equiv \langle u|v\rangle, \qquad \text{for } u,v \in E.,$$

The bilinearity of $\langle \cdot|\cdot\rangle$ then implies that the "bra" (in Dirac's terminology) $\langle u| \equiv \langle u|\cdot\rangle$ represents an element of the dual $E^*$ of $E$:

$$\begin{aligned} \langle u| \,:\, E &\longrightarrow \mathbb{R} \\ v &\longmapsto \langle u|v\rangle. \end{aligned} \tag{D.21}$$

Accordingly we can consider the so-called *musical homomorphism* $^\flat \equiv{}^\flat(\Omega)$ *associated to the bilinear form $\Omega$ on $E$,*

$$\begin{aligned} ^\flat \,:\, E &\longrightarrow E^* \\ u &\longmapsto {}^\flat u \equiv \langle u|, \end{aligned} \tag{D.22}$$

whose linearity again follows from the bilinearity of $\langle \cdot|\cdot\rangle$. In terms of this notation, the weak non-degeneracy condition (D.20) for the bilinear form $\Omega$ is tantamount to saying that

$$^\flat u(v) = 0, \text{ for all } v \in E \text{ implies that } u = 0.$$

In other words, $^\flat u = 0$ implies that $u = 0$ which is equivalent to saying that *the linear map $^\flat$ associated to $\Omega$ is injective.* For a finite-dimensional vector space $E$, it follows from the injectivity of the linear map $^\flat$ and from $\dim E = \dim E^*$ that this map is bijective, i.e. $^\flat$ is an isomorphism. However, this is not ensured for an infinite-dimensional vector space. If the map $^\flat$ is an isomorphism, then the symplectic structure on $E$ is said to be *strongly non-degenerate.* However, for most symplectic structures appearing in infinite-dimensional Hamiltonian systems, this structure is only weakly non-degenerate. We will discuss several examples in the next subsection.

We note that for a (finite or infinite-dimensional) manifold $P$ endowed with a closed 2-form $\Omega$, the map (D.22) is given by

$$\begin{aligned} ^\flat \,:\, TP &\longrightarrow T^*P \\ X &\longmapsto i_X \Omega, \end{aligned} \tag{D.23}$$

where $X$ is a vector field on $P$ and $i_X P$ is the 1-form on $P$ obtained by contracting $\Omega$ with $X$.

## D.7 Hamiltonian vector fields and Hamiltonian equations

The considerations that we made for classical mechanics in equations (C.16)-(C.22) carry over (up to technical subtleties) to the case of classical field theory, i.e. to infinite-dimensional Hamiltonian systems. For simplicity, we consider the case of a single real scalar field $\varphi \in C^\infty(\mathbb{R}^d)$

and suppose that the associated momentum $\pi$ represents a smooth function on $\mathbb{R}^d$ with compact support, i.e. $\pi \in C_0^\infty(\mathbb{R}^d)$. Then, we have a well-defined weakly non-degenerate pairing

$$\langle \cdot, \cdot \rangle : C_0^\infty(\mathbb{R}^d) \times C^\infty(\mathbb{R}^d) \longrightarrow \mathbb{R}$$

$$(\pi, \varphi) \longmapsto \langle \pi, \varphi \rangle \equiv \int_{\mathbb{R}^d} d^d x \, \pi(\vec{x}) \varphi(\vec{x}). \tag{D.24}$$

*The symplectic 2-form (D.17) (with $a = 1$ and $\varphi^1 \equiv \varphi, \pi_1 \equiv \pi$) then represents a weakly non-degenerate symplectic structure on the phase space $E \equiv C^\infty(\mathbb{R}^d) \times C_0^\infty(\mathbb{R}^d)$: with the usual component field notation $(\varphi_1, \pi_1)$ for the*

$$\text{vector field} \quad \int_{\mathbb{R}^d} d^d x \left[ \varphi_1(\vec{x}) \frac{\delta}{\delta \varphi(\vec{x})} + \pi_1(\vec{x}) \frac{\delta}{\delta \pi(\vec{x})} \right], \quad \text{on } E, \tag{D.25}$$

we have

$$\Omega : E \times E \longrightarrow \mathbb{R}$$

$$\big((\varphi_1, \pi_1), (\varphi_2, \pi_2)\big) \longmapsto \Omega\big((\varphi_1, \pi_1), (\varphi_2, \pi_2)\big) \equiv \langle \pi_2, \varphi_1 \rangle - \langle \pi_1, \varphi_2 \rangle = \int_{\mathbb{R}^d} d^d x \, (\pi_2 \varphi_1 - \pi_1 \varphi_2). \tag{D.26}$$

Quite generally, one has the following results [348]. The cotangent bundle $P \equiv T^*\mathcal{Q}$ of a Banach manifold $\mathcal{Q}$ carries a canonical symplectic structure; this structure is strongly non-degenerate if the manifold $\mathcal{Q}$ is modeled on a Banach space $E$ that is *reflexive* (i.e. $(E^*)^* \cong E$), otherwise it is only weakly non-degenerate.

On the (linear) weak symplectic manifold $(E, \Omega)$ we have (as in the finite-dimensional case, see Eqn. (C.19)) the following

Poisson bracket on a symplectic manifold $(P, \Omega)$: $\qquad \boxed{\{F, G\} = \Omega(X_F, X_G),} \tag{D.27}$

for smooth functionals $F, G$ on $P \equiv E$. Here, the *Hamiltonian vector field* $X_H$ associated to a smooth functional $H$ on $P = E$ is given as in Eqn. (C.22):

Hamiltonian vector field associated to the functional $H$: $\quad X_H = \int_{\mathbb{R}^d} d^d x \left[ \frac{\delta H}{\delta \pi} \frac{\delta}{\delta \varphi} - \frac{\delta H}{\delta \varphi} \frac{\delta}{\delta \pi} \right]$ \tag{D.28}

(with $\delta H / \delta \pi \in C^\infty(\mathbb{R}^d)$ and $\delta H / \delta \varphi \in C_0^\infty(\mathbb{R}^d)$). Substitution of this expression into (D.27) (with $\Omega$ given by (D.26) or equivalently by (D.17), i.e. $\Omega = \int_{\mathbb{R}^d} d^d x \, \delta \varphi \wedge \delta \pi$) then yields

$$\{F, G\} = \int_{\mathbb{R}^d} d^d x \left( \frac{\delta F}{\delta \varphi} \frac{\delta G}{\delta \pi} - \frac{\delta F}{\delta \pi} \frac{\delta G}{\delta \varphi} \right), \tag{D.29}$$

i.e. the familiar expression (D.7). In practice, one assumes that, for any functional $H$, the functional derivatives $\delta H / \delta \varphi$ and $\delta H / \delta \pi$ are smooth functions which decrease strongly for $|\vec{x}| \to \infty$ so that one can perform partial integrations without generating boundary terms.

We note that the relation (C.16) defining the Hamiltonian vector field in the finite-dimensional case presently writes $i_{X_H} \Omega = \delta H$ and that the Hamiltonian equations $\dot{\Phi}^I = \{\Phi^I, H\} = -\{H, \Phi^I\} = X_H(\Phi^I)$ yield the familiar expression for the time evolution of $(\Phi^I) \equiv (\varphi, \pi)$: if we consider the symplectic form $\Omega$ given by $\Omega = \int_{\mathbb{R}^d} d^d x \, \delta \varphi \wedge \delta \pi$ or equivalently by (D.26), we get the

Hamiltonian equations: $\qquad \dot{\varphi} = \frac{\delta H}{\delta \pi}, \quad \dot{\pi} = -\frac{\delta H}{\delta \varphi}. \tag{D.30}$

On a weak symplectic vector space $(E, \Omega)$, the relation $\{F, G\} = 0$ for all admissible functionals $G$ is (by virtue of (D.27)) tantamount to $\Omega(X_F, X_G) = 0$ for all vector fields $X_G$. (Here, the tangent vectors $X$ to the vector space $E$ represent themselves vectors belonging to $E$.) Now, the weak degeneracy of $\Omega$ implies that $X_F = 0$ (for the vector field $X_F$ defined by $i_{X_F} \Omega = \delta F$), i.e. $F$ is constant. Thus, we recover (in the symplectic setting) the non-degeneracy condition for a Poisson bracket that we introduced in Eqn. (D.11).

As we noted in our discussion of Poisson structures, the Poisson bracket (D.12) introduced for the KdV equation (for smooth functions $x \mapsto \delta F/\delta u(x)$ and $x \mapsto \delta G/\delta u(x)$ decreasing at infinity) is non-degenerate and thereby equivalent to a weak symplectic structure, the latter being explicitly given by (D.19). Similarly, the Poisson bracket (D.13) introduced for the case of periodic functions $x \mapsto u(x)$ is non-degenerate upon restriction to a level surface (D.14) and thus equivalent to a weak symplectic structure.

## D.8 On the Darboux theorem in the infinite-dimensional case

A way to formulate the Darboux theorem on a finite-dimensional symplectic manifold $(M, \omega)$ is to say that in the vicinity of any point of $M$ one can find local coordinates $(q^i, p_i)_{i=1,\dots,n}$ with respect to which the symplectic form $\omega$ is constant, i.e. $\omega = dq^i \wedge dp_i$, see Subsection C.3. In 1969, A. Weinstein [349] has generalized this theorem to the case of an infinite-dimensional Banach manifold which is endowed with a strongly non-degenerate symplectic form – see reference [350] for further discussion. Shortly thereafter, J. Marsden showed (by providing a counterexample) that the theorem fails to hold in general for the case of a *weakly* non-degenerate form [348]. In 1999, D. Bambusi [351] gave necessary and sufficient conditions for the existence of local Darboux coordinates for a weakly non-degenerate symplectic form on a Banach manifold which is modeled on a reflexive Banach space. For some related results we refer to the recent works [352, 353].

In summary, *in the physically most interesting instance of a weakly non-degenerate symplectic form, the issue of Darboux charts is non-trivial and fairly technical.*

## D.9 About the action and the variational principle

The existence of the action functional (defined as an integral over unbounded space-time $M$, e.g. Minkowski space-time $\mathbb{R}^n$) sensibly depends on the properties of fields and of the Lagrangian density, e.g. see reference [354]. Even if one makes the familiar assumption that fields fall off rapidly at spatial infinity, the integral over time generally diverges. From the mathematical point of view, this divergence actually ensures the existence of a non-trivial symplectic 2-form $\Omega$, see section 2.2 of reference [71].

Despite this existence problem of the action functional on $\mathbb{R}^n$, the Euler-Lagrange equations associated to a Lagrangian density $\mathcal{L}$ may be viewed as well defined stationary points of all "local action functionals"

$$S_B[\varphi] \equiv \int_B d^n x \, \mathcal{L}\big(x, \varphi(x), (\partial_\mu \varphi)(x)\big), \tag{D.31}$$

where $B \subset \mathbb{R}^n$ are *bounded* open subsets of $\mathbb{R}^n$ and where the fields $\varphi$ are assumed to be continuously differentiable, real-valued functions on $B$ which vanish on the boundary of $B$ [354]. The integral (D.31) can be obtained from $S[\varphi] \equiv \int_M d^n x \, \mathcal{L}$ following E. C. G. Stückelberg by multiplying $\mathcal{L}$ by a function $g \in C^\infty(M)$ of compact support [343]. Another approach [71] consists in viewing the variational condition

$$0 = \frac{\delta S}{\delta \varphi}\bigg|_{\varphi_0} \equiv \frac{dS}{d\varepsilon}[\varphi_\varepsilon]\bigg|_{\varepsilon=0}$$

(where $\varepsilon \mapsto \varphi_\varepsilon$ denotes a one-parameter family of fields), as an abusive notation for

$$\int_{\mathbb{R}^n} d^n x \, \frac{d\mathcal{L}}{d\varepsilon}(\varphi_\varepsilon)\Big|_{\varepsilon=0} = 0,$$

where $\varphi_\varepsilon$ does not depend on $\varepsilon$ outside of a compact set $B \subset \mathbb{R}^n$ (or more generally a compact set $B \subset M$ for a space-time manifold $M$). Thus, *the derivation of field equations from the variational principle admits a solid mathematical foundation (by resorting to compact subsets)* despite the divergence problems of the action functional.

### D.10  On infinite-dimensional vector spaces and manifolds

Classical fields locally parametrize infinite-dimensional manifolds. Thus, the general properties of these fields determine the class of manifolds to be considered and in particular the class of infinite-dimensional vector spaces on which these manifolds are to be modeled. The *pragmatic point of view adopted in theoretical physics* is to dispense largely with mathematical rigor in this respect and to assume that fields are given by smooth functions on space-time while eventually approximating discontinuous or generalized functions in an appropriate way by sequences of smooth functions [21].

From the mathematical point of view, a smooth real infinite-dimensional manifold is a manifold $P$ for which the local coordinates belong to some real infinite-dimensional vector space $E$. Depending on the type of vector space that is considered, different classes of smooth infinite-dimensional manifolds can be introduced [145, 350, 355, 356]. A case which has been studied a lot during the fifties, sixties and seventies [118, 350, 357] is the one of *Banach manifolds* where $E$ represents a real infinite-dimensional *Banach space,* i.e. a real vector space which is endowed with a norm and which is complete with respect to this norm. We recall that the Banach space $E$ is said to be reflexive if $(E^*)^* \cong E$. Every finite-dimensional real vector space endowed with a norm represents a finite-dimensional Banach space. An important example of a non-reflexive Banach space (that we mentioned in the previous subsection in relationship with the variational principle) is the space $C^1(B)$ of continuously differentiable functions on a compact subset $B \subset \mathbb{R}^n$. A particular class of reflexive Banach spaces is given by the real Hilbert spaces for which one considers the norm induced by the inner product: as special cases of the latter one then has the separable Hilbert spaces like $L^2(\mathbb{R}^d)$ which admit a countably infinite orthornormal basis. Another example for a real separable Hilbert space of infinite dimension is given by the $L^2$-type *Sobolev space* which plays an important role in the theory of partial differential equations [358, 359]: for $m \in \mathbb{N}$, one considers the space

$$H^m(\mathbb{R}) \equiv \{f \in L^2(\mathbb{R}) \,|\, f', f'', \ldots, f^{(m)} \in L^2(\mathbb{R})\}, \tag{D.32}$$

endowed with the inner product[34] $\langle f, g \rangle_{H^m} \equiv \sum_{k=0}^{m} \langle f^{(k)}, g^{(k)} \rangle_{L^2}$.

Unfortunately, some natural spaces of physical fields are not Banach spaces. For instance the vector space $C^\infty(\mathbb{R}, \mathbb{R})$ of smooth real-valued functions on $\mathbb{R}$ cannot be turned into a Banach space, i.e. there exists no norm for this space so that all functions (e.g. the exponential function) have a finite norm and that the space is complete with respect to this norm. Similarly the vector space $\mathcal{D}(\mathbb{R})$ of smooth functions on $\mathbb{R}$ with compact support or the Schwartz space $\mathcal{S}(\mathbb{R})$ of smooth strongly decreasing test functions (which spaces play a fundamental role in

---

[34]In expression (D.32), the derivatives are to be understood in the distributional sense. Thus, $f' \in L^2(\mathbb{R})$ means that there exists a square-integrable function which is denoted by $f' : \mathbb{R} \to \mathbb{R}$ and defined by the relation $\int_{\mathbb{R}} dx \, f' \varphi \equiv -\int_{\mathbb{R}} dx \, f \varphi'$ for all smooth test functions, i.e. $\varphi : \mathbb{R} \to \mathbb{R}$ of compact support, $\varphi \in \mathcal{D}(\mathbb{R}) \equiv C_0^\infty(\mathbb{R})$: this function $f'$ is, if it exists, uniquely defined up to a set of Lesbegue measure zero, i.e. it represents a well defined element of $L^2(\mathbb{R})$. We note that the functions belonging to a Sobolev space have some, though not too great smoothness properties; indeed they can still be rather badly behaved, e.g. be discontinuous and/or unbounded.

the theory of distributions) do not admit a norm with respect to which they are complete. The study of spaces of smooth functions led to the introduction of the notion of *Fréchet space* (i.e. a topological vector space which is metrizable and complete[35] [360]) as well as manifolds which are modeled on such a vector space. Indeed, differential calculus can be formulated in these spaces to a large extent, see [356] and references therein. We note that Poisson brackets on a manifold modeled on such a locally convex vector space have recently been investigated [361].

The various classes of infinite-dimensional manifolds that we just mentioned originated from the study of specific mathematical or physical problems and were designed to deal with these applications. While all of these approaches have there interest and have allowed to establish important mathematical results, a different point of view has been put forward in the eighties by A. Frölicher and A. Kriegl [362] in their quest for formulating **differential calculus in an infinite-dimensional non-normed vector space,** see the monograph [355] and the review [363]. The underlying idea is to introduce a differential calculus which is as easy to use as possible and that the applications requiring further assumptions should then be treated in a setting depending on the specific problem. By definition, a *convenient manifold* is a smooth real manifold that is modeled on a so-called *convenient vector space* $E$: the latter is a locally convex topological vector space such that a curve $\gamma : \mathbb{R} \to E$ is smooth if and only if $\lambda \circ \gamma$ is smooth for all continuous linear functionals $\lambda$ on $E$. For the particular case of Fréchet spaces like $C^{\infty}(\mathbb{R})$, the *convenient calculus* coincides with the Gateaux approach to differentiation, but for more general model vector spaces it yields a notion of smooth map which does not necessarily imply continuity. For the relationship between different definitions of infinite-dimensional manifolds and notions of smoothness (in particular the *Bastiani calculus* which was introduced by Andrée Bastiani in her PhD thesis [364] and which is considered in works of field theory [11, 145]), we refer to [343, 356, 365, 366] and the appendix of [367].

By way of conclusion, we may say that, in dealing with geometric structures on a space of classical fields in theoretical physics, a minimum of mathematical rigor is required to take into account physically important aspects like degeneracies. *The specification of a rigorous analytic framework eventually depends on the applications one has in mind* and the high degree of technicality involved requires a fair amount of motivation and a well chosen bibliography. Here, we only mention the introductory textbook [368] which is devoted to calculus in normed vector spaces and more specifically the recent work [343] addressing physical aspects, in particular the concept of locality which is fundamental in field theory [11]. In fact, the authors of reference [11] and [343] argue on physical and mathematical grounds that the *choice $E = C^{\infty}(M)$ for the space of classical fields on a space-time manifold $M$* is appropriate for classical field theories and their quantization, and that the *Bastiani differentiability* (which corresponds to the definition of functional derivatives in physics) is best adapted for the needs of physics. Since $C^{\infty}(M)$ represents a Fréchet space, the Bastiani differentiability of functionals is equivalent to the *convenient differentiability* mentioned above.

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
