# Peer review of "Covariant canonical formulations of classical field theories"

_SciPost Physics Lecture Notes, doi:SciPost Phys. Lect. Notes 77 (2023)_

## Round 2 · Referee Report · Anonymous · 2022-3-30

Report

This is an interesting review article on the so-called covariant canonical approaches to classical field theories. The review is well-written and has the advantage of presenting various approaches scattered in existing literature in a unified picture with coherent notations. Because this is mainly a review article, I will not focus on possible criticisms of shortcoming of the various approaches described, but instead list several points which I think should be revised in order to improve the clarify of the manuscript. I recommend the manuscript for publication in SciPost, provided the following points are addressed and clarified:

- Generally speaking it could be useful to clarify the role of boundaries at various points throughout the manuscript. It seems to me that in most of the manuscript the author is considering that boundaries are absent, although they are clearly important for the discussions of e.g. section 6. The author could for example state more precisely in which section boundaries are present or not, and if they are absent, how the various results would extend in the case of non-trivial boundaries. For example, since the Peierls bracket discussed in section 3 relies on the use of Green functions, it seems that the latter would be strongly affected by the presence of boundaries. Similarly, in the middle of page 38 (paragraph ‘’From the invariance…’’) it is claimed that (4.39) and (4.45) are equivalent, but it is not clear whether this holds because there are no boundaries. It also does not seem that reference [111] cites at this point is the most appropriate reference for this statement. Similarly, in (4.12) for the proof of invariance of the symplectic 2-form under choice of \Sigma, the author is then seemingly introducing a boundary, but invoking the condition that J^\mu vanishes on the boundary to get the desired result. I agree that this is a possibility, but once again, I think that this part would benefit from a clearer discussion of the role of boundaries. It seems for example that references such as https://arxiv.org/abs/1906.08616 discuss the role of boundaries and relation with the Peierls bracket.

- in section 6.5 it is not clear to me what the author means by ‘’asymptotic symmetries’’. There is no notion of ‘’asymptotia’’ defined in this part, especially in the subsection about general relativity (where there is also no discussion on ‘’symmetries’’).

- the author could consider including a discussion on the so-called Weiss variational principle (https://arxiv.org/abs/1708.04489) which also gives rise to a covariant Hamiltonian.

---

## Round 2 · Referee Report · Anonymous · 2022-6-3

Strengths

The review is pedagogical with particular emphasis on the development of the field with rather exhaustive references.

Weaknesses

Some technical concepts are sometimes mentioned without definition and the boundary between what is pedagogically described and what is only mentioned as a reference is not always sharp.

Report

In this review the author gives a comprehensive description of several approaches to the covariant canonical formulation of relativistic field theories with constraints. It is often necessary to use the hamiltonian formulation of a theory to describe its Cauchy problem, its conserved charges, and eventually its quantization. The lack of Lorentz invariance and more generally diffeomorphism invariance is often a technical drawback that has lead mathematical physicists to define a covariant canonical formulation of classical field theory. This review describes these attempts with particular emphasis on the historical evolution of the field. The three approaches described in the review are the multiphase or multisymplectic approach, the covariant phase space approach and finally the variational bicomplexe that somehow encompasses the two other approaches. The review emphasizes that these constructions are all equivalent, including the Peierls bracket (a Lorentz invariant bracket defined such as to reproduce the commutator of causal quantum fields). I guess that this equivalence is only at the formal level usually required by physics analysis, but the mathematical objects may differ in a rigorous definition since the truncated phase space seems to be defined with the topology of a finite dimensional jet space whereas the covariant phase space is defined with the appropriate topology on the function space of solutions to the equations of motion, which may differ from the topology of the infinite jet space in the bicomplexe. The review is well written and can certainly provide an interesting reference in the literature that deserves to be published.

Nevertheless, I believe the presentation could perhaps be improved.

I found the introduction and in particular the overview rather hard to read. The first presentation of the different approaches could be more digest if the author was not putting so much technical details on the evolution of the field and on comparaisons of objects that are not yet defined in the review. I appreciate that the author is very precise about references, but it may help to postpone some of the historical points to section 1.4. Actually I have not found the precise definition of the jet bundle affine dual in the review whereas it is already mentioned twice in page 2. If it is only going to be defined as the extended multi-phase space, including both the canonical momentum vector field and the Hamiltonian as a separated variable, I believe that this more mathematical object could only be mentioned in section 2 with the appropriate reference. The description of line 2.b is also particularly cryptic. Since the author decides to enumerate the various approaches according to the way they appear in the flow diagram, the flow diagram should probably appear in the same subsection. The reference to Appendix C in the discussion of line 1 is not very clear, probably the author means that a review of the Hamiltonian formulation can be found in Appendix C, but I think this should be said in words, perhaps in a footnote.

This may be a personal opinion, but I was slightly disappointed that no more details would be given in section 3.3 about the Peierls bracket of conserved charges in general relativity, and similarly in 6.1.3 for corner charges in general relativity. These examples are particularly relevant to recent research developments in asymptotic symmetry and I think they would deserve to be included in the review.

Sometimes the author briefly mention something without defining it and nevertheless refers to it afterward just like if he did. This the case for the jet bundle affine dual in the introduction as mentioned above, but also for the Koszul-Tate differential mentioned in (6.15) and then referred to in (6.38). The Koszul-Tate complexe is mentioned three times in the review but never defined. I believe it would be appropriate to give the definition in (6.15) and maybe its relation to the antifield formalism of section 6.6.2.

I have not understood why the author needs to linearise in (6.96) to define the color charge in (6.100).

Despite these minor queries, I believe this review is of significant interest to the community and deserves to be published in Scipost.

Requested changes

1) I found the introduction and in particular the overview rather hard to read. The first presentation of the different approaches could be more digest if the author was not putting so much technical details on the evolution of the field and on comparaisons of objects that are not yet defined in the review. I appreciate that the author is very precise about references, but it may help to postpone some of the historical points to section 1.4.

2) Actually I have not found the precise definition of the jet bundle affine dual in the review whereas it is already mentioned twice in page 2. If it is only going to be defined as the extended multi-phase space, including both the canonical momentum vector field and the Hamiltonian as a separated variable, I believe that this more mathematical object could only be mentioned in section 2 with the appropriate reference.

2) The description of line 2.b is also particularly cryptic.

3) Since the author decides to enumerate the various approaches according to the way they appear in the flow diagram, the flow diagram should probably appear in the same subsection.

4) The reference to Appendix C in the discussion of line 1 is not very clear, probably the author means that a review of the Hamiltonian formulation can be found in Appendix C, but I think this should be said in words, perhaps in a footnote.

5) This may be a personal opinion, but I was slightly disappointed that no more details would be given in section 3.3 about the Peierls bracket of conserved charges in general relativity, and similarly in 6.1.3 for corner charges in general relativity. These examples are particularly relevant to recent research developments in asymptotic symmetry and I think they would deserve to be included in the review.

6) Sometimes the author briefly mention something without defining it and nevertheless refers to it afterward just like if he did. This the case for the jet bundle affine dual in the introduction as mentioned above, but also for the Koszul-Tate differential mentioned in (6.15) and then referred to in (6.38). The Koszul-Tate complexe is mentioned three times in the review but never defined. I believe it would be appropriate to give the definition in (6.15) and maybe its relation to the antifield formalism of section 6.6.2.

7) I have not understood why the author needs to linearise in (6.96) to define the color charge in (6.100). Could you explain why.

---

## Round 3 · Referee Report · Anonymous · 2023-7-17

Report

This revised version addresses in a constructive manner the various questions and comments which were raised, so I recommend it for publication.

---

## Round 3 · Referee Report · Anonymous · 2023-10-20

Strengths

The review is pedagogical with particular emphasis on the development of the field with rather exhaustive references.

Weaknesses

Does not discuss much quantization.

Report

Let me thank the author for doing many corrections. I believe the introduction is clearer now, and I appreciated the additions in sections 4.3 to 4.5, 7.7 and Appendix B. I recommend the paper for publication.

Requested changes

Let me just mention a minor misprint above equation 4.15 were the text is strangely cut.

---

## Round 3 · Author Response

I sincerely thank the two referees for their careful review and positive feedback
as well as for the detailed comments and insightful suggestions.
The latter led me to revise and improve several parts or points of the text.
I have also taken this opportunity to update and complete the references.
I hope that these additions, modifications and comments represent satisfactory answers to
all of the points raised by the referees.

---

## Round 3 · List of Changes

I first spell out the major changes made in the text:

- Points 1-4 of referee 2: To make the introductory part easier to read, I have reorganized the introduction
by postponing the historical evolution and synthetic overview to a new section 2 entitled "Pre-/overview of results
and historical evolution of the subject".
As emphasized there, the goal of this section is to try to convey already some of the basic concepts and ideas
(within their historical context) while postponing the details to the
later parts of the notes. Accordingly, I have added references for the
mathematical notions (that are mentioned here)
to the adequate equations or subsections
of the text where these notions are discussed.
These indications should help the reader to go right away to the technical details
if he wishes to do so in this overview part of the notes.
Concerning this part, I agree with referee 2 that the description of the multisymplectic approach is somewhat cryptic
from the mathematical point of view. In fact, I have tried to put forward the physical aspects
while referring to various extensive mathematical reviews which focus on the latter aspects.
Following the suggestion of referee 2, I have postponed the mention of the "twisted affine dual" to section 3 (footnote 3).

- Point 5 of referee 2: For the Peierls bracket, I have expanded the subsection "Geometric symmetries and conservation laws"
(now subsection 4.3) so as to elaborate on the given example and on the bracket of charges.
Moreover, I added some comments concerning the mathematical underpinnings (new subsection 4.4).
The definition and construction of gravitational charges is now discussed in some detail in the new subsections 7.7 and 7.8.

- Point 6 of referee 2: Since the Koszul-Tate differential was mentioned (but not defined) in several places of the notes,
I have included a new appendix B which provides a synthetic introduction to the various differentials in field space that are
considered in the main body of the text: the horizontal differential, the BRST differential, the Koszul-Tate differential
and the BV differential. Hopefully, this addition is also useful in its own.

- Referee 1 raises the interesting and important issue of boundaries.
Concerning the boundaries of space-time at infinity or at a finite distance, I have added a
new subsection 1.3 in the introduction. (Here, I have referred in particular to the quite recent review
"From asymptotic symmetries to the corner proposal" by L. Ciambelli: the introduction of this review
provides several hundred references which address the issue of boundary conditions and terms
as well as the relationship with theories in the bulk. Moreover, it elaborates in detail on the particular approach
of the corner proposal.)
In the main part of the text, I have followed the suggestion of referee 1 by adding
some comments on the issue of boundaries
within the different approaches that are considered in the notes (new subsections 3.5, 4.5, 5.5, 6.7 and 7.8
as well as an elaboration on boundary conditions and boundary actions in general relativity on pages 47-51).

- Since most of the recent literature on the covariant phase space approach to gravitational theories
is based on the work of R. Wald and his collaborators, I have added an introduction to this topic
while considering the recently given geometric reformulation of this work (new section 7.7).
This section also addresses the relationships with the results discussed in other parts of the notes.

Concerning some details raised by the referees:

- Point 7 of referee 2: As I have now spelled out in the text, eq.(7.97) represents the (complete) expansion
of the Lagrangian density with respect to the coupling
constant g viewed as a deformation parameter. Only the part L _0 describing the dynamics of a collection
of free Abelian fields is invariant under the given symmetry transformation (7.99) thus leading to the conservation of the
associated ``color charges'' (7.101). As noted after eq.(7.101), the latter have the same expression as (7.66)
for vanishing background charges. (Other surface charges as well as their algebra are discussed in the cited reference [242].)

- Referee 1 made a remark concerning the equivalence of different expressions for the symplectic two-form of gravity
(which are now numbered by (5.44) and (5.55)): as he suggested I added an earlier reference for this result (namely the work
of J. Lee and R. Wald) and I made more precise the role of the boundary term in the action.

- Referee 1 emphasized the role of boundaries and boundary conditions for establishing (in eq.(5.12))
the independence
of the symplectic two-form on the hypersurface Sigma over which one integrates.
The new subsection 7.8 now treats some basic aspects of manifolds with a boundary
notably following the work of D. Harlow and J. Q. Wu. The results obtained by these authors
concerning the equivalence of Poisson brackets and Peierls brackets are now mentioned in the new subsection 5.5.

- Concerning the remark of referee 1 on the subsection "Asymptotic symmetries" (now numbered as subsection 7.5):
At the beginning of this part, I have added a parenthesis concerning the ``asymptotia''.
Moreover, in the paragraph dealing with general relativity I added a parenthesis (before eq.(7.71))
to emphasize that the Killing vector fields (of the background metric) parametrize the asymptotic symmetries.

- Following the suggestion of referee 1, I have included some comments and relevant references to the
so-called Weiss variational principle (after equation (6.44) as well as on top of page 122).
Indeed, as it is explicitly pointed out in reference [234], this variational principle amounts to a derivation
of Noether's first theorem if the infinitesimal variations are viewed as infinitesimal symmetry transformations
of an invariant action functional. Thus it leads to the canonical energy-momentum tensor in field theory
(which is referred to as ``Hamiltonian Complex'' or ``Hamiltonian tensor'' in reference [234]).

Resubmission 2109.07330v4 on 26 October 2023

---

## Round 4 · List of Changes

Text cut modified above eq. (4.15)
(as requested by one of the referee's)

You are currently on this page

Resubmission 2109.07330v4 on 26 October 2023

---

## Editorial Decision

published